# Heterogeneous Sufficient Dimension Reduction and Subspace Clustering

**Lei Yan** [1]   **Xin Zhang** [1]   **Qing Mai** [1]

## Abstract

Scientific and engineering applications are often heterogeneous, making it beneficial to account for latent clusters or sub-populations when learning low-dimensional subspaces in supervised learning, and vice versa. In this paper, we combine the concept of subspace clustering with model-based sufficient dimension reduction and thus generalize the sufficient dimension reduction framework from homogeneous regression setting to heterogeneous data applications. In particular, we propose the mixture of principal fitted components (mixPFC) model, a novel framework that simultaneously achieves clustering, subspace estimation, and variable selection, providing a unified solution for high-dimensional heterogeneous data analysis. We develop a group Lasso penalized expectation-maximization (EM) algorithm and obtain its non-asymptotic convergence rate. Through extensive simulation studies, mixPFC demonstrates superior performance compared to existing methods across various settings. Applications to real world datasets further highlight its effectiveness and practical advantages.

## 1. Introduction

Reducing high-dimensional data to a low-dimensional representation is one of the most important steps in multivariate statistics and various applied sciences. Typically, this is achieved by projecting data onto a single low-dimensional subspace. Among unsupervised dimension reduction methods, principal component analysis (PCA) is probably the most popular one. However, when applied in the context of regressing a univariate response $Y$ on $p$-dimensional predictor $\mathbf{X}$, PCA faces three critical limitations: data heterogeneity, loss of information on regression, and the curse of

dimensionality. First, real-world data often lie in a union of multiple subspaces with unknown membership, reflecting underlying latent sub-populations. Second, PCA, as an unsupervised method, equates variation with information and thus disregards the specific relationship between $Y$ and $\mathbf{X}$. Third, in the high-dimensional setting where $p$ is much larger than the sample size $n$, the subspace estimated by PCA can be highly unreliable, or even orthogonal to the true subspace (Baik & Silverstein, 2006; Paul, 2007).

Addressing the challenges of heterogeneity and high dimensionality is crucial for statistical and machine learning methods. Subspace clustering is a family of powerful methods for clustering the data into multiple subspaces(Vidal, 2011; Soltanolkotabi & Candés, 2012). Most subspace clustering methods generalize PCA and factor analysis from a single subspace to a union of multiple subspaces (Agarwal & Mustafa, 2004; Vidal et al., 2005; Yan & Pollefeys, 2006; Tron & Vidal, 2007; Favaro et al., 2011; Elhamifar & Vidal, 2013). However, like PCA, these methods ignore the response, leading to inevitable loss of regression relevant information. Furthermore, subspace clustering methods often assume clean observations with small random noise (Kanatani, 2001; Vidal et al., 2005), meaning data points are expected to lie nearly exactly within a subspace. When the observations are subject to significant random errors, these methods frequently fail to accurately identify clusters.

Conversely, sufficient dimension reduction (SDR) provides a supervised framework by projecting $\mathbf{X}$ onto a low-dimensional subspace while preserving all relevant regression information. Formally, we have

$$Y \perp\!\!\!\perp \mathbf{X} \mid \mathbf{P}_{\mathcal{S}}\mathbf{X}, \tag{1}$$

where $\mathbf{P}_{\mathcal{S}}$ is the projection matrix onto $\mathcal{S}$. The intersection of all the subspaces satisfying (1) is called the central subspace. Recent advances in deep learning-based SDR methods (Banijamali et al., 2018; Liang et al., 2022; Kapla et al., 2022; Huang et al., 2024; Chen et al., 2024) have demonstrated the potential to capture complex nonlinear structures in high-dimensional data.

While SDR guarantees regression sufficiency (Li, 1991; Cook & Forzani, 2008), it overlooks the heterogeneous nature of scientific and engineering applications. Incor-

[1]Department of Statistics, Florida State University, Tallahassee, Florida, United States. Correspondence to: Xin Zhang <henry@stat.fsu.edu>.

*Proceedings of the 42^{nd} International Conference on Machine Learning*, Vancouver, Canada. PMLR 267, 2025. Copyright 2025 by the author(s).

porating heterogeneity into SDR has the potential to enhance dimension reduction by uncovering latent subpopulations. Motivated by this, we integrate the subspace clustering concept with model-based SDR.

**Contributions.** In this paper, we propose a novel mixture of principal fitted component (mixPFC) model, designed for simultaneous clustering, variable selection, and dimension reduction. It has the following major contributions.

- **Supervised Subspace Clustering:** By extending subspace clustering into a supervised framework, mixPFC identifies subspaces that preserve regression information, addressing both heterogeneity and predictive accuracy. This contrasts sharply with unsupervised approaches (Elhamifar & Vidal, 2013; Ji et al., 2017; Cai et al., 2022). Leveraging response information, mixPFC allows for exact overlap between subspaces, overcoming limitations of classical methods that require separation conditions (e.g., minimal angle condition in Soltanolkotabi & Candés (2012)). Building upon SDR, mixPFC targets central subspaces rather than subspaces where $\mathbf{X}$ resides. This ensures the low-dimensional representation of $\mathbf{X}$ preserves all the information relevant to regression. Moreover, unlike most subspace clustering methods that separate clustering from subspace estimation, mixPFC performs both tasks jointly in a unified framework.

- **High-Dimensional Estimation:** We develop a group penalized expectation-maximization (EM) algorithm for the mixPFC model. Extending SDR to high dimensions is a challenging and nascent research area (Lin et al., 2018; 2019; Tan et al., 2018; 2020; Zeng et al., 2024). Existing approaches often require inverting $p \times p$ matrices or estimating parameters in a $p^2$-dimensional space, which poses scalability challenges. Our model, designed to estimate multiple heterogeneous subspaces in different unknown subpopulations, is much more complicated. To address this challenge, we formulate the subspace estimation as a convex optimization over an approximately $p$-dimensional parameter space. We further incorporate a group Lasso penalty (Yuan & Lin, 2006) for coordinate-independent variable selection (Chen et al., 2010).

- **Theoretical Guarantees:** We establish theoretical results for the proposed group penalized EM algorithm. While classical EM theories only guaranteed asymptotic convergence to a fixed point, we derive a non-asymptotic result that mixPFC converges geometrically to a fixed point that is within statistical precision of the unknown **true** parameter. This stronger type of guarantee has emerged only recently (Balakr-

ishnan et al., 2017). Unlike many existing proofs in high-dimensional EM algorithms, our analysis does not require sample splitting (Kwon et al., 2019) and allows a relatively general model. Specifically, we derive a non-asymptotic convergence rate for a two-mixture principal fitted components model with unknown mixing proportions and without restrictions on the minimum angle between subspaces.

## 2. Mixture of Principal Fitted Components

We extend the framework of sufficient dimension reduction by introducing a latent variable to model heterogeneity in data. In particular, we consider univariate (continuous or discrete) response $Y \in \mathbb{R}$, multivariate predictor $\mathbf{X} \in \mathbb{R}^p$, and a latent categorical variable $W \in \{1, 2, \ldots, K\}$. We aim to estimate $K$ subspaces $\mathcal{S}_w$, $w = 1, \ldots, K$, such that

$$
\begin{aligned}
Y &\perp\!\!\!\perp \mathbf{X} \mid (\mathbf{P}_{\mathcal{S}_w}\mathbf{X}, W = w), \\
\Pr(W \mid Y, \mathbf{X}) &= \Pr(W \mid Y, \mathbf{P}_{\mathcal{S}}\mathbf{X})
\end{aligned}
\tag{2}
$$

where $\mathcal{S} = \sum_{w=1}^{K} \mathcal{S}_w \subseteq \mathbb{R}^p$ offers the usual SDR as seen in the literature and each $\mathbf{P}_{\mathcal{S}_w}$ is the projection matrix onto $\mathcal{S}_w$ to capture the relationship between $Y$ and $\mathbf{X}$ within cluster $w$. Let $\boldsymbol{\beta} \in \mathbb{R}^{p \times d}$ denote a basis matrix of $\mathcal{S}$. Then, the projected data $\boldsymbol{\beta}^T\mathbf{X}$ contains all relevant information in $\mathbf{X}$ to be combined with response information $Y$ for clustering data into $K$ clusters. When $W$ is observable, the smallest space $\mathcal{S}_w$ is known as the conditional central subspace, and is a building block for studying the partial central subspace (Chiaromonte et al., 2002). However, our problem is much more challenging because $W$ is latent and has to be inferred from data. Moreover, unlike existing partial/conditional central subspace methods, we further incorporate variable selection for high-dimensional studies.

We propose the mixPFC model, as a generative mixture of principal fitted components (PFC),

$$
\begin{aligned}
\mathbf{X} \mid (Y, W = w) &\sim N(\boldsymbol{\mu}_w + \boldsymbol{\Gamma}_w\mathbf{f}(Y), \boldsymbol{\Delta}), \\
\Pr(W = w) &= \pi_w, \ w = 1, \ldots, K,
\end{aligned}
\tag{3}
$$

where $\boldsymbol{\mu}_w \in \mathbb{R}^p$ is the center of each cluster, $\boldsymbol{\Gamma}_w \in \mathbb{R}^{p \times q}$ is the coefficient matrix that represents the relationship between $Y$ and $\mathbf{X}$ in each cluster, $\mathbf{f}(\cdot) = (f_1(\cdot), \ldots, f_q(\cdot))^T : \mathbb{R} \mapsto \mathbb{R}^q$ is a set of pre-specified fitting functions that introduces non-linear relationships, $\boldsymbol{\Delta} \in \mathbb{R}^{p \times p}$ is symmetric positive definite matrix, and $\pi_w > 0$ is the mixture probabilities with $\sum_{w=1}^{K} \pi_w = 1$.

The mixPFC model unifies and generalizes many model-based clustering and model-based SDR approaches. When $K = 1$, the mixPFC reduces to the PFC model (Cook & Forzani, 2008), which further reduces to the probabilistic principal component analysis (PCA) model (Tipping & Bishop, 1999) by restricting $\boldsymbol{\Delta} = \sigma^2\mathbf{I}_p$ and replacing $\mathbf{f}(Y)$

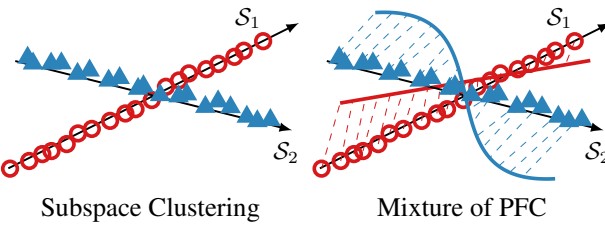

*Figure 1.* The mixPFC model enhances subspace clustering by considering how the response (solid line) changes as the predictor varies within each cluster: linear response effects in the first mixture and non-linear in the second.

as latent variables $\boldsymbol{\nu} \sim N(0, \mathbf{I}_q)$. Therefore, our mixPFC, by allowing general covariance structures and incorporating response information, is a supervised generalization of the probabilistic PCA. On the other hand, if we completely remove the effect of the response $\boldsymbol{\Gamma}_w \mathbf{f}(Y)$, the mixPFC becomes the Gaussian mixture model (GMM) (McLachlan et al., 2019; Cai et al., 2019). If we further have $\boldsymbol{\Delta} = \sigma^2 \mathbf{I}_p$, mixPFC is a model-based interpretation of K-means clustering (Forgy, 1965; MacQueen et al., 1967). When $Y$ is categorical, the mixPFC model reduces to the mixture discriminant analysis model (Hastie & Tibshirani, 1996; Fraley & Raftery, 2002).

The mixPFC can also be viewed as an extension of subspace clustering. Figure 1 demonstrates the advantage of mixPFC over subspace clustering. In subspace clustering, we have a fully unsupervised problem on $\mathbf{X}$, but in mixPFC we further have the response $Y$ to guide our clustering, which can be very informative. Consequently, subspace clustering requires a minimal angle condition for identifiability (Soltanolkotabi & Candés, 2012), but in mixPFC we allow non-overlap, partial overlap, and complete overlap between $\mathcal{S}_j$ and $\mathcal{S}_k$ for any $(j, k)$. With the response $Y$, even when $\mathcal{S}_j = \mathcal{S}_k$, we can still have $\boldsymbol{\Gamma}_j \neq \boldsymbol{\Gamma}_k$ to ensure cluster identifiability, although $\mathrm{span}(\boldsymbol{\Gamma}_j) = \mathrm{span}(\boldsymbol{\Gamma}_k)$. A more concrete demonstration is given in Figure 2, where we have conducted two toy example simulations based on the proposed model (3). The first toy example in Figure 2 (a) has $\mathcal{S}_1 = \mathcal{S}_2$, but the response $Y$ has different relationships with $\mathbf{X}$ in the two clusters. The subspace clustering would completely fail to identify the subspace or to cluster data while mixPFC works well and produces near perfect subspace estimation and clustering results. In Figure 2 (b), we have another simulation where $\mathcal{S}_1 \perp \mathcal{S}_2$, which is an ideal setup for subspace clustering methods. We applied our proposed method and two popular subspace clustering methods: random sample consensus (RANSAC, Tron & Vidal (2007)) and sparse subspace clustering (SSC, Elhamifar & Vidal (2013)). The clustering errors by subspace clustering methods, 8.4% (SSC) and 13.0% (RANSAC), are reduced by mixPFC to 3% thanks to the additional response super-

vision.

Finally, the following proposition identifies the key parameter for fitting mixPFC.

**Proposition 2.1.** *Under model* (3), *the smallest subspaces satisfying* (2) *are* $\mathcal{S}_w = \mathrm{span}(\boldsymbol{\Delta}^{-1}\boldsymbol{\Gamma}_w)$, $w = 1, \dots, K$. *Consequently,* $d_w = \dim(\mathcal{S}_w) = \mathrm{rank}(\boldsymbol{\Gamma}_w)$ *and* $d = \dim(\sum_{w=1}^{K} \mathcal{S}_w) \leq \sum_{w=1}^{K} d_w$.

The rank of $\boldsymbol{\Gamma}_w \in \mathbb{R}^{p \times q}$, $d_w$, could be smaller than $q$, the number of functions in $\mathbf{f}$. Our model-based SDR approach for handling heterogeneity in data is now rigorously connected to the central subspace notion in general (2). Based on the maximum likelihood estimation (MLE) for PFC model parameters (Cook & Forzani, 2008), we derive the MLE for $\mathcal{S}_w$ and, more importantly, a penalized EM algorithm for high-dimensional data.

## 3. Group-Penalized EM Algorithm

Let $\{(\mathbf{X}_i, Y_i)\}_{i=1}^{n}$ be $n$ independent data points from mixPFC (3), $\boldsymbol{\theta} = (\boldsymbol{\Delta}, \pi_w, \boldsymbol{\mu}_w, \mathcal{S}_w, w = 1, \dots, K)$ be the set of unknown model parameters. In low dimensions, all the parameters can be estimated by the EM algorithm.

The EM algorithm aims to maximize the log-likelihood of $\mathbf{X} \mid Y$ over $\boldsymbol{\theta}$, by iteratively alternating between an Expectation-step (E-step) and a Maximization-step (M-step). The conditional log-likelihood of $\mathbf{X} \mid Y$ is

$$l(\boldsymbol{\theta}) = \sum_{i=1}^{n} \log \left( \sum_{w=1}^{K} \pi_w N(\mathbf{X}_i \mid \boldsymbol{\mu}_w + \boldsymbol{\Gamma}_w \mathbf{f}_i, \boldsymbol{\Delta}) \right),$$

where $\mathbf{f}_i = \mathbf{f}(Y_i)$ and $N(\cdot \mid \boldsymbol{\mu}, \boldsymbol{\Delta})$ is the probability density function of a multivariate normal distribution with mean $\boldsymbol{\mu}$ and covariance $\boldsymbol{\Delta}$. In the E-step, we compute the expectation of the log-likelihood function of $\boldsymbol{\theta}$ with respect to the conditional distribution of $W$ given $\{(\mathbf{X}_i, Y_i)\}_{i=1}^{n}$:

$$\begin{aligned} Q(\boldsymbol{\theta}|\widehat{\boldsymbol{\theta}}^{(t)}) = \sum_{i=1}^{n} \sum_{w=1}^{K} \gamma_{iw}(\widehat{\boldsymbol{\theta}}^{(t)})[\log(\pi_w) \\ + \log(N(\mathbf{X}_i \mid \mu_w + \boldsymbol{\Gamma}_w \mathbf{f}_i, \boldsymbol{\Delta}))], \end{aligned} \quad (4)$$

where $\gamma_{iw}(\widehat{\boldsymbol{\theta}}^{(t)}) = \mathrm{Pr}(W_i = w \mid \widehat{\boldsymbol{\theta}}^{(t)}, \mathbf{X}_i, Y_i)$. Assuming the cluster means $\boldsymbol{\mu}_w$ are equal, the estimated probability $\gamma_{iw}(\widehat{\boldsymbol{\theta}}^{(t)})$ is given by

$$\gamma_{iw}(\widehat{\boldsymbol{\theta}}^{(t)})^{-1} = \sum_{j \neq w} \frac{\widehat{\pi}_j^{(t)}}{\widehat{\pi}_w^{(t)}} \exp\{(\mathbf{X}_i - 1/2[(\widehat{\boldsymbol{\Gamma}}_j^{(t)} + \widehat{\boldsymbol{\Gamma}}_w^{(t)})\mathbf{f}_i])^T$$
$$(\widehat{\boldsymbol{\Delta}}^{(t)})^{-1}(\widehat{\boldsymbol{\Gamma}}_j^{(t)} - \widehat{\boldsymbol{\Gamma}}_w^{(t)})\mathbf{f}_i\} + 1$$

Then, in the M-step, we update $\widehat{\boldsymbol{\theta}}^{(t+1)} = \mathrm{argmax}_{\boldsymbol{\theta}} Q(\boldsymbol{\theta}|\widehat{\boldsymbol{\theta}}^{(t)})$ by maximizing (4).

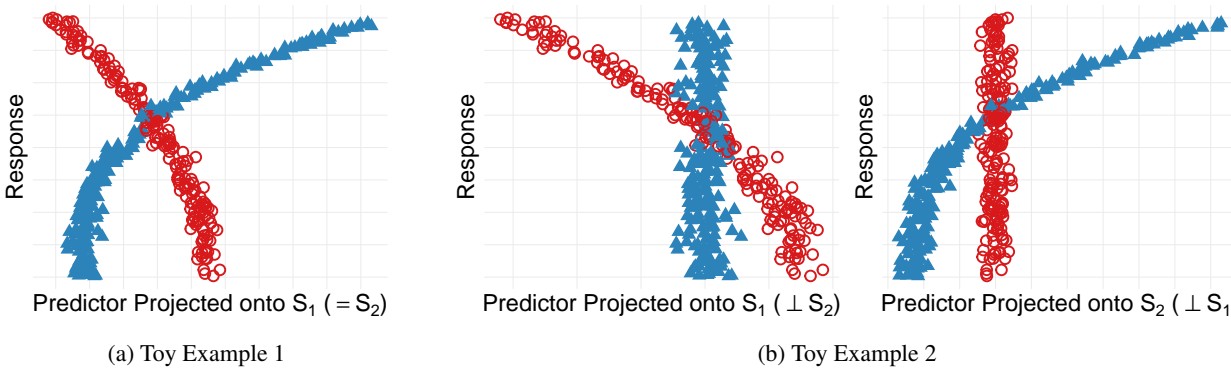

(a) Toy Example 1          (b) Toy Example 2

*Figure 2.* Two toy simulation examples, with $p = 6$ and two mixtures of size $n_1 = n_2 = 200$. The proposed mixPFC method works well in both extremes of setups: (a) Subspaces coincide, i.e, $\mathcal{S}_1 = \mathcal{S}_2$, but mixtures are well-separated by response variability. In this case, subspace clustering completely fails. (b) Subspaces orthogonal to each other, i.e, $\mathcal{S}_1 \perp \mathcal{S}_2$. For both examples, we plot the response versus the estimated linear reductions of the predictors based on the mixPFC method.

The above standard EM algorithm is infeasible for high-dimensional problems. The inverse of $p \times p$ covariance matrix $\mathbf{\Delta}$ serves as the cornerstone of the EM algorithm. In high dimensions, it is impractical to use $\mathbf{\Delta}^{-1}$ repeatedly in the EM updates. However, it can be seen that the probabilities $\gamma_{iw}(\widehat{\boldsymbol{\theta}}^{(t)})$ are evaluated on the linear function of $\mathbf{X}_i$: $\mathbf{X}_i^T(\widehat{\mathbf{\Delta}}^{(t)})^{-1}(\widehat{\mathbf{\Gamma}}_j^{(t)} - \widehat{\mathbf{\Gamma}}_w^{(t)})$. By Proposition 2.1, $\mathbf{\Delta}^{-1}(\mathbf{\Gamma}_j - \mathbf{\Gamma}_w)$ is contained in the central subspace. Hence, there is no loss of information to first project $\mathbf{X}_i$ onto the central subspace $\mathcal{S}$ and then calculate the probability based on reduced predictors, avoiding the $p \times p$ matrix inversion $\mathbf{\Delta}^{-1}$. Specifically, given a basis matrix $\boldsymbol{\beta} \in \mathbb{R}^{p \times d}$ of $\mathcal{S}$, we focus on the reduced predictors $\boldsymbol{\beta}^T \mathbf{X} = \boldsymbol{\beta}^T \mathbf{\Gamma}_w \mathbf{f}(Y) + \boldsymbol{\beta}^T \boldsymbol{\epsilon} \in \mathbb{R}^d$. This is a mixture linear regression problem $\mathbf{Z} = \mathbf{A}_w \mathbf{f}(Y) + \boldsymbol{\xi}$, where $\mathbf{Z} = \boldsymbol{\beta}^T \mathbf{X}$, $\mathbf{A}_w = \boldsymbol{\beta}^T \mathbf{\Gamma}_w$, and $\boldsymbol{\xi} \sim N(0, \mathbf{\Delta}^*)$ with $\mathbf{\Delta}^* = \boldsymbol{\beta}^T \mathbf{\Delta} \boldsymbol{\beta} \in \mathbb{R}^{d \times d}$. Then the updating equation for the probabilities can be simplified to

$$\gamma_{iw}(\widehat{\boldsymbol{\theta}}^{(t)})^{-1} = \sum_{j \neq w} \frac{\widehat{\pi}_j^{(t)}}{\widehat{\pi}_w^{(t)}} \exp\left\{ (\mathbf{Z}_i^{(t)} - \frac{1}{2}[(\widehat{\mathbf{A}}_j^{(t)} \right.$$
$$\left. + \widehat{\mathbf{A}}_w^{(t)}) \mathbf{f}_i)])^T ((\widehat{\mathbf{\Delta}}^*)^{(t)})^{-1} \left( \widehat{\mathbf{A}}_j^{(t)} - \widehat{\mathbf{A}}_w^{(t)} \right) \mathbf{f}_i \right\} + 1. \tag{5}$$

Given $\boldsymbol{\beta}$, the closed-form updates for $\mathbf{\Delta}^*$, $\mathbf{A}_w$, and $\pi_w$ are straightforward to derive. The most challenging part remaining is how to obtain an accurate estimator of the central subspace.

For high-dimensional predictors, we consider the following groupwise penalized estimation of the basis matrix of each subspace $\boldsymbol{\beta}_w$. First of all, we recognize that $\text{span}(\boldsymbol{\beta}_w) \equiv \text{span}(\mathbf{\Delta}^{-1} \mathbf{\Gamma}_w) = \text{span}(\mathbf{\Sigma}_w^{-1} \mathbf{U}_w)$, where $\mathbf{\Sigma}_w = \text{cov}(\mathbf{X} \mid W = w) \in \mathbb{R}^{p \times p}$ and $\mathbf{U}_w = \text{cov}(\mathbf{X}, \mathbf{f}(Y) \mid W = w) \in \mathbb{R}^{p \times q}$ are the covariance matrices. The iterative sample es-

timates in EM updates are computed as

$$\widehat{\mathbf{\Sigma}}_w^{(t)} = \frac{1}{n} \sum_{i=1}^n \gamma_{iw}(\widehat{\boldsymbol{\theta}}^{(t)})(\mathbf{X}_i - \widehat{\boldsymbol{\mu}}_w^{(t)})(\mathbf{X}_i - \widehat{\boldsymbol{\mu}}_w^{(t)})^T,$$

$$\widehat{\mathbf{U}}_w^{(t)} = \frac{1}{n} \sum_{i=1}^n \gamma_{iw}(\widehat{\boldsymbol{\theta}}^{(t)})(\mathbf{X}_i - \widehat{\boldsymbol{\mu}}_w^{(t)})(\mathbf{f}_i - \overline{\mathbf{f}})^T.$$

where $\widehat{\boldsymbol{\mu}}_w^{(t)} = (\sum_i \gamma_{iw}(\widehat{\boldsymbol{\theta}}^{(t)}))^{-1} \sum_i \gamma_{iw}(\widehat{\boldsymbol{\theta}}^{(t)}) \mathbf{X}_i$ and $\overline{\mathbf{f}} = 1/n \sum_i \mathbf{f}_i$. Then we solve the convex optimization problem,

$$\widehat{\mathbf{B}}_w^{(t)} = \underset{\mathbf{B}_w \in \mathbb{R}^{p \times q}}{\operatorname{argmin}} \frac{1}{2} \operatorname{tr}(\mathbf{B}_w^T \widehat{\mathbf{\Sigma}}_w^{(t)} \mathbf{B}_w)$$
$$- \operatorname{tr}\{(\widehat{\mathbf{U}}_w^{(t)})^T \mathbf{B}_w\} + \lambda \|\mathbf{B}_w\|_{2,1}, \tag{6}$$

where $\lambda > 0$ is tuning parameter and the $L_{2,1}$ penalty $\|\mathbf{B}_w\|_{2,1} = \sum_{i=1}^p (\sum_{j=1}^q (\mathbf{B}_w)_{ij}^2)^{1/2}$ is coordinate-independent (Chen et al., 2010). The problem in (6) is convex. We develop a groupwise coordinate descent algorithm to solve it efficiently. Note that $\mathbf{B}_w \in \mathbb{R}^{p \times q}$'s are naturally rank deficient, with $q \geq d_w$. Therefore, at the convergence of the penalized EM algorithm 1, we use the span of the top-$d_w$ left singular vectors of $\widehat{\mathbf{B}}_w$ to be the subspace estimates $\widehat{\mathcal{S}}_w$.

As shown in the original PFC paper (Cook & Forzani, 2008), subspace estimation remains consistent under misspecification of $\mathbf{f}(Y)$, provided $\mathbf{f}(Y)$ is sufficiently correlated with the true function. Our mixPFC model inherits this property, ensuring validity across a broad class of functions. In practice, polynomials or splines are standard choices. The initialization method, selection of $K$, along with mixPFC-ISO—an alternative algorithm tailored for isotropic covariance matrices—is detailed in Section B of the appendix. The code is available on GitHub at `https://github.com/leiyan-ly/mixPFC`.

---

**Algorithm 1** Penalized EM algorithm for mixture PFC

---

**Input:** Data $\{(\mathbf{X}_i, Y_i)\}_{i=1}^n$, fitting function $\mathbf{f}(\cdot)$
Initialize $\widehat{\gamma}_{iw}(\boldsymbol{\theta}^0)$ and center $\mathbf{f}_i$
**repeat**

E-Step: $\gamma_{iw}(\widehat{\boldsymbol{\theta}}^{(t)}) = \dfrac{\widehat{\pi}_w^{(t)}}{\widehat{\pi}_w^{(t)} + \sum_{j \neq w} \widehat{\pi}_j^{(t)} \exp\{(\mathbf{Z}_i^{(t)} - \frac{1}{2}[(\widehat{\mathbf{A}}_j^{(t)} + \widehat{\mathbf{A}}_w^{(t)})\mathbf{f}_i])^T ((\widehat{\boldsymbol{\Delta}}^*)^{(t)})^{-1}(\widehat{\mathbf{A}}_j^{(t)} - \widehat{\mathbf{A}}_w^{(t)})\mathbf{f}_i\}}$

M-Step: $\widehat{\mathbf{B}}_w^{(t+1)} = \mathrm{argmin}_{\mathbf{B}_w} \frac{1}{2}\mathrm{tr}(\mathbf{B}_w^T \widehat{\boldsymbol{\Sigma}}_w^{(t)} \mathbf{B}_w) - \mathrm{tr}\{(\widehat{\mathbf{U}}_w^{(t)})^T \mathbf{B}_w\} + \lambda \|\mathbf{B}_w\|_{2,1}$

$\widehat{\pi}_w^{(t+1)} = 1/n \sum_{i=1}^n \gamma_{iw}(\widehat{\boldsymbol{\theta}}^{(t)})$, $\widehat{\boldsymbol{\mu}}_w^{(t+1)} = 1/\sum_{i=1}^n [\gamma_{iw}(\widehat{\boldsymbol{\theta}}^{(t)})] \sum_{i=1}^n \gamma_{iw}(\widehat{\boldsymbol{\theta}}^{(t)})\mathbf{X}_i$

**until** converge
**Output:** $\widehat{\pi}_w, \widehat{\mathcal{S}}_w$

---

## 4. Theory

### 4.1. Preliminary

We begin this section with some notations. For numbers $a$ and $b$, $a \vee b$ and $a \wedge b$ means $\max\{a, b\}$ and $\min\{a, b\}$. For an integer $n$, $[n]$ denotes the set $\{1, \ldots, n\}$. For a vector $\mathbf{x} = (x_1, \ldots, x_p)^T$, $\|\mathbf{x}\|_1 = \sum_{i=1}^p |x_i|$, and $\|\mathbf{x}\|_2 = \sqrt{\sum_{i=1}^p x_i^2}$. For a matrix $\mathbf{A} = (a_{ij})$, $\sigma_{\min}(\mathbf{A})$ and $\sigma_{\max}(\mathbf{A})$ represent the smallest and largest singular values of $\mathbf{A}$, $\mathcal{S}_{\mathbf{A}}$ denote the column space of $\mathbf{A}$. The Frobenius norm, and spectral norm of $\mathbf{A}$ are defined as $\|\mathbf{A}\|_F = \sqrt{\mathrm{tr}(\mathbf{A}^T\mathbf{A})}$, $\|\mathbf{A}\|_2 = \sigma_{\max}(\mathbf{A})$. The Frobenius inner product between two matrices $\mathbf{A}$ and $\mathbf{B}$ is $\langle \mathbf{A}, \mathbf{B} \rangle_F = \mathrm{tr}(\mathbf{A}^T\mathbf{B})$. For matrices $\mathbf{A}$ and $\mathbf{B}$ with same column rank $d$, the distance between subspaces is $\mathcal{D}(\mathcal{S}_{\mathbf{A}}, \mathcal{S}_{\mathbf{B}}) = \|\mathbf{P}_{\mathbf{A}} - \mathbf{P}_{\mathbf{B}}\|_F / \sqrt{2d}$. For a set $\mathcal{A} \subseteq \{1, \ldots, p\}$, $\mathcal{A}^c$ and $|\mathcal{A}|$ denote its complement and cardinality. For two sequences of positive numbers $\{a_n\}$ and $\{b_n\}$, $a_n \lesssim b_n$ or $a_n = O(b_n)$ means $a_n/b_n \leq C < \infty$, and $a_n = o(b_n)$ means that $a_n/b_n \to 0$ as $n \to \infty$. Let $\mathbb{S}^{pq-1}$ be the unit sphere. For a positive integer $s < p/(3q)$, let $\mathcal{L}(s) = \{\mathbf{u} \in \mathbb{R}^{pq} : \|\mathbf{u}_{\widetilde{\mathcal{S}}_1^c}\|_1 \leq (\sqrt{sq} + 2q\sqrt{3s})\|\mathbf{u}_{\widetilde{\mathcal{S}}_1}\|_2 + \sqrt{sq}\|\mathbf{u}\|_2$, for some $\widetilde{\mathcal{S}}_1 \subset [pq]$ with $|\widetilde{\mathcal{S}}_1| = 3sq$ and $\mathcal{L}_p(s) = \mathcal{L}(s)_{1:p} = \{\mathbf{u}_{1:p} : \mathbf{u} \in \mathcal{L}(s)\}$. For a matrix $\mathbf{A} \in \mathbb{R}^{p \times q}$, define $\|\mathbf{A}\|_{F,s} = \sup_{\mathbf{u} \in \mathbb{R}^{p \times q}, \mathrm{vec}(\mathbf{u}) \in \mathcal{L}(s) \cap \mathbb{S}^{pq-1}} \langle \mathbf{A}, \mathbf{u} \rangle_F$.

We conduct the theoretical analysis under the assumption that $Y_i$ is fixed, $\mathbf{f}(\cdot)$ is known, and $\boldsymbol{\mu}_w = 0$. We focus on the case where $K = 2$, a common assumption in high-dimensional EM algorithm analysis (Cai et al., 2019; Wang et al., 2024). We further assume that $\boldsymbol{\Delta} = \sigma^2 \mathbf{I}_p$ with $\sigma$ known. Treating the covariance matrix as a known parameter is also standard in theoretical studies of simpler models such as mixture linear regression (Klusowski et al., 2019; Wang et al., 2024) and the Gaussian mixture model (Xu et al., 2016; Cai et al., 2019). Without loss of generality, we set $\sigma^2 = 1$. Under these assumptions, we re-define the parameter as $\boldsymbol{\theta} = (\pi_1, \boldsymbol{\Gamma}_1, \boldsymbol{\Gamma}_2)$, since $\pi_2 = 1 - \pi_1$ and $\mathcal{S}_w = \mathrm{span}(\boldsymbol{\Gamma}_w), w = 1, 2$. With this setup, we analyze

the theoretical properties of a simplified version of Algorithm 1, which is detailed in Algorithm 4 in the appendix. Let $\boldsymbol{\theta}^*$ denote the true value of $\boldsymbol{\theta}$, and $\widehat{\boldsymbol{\theta}}^{(t)}$ represent the estimate of $\boldsymbol{\theta}$ at the $t$-th iteration. The true parameter space is defined as

$$\boldsymbol{\Theta}^* = \{\boldsymbol{\theta}^* : \pi_1^* \in (c_\pi, 1 - c_\pi), \|\mathrm{vec}(\boldsymbol{\Gamma}_w^*)\|_0 \leq sq,$$
$$\|\mathbf{B}_w^*\|_F \leq M_a, \|\boldsymbol{\Gamma}_w^*\|_F \leq M_b, w = 1, 2\},$$

where each condition has a natural interpretation. The condition $\pi_1^* \in (c_\pi, 1 - c_\pi)$ ensures each latent cluster has a sufficiently large sample size. The sparsity condition $\|\mathrm{vec}(\boldsymbol{\Gamma}_w^*)\|_0 \leq sq$ reflects group sparsity structure, and $\|\boldsymbol{\Gamma}_w^*\|_F \leq M_b$ are used in the literature on mixture linear regression (Yi & Caramanis, 2015; Wang et al., 2024). The parameter $\mathbf{B}_w^*$ is the true solution to the optimization problem (6), and is defined later in the text.

Since $\sigma^2 = 1$ and $K = 2$, the conditional probability $\Pr(W_i = 1 | \boldsymbol{\theta}, \mathbf{X}_i, Y_i)$ is simplified as

$$\gamma_{i1}(\boldsymbol{\theta})^{-1} = (1/\pi_1 - 1)\exp\{[\mathbf{X}_i - 1/2(\boldsymbol{\Gamma}_2 + \boldsymbol{\Gamma}_1)\mathbf{f}_i]^T$$
$$(\boldsymbol{\Gamma}_2 - \boldsymbol{\Gamma}_1)\mathbf{f}_i\} + 1,$$

and let $\gamma_{i2}(\boldsymbol{\theta}) = \Pr(W_i = 2 | \boldsymbol{\theta}, \mathbf{X}_i, Y_i) = 1 - \gamma_{i1}(\boldsymbol{\theta})$. The following quantities are used repeatedly in the theoretical analysis:

$$\widehat{\pi}_w(\boldsymbol{\theta}) = \frac{1}{n}\sum_{i=1}^n \gamma_{iw}(\boldsymbol{\theta}), \quad \pi_w(\boldsymbol{\theta}) = \mathrm{E}[\widehat{\pi}_w(\boldsymbol{\theta})],$$

$$\widehat{\mathbf{U}}_w(\boldsymbol{\theta}) = \frac{1}{n}\sum_{i=1}^n \gamma_{iw}(\boldsymbol{\theta})\mathbf{X}_i\mathbf{f}_i^T, \quad \mathbf{U}_w(\boldsymbol{\theta}) = \mathrm{E}[\widehat{\mathbf{U}}_w(\boldsymbol{\theta})],$$

$$\widehat{\boldsymbol{\Sigma}}_w(\boldsymbol{\theta}) = \frac{1}{n}\sum_{i=1}^n \gamma_{iw}(\boldsymbol{\theta})\mathbf{X}_i\mathbf{X}_i^T, \quad \boldsymbol{\Sigma}_w(\boldsymbol{\theta}) = \mathrm{E}[\widehat{\boldsymbol{\Sigma}}_w(\boldsymbol{\theta})],$$

where the expectation is with respect to $\mathbf{X}_i, i = 1, 2 \ldots, n$. We define $\mathbf{B}_w^* = (\boldsymbol{\Sigma}_w^*)^{-1}\mathbf{U}_w^*$, where $\boldsymbol{\Sigma}_w^* = \boldsymbol{\Sigma}_w(\boldsymbol{\theta}^*)$ and $\mathbf{U}_w^* = \mathbf{U}_w(\boldsymbol{\theta}^*)$. Let $\widehat{\boldsymbol{\beta}}_w^{(t)}$ and $\boldsymbol{\beta}_w^*$ represent top-$d_w$ left singular vectors of $\widehat{\mathbf{B}}_w^{(t)}$ and $\mathbf{B}_w^*$. Then we have $\mathcal{S}_w = \mathcal{S}_{\boldsymbol{\beta}_w^*}$. Let $M_n(\boldsymbol{\theta}) = \{\widehat{\pi}_w(\boldsymbol{\theta}), \widehat{\mathbf{U}}_w(\boldsymbol{\theta}), \widehat{\boldsymbol{\Sigma}}_w(\boldsymbol{\theta}), w = $

1, 2} represent the sample-based estimates and $M(\boldsymbol{\theta}) = \{\pi_w(\boldsymbol{\theta}), \mathbf{U}_w(\boldsymbol{\theta}), \boldsymbol{\Sigma}_w(\boldsymbol{\theta}), w = 1, 2\}$ be the population counterpart. To quantify the differences between two sets of parameters, we introduce a distance $d_F(M(\boldsymbol{\theta}_1), M(\boldsymbol{\theta}_2))$, defined as

$$\max_{w=1,2}\{|\pi_w(\boldsymbol{\theta}_1) - \pi_w(\boldsymbol{\theta}_2)| \vee \|\mathbf{U}_w(\boldsymbol{\theta}_1) - \mathbf{U}_w(\boldsymbol{\theta}_2)\|_F$$
$$\vee \|(\boldsymbol{\Sigma}_w(\boldsymbol{\theta}_1) - \boldsymbol{\Sigma}_w(\boldsymbol{\theta}_2))\mathbf{B}_w^*\|_F\}.$$

Let $\Omega = \sqrt{\mathrm{tr}[(\boldsymbol{\Gamma}_2^* - \boldsymbol{\Gamma}_1^*)\widehat{\boldsymbol{\Sigma}}_{\mathbf{f}}(\boldsymbol{\Gamma}_2^* - \boldsymbol{\Gamma}_1^*)^T]}$ denote the signal strength of the mixture PFC model, where $\widehat{\boldsymbol{\Sigma}}_{\mathbf{f}} = 1/n \sum_{i=1}^n \mathbf{f}_i \mathbf{f}_i^T$. We define the contraction basin $\mathcal{B}_{\mathrm{con}}(\boldsymbol{\theta}^*)$ as the set:

$$\mathcal{B}_{\mathrm{con}}(\boldsymbol{\theta}^*) = \{\boldsymbol{\theta} : \pi_w \in (c_0, 1 - c_0), \|\boldsymbol{\Gamma}_w - \boldsymbol{\Gamma}_w^*\|_F \le C_b\Omega,$$
$$(1 - C_d)\Omega^2 \le |\mathrm{tr}(\boldsymbol{\delta}_w(\boldsymbol{\Gamma})\widehat{\boldsymbol{\Sigma}}_{\mathbf{f}}(\boldsymbol{\Gamma}_2 - \boldsymbol{\Gamma}_1)^T)|$$
$$\le (1 + C_d)\Omega^2,$$
$$\mathrm{vec}(\boldsymbol{\Gamma}_w - \boldsymbol{\Gamma}_w^*) \in \mathcal{L}(s), w = 1, 2\},$$

where $c_0 \le c_\pi$ and $\boldsymbol{\delta}_w(\boldsymbol{\Gamma}) = \boldsymbol{\Gamma}_w^* - (\boldsymbol{\Gamma}_2 + \boldsymbol{\Gamma}_1)/2$. The contraction basin requires that $\boldsymbol{\theta}$ is not far away from the true parameter $\boldsymbol{\theta}^*$. Under the conditions shown later, an initialization $\widehat{\boldsymbol{\theta}}^{(0)}$ within the contraction basin guarantees that all subsequent estimators $\widehat{\boldsymbol{\theta}}^{(t)}$ remain in the contraction basin throughout the iterative process.

## 4.2. Main Results

We need some technical conditions before stating the theoretical results.

(C1) The singular values of $\widehat{\boldsymbol{\Sigma}}_{\mathbf{f}} = 1/n \sum_{i=1}^n \mathbf{f}_i \mathbf{f}_i^T$ satisfy that $M_1 \le \sigma_{\min}(\widehat{\boldsymbol{\Sigma}}_{\mathbf{f}}) \le \sigma_{\max}(\widehat{\boldsymbol{\Sigma}}_{\mathbf{f}}) \le M_2$, and $M_3 \le \min_{1 \le i \le n} \|\mathbf{f}_i\|_2 \le \max_{1 \le i \le n} \|\mathbf{f}_i\|_2 \le M_4$.

(C2) The initialization $\boldsymbol{\theta}^{(0)}$ satisfies that $d_F(\boldsymbol{\theta}^{(0)}, \boldsymbol{\theta}^*) \vee \|\mathbf{B}_1^{(0)} - \mathbf{B}_1^*\|_F \vee \|\mathbf{B}_2^{(0)} - \mathbf{B}_2^*\|_F < r\Omega$, and $\mathrm{vec}(\boldsymbol{\Gamma}_w^{(0)} - \boldsymbol{\Gamma}_w^*) \in \mathcal{L}(s)$, with $r < |c_0 - c_\pi|/\Omega \wedge C_b \wedge \frac{1}{a}(\sqrt{C_d - 1/(4\sqrt{M_1})} + \frac{b^2}{4a^2} - \frac{b}{2a})$, $a^2 = 2M_2^{3/2}/\sqrt{M_1}$, $b = 2\sqrt{M_2} + [M_2 + \sqrt{M_2}/2]/\sqrt{M_1}$.

(C3) There exists a sufficiently large constant $M_5 > 0$, which does not depend on $n, p, s$, such that $\sigma_{d_w}(\mathbf{B}_w^*) \ge M_5 \ge \sqrt{sq^3(\log n)^2 \log p/n}$.

(C4) $\Omega \ge C_1(c_0, C_b, M_b, M_i; i = 1, \ldots, 4)$ for a constant that is only depends on $c_0$, $M_b$, $C_b$, and $M_i, i = 1, \ldots, 4$, and $C_b < C_2(M_2)$ for a constant only depends on $M_2$.

(C5) $n > C_3 sq^3 \log(p)$ for a sufficiently large constant $C_3$.

Condition (C1) is mild since $\mathbf{f}_i$ is a $q$-dimensional vector, where $q$ is a small fixed number that does not grow with $n$ and $p$. Condition (C2) ensures the initialization lies within the contraction basin, which guarantees the estimates produced at each step of the EM algorithm stay in the contraction basin. It is a common condition in mixture models (Cai et al., 2019; Wang et al., 2024). Condition (C3) requires that the nonzero singular values of $\mathbf{B}_w^*$ are sufficiently separated from zero. This is a standard assumption in the theoretical analysis of high dimensional SDR problems (Zeng et al., 2024). Condition (C4) has two requirements. The first one is that the signal strength is larger than a constant that does not depend on $n$ and $p$ such that the two mixtures are distinguishable. This assumption is widely used in mixture linear model (Zhang et al., 2020; Wang et al., 2024). The second is that, for the parameters $\boldsymbol{\Gamma}_w$ within the contraction basin, the distance $\|\boldsymbol{\Gamma}_w - \boldsymbol{\Gamma}_w^*\|_F$ is bounded by the signal strength multiplied by a universal constant independent of $n$ and $p$. Condition (C5) is a common assumption in high dimensions on the relationship among $n, p, s$ to guarantee consistent estimation (Meinshausen & Yu, 2009; Cai et al., 2019). Specifically, it implies that the restrictive eigenvalue condition $\inf_{\mathbf{u} \in \mathcal{L}_p(s) \cap \mathbb{S}^{p-1}} |\mathbf{u}^T \frac{1}{n} \sum_{i=1}^n \mathbf{X}_i \mathbf{X}_i^T \mathbf{u}| > \tau_1$ holds with high probability for a positive constant $\tau_1$.

Next, we state the main result for the subspace estimation error of mixPFC in Theorem 4.1, with its proof provided in Section C in the appendix.

**Theorem 4.1.** *Under conditions (C1)-(C5), there exists a constant $0 < \kappa < 1/2$, such that $\widehat{\mathbf{B}}_w^{(t)}$ satisfies, with probability at least $1 - o(1)$,*

$$\|\widehat{\mathbf{B}}_w^{(t)} - \mathbf{B}_w^*\|_F \lesssim \underbrace{\kappa^t (d_F(\widehat{\boldsymbol{\theta}}^{(0)}, \boldsymbol{\theta}^*) \vee \|\widehat{\mathbf{B}}_1^{(0)} - \mathbf{B}_1^*\|_F \vee \|\widehat{\mathbf{B}}_2^{(0)} - \mathbf{B}_2^*\|_F)}_{computational\ error} + \underbrace{\sqrt{\frac{sq^3(\log n)^2 \log p}{n}}}_{statistical\ error}.$$

*Consequently, for $t \ge (-\log \kappa)^{-1} \log\{n(d_F(\widehat{\boldsymbol{\theta}}^{(0)}, \boldsymbol{\theta}^*) \vee \|\widehat{\mathbf{B}}_1^{(0)} - \mathbf{B}_1^*\|_F \vee \|\widehat{\mathbf{B}}_2^{(0)} - \mathbf{B}_2^*\|_F)\}$,*

$$\|\widehat{\mathbf{B}}_w^{(t)} - \mathbf{B}_w^*\|_F, \mathcal{D}(\mathcal{S}_{\widehat{\boldsymbol{\beta}}_w^{(t)}}, \mathcal{S}_{\boldsymbol{\beta}_w^*}) \lesssim \sqrt{\frac{sq^3(\log n)^2 \log p}{n}}.$$

Theorem 4.1 is the first theoretical result in high-dimensional heterogeneous SDR. Compared to the high-dimensional PFC result in Zeng et al. (2024), our convergence rate is slower by a factor of $\log n$, reflecting the added complexity of unknown cluster labels. Importantly, Theorem 4.1 holds under unequal proportions and arbitrary subspace angles, making it highly non-trivial. Even in low-dimensional settings, existing EM theory often requires additional assumptions such as equal proportions (Gaussian mixtures (Xu et al., 2016)), or symmetric coefficients (mixtures of linear regression (Zhu et al., 2017)). Additionally, our analysis does not rely on sample splitting, a common technique in the literature (Yi et al., 2014; Yi & Caramanis, 2015; Zhang et al., 2020) that divides the data into many batches and uses a new batch of samples in each iteration to make random samples and current parameter estimates independent. Sample splitting, while theoretically convenient, is suboptimal in practice as it decreases estimation efficiency and is rarely used in real-world applications. Recent work by Wang et al. (2024) derived a rate of $\sqrt{s(\log n)^2 \log p/n}$ without data splitting for mixture of linear regression. However, the mixture of PFC is inherently more complex. Our rate is slower by a factor of $q^{3/2}$ due to the dependence on the $q$-dimensional vector $\mathbf{f}(Y)$. Similarly, compared to Gaussian mixture model (Cai et al., 2019), the convergence rate is slower by a factor of $q^{3/2} \log n$ due to the involvement of $\gamma_{iw}$ in $\boldsymbol{\Sigma}_w(\boldsymbol{\theta})$ and function $\mathbf{f}(Y)$.

Starting with an initial value within the contraction basin, Theorem 4.1 shows that the proposed algorithm converges to the true parameters at a rate containing both computational and statistical errors. The computational error, expressed as $\kappa^t(d_F(\widehat{\boldsymbol{\theta}}^{(0)}, \boldsymbol{\theta}^*) \vee \|\widehat{\mathbf{B}}_1^{(0)} - \mathbf{B}_1^*\|_F \vee \|\widehat{\mathbf{B}}_2^{(0)} - \mathbf{B}_2^*\|_F)$, diminishes exponentially as $t \to \infty$ since $0 < \kappa < 1/2$. The statistical error, $\sqrt{sq^3(\log n)^2 \log p/n}$, represents the irreducible estimation error and persists regardless of the number of EM iterations. For sufficiently large $t \geq (-\log \kappa)^{-1} \log\{n(d_F(\widehat{\boldsymbol{\theta}}^{(0)}, \boldsymbol{\theta}^*) \vee \|\widehat{\mathbf{B}}_1^{(0)} - \mathbf{B}_1^*\|_F \vee \|\widehat{\mathbf{B}}_2^{(0)} - \mathbf{B}_2^*\|_F)\}$, the computation error becomes negligible relative to the statistical error. Beyond this step, additional iterations do not improve the estimators. Notably, since this threshold grows only logarithmically with $n$, the Algorithm 1 achieves accurate estimation in practice within a limited number of iterations.

The computational cost per EM iteration is $O(nKpq + nK^3q^3 + KTpnq + nKp^2)$, where $T$ denotes the number of iterations to solve the penalized optimization problem (6). Given that $q$ is a small number that does not grow with $n, p, K$, the overall complexity of Algorithm 1 is $O(\log(n)(nK^3 + KTnp + Knp^2))$ with the dominated term $O(\log(n)Knp^2)$ from covariance estimation. This remains tractable even for large $K$ or $p$.

## 5. Numerical Results

### 5.1. Simulations

We compare the mixPFC and mixPFC-ISO against existing methods in clustering accuracy, subspace estimation, and variable selection. Since no existing method simultaneously classifies the data and estimates subspaces, we evaluate our methods against subspace clustering approaches for clustering error rates and high-dimensional SDR methods for subspace estimation and variable selection. The subspace clustering methods considered include LSA (Yan & Pollefeys, 2006), SSC (Elhamifar & Vidal, 2013), LRSC (Favaro et al., 2011), GPCA (Vidal et al., 2005), and RANSAC (Tron & Vidal, 2007). GPCA is applied only to the important variables due to computational constraints. Additionally, $K$-means and hierarchical clustering, are included and applied to both $\mathbf{X}$ and $\mathbf{X} \circ Y$, where the $i$-th row of $\mathbf{X} \circ Y$ is defined as $Y_i \times \mathbf{X}_i$. For variable selection and subspace estimation accuracy, we include LassoSIR (Lin et al., 2019), SEAS-SIR, and SEAS-PFC (Zeng et al., 2024). Clustering results are presented in this section, with subspace estimation and variable selection results provided in Section A of the appendix.

We consider four settings for central subspaces, denoted as models M1-M4, to examine different configurations of mixtures. Models M1-M3 have $K = 2$ mixtures with different degrees of overlap between two subspaces. Specifically, the subspaces are identical in M1, orthogonal in M2, and oblique in M3. Model M4 randomly generates multiple subspaces ($K > 2$), which tend to be nearly orthogonal to each other sine $s = 10$. The dimension $d_w$ of each subspace is 1 for M1 and M2, and 2 for M3 and M4. The active set is defined as $\mathcal{A}_w = \{1, \ldots, 6\}$ for M1-M3, and $\mathcal{A}_w = \{1, \ldots, 10\}$ for M4. Across all models, we set $\boldsymbol{\mu}_w = 0$, $\mathbf{f}(Y) = (Y, |Y|)^T$, and $\pi_w = 1/K$. After the basis matrices $\boldsymbol{\beta}_w$ of central subspaces are generated according to M1-M4 (parameters provided in Section A of the appendix), we set $\boldsymbol{\Gamma}_w = \boldsymbol{\Delta}\boldsymbol{\beta}_w\boldsymbol{\eta}_w$, where $\boldsymbol{\eta}_w \in \mathbb{R}^{d_w \times q}$ links the central subspace and function $\mathbf{f}(Y)$. Imbalanced clusters and non-linear functions are also examined in Section A of the appendix.

The sample size is fixed at $n = 200K$ with $p = 1000$, and for each simulation setting, 100 independent datasets are generated. To explore the influence of different covariance structures, we examine four configurations: $0.1\mathbf{I}_p, \mathbf{I}_p, \mathrm{AR}(0.3), \mathrm{AR}(0.5)$, where $\mathrm{AR}(r)$ represents the auto-regressive covariance structure $(\boldsymbol{\Delta})_{ij} = r^{|i-j|}$ for $i, j = 1, \ldots, p$.

Table 1 summarizes clustering error rates. As expected, mixPFC achieves substantially lower error rates than all subspace clustering methods across most model settings. When $K = 2$, mixPFC has error rates of around 10%

across all settings, dropping below 5% in certain cases. Notably, mixPFC-ISO demonstrates superior performance with even lower error rates for model M2. For M2 with $\Delta = 0.1\mathbf{I}_p$, subspace clustering methods exhibit comparable or slightly lower error rates. This is likely due to favorable conditions for subspace clustering methods, where the subspaces are orthogonal and the random errors are minimal. When $K > 2$, error rates remain impressively low, under 3% for $K = 3$, probably due to enhanced signal strength from setting $s = 10$. When $K = 5$, error rates rise to around 15%, likely due to the increased difficulty in generating high-quality initial values for larger $K$.

### 5.2. Real Data Analysis

The Australian Institute of Sport (AIS) dataset, available in the R package `dr`, contains lean body mass data for 102 male and 100 female athletes. The objective is to investigate the relationship between lean body mass and 8 predictors, including height, weight, and red cell count. Given that body composition varies between males and females (Bredella, 2017), the AIS data likely includes two distinct subpopulations. Figure 3 (a) shows summary plots for males and females when sex is observed, highlighting distinct fitted lines for each group. Figure 3 (b) demonstrates that mixPFC effectively identifies the two subpopulations, achieving an error rate of 0.074.

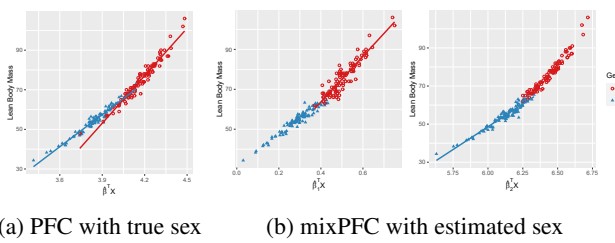

(a) PFC with true sex          (b) mixPFC with estimated sex

*Figure 3.* Summary plots for the AIS dataset. (a) Fitted lines for males and females when the sex variable is observed, illustrate distinct subpopulation trends. (b) Results from mixPFC, which accurately identifies the two subpopulations with an error rate of 0.074.

The Cancer Cell Line Encyclopedia (CCLE) dataset contains 8-point dose-response curves for 24 chemical compounds across over 400 cell lines, with 18,926 gene expression features for each cell line, accessible at https://sites.broadinstitute.org/ccle. Due to inconsistencies in cell lines across compounds, we focus on two popular cancer treatments: Nutlin-3 ($n = 480$) and AZD6244 ($n = 479$). Following (Wang et al., 2024; Li et al., 2019), we use the logarithm of the area under the dose-response curve as the response, representing drug sensitivity. The top $p = 500$ genes with the highest absolute correlations with the responses are selected for analysis. Given the inherent complexity of cancer, the CCLE data is expected to be heterogeneous.

The dataset is randomly partitioned into 80% training and 20% testing samples, with 100 repetitions. Table 2 reports the prediction mean squared errors (PMSE) and the number of selected variables $\widehat{s}$ for each method, with the number of clusters set to 3 and 5 for Nutlin-3 and AZD6244 when using mixPFC. For both compounds, mixPFC significantly reduces prediction error compared to homogeneous methods, suggesting heterogeneity in the data. Notably, mixPFC does not select more variables than Lasso and the three homogeneous SDR methods.

Figure 4 shows summary plots of the response against reduced predictors projected onto each subspace for Nutlin. Within each cluster, the response exhibits approximately linear relationships with the projected predictors. The lack of clear patterns when points are projected onto subspaces outside their cluster further highlights the heterogeneity of data. The plot for AZD6244 and additional real data analysis are provided in Section A in the appendix.

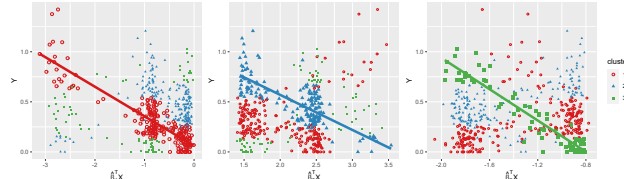

*Figure 4.* The scatter-plot of the response $Y$ versus $\widehat{\boldsymbol{\beta}}_w^T \mathbf{X}$ for the drug Nutlin. The solid line is fitted using samples in the given cluster.

## 6. Discussion

In this work, we proposed a mixture of PFC model, which combines subspace clustering with SDR methods to handle heterogeneous and high-dimensional data. An efficient group Lasso penalized EM algorithm has been developed to simultaneously perform clustering, subspace estimation, and variable selection. Theoretical analysis revealed an encouraging non-asymptotic convergence rate, offering insight into the empirical success of the algorithm.

A key aspect of our theoretical framework is its development for $K = 2$ cluster scenario. Generalizing theories for multi-cluster EM algorithms remains an important yet challenging direction. Recent advances focus on low-dimensional settings (Yan et al., 2017; Tian et al., 2024), and to the best of our knowledge, no general theory exists for high-dimensional multi-cluster EM. Addressing this open question likely requires fundamentally new tools to handle complex parameter spaces and interactions between $K$ subspaces.

Further discussion and potential extensions are provided in

*Table 1.* Averages and standard errors of clustering error rates with $n = 200K, p = 1000$.

| mixPFC-ISO | mixPFC | RANSAC | LSA | LRSC | SSC | GPCA | $K$-means $\mathbf{X}$ | $K$-means $\mathbf{X} \circ Y$ | hclust $\mathbf{X}$ | hclust $\mathbf{X} \circ Y$ |
|---|---|---|---|---|---|---|---|---|---|---|
| | | | | M1: $K = 2$, $\mathbf{\Delta} = 0.1\mathbf{I}_p, \mathbf{I}_p$, AR(0.3), AR(0.5) | | | | | | |
| 5.0 (0.8) | 3.4 (0.7) | 48 (0.1) | 44.6 (0.2) | 47.8 (0.2) | 47.9 (0.1) | 47.9 (0.1) | 44.2 (0.2) | 20.2 (0.3) | 45.8 (0.2) | 14 (0.2) |
| 12.1 (1.0) | 10.3 (1.1) | 48.1 (0.2) | 47.9 (0.2) | 48 (0.2) | 48 (0.1) | 48.1 (0.1) | 45.2 (0.2) | 29.1 (0.4) | 48.1 (0.2) | 38.2 (0.5) |
| | 6.0 (0.6) | 48.0 (0.2) | 47.6 (0.2) | 48.0 (0.2) | 47.9 (0.1) | 48.2 (0.1) | 44.7 (0.1) | 23.1 (0.3) | 47 (0.2) | 23.9 (0.3) |
| | 4.7 (0.4) | 47.9 (0.2) | 47.0 (0.2) | 47.8 (0.2) | 48.2 (0.1) | 48.0 (0.1) | 44.5 (0.2) | 21.6 (0.3) | 46.6 (0.2) | 18.3 (0.3) |
| | | | | M2: $K = 2$, $\mathbf{\Delta} = 0.1\mathbf{I}_p, \mathbf{I}_p$, AR(0.3), AR(0.5) | | | | | | |
| 9.9 (1.5) | 16.4 (2.0) | 28.3 (1.2) | 4.9 (0.1) | 48.1 (0.1) | 11.7 (0.2) | 48 (0.1) | 47.6 (0.2) | 27.5 (0.3) | 46.9 (0.2) | 19.1 (0.3) |
| 13.9 (0.8) | 16.3 (1.4) | 47.6 (0.2) | 43.5 (0.3) | 47.7 (0.2) | 47.2 (0.2) | 48.2 (0.1) | 47.5 (0.2) | 36.5 (0.3) | 47.4 (0.2) | 46.2 (0.3) |
| | 11.5 (0.9) | 46.2 (0.3) | 31.2 (0.4) | 48.3 (0.1) | 44.9 (0.3) | 48.2 (0.1) | 37 (0.4) | 32.2 (0.3) | 44.2 (0.3) | 42.6 (0.5) |
| | 11.4 (0.9) | 43.4 (0.6) | 21.5 (0.3) | 48.0 (0.1) | 43.5 (0.4) | 48.0 (0.2) | 33.9 (0.2) | 31.1 (0.3) | 40.2 (0.3) | 34.9 (0.5) |
| | | | | M3: $K = 2$, $\mathbf{\Delta} = 0.1\mathbf{I}_p, \mathbf{I}_p$, AR(0.3), AR(0.5) | | | | | | |
| 5.9 (1.1) | 3.6 (1.2) | 39.7 (0.7) | 40.0 (0.8) | 47.5 (0.2) | 13.2 (0.2) | 47.8 (0.2) | 44.6 (0.2) | 22.3 (0.3) | 44.5 (0.4) | 15.1 (0.2) |
| 11.1 (1.5) | 8.4 (1.4) | 47.8 (0.2) | 47.7 (0.2) | 48.0 (0.1) | 48.0 (0.1) | 47.9 (0.2) | 45.9 (0.2) | 30.6 (0.4) | 47.9 (0.1) | 41.2 (0.4) |
| | 7.2 (1.4) | 47.8 (0.2) | 47.3 (0.2) | 47.8 (0.2) | 47.5 (0.2) | 47.9 (0.2) | 47.2 (0.1) | 24.8 (0.3) | 47.9 (0.1) | 26.3 (0.3) |
| | 4 (0.9) | 47.0 (0.2) | 47.2 (0.1) | 48.1 (0.1) | 47.7 (0.2) | 47.8 (0.2) | 46.7 (0.1) | 24.1 (0.3) | 47.7 (0.2) | 21.1 (0.3) |
| | | | | M4: $K = 3$, $\mathbf{\Delta} = 0.1\mathbf{I}_p, \mathbf{I}_p$, AR(0.3), AR(0.5) | | | | | | |
| 0 (0) | 0 (0) | 44.1 (1) | 25.4 (0.5) | 18.2 (0.6) | 6.5 (0.1) | 63.3 (0.1) | 58.3 (0.2) | 52.1 (0.2) | 59.4 (0.4) | 29.4 (0.3) |
| 3.7 (0.1) | 2.9 (0.1) | 62.3 (0.2) | 57.2 (0.2) | 64 (0.1) | 60.5 (0.3) | 63.4 (0.1) | 62.8 (0.2) | 53.3 (0.2) | 68.8 (0.3) | 53.7 (0.4) |
| | 2.5 (0.1) | 59.8 (0.3) | 56.1 (0.3) | 64 (0.1) | 57.5 (0.3) | 63.4 (0.1) | 61.8 (0.2) | 50.1 (0.3) | 66 (0.3) | 46.6 (0.4) |
| | 2.2 (0.1) | 56.9 (0.3) | 54.1 (0.2) | 64 (0.1) | 59.6 (0.3) | 63.7 (0.1) | 62.8 (0.2) | 47.8 (0.3) | 63.3 (0.3) | 42.3 (0.5) |
| | | | | M4: $K = 5$, $\mathbf{\Delta} = 0.1\mathbf{I}_p, \mathbf{I}_p$, AR(0.3), AR(0.5) | | | | | | |
| 0.4 (0.3) | 0.2 (0.2) | 50.1 (0.7) | 38.8 (0.6) | 32.6 (0.8) | 13.8 (0.2) | 74.1 (0.2) | 63.4 (0.2) | 46.5 (0.4) | 64.8 (0.2) | 25.5 (0.2) |
| 14.0 (1.2) | 10.2 (1.6) | 75.2 (0.1) | 66.7 (0.2) | 76.9 (0.1) | 75.3 (0.1) | 76 (0.1) | 65.7 (0.1) | 54.3 (0.4) | 75.6 (0.1) | 71.5 (0.3) |
| | 15.7 (2.0) | 73.6 (0.2) | 65.1 (0.2) | 76.8 (0.1) | 74.6 (0.1) | 76 (0.1) | 62.7 (0.1) | 53.7 (0.3) | 72.8 (0.1) | 66.1 (0.3) |
| | 24.4 (2.2) | 72.4 (0.2) | 65 (0.2) | 77 (0.1) | 75.8 (0.1) | 76.1 (0.1) | 63.1 (0.2) | 55.4 (0.2) | 71.1 (0.1) | 62 (0.3) |

*Table 2.* The averages of the prediction errors, the sparsity level $\widehat{s}$, and the corresponding standard errors based on 100 replicates.

| | mixPFC | SEAS-SIR | SEAS-PFC | LassoSIR | Lasso |
|---|---|---|---|---|---|
| | | | Nutlin-3 | | |
| PMSE $\times 100$ | 8.9 (0.4) | 18.7 (0.4) | 18.8 (0.4) | 18.2 (0.3) | 17.6 (0.3) |
| $\widehat{s}$ | 31.5 (2.5) | 48.3 (1.3) | 33.3 (1.8) | 32 (0.9) | 39.1 (1.1) |
| | | | AZD6244 | | |
| PMSE $\times 100$ | 45.6 (1.7) | 108.8 (2.2) | 107.1 (2.2) | 83.2 (1.6) | 77.6 (1.5) |
| $\widehat{s}$ | 77.8 (3.2) | 78.3 (1.6) | 58.8 (1.5) | 66.8 (0.7) | 78.6 (0.7) |

Section G in the appendix.

## Acknowledgements

The authors thank reviewers for constructive comments. The research was partly supported by grant DMS-2053697 from the U.S. National Science Foundation.

## Impact Statement

This paper presents work whose goal is to advance the field of Machine Learning. There are many potential societal consequences of our work, none which we feel must be specifically highlighted here.

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

This appendix provides additional numerical results, implementation details, and technical proofs supporting the theoretical analysis. Section A presents detailed results from the numerical studies. Section B outlines implementation specifics, including the initialization method, selection of the number of clusters, and the mixPFC-ISO algorithm. The theoretical analysis of Theorem 4.1 relies on two key lemmas, with ancillary results provided in Section F. The proofs of the two key lemmas are detailed in Sections D and E, while Section C contains the full proof of the theorem. We discuss potential extensions in Section G.

## A. Additional Numerical Results

### A.1. Simulations

The parameters for M1-M4 in the simulation section are generated as follows.

(M1) The non-zero coefficients of $\boldsymbol{\beta}_1$ are $\boldsymbol{\beta}_{1i} = 2.3, i = 1, \ldots 6$, and then set $\boldsymbol{\beta}_2 = -\boldsymbol{\beta}_1$.

(M2) The non-zero coefficients of $\boldsymbol{\beta}_1$ and $\boldsymbol{\beta}_2$ are $\boldsymbol{\beta}_{1i} = 2.3, 1 \le i \le 6$, and $\boldsymbol{\beta}_{2i} = 2.3$ for $i = 1, 3, 5$, $\boldsymbol{\beta}_{2i} = -2.3$ for $i = 2, 4, 6$.

(M3) The basis matrices $\boldsymbol{\beta}_1$ and $\boldsymbol{\beta}_2$ have two columns. The non-zero rows of $\boldsymbol{\beta}_1$ and $\boldsymbol{\beta}_2$ are

$$(\boldsymbol{\beta}_1)^T_{1:6\cdot} = \begin{pmatrix} 2.3 & 2.3 & 2.3 & 2.3 & 2.3 & 2.3 \\ 2.3 & -2.3 & 2.3 & -2.3 & 2.3 & -2.3 \end{pmatrix},$$
$$(\boldsymbol{\beta}_2)^T_{1:6\cdot} = \begin{pmatrix} 2.3 & -2.3 & -2.3 & -2.3 & -2.3 & -2.3 \\ 2.3 & 2.3 & -2.3 & 2.3 & -2.3 & 2.3 \end{pmatrix}.$$

(M4) The basis matrices $\boldsymbol{\beta}_w, w = 1, \ldots, K$ have two columns. For each cluster $w$, the non-zero elements are generated as

$$(\boldsymbol{\beta}_w)_{ij} \sim \text{Unif}([-2.5, -2.1] \cup [2.1, 2.5]), i \le 10, j = 1, 2, w = 1, 2, \ldots, K.$$

We consider four scenarios of simulations as described in Table A.3. The clustering error rates for S1 and S4 are presented in the main paper. The error rates for scenarios S2 and S3 are summarized in Table A.4&A.5.

To assess the subspace estimation and variable selection accuracy, we define the following criteria: the distance between estimated and actual subspaces is defined as $\mathcal{D}_w = \mathcal{D}(\mathcal{S}_{\boldsymbol{\beta}_w}, \mathcal{S}_{\widehat{\boldsymbol{\beta}}_w}) = \|\mathbf{P}_{\boldsymbol{\beta}_w} - \mathbf{P}_{\widehat{\boldsymbol{\beta}}_w}\|_F / \sqrt{2d_w}, w = 1, \ldots, K$; error rate ER is the fraction of incorrectly classified samples; true positive rate (TPR) and false positive rate (FPR) are defined as $\text{TPR}_w = |\widehat{\mathcal{A}}_w \bigcap \mathcal{A}_w|/|\mathcal{A}_w|$ and $\text{TPR}_w = |\widehat{\mathcal{A}}_w \bigcap \mathcal{A}_w^c|/|\mathcal{A}_w^c|$. When $K > 2$, instead of reporting each $\mathcal{D}_w$, $\text{TPR}_w$ and $\text{FPR}_w$, we calculate the average subspace distance $\overline{\mathcal{D}} = \sum_{w=1}^K \mathcal{D}_w/K$, average TPR and FPR $\overline{\text{TPR}} = \sum_{w=1}^K \text{TPR}_w /K$ and $\overline{\text{FPR}} = \sum_{w=1}^K \text{FPR}_w /K$, and TPR and FPR of the union of selected variables across all clusters $\widetilde{\text{TPR}} = |(\bigcup \widehat{\mathcal{A}}_w) \bigcap (\bigcup \mathcal{A}_w)|/|(\bigcup \mathcal{A}_w)|$ and $\widetilde{\text{FPR}} = |(\bigcup \widehat{\mathcal{A}}_w) \bigcap (\bigcup \mathcal{A}_w^c)|/|(\bigcup \mathcal{A}_w^c)|$.

Table A.6&A.7 summarize variable selection and subspace estimation results for scenario S1&S4 with covariance $\boldsymbol{\Delta} = \mathbf{I}$ and AR(0.3). The corresponding results for $\boldsymbol{\Delta} = 0.1\mathbf{I}$ and AR(0.5) are provided in Table A.8&A.9. Results for S2 and S3 can be found in Table A.10-A.11 and Table A.12-A.13, respectively. Under scenario S1, mixPFC demonstrates strong performance, effectively identifying the important variables and accurately estimating central subspaces. It achieves true positive rates $\text{TPR}_w$ greater than 95%, false positive rates $\text{FPR}_w$ below 1%, and subspace estimation errors $\mathcal{D}_w$ around 0.3 when $\boldsymbol{\Delta} = \mathbf{I}$ and AR(0.3). However, as variables correlation increases ($\boldsymbol{\Delta} = \text{AR}(0.5)$), mixPFC shows reduced accuracy in subspace estimation with $\mathcal{D}_w$ increasing to approximately 0.5, though it remains effective in variable selection. When the covariance matrix contains small elements ($\boldsymbol{\Delta} = 0.1\mathbf{I}$), mixPFC performance declines in both variable selection and subspace estimation. This reduction occurs because estimating $\mathcal{S}_w = \boldsymbol{\Delta}^{-1}\mathcal{S}_{\boldsymbol{\Gamma}_w}$ becomes challenging when all elements of $\boldsymbol{\Delta}$ have small magnitude. In such cases, mixPFC-ISO, which assumes isotropic errors, outperforms mixPFC when $\boldsymbol{\Delta} = 0.1\mathbf{I}$ and $\mathbf{I}$.

In the unbalanced clusters scenario S2, subspace estimation errors are smaller for clusters with more samples, while variable selection and error rates remain consistent with S1. In scenario S3, featuring a nonlinear fitting function, both mixPFC and mixPFC-ISO perform poorly in variable selection and subspace estimation for model M3. Even SEAS-PFC using true clusters offers little improvement. However, the error rates remain well controlled. The performance under M1 and M2 is

similar to S1. In the multi-cluster setting S4, the mixPFC shows 10% - 20% reduction in $\text{TPR}_w$ for each cluster compared to S1. But, the union of selected variables typically includes all the important variables.

When compared with LassoSIR, SEAS-SIR, and SEAS-PFC fitted on estimated clusters, mixPFC has similar $\text{TPR}_w$ and $\text{FPR}_w$ but larger subspace estimation errors. This is likely because the algorithm incorporates some misleading information from other clusters through $\gamma_{iw}$. Among the high-dimensional SDR methods, SEAS-SIR and SEAS-PFC consistently outperformed LassoSIR. Therefore, it is recommended to refit with SEAS-PFC after classifying the data using mixPFC. Notably, fitting with true clusters offers only marginal improvements—typically just a few percentage points—over using clusters estimated by mixPFC.

*Table A.3.* Parameters settings of four sets of simulations. The linking matrices $\boldsymbol{\eta}_w = (1, 0.3)$ for M1 and M2, and $\boldsymbol{\eta}_w = \mathbf{I}$ for M3 and M4.

| Scenario | Model | $\pi$ | Y | $\mathbf{f}(Y)$ |
|---|---|---|---|---|
| S1 | M1-M3 | $(0.5, 0.5)$ | $\text{Unif}(-1, 1)$ | $(Y, \|Y\|)^T$ |
| S2 | M1-M3 | $(0.7, 0.3)$ | $\text{Unif}(-1, 1)$ | $(Y, \|Y\|)^T$ |
| S3 | M1-M3 | $(0.5, 0.5)$ | $\text{Unif}(0, 2)$ | $(Y^2/3, Y^3/6)^T$ |
| S4 | M4 $K = 3, 5$ | $(1/K, \ldots, 1/K)$ | $\text{Unif}(-1, 1)$ | $(Y, \|Y\|)^T$ |

*Table A.4.* Averages and standard errors of clustering error rates for scenario S2 with $n = 400, p = 1000, K = 2$.

| Method | mixPFC-ISO | mixPFC | RANSAC | LSA | LRSC | SSC | GPCA | $K$-means $\mathbf{X}$ | $K$-means $\mathbf{X} \circ Y$ | hclust $\mathbf{X}$ | hclust $\mathbf{X} \circ Y$ |
|---|---|---|---|---|---|---|---|---|---|---|---|
| | | | | M1: $n = 400, p = 1000, \boldsymbol{\Delta} = 0.1\mathbf{I}, \mathbf{I}, \text{AR}(0.3), \text{AR}(0.5)$ | | | | | | |
| ER(%) | 7.7 (1.3) | 3.1 (0.7) | 40.8 (0.5) | 48.1 (0.1) | 47.7 (0.2) | 47.4 (0.2) | 47.3 (0.2) | 48.7 (0.1) | 33.1 (0.6) | 46.1 (0.3) | 10.3 (0.5) |
| | 13.1 (1.3) | 7.2 (0.6) | 48 (0.2) | 41.2 (0.4) | 42.5 (0.4) | 47.9 (0.1) | 47.4 (0.2) | 48.6 (0.1) | 42.7 (0.7) | 43 (0.5) | 27.8 (0.9) |
| | | 6.9 (0.9) | 46.4 (0.3) | 44.5 (0.3) | 47.8 (0.1) | 47.7 (0.2) | 45 (0.4) | 48.5 (0.1) | 35.9 (0.7) | 46.5 (0.3) | 18.8 (0.8) |
| | | 4.4 (0.5) | 43.8 (0.4) | 45.5 (0.3) | 48 (0.1) | 48 (0.1) | 44.9 (0.4) | 48.7 (0.1) | 35.2 (0.6) | 45.9 (0.3) | 13.5 (0.5) |
| | | | | M2: $n = 400, p = 1000, \boldsymbol{\Delta} = 0.1\mathbf{I}, \mathbf{I}, \text{AR}(0.3), \text{AR}(0.5)$ | | | | | | |
| ER(%) | 12.9 (1.7) | 14.4 (1.9) | 29.9 (1.1) | 5.4 (0.1) | 47.6 (0.2) | 11.3 (0.2) | 47 (0.2) | 48.3 (0.1) | 41.7 (0.3) | 45.3 (0.3) | 13.9 (0.6) |
| | 13.2 (0.5) | 14.4 (1.3) | 47.4 (0.2) | 40.6 (0.5) | 39.5 (0.5) | 46.9 (0.2) | 47.9 (0.2) | 48.3 (0.1) | 45.9 (0.6) | 40.8 (0.6) | 30.7 (0.4) |
| | | 10 (0.7) | 44.8 (0.4) | 33.6 (0.3) | 48.1 (0.2) | 46.1 (0.3) | 44.2 (0.4) | 46.5 (0.2) | 45.6 (0.2) | 47.2 (0.2) | 29.9 (0.7) |
| | | 9.4 (0.8) | 40.8 (0.7) | 24.4 (0.2) | 47.9 (0.2) | 46.2 (0.3) | 44 (0.4) | 44.2 (0.2) | 44.4 (0.3) | 48 (0.2) | 31.8 (1.2) |
| | | | | M3: $n = 400, p = 1000, \boldsymbol{\Delta} = 0.1\mathbf{I}, \mathbf{I}, \text{AR}(0.3), \text{AR}(0.5)$ | | | | | | |
| ER(%) | 8.0 (1.2) | 2.1 (0.9) | 37.8 (0.8) | 46.1 (0.3) | 46.8 (0.2) | 14.4 (0.5) | 47.1 (0.2) | 45.1 (0.2) | 35.7 (0.5) | 43 (0.5) | 11.7 (0.5) |
| | 10.9 (1.2) | 3.3 (0.7) | 47.9 (0.2) | 45.8 (0.3) | 39.4 (0.3) | 47.6 (0.2) | 46.9 (0.2) | 46.8 (0.2) | 44.4 (0.7) | 41.8 (0.6) | 29.7 (0.8) |
| | | 3.9 (1) | 45.6 (0.3) | 46.9 (0.2) | 48.1 (0.1) | 47.7 (0.2) | 44 (0.4) | 48 (0.1) | 39.3 (0.5) | 46.7 (0.2) | 20 (0.8) |
| | | 4.8 (1) | 42.4 (0.4) | 46.6 (0.2) | 48.2 (0.2) | 48.1 (0.2) | 44.4 (0.4) | 48.2 (0.1) | 38.2 (0.4) | 46.6 (0.2) | 14.7 (0.5) |

*Table A.5.* Averages and standard errors of clustering error rates for scenario S3 with $n = 400, p = 1000, K = 2$.

| Method | mixPFC-ISO | mixPFC | RANSAC | LSA | LRSC | SSC | GPCA | $K$-means $\mathbf{X}$ | $K$-means $\mathbf{X} \circ Y$ | hclust $\mathbf{X}$ | hclust $\mathbf{X} \circ Y$ |
|---|---|---|---|---|---|---|---|---|---|---|---|
| | | | | M1: $n = 400, p = 1000, \boldsymbol{\Delta} = 0.1\mathbf{I}_p, \mathbf{I}_p, \text{AR}(0.3), \text{AR}(0.5)$ | | | | | | |
| ER(%) | 6.1 (0.8) | 3.8 (0.1) | 47.7 (0.2) | 42.7 (0.2) | 48.3 (0.1) | 47.9 (0.2) | 48 (0.2) | 39.9 (0.2) | 39.6 (0.2) | 42.5 (0.3) | 35.8 (0.2) |
| | 14.2 (0.5) | 14.3 (0.8) | 48 (0.2) | 47.8 (0.2) | 47.9 (0.1) | 47.9 (0.2) | 47.9 (0.2) | 44 (0.2) | 43.6 (0.3) | 47.9 (0.2) | 45.4 (0.3) |
| | | 9.1 (0.5) | 47.9 (0.1) | 47 (0.2) | 48.2 (0.1) | 48.1 (0.2) | 41 (0.2) | 40.9 (0.2) | 47.6 (0.2) | 40.6 (0.3) |
| | | 7.8 (0.6) | 48 (0.1) | 44.3 (0.3) | 48.1 (0.1) | 48 (0.1) | 48.2 (0.1) | 40.6 (0.1) | 39.6 (0.2) | 46.3 (0.2) | 37.7 (0.2) |
| | | | | M2: $n = 400, p = 1000, \boldsymbol{\Delta} = 0.1\mathbf{I}_p, \mathbf{I}_p, \text{AR}(0.3), \text{AR}(0.5)$ | | | | | | |
| ER(%) | 6.9 (0.8) | 11.1 (1.6) | 40 (0.8) | 18.7 (1.2) | 48.1 (0.2) | 16.9 (0.2) | 48.2 (0.1) | 46.7 (0.2) | 40.4 (0.2) | 45.3 (0.4) | 38.1 (0.2) |
| | 21.7 (0.7) | 19.9 (0.9) | 48 (0.2) | 47.9 (0.2) | 48 (0.1) | 47.9 (0.2) | 47.8 (0.2) | 47.7 (0.2) | 45.2 (0.3) | 48 (0.2) | 47.2 (0.2) |
| | | 17.6 (0.8) | 47.5 (0.2) | 42 (0.2) | 47.9 (0.2) | 47.6 (0.2) | 48.1 (0.1) | 46.3 (0.3) | 40.8 (0.2) | 47.5 (0.2) | 47 (0.2) |
| | | 16.7 (0.7) | 46.3 (0.2) | 38.4 (0.2) | 47.8 (0.2) | 46.8 (0.2) | 47.9 (0.2) | 41.4 (0.4) | 39.6 (0.2) | 44.3 (0.3) | 44.9 (0.3) |
| | | | | M3: $n = 400, p = 1000, \boldsymbol{\Delta} = 0.1\mathbf{I}_p, \mathbf{I}_p, \text{AR}(0.3), \text{AR}(0.5)$ | | | | | | |
| ER(%) | 17.9 (1.8) | 22.6 (2.2) | 33.1 (1.1) | 11.1 (0.2) | 47.6 (0.2) | 12.6 (0.2) | 48.1 (0.2) | 39.9 (0.1) | 39.8 (0.2) | 39.7 (0.3) | 35.2 (0.2) |
| | 20 (1.2) | 13.8 (1.1) | 47.6 (0.2) | 46.5 (0.2) | 47.9 (0.2) | 47.3 (0.2) | 47.8 (0.2) | 42.9 (0.2) | 42.1 (0.3) | 47.7 (0.2) | 43.9 (0.3) |
| | | 13.1 (1.2) | 47.7 (0.2) | 45.5 (0.2) | 48.1 (0.1) | 47.5 (0.2) | 48.1 (0.2) | 44 (0.2) | 40.2 (0.3) | 47.5 (0.2) | 42.6 (0.3) |
| | | 11.5 (1.1) | 47.2 (0.3) | 44.2 (0.2) | 48 (0.2) | 46.9 (0.2) | 48.1 (0.2) | 44.8 (0.2) | 39 (0.2) | 47.4 (0.2) | 39.4 (0.3) |

*Table A.6.* Simulations results for scenario S1 with $n = 400, p = 1000, K = 2$. The table reports averages and standard errors of TPR, FPR, and subspace distance for each cluster.

| Method | $\mathbf{\Delta = I}$ | | | $\mathbf{\Delta} = \mathrm{AR}(0.3)$ | | |
|---|---|---|---|---|---|---|
| | $\mathrm{TPR}_w(\%)$ | $\mathrm{FPR}_w(\%)$ | $\mathcal{D}_w \times 100$ | $\mathrm{TPR}_w(\%)$ | $\mathrm{FPR}_w(\%)$ | $\mathcal{D}_w \times 100$ |
| | | | M1 | | | |
| mixPFC | 95, 96.7 (1.7, 1.4) | 2, 1.6 (0.6, 0.5) | 30, 29.4 (2.2, 2.1) | 98.7, 98.3 (0.7, 1) | 0.2, 0.4 (0.2, 0.3) | 35, 35 (1.4, 1.4) |
| mixPFC-ISO | 100, 100 (0, 0) | 0.4, 0.4 (0, 0) | 6.3, 6.4 (0.2, 0.2) | | | |
| LassoSIR$^t$ | 100, 100 (0, 0) | 1.5, 1.7 (0.1, 0.1) | 30.6, 31.7 (1, 1) | 99.5, 99.7 (0.3, 0.2) | 1.3, 1.1 (0.1, 0.1) | 41.2, 41.1 (1.2, 1.1) |
| LassoSIR$^m$ | 94.7, 96 (1.8, 1.5) | 2.3, 2 (0.4, 0.3) | 38.2, 37 (2.1, 2.1) | 98.5, 97.7 (0.8, 1.1) | 1.2, 1.2 (0.2, 0.2) | 41.2, 41 (1.5, 1.4) |
| SEAS-SIR$^t$ | 100, 100 (0, 0) | 0.5, 0.6 (0.1, 0.1) | 7.5, 7.5 (0.4, 0.3) | 100, 100 (0, 0) | 0.1, 0.1 (0, 0) | 13.9, 14.5 (0.3, 0.4) |
| SEAS-SIR$^m$ | 99.1, 99.5 (0.5, 0.4) | 0.7, 0.8 (0.1, 0.2) | 10.7, 10.6 (1.3, 1.3) | 99.8, 99.3 (0.2, 0.5) | 0.2, 0.2 (0.1, 0.1) | 15, 15.8 (0.6, 1) |
| SEAS-PFC$^t$ | 100, 100 (0, 0) | 1.2, 1.4 (0.2, 0.2) | 8, 8 (0.4, 0.4) | 100, 100 (0, 0) | 0.2, 0.2 (0, 0) | 13.5, 14.2 (0.3, 0.4) |
| SEAS-PFC$^m$ | 96.7, 96.8 (1.5, 1.4) | 1.2, 1.6 (0.4, 0.3) | 14.9, 13.9 (2.4, 2) | 99.8, 99.7 (0.2, 0.3) | 0.3, 0.5 (0.1, 0.2) | 15.1, 15.7 (1, 1.2) |
| | | | M2 | | | |
| mixPFC | 95.3, 94.5 (1.4, 1.5) | 0.7, 0.8 (0.1, 0.1) | 31.7, 31.8 (1.9, 1.9) | 98.8, 100 (0.5, 0) | 0, 1.3 (0, 0.1) | 37.3, 24.1 (1.4, 0.9) |
| mixPFC-ISO | 98.7, 98.8 (0.8, 0.6) | 0.5, 0.8 (0.1, 0.2) | 9.2, 10 (1.3, 1.6) | | | |
| LassoSIR$^t$ | 100, 100 (0, 0) | 1.5, 1.3 (0.1, 0.1) | 30.6, 29.8 (1, 1) | 99.5, 100 (0.3, 0) | 1.3, 2.4 (0.1, 0.2) | 41.2, 29.9 (1.2, 1) |
| LassoSIR$^m$ | 92.5, 91.7 (1.7, 1.9) | 1.7, 1.8 (0.2, 0.2) | 40.9, 39.2 (2.1, 2.1) | 97, 99 (1.1, 0.5) | 1.1, 2.4 (0.1, 0.2) | 43.5, 32.1 (1.5, 1.3) |
| SEAS-SIR$^t$ | 100, 100 (0, 0) | 0.5, 0.6 (0.1, 0.1) | 7.5, 7.8 (0.4, 0.3) | 100, 100 (0, 0) | 0.1, 1.8 (0, 0.1) | 13.9, 30.9 (0.3, 0.5) |
| SEAS-SIR$^m$ | 93.8, 92.7 (1.6, 1.7) | 0.4, 0.5 (0.1, 0.1) | 19.4, 19.2 (2.4, 2.4) | 98.2, 97.3 (0.8, 1.1) | 0.1, 1.8 (0, 0.2) | 18, 33.8 (1.3, 1.1) |
| SEAS-PFC$^t$ | 100, 100 (0, 0) | 1.2, 1.3 (0.2, 0.2) | 8, 8.1 (0.4, 0.3) | 100, 100 (0, 0) | 0.2, 0.9 (0, 0.1) | 13.5, 28.4 (0.3, 0.5) |
| SEAS-PFC$^m$ | 92.8, 92.3 (1.7, 1.7) | 1.1, 0.9 (0.2, 0.2) | 19.6, 19.6 (2.4, 2.4) | 97.5, 97.7 (1.1, 1) | 0.2, 1 (0, 0.1) | 17.7, 30.9 (1.4, 1.1) |
| | | | M3 | | | |
| mixPFC | 94.2, 94.3 (1.7, 1.6) | 0.2, 0.1 (0.1, 0.1) | 29.5, 29.6 (2.2, 2.1) | 96.8, 98.5 (0.9, 0.6) | 0.3, 0.4 (0.2, 0.1) | 30, 35.7 (1.7, 1.4) |
| mixPFC-ISO | 98.8, 97.2 (1, 1.6) | 0.7, 0.7 (0.1, 0.2) | 21.4, 22.1 (1.7, 2) | | | |
| LassoSIR$^t$ | 100, 100 (0, 0) | 6.6, 6.6 (0.2, 0.2) | 51.1, 50.5 (0.8, 0.9) | 99.8, 100 (0.2, 0) | 9.8, 8.4 (0.2, 0.2) | 69.3, 68.5 (0.5, 0.5) |
| LassoSIR$^m$ | 94.8, 96.3 (1.5, 1.2) | 7.6, 7.2 (0.3, 0.3) | 59, 57.4 (1.4, 1.5) | 98.3, 98.7 (0.6, 0.6) | 9.3, 8.1 (0.2, 0.2) | 71, 70.3 (0.7, 0.7) |
| SEAS-SIR$^t$ | 100, 100 (0, 0) | 2.2, 2 (0.3, 0.3) | 9.8, 9.6 (0.3, 0.3) | 100, 100 (0, 0) | 0.2, 0.3 (0, 0) | 18, 31.2 (0.4, 0.3) |
| SEAS-SIR$^m$ | 94.1, 92.6 (1.5, 1.7) | 1.4, 2.2 (0.2, 0.3) | 21.8, 23.2 (2.3, 2.4) | 95.3, 95 (1.6, 1.6) | 0.1, 0.2 (0, 0) | 26.7, 37.7 (2, 1.6) |
| SEAS-PFC$^t$ | 100, 100 (0, 0) | 1.5, 1.9 (0.2, 0.3) | 9.1, 9.7 (0.3, 0.3) | 100, 100 (0, 0) | 0.2, 0.3 (0, 0) | 17.7, 30.8 (0.4, 0.3) |
| SEAS-PFC$^m$ | 91.5, 89.2 (1.9, 2.2) | 1.2, 1.3 (0.2, 0.2) | 22.4, 23.6 (2.5, 2.6) | 94.3, 93 (1.7, 1.9) | 0.2, 0.2 (0.1, 0) | 26.7, 37.6 (2, 1.6) |

*Table A.7.* Simulations results for scenario S4 with $n = 400K, p = 1000, K = 3, 5$. The table reports averages and standard errors of TPR, FPR, and subspace distance, calculated as the mean values across $K$ clusters.

| Method | $\mathbf{\Delta = I}$ | | | | | $\mathbf{\Delta} = \mathrm{AR}(0.3)$ | | | | |
|---|---|---|---|---|---|---|---|---|---|---|
| | $\overline{\mathrm{TPR}}(\%)$ | $\overline{\mathrm{FPR}}(\%)$ | $\widetilde{\mathrm{TPR}}(\%)$ | $\widetilde{\mathrm{FPR}}(\%)$ | $\overline{\mathcal{D}} \times 100$ | $\overline{\mathrm{TPR}}(\%)$ | $\overline{\mathrm{FPR}}(\%)$ | $\widetilde{\mathrm{TPR}}(\%)$ | $\widetilde{\mathrm{FPR}}(\%)$ | $\overline{\mathcal{D}} \times 100$ |
| | | | | | M4 $K = 3$ | | | | | |
| mixPFC | 99.6 (0.1) | 0 (0) | 100 (0) | 0 (0) | 27.2 (0.4) | 86.7 (0.8) | 0.3 (0.1) | 100 (0) | 1 (0.2) | 45.5 (0.6) |
| mixPFC-ISO | 100 (0) | 0 (0) | 100 (0) | 0.1 (0) | 14.7 (0.2) | | | | | |
| LassoSIR$^t$ | 100 (0) | 6.1 (0.1) | 100 (0) | 17.3 (0.3) | 53.1 (0.4) | 99.4 (0.1) | 6 (0.1) | 100 (0) | 17 (0.2) | 61.2 (0.3) |
| LassoSIR$^m$ | 100 (0) | 6.1 (0.1) | 100 (0) | 17.2 (0.3) | 52.9 (0.4) | 99.4 (0.2) | 6.1 (0.1) | 100 (0) | 17.4 (0.3) | 61.7 (0.3) |
| SEAS-SIR$^t$ | 100 (0) | 2.5 (0.1) | 100 (0) | 7.5 (0.4) | 11.3 (0.2) | 92 (0.6) | 0.3 (0.1) | 100 (0) | 0.9 (0.2) | 43.8 (0.7) |
| SEAS-SIR$^m$ | 100 (0) | 2.3 (0.1) | 100 (0) | 6.7 (0.4) | 12.1 (0.2) | 90.4 (0.6) | 0.3 (0.1) | 100 (0) | 0.8 (0.2) | 45.4 (0.7) |
| SEAS-PFC$^t$ | 100 (0) | 1 (0.1) | 100 (0) | 3 (0.3) | 9.7 (0.2) | 90.3 (0.8) | 0.1 (0) | 100 (0) | 0.3 (0) | 44 (0.8) |
| SEAS-PFC$^m$ | 100 (0) | 0.8 (0.1) | 100 (0) | 2.4 (0.3) | 10.5 (0.2) | 89.4 (0.8) | 0.1 (0) | 100 (0) | 0.3 (0) | 45.6 (0.8) |
| | | | | | M4 $K = 5$ | | | | | |
| mixPFC | 98 (0.4) | 0.1 (0) | 100 (0) | 0.4 (0.1) | 32.3 (1) | 88 (0.7) | 0.8 (0.1) | 100 (0) | 4 (0.4) | 50 (1.2) |
| mixPFC-ISO | 97.2 (0.6) | 0.2 (0) | 100 (0) | 0.9 (0.2) | 25.6 (1.4) | | | | | |
| LassoSIR$^t$ | 100 (0) | 6.1 (0.1) | 100 (0) | 27 (0.3) | 53.6 (0.3) | 99.8 (0.1) | 6.2 (0.1) | 100 (0) | 27.4 (0.4) | 62.5 (0.3) |
| LassoSIR$^m$ | 98.3 (0.5) | 6.3 (0.1) | 100 (0) | 27.6 (0.5) | 57.5 (1.1) | 93.7 (1) | 6.8 (0.2) | 99.8 (0.1) | 29.6 (0.6) | 69.9 (1.2) |
| SEAS-SIR$^t$ | 100 (0) | 2.5 (0.1) | 100 (0) | 11.8 (0.6) | 11.4 (0.2) | 94.6 (0.4) | 0.4 (0) | 100 (0) | 1.8 (0.2) | 42.9 (0.5) |
| SEAS-SIR$^m$ | 99.3 (0.2) | 2.6 (0.1) | 100 (0) | 12.2 (0.5) | 17.3 (1.3) | 88.6 (0.8) | 0.8 (0.1) | 100 (0) | 3.8 (0.5) | 50.3 (1.1) |
| SEAS-PFC$^t$ | 100 (0) | 1.2 (0.1) | 100 (0) | 5.9 (0.4) | 9.6 (0.2) | 91.8 (0.5) | 0.3 (0) | 100 (0) | 1.2 (0.1) | 44.3 (0.5) |
| SEAS-PFC$^m$ | 98.7 (0.3) | 1.1 (0.1) | 100 (0) | 5.4 (0.4) | 15.5 (1.3) | 86.3 (0.9) | 0.5 (0.1) | 100 (0) | 2.5 (0.5) | 50.9 (1) |

*Table A.8.* Simulations results for scenario S1 with $n = 400, p = 1000, K = 2$. The table reports averages and standard errors of TPR, FPR, and subspace distance for each cluster.

| Method | $\Delta = 0.1\mathbf{I}$ | | | $\Delta = \mathrm{AR}(0.5)$ | | |
|---|---|---|---|---|---|---|
| | $\mathrm{TPR}_w(\%)$ | $\mathrm{FPR}_w(\%)$ | $\mathcal{D}_w \times 100$ | $\mathrm{TPR}_w(\%)$ | $\mathrm{FPR}_w(\%)$ | $\mathcal{D}_w \times 100$ |
| | | | M1 | | | |
| mixPFC | 97.5, 98 (0.9, 0.7) | 0, 0 (0, 0) | 39, 39.2 (1.2, 1.3) | 96.3, 96.5 (1.1, 0.8) | 0.9, 1.1 (0.1, 0.3) | 50.3, 48.3 (1.3, 1.2) |
| mixPFC-ISO | 100, 100 (0, 0) | 0.1, 0.1 (0, 0) | 1.7, 1.6 (0.1, 0.1) | | | |
| LassoSIR$^t$ | 98.8, 99.5 (0.4, 0.3) | 0.8, 0.9 (0.1, 0.1) | 43.9, 44.8 (1.2, 1.1) | 96.2, 95.3 (0.7, 0.8) | 1.1, 1 (0.1, 0.1) | 52.8, 51.7 (1.2, 1.2) |
| LassoSIR$^m$ | 98, 98.5 (0.7, 0.7) | 0.8, 0.9 (0.1, 0.1) | 45.2, 45.5 (1.3, 1.3) | 96, 94.5 (0.8, 1.2) | 1, 1.2 (0.1, 0.2) | 51.7, 52.8 (1.2, 1.4) |
| SEAS-SIR$^t$ | 100, 100 (0, 0) | 0.1, 0.1 (0, 0) | 2.7, 2.5 (0.2, 0.2) | 100, 100 (0, 0) | 1.1, 1 (0.1, 0.1) | 19.7, 19.7 (0.4, 0.4) |
| SEAS-SIR$^m$ | 100, 100 (0, 0) | 0.1, 0.2 (0, 0.1) | 2.9, 3 (0.2, 0.2) | 99.8, 99.8 (0.2, 0.2) | 0.8, 1 (0.1, 0.2) | 20.5, 20.6 (0.6, 0.8) |
| SEAS-PFC$^t$ | 100, 100 (0, 0) | 0, 0 (0, 0) | 2.8, 2.4 (0.2, 0.2) | 100, 100 (0, 0) | 0.1, 0.1 (0, 0) | 19.8, 19.5 (0.5, 0.4) |
| SEAS-PFC$^m$ | 100, 100 (0, 0) | 0.2, 0 (0.1, 0) | 3, 3.1 (0.3, 0.3) | 100, 99.2 (0, 0.6) | 0.1, 0.3 (0, 0.2) | 20.5, 20.6 (0.6, 1) |
| | | | M2 | | | |
| mixPFC | 87.8, 87.5 (1.8, 1.9) | 0.2, 0.2 (0.1, 0.1) | 50.6, 49.4 (1.5, 1.7) | 95.5, 100 (1, 0) | 0.8, 2.6 (0.1, 0.2) | 54.2, 34.1 (1.6, 1) |
| mixPFC-ISO | 100, 99.8 (0, 0.2) | 0.2, 0.2 (0, 0) | 9.8, 10.3 (2, 2.1) | | | |
| LassoSIR$^t$ | 98.8, 99.3 (0.4, 0.3) | 0.8, 1.1 (0.1, 0.1) | 44.3, 45.1 (1.2, 1.2) | 96.3, 97.7 (0.7, 0.6) | 1.1, 3.8 (0.1, 0.2) | 52.9, 40.4 (1.2, 1.1) |
| LassoSIR$^m$ | 84.7, 84.7 (2.3, 2.3) | 1.2, 1.1 (0.1, 0.1) | 57.2, 55 (1.9, 1.7) | 91.7, 96.3 (1.4, 0.9) | 1, 4 (0.1, 0.2) | 55.3, 43.7 (1.4, 1.4) |
| SEAS-SIR$^t$ | 100, 100 (0, 0) | 0.1, 0.1 (0, 0) | 2.9, 2.6 (0.2, 0.1) | 100, 89.5 (0, 1.3) | 1.1, 2.5 (0.1, 0.2) | 19.7, 63.8 (0.4, 0.5) |
| SEAS-SIR$^m$ | 86.3, 86.2 (2.2, 2.2) | 0.1, 0.1 (0, 0) | 23.1, 23.1 (3.1, 3.1) | 99, 89.8 (0.5, 1.5) | 0.8, 2.4 (0.1, 0.2) | 23.6, 65.1 (1.2, 0.6) |
| SEAS-PFC$^t$ | 100, 100 (0, 0) | 0, 0 (0, 0) | 2.8, 2.6 (0.2, 0.2) | 100, 95.7 (0, 0.9) | 0.1, 1.9 (0, 0.2) | 19.8, 59.3 (0.5, 0.5) |
| SEAS-PFC$^m$ | 85.5, 85.3 (2.3, 2.3) | 0, 0 (0, 0) | 23.3, 23.1 (3.1, 3.1) | 98.2, 92 (0.7, 1.4) | 0.1, 1.7 (0, 0.2) | 23.8, 60.6 (1.3, 0.6) |
| | | | M3 | | | |
| mixPFC | 92.3, 92.5 (2, 1.9) | 0, 0.3 (0, 0.1) | 39.2, 37.9 (1.8, 1.9) | 96.7, 97.5 (1.1, 0.8) | 0.5, 0.9 (0.1, 0.3) | 40.5, 50.1 (1.2, 1.2) |
| mixPFC-ISO | 100, 100 (0, 0) | 0.4, 0.4 (0, 0) | 10.8, 10.7 (1.5, 1.5) | | | |
| LassoSIR$^t$ | 100, 100 (0, 0) | 4.5, 4.8 (0.2, 0.2) | 57.3, 58.5 (0.8, 0.8) | 98.7, 98.5 (0.5, 0.5) | 8.8, 6.4 (0.2, 0.3) | 77.9, 75.3 (0.4, 0.4) |
| LassoSIR$^m$ | 95.5, 95.3 (1.7, 1.7) | 4.6, 4.7 (0.2, 0.2) | 59.8, 60.1 (1.3, 1.2) | 96.5, 98.2 (0.9, 0.7) | 9.1, 6.8 (0.2, 0.3) | 78.5, 76 (0.5, 0.4) |
| SEAS-SIR$^t$ | 100, 100 (0, 0) | 0, 0 (0, 0) | 3.8, 3.6 (0.2, 0.2) | 100, 87.5 (0, 1.3) | 0.8, 0.6 (0.1, 0.2) | 34.3, 53.4 (0.6, 0.9) |
| SEAS-SIR$^m$ | 95.2, 96 (1.7, 1.5) | 0, 0 (0, 0) | 9.9, 9.7 (2.1, 2) | 99.5, 89 (0.3, 1.3) | 0.7, 0.8 (0.1, 0.2) | 36.6, 53.3 (1, 0.9) |
| SEAS-PFC$^t$ | 100, 100 (0, 0) | 0, 0 (0, 0) | 3, 2.9 (0.1, 0.1) | 100, 83 (0, 1.5) | 0.1, 0.1 (0, 0) | 33.1, 54.1 (0.5, 0.9) |
| SEAS-PFC$^m$ | 94.8, 94.7 (1.8, 1.8) | 0, 0 (0, 0) | 9.2, 9.1 (2.1, 2.2) | 98.3, 86.3 (1, 1.6) | 0.1, 0.2 (0, 0.1) | 35.7, 53.2 (1.1, 1) |

*Table A.9.* Simulations results for scenario S4 with $n = 400K^*, p = 1000, K = 3, 5$. The table reports averages and standard errors of TPR, FPR, and subspace distance, calculated as the mean values across $K$ clusters.

| Method | $\Delta = 0.1\mathbf{I}$ | | | | | $\Delta = \mathrm{AR}(0.5)$ | | | | |
|---|---|---|---|---|---|---|---|---|---|---|
| | $\overline{\mathrm{TPR}}(\%)$ | $\overline{\mathrm{FPR}}(\%)$ | $\widetilde{\mathrm{TPR}}(\%)$ | $\widetilde{\mathrm{FPR}}(\%)$ | $\overline{\mathcal{D}} \times 100$ | $\overline{\mathrm{TPR}}(\%)$ | $\overline{\mathrm{FPR}}(\%)$ | $\widetilde{\mathrm{TPR}}(\%)$ | $\widetilde{\mathrm{FPR}}(\%)$ | $\overline{\mathcal{D}} \times 100$ |
| | | | | | M4 $K^* = 3$ | | | | | |
| mixPFC | 65.9 (0.9) | 0 (0) | 93.5 (0.8) | 0 (0) | 62.9 (0.7) | 77.9 (0.7) | 0.7 (0.1) | 99.7 (0.2) | 2 (0.2) | 57 (0.5) |
| mixPFC-ISO | 100 (0) | 0.1 (0) | 100 (0) | 0.3 (0) | 3.3 (0.1) | | | | | |
| LassoSIR$^t$ | 98.7 (0.2) | 4.4 (0.1) | 100 (0) | 12.7 (0.3) | 68 (0.3) | 96.5 (0.3) | 6.1 (0.1) | 100 (0) | 17.3 (0.3) | 70.2 (0.3) |
| LassoSIR$^m$ | 98.8 (0.2) | 4.5 (0.1) | 100 (0) | 12.9 (0.3) | 68.2 (0.4) | 96.4 (0.3) | 6.1 (0.1) | 100 (0) | 17.2 (0.3) | 70.2 (0.3) |
| SEAS-SIR$^t$ | 99.9 (0) | 0 (0) | 100 (0) | 0 (0) | 12.5 (0.5) | 76.4 (0.5) | 0.6 (0) | 100 (0) | 1.7 (0.1) | 61.8 (0.2) |
| SEAS-SIR$^m$ | 100 (0) | 0 (0) | 100 (0) | 0 (0) | 12 (0.4) | 75.3 (0.5) | 0.5 (0) | 100 (0) | 1.5 (0.1) | 62.2 (0.2) |
| SEAS-PFC$^t$ | 100 (0) | 0 (0) | 100 (0) | 0 (0) | 9.1 (0.3) | 72 (0.4) | 0 (0) | 100 (0) | 0.1 (0) | 61.9 (0.1) |
| SEAS-PFC$^m$ | 100 (0) | 0 (0) | 100 (0) | 0 (0) | 8.9 (0.3) | 72.4 (0.5) | 0 (0) | 100 (0) | 0.1 (0) | 61.9 (0.1) |
| | | | | | M4 $K^* = 5$ | | | | | |
| mixPFC | 65.7 (0.8) | 0 (0) | 97.8 (0.4) | 0 (0) | 64.6 (0.7) | 76.8 (0.7) | 1 (0.1) | 100 (0) | 5.1 (0.4) | 65.6 (0.9) |
| mixPFC-ISO | 100 (0) | 0.1 (0) | 100 (0) | 0.6 (0) | 3.8 (0.4) | | | | | |
| LassoSIR$^t$ | 98.9 (0.1) | 4.4 (0.1) | 100 (0) | 19.9 (0.3) | 67.7 (0.3) | 97.5 (0.2) | 6.1 (0.1) | 100 (0) | 27.1 (0.4) | 70.7 (0.2) |
| LassoSIR$^m$ | 98.8 (0.2) | 4.4 (0.1) | 100 (0) | 20.3 (0.3) | 68 (0.3) | 84.8 (1.4) | 6.5 (0.1) | 99.9 (0.1) | 28.5 (0.5) | 80.1 (0.9) |
| SEAS-SIR$^t$ | 99.9 (0) | 0 (0) | 100 (0) | 0 (0) | 10 (0.3) | 80.3 (0.4) | 0.4 (0) | 100 (0) | 1.7 (0.1) | 62.8 (0.1) |
| SEAS-SIR$^m$ | 99.8 (0.1) | 0 (0) | 100 (0) | 0 (0) | 10.4 (0.4) | 74.6 (0.8) | 0.8 (0.1) | 100 (0) | 4 (0.5) | 68.1 (0.7) |
| SEAS-PFC$^t$ | 100 (0) | 0 (0) | 100 (0) | 0 (0) | 7.4 (0.3) | 77.3 (0.3) | 0 (0) | 100 (0) | 0.2 (0) | 61.5 (0.1) |
| SEAS-PFC$^m$ | 99.9 (0.1) | 0 (0) | 100 (0) | 0 (0) | 7.7 (0.3) | 74.5 (0.7) | 0.8 (0.1) | 100 (0) | 3.7 (0.7) | 67 (0.6) |

*Table A.10.* Simulations results for scenario S2 with $n = 400, p = 1000, K = 2$. The table reports averages and standard errors of TPR, FPR, and subspace distance for each cluster.

| Method | $\boldsymbol{\Delta} = 0.1\mathbf{I}$ | | | $\boldsymbol{\Delta} = \mathbf{I}$ | | |
|---|---|---|---|---|---|---|
| | TPR(%) | FPR(%) | $\mathcal{D} \times 100$ | TPR(%) | FPR(%) | $\mathcal{D} \times 100$ |
| | | | M1 | | | |
| mixPFC | 97.8, 97.3 (0.7, 0.9) | 0, 0 (0, 0) | 35.6, 35.9 (1.3, 1.4) | 98.2, 98.8 (1, 0.6) | 0.5, 1 (0.1, 0.3) | 23.7, 23.9 (1.5, 1.6) |
| mixPFC-ISO | 100, 100 (0, 0) | 0.2, 0.3 (0, 0) | 1.9, 1.9 (0.1, 0.1) | 100, 99.8 (0, 0.2) | 0.8, 0.8 (0.2, 0.2) | 8, 8.5 (0.8, 0.9) |
| LassoSIR$^t$ | 100, 96.3 (0, 0.7) | 0.8, 0.8 (0.1, 0.1) | 37.3, 52 (1.2, 1.3) | 100, 100 (0, 0) | 1.5, 1.4 (0.1, 0.1) | 24.9, 41.1 (0.9, 1.2) |
| LassoSIR$^m$ | 97, 98.2 (0.9, 0.7) | 0.9, 0.8 (0.1, 0.1) | 45.1, 47.4 (1.7, 1.7) | 98.8, 98.8 (0.6, 0.7) | 1.7, 1.6 (0.2, 0.2) | 36.1, 37 (1.7, 1.9) |
| SEAS-SIR$^t$ | 100, 100 (0, 0) | 0, 0.1 (0, 0) | 2.1, 3.3 (0.1, 0.2) | 100, 100 (0, 0) | 1.5, 0.4 (0.2, 0.1) | 6.1, 9.8 (0.2, 0.4) |
| SEAS-SIR$^m$ | 100, 100 (0, 0) | 0.1, 0.1 (0.1, 0) | 2.8, 3.1 (0.2, 0.2) | 99.8, 99.3 (0.2, 0.7) | 1, 1.2 (0.2, 0.2) | 9.4, 10.2 (0.7, 1.1) |
| SEAS-PFC$^t$ | 100, 100 (0, 0) | 0, 0 (0, 0) | 2, 3 (0.1, 0.2) | 100, 100 (0, 0) | 1.4, 0.8 (0.3, 0.1) | 6.4, 9.1 (0.3, 0.3) |
| SEAS-PFC$^m$ | 100, 100 (0, 0) | 0.1, 0.1 (0.1, 0) | 2.7, 3.4 (0.2, 0.5) | 98.7, 98.2 (1.1, 1.3) | 0.9, 1.1 (0.2, 0.2) | 9.9, 10.8 (1.3, 1.5) |
| | | | M2 | | | |
| mixPFC | 89.8, 89 (1.7, 1.6) | 0.2, 0.3 (0.1, 0.1) | 43.4, 50.8 (1.9, 1.6) | 97.2, 96 (0.9, 1.3) | 0.9, 0.9 (0.1, 0.1) | 26.6, 37 (1.9, 1.9) |
| mixPFC-ISO | 100, 97.5 (0, 1.1) | 0.4, 0.3 (0.1, 0) | 9.6, 17.1 (1.7, 2.9) | 100, 98.7 (0, 0.8) | 0.2, 1.8 (0, 0.3) | 6.4, 13.3 (0.7, 1.5) |
| LassoSIR$^t$ | 100, 96.5 (0, 0.7) | 0.8, 0.8 (0.1, 0.1) | 37.2, 53.4 (1.2, 1.3) | 100, 100 (0, 0) | 1.5, 1.4 (0.1, 0.1) | 24.9, 41.3 (0.9, 1.3) |
| LassoSIR$^m$ | 89.2, 83.7 (1.9, 2.2) | 0.9, 0.9 (0.1, 0.1) | 47.5, 61 (2, 1.6) | 95.7, 92.2 (1.3, 1.7) | 1.7, 1.7 (0.2, 0.1) | 34, 53.5 (2, 2) |
| SEAS-SIR$^t$ | 100, 100 (0, 0) | 0, 0.1 (0, 0) | 2.1, 3.5 (0.1, 0.2) | 100, 100 (0, 0) | 1.5, 0.6 (0.2, 0.1) | 6.1, 10.4 (0.2, 0.4) |
| SEAS-SIR$^m$ | 90.3, 87.7 (1.9, 2.2) | 0, 0 (0, 0) | 20.2, 21.5 (3, 3) | 97.3, 93.9 (0.7, 1.5) | 1.4, 0.4 (0.3, 0.1) | 16.7, 22.2 (2.2, 2.3) |
| SEAS-PFC$^t$ | 100, 100 (0, 0) | 0, 0 (0, 0) | 2, 3.2 (0.1, 0.2) | 100, 100 (0, 0) | 1.4, 0.8 (0.3, 0.1) | 6.4, 9.8 (0.3, 0.4) |
| SEAS-PFC$^m$ | 89.8, 87 (1.9, 2.2) | 0, 0 (0, 0) | 19.9, 21.5 (2.9, 3) | 95.7, 92 (1.2, 1.9) | 1, 0.7 (0.2, 0.1) | 17.8, 23.5 (2.3, 2.5) |
| | | | M3 | | | |
| mixPFC | 96.3, 93.2 (1.2, 1.2) | 0.1, 0.1 (0.1, 0) | 35.9, 40.1 (1.4, 1.8) | 98.8, 96.8 (0.4, 1.2) | 0.1, 0.2 (0.1, 0.1) | 19.4, 27 (1.4, 1.4) |
| mixPFC-ISO | 100, 100 (0, 0) | 0.5, 0.6 (0.1, 0) | 10.9, 14.3 (1.3, 1.6) | 100, 92.7 (0, 2.5) | 0.4, 1.7 (0.1, 0.1) | 17.1, 31.7 (1.3, 2.3) |
| LassoSIR$^t$ | 100, 99.3 (0, 0.3) | 4.2, 4 (0.2, 0.1) | 48.8, 69.1 (0.8, 0.7) | 100, 100 (0, 0) | 7.4, 5.2 (0.2, 0.1) | 42.5, 63.1 (0.8, 0.8) |
| LassoSIR$^m$ | 98.5, 98.3 (0.8, 0.6) | 4.6, 4.2 (0.2, 0.2) | 51.8, 70.5 (1.3, 1) | 98.8, 99.5 (0.6, 0.3) | 7.5, 5.6 (0.2, 0.2) | 45.9, 65 (1.2, 0.9) |
| SEAS-SIR$^t$ | 100, 100 (0, 0) | 0, 0.4 (0, 0.1) | 3, 4.7 (0.2, 0.3) | 100, 100 (0, 0) | 2, 1.1 (0.2, 0.1) | 7.8, 12.9 (0.2, 0.3) |
| SEAS-SIR$^m$ | 99.3, 97.3 (0.5, 1.2) | 0, 0.3 (0, 0.1) | 6.7, 8.8 (1.3, 1.6) | 99.2, 98 (0.5, 0.8) | 1.6, 1.1 (0.2, 0.1) | 12.3, 18.3 (1.5, 1.6) |
| SEAS-PFC$^t$ | 100, 100 (0, 0) | 0, 0 (0, 0) | 2.3, 3.7 (0.1, 0.2) | 100, 100 (0, 0) | 1.5, 0.8 (0.3, 0.1) | 7.5, 11.3 (0.3, 0.3) |
| SEAS-PFC$^m$ | 98.2, 97.2 (1.1, 1.2) | 0.1, 0 (0.1, 0) | 7, 8.3 (1.7, 1.7) | 97.5, 97 (1.2, 1.1) | 1.1, 0.7 (0.2, 0.1) | 12.7, 17.1 (1.7, 1.6) |

*Table A.11.* Simulations results for scenario S2 with $n = 400, p = 1000, K = 2$. The table reports averages and standard errors of TPR, FPR, and subspace distance for each cluster.

| Method | $\boldsymbol{\Delta} = \mathrm{AR}(0.3)$ | | | $\boldsymbol{\Delta} = \mathrm{AR}(0.5)$ | | |
|---|---|---|---|---|---|---|
| | TPR(%) | FPR(%) | $\mathcal{D} \times 100$ | TPR(%) | FPR(%) | $\mathcal{D} \times 100$ |
| | | | M1 | | | |
| mixPFC | 96, 98 (1.4, 0.9) | 0.1, 0 (0, 0) | 35.7, 34 (1.6, 1.3) | 96.2, 95 (0.8, 1.1) | 0.8, 0.7 (0.1, 0.1) | 46.2, 44.4 (1.4, 1.4) |
| LassoSIR$^t$ | 100, 97 (0, 0.6) | 1.2, 1.2 (0.1, 0.1) | 33.2, 50.3 (1.1, 1.4) | 99.7, 91.3 (0.2, 1) | 0.9, 0.9 (0.1, 0.1) | 40.7, 60.9 (1.2, 1.1) |
| LassoSIR$^m$ | 97.3, 98.8 (0.9, 0.5) | 1.2, 1.5 (0.1, 0.2) | 48.3, 44.9 (2, 1.8) | 95.5, 94.8 (0.8, 0.8) | 0.9, 0.7 (0.1, 0.1) | 52.8, 50.2 (1.7, 1.7) |
| SEAS-SIR$^t$ | 100, 100 (0, 0) | 1.2, 0.5 (0.2, 0.1) | 13.5, 15.5 (0.3, 0.4) | 100, 100 (0, 0) | 0.3, 0.1 (0, 0) | 18.8, 19.8 (0.2, 0.4) |
| SEAS-SIR$^m$ | 100, 100 (0, 0) | 0.9, 1.2 (0.2, 0.2) | 15.9, 15.1 (0.6, 0.4) | 100, 100 (0, 0) | 0.2, 0.2 (0, 0) | 19.5, 20.1 (0.3, 0.5) |
| SEAS-PFC$^t$ | 100, 100 (0, 0) | 0.1, 0.8 (0, 0.1) | 13.4, 14.9 (0.3, 0.4) | 100, 100 (0, 0) | 0.1, 0.1 (0, 0) | 18.7, 19.7 (0.2, 0.4) |
| SEAS-PFC$^m$ | 99.3, 100 (0.7, 0) | 0.4, 0.3 (0.1, 0.1) | 16.3, 15.8 (1, 1.1) | 100, 100 (0, 0) | 0.1, 0.1 (0, 0) | 19.4, 20.1 (0.3, 0.4) |
| | | | M2 | | | |
| mixPFC | 98.5, 99.8 (0.8, 0.2) | 0, 1.4 (0, 0.2) | 32.4, 32 (1.3, 1) | 96.3, 97.3 (1.1, 0.9) | 0.5, 3 (0, 0.2) | 41, 47.5 (1.3, 1.2) |
| LassoSIR$^t$ | 100, 100 (0, 0) | 1.2, 2.3 (0.1, 0.1) | 33.2, 42.2 (1.1, 1.1) | 99.7, 88.8 (0.2, 1.1) | 0.9, 3.4 (0.1, 0.1) | 40.8, 61.4 (1.2, 1.2) |
| LassoSIR$^m$ | 97.7, 97.8 (1, 1) | 1, 2 (0.1, 0.1) | 35.2, 48.7 (1.3, 1.4) | 95.8, 85.7 (1.3, 1.5) | 1.1, 3 (0.1, 0.1) | 45.9, 64.5 (1.6, 1.2) |
| SEAS-SIR$^t$ | 100, 100 (0, 0) | 1.2, 1.4 (0.2, 0.1) | 13.5, 38.1 (0.3, 0.5) | 100, 74.7 (0, 1.4) | 0.3, 1.2 (0, 0.1) | 19, 69.9 (0.2, 0.5) |
| SEAS-SIR$^m$ | 99.3, 97.8 (0.3, 0.8) | 1.7, 1.1 (0.3, 0.1) | 16.2, 41.3 (0.9, 1) | 99.7, 74.2 (0.2, 1.5) | 0.2, 1.2 (0, 0.1) | 20.6, 71.3 (0.6, 0.6) |
| SEAS-PFC$^t$ | 100, 100 (0, 0) | 0.1, 0.8 (0, 0.1) | 13.4, 33.4 (0.3, 0.5) | 100, 85.2 (0, 1.4) | 0.1, 1.2 (0, 0.1) | 18.9, 64.9 (0.2, 0.7) |
| SEAS-PFC$^m$ | 99.3, 97.8 (0.3, 1) | 0.1, 0.8 (0, 0.1) | 16, 36 (1, 1) | 99.3, 83.3 (0.3, 1.7) | 0.1, 1.2 (0, 0.1) | 20.8, 66 (0.6, 0.7) |
| | | | M3 | | | |
| mixPFC | 98.8, 98.5 (0.5, 0.8) | 0.3, 0.1 (0.2, 0.1) | 25.1, 36.6 (1.5, 1.1) | 96.2, 95.7 (1.4, 1.1) | 0.1, 0.8 (0, 0.2) | 37, 54.3 (1.5, 1) |
| LassoSIR$^t$ | 100, 99.3 (0, 0.3) | 12.1, 5.6 (0.2, 0.1) | 62.8, 75.9 (0.5, 0.4) | 99.8, 95.2 (0.2, 0.8) | 10.9, 4.9 (0.3, 0.1) | 74.1, 80.2 (0.3, 0.4) |
| LassoSIR$^m$ | 98.2, 98.3 (0.9, 0.6) | 12, 5.8 (0.2, 0.2) | 64.8, 76.5 (0.8, 0.5) | 97.2, 93.8 (1.1, 1) | 11, 5.1 (0.3, 0.2) | 75.3, 80.6 (0.6, 0.5) |
| SEAS-SIR$^t$ | 100, 99.8 (0, 0.2) | 0.6, 0.8 (0.2, 0.1) | 18.3, 33.1 (0.4, 0.4) | 100, 86 (0, 1.3) | 0.4, 0.1 (0, 0) | 32.7, 54.5 (0.5, 0.9) |
| SEAS-SIR$^m$ | 97.3, 96.7 (1.2, 1.3) | 1.1, 1 (0.3, 0.1) | 22.6, 36.1 (1.4, 1.2) | 97.5, 84.3 (1, 1.5) | 0.3, 0.1 (0, 0) | 36.5, 56.1 (1.2, 1) |
| SEAS-PFC$^t$ | 100, 99.2 (0, 0.5) | 0.1, 0.7 (0, 0.1) | 17.6, 32.5 (0.3, 0.6) | 100, 83.8 (0, 1.4) | 0.1, 0.1 (0, 0) | 32, 54.6 (0.5, 0.9) |
| SEAS-PFC$^m$ | 96.8, 95.3 (1.4, 1.6) | 0.1, 0.8 (0, 0.1) | 21.7, 36.3 (1.5, 1.3) | 96.5, 81.8 (1.4, 1.6) | 0.1, 0.1 (0, 0) | 35.8, 56.3 (1.3, 1) |

*Table A.12.* Simulations results for scenario S3 with $n = 400, p = 1000, K = 2$. The table reports averages and standard errors of TPR, FPR, and subspace distance for each cluster.

| Method | $\boldsymbol{\Delta} = 0.1\mathbf{I}$ | | | $\boldsymbol{\Delta} = \mathbf{I}$ | | |
|---|---|---|---|---|---|---|
| | TPR(%) | FPR(%) | $\mathcal{D} \times 100$ | TPR(%) | FPR(%) | $\mathcal{D} \times 100$ |
| | | | M1 | | | |
| mixPFC | 96.8, 95.8 (1.2, 1.4) | 0, 0 (0, 0) | 38, 37.7 (1.3, 1.4) | 95.7, 95.7 (1.6, 1.7) | 1.4, 1.3 (0.3, 0.3) | 37.8, 36 (1.9, 2) |
| mixPFC-ISO | 100, 100 (0, 0) | 0.6, 0.6 (0, 0) | 2.6, 2.7 (0.1, 0.1) | 99.7, 99.7 (0.3, 0.3) | 1.4, 1.5 (0.1, 0.2) | 13.2, 13.5 (1, 1.1) |
| LassoSIR$^t$ | 100, 99.5 (0, 0.3) | 1, 1.2 (0.1, 0.1) | 38.1, 43.8 (1.1, 1.3) | 100, 100 (0, 0) | 2.5, 2.6 (0.2, 0.2) | 36.5, 40.8 (1.2, 1.2) |
| LassoSIR$^m$ | 99.7, 99.7 (0.2, 0.2) | 1.2, 0.9 (0.1, 0.1) | 43.1, 39.5 (1.2, 1.2) | 96.7, 96.2 (1.6, 1.7) | 3.1, 3.3 (0.3, 0.3) | 45.8, 45.3 (1.8, 1.9) |
| SEAS-SIR$^t$ | 100, 100 (0, 0) | 0, 0 (0, 0) | 3.6, 3.7 (0.2, 0.2) | 100, 100 (0, 0) | 0.9, 1.1 (0.1, 0.2) | 11.1, 13.2 (0.4, 0.4) |
| SEAS-SIR$^m$ | 100, 100 (0, 0) | 0, 0 (0, 0) | 3.8, 3.9 (0.2, 0.2) | 98.1, 98.3 (1.3, 1.2) | 1.6, 1.6 (0.2, 0.2) | 17.1, 16 (1.3, 1.4) |
| SEAS-PFC$^t$ | 100, 100 (0, 0) | 0.2, 0.3 (0, 0.1) | 3.4, 3.5 (0.2, 0.2) | 100, 100 (0, 0) | 0.9, 1.1 (0.2, 0.2) | 10.8, 12.9 (0.4, 0.4) |
| SEAS-PFC$^m$ | 100, 100 (0, 0) | 0.3, 0.3 (0.1, 0) | 3.7, 3.5 (0.2, 0.2) | 97, 96.8 (1.7, 1.6) | 1.6, 1.7 (0.3, 0.3) | 19.5, 18.1 (1.8, 1.9) |
| | | | M2 | | | |
| mixPFC | 91.7, 90.5 (1.6, 1.7) | 0.8, 0.6 (0.3, 0.4) | 42.7, 45.9 (1.8, 1.9) | 95.3, 95 (1.6, 1.4) | 1.1, 1.7 (0.2, 0.2) | 36, 39.8 (1.7, 1.7) |
| mixPFC-ISO | 100, 99.5 (0, 0.5) | 4.4, 0.8 (1.4, 0) | 8.5, 7 (2.1, 1.5) | 98.3, 97.7 (0.8, 0.9) | 1.7, 2.5 (0.2, 0.4) | 16.9, 19.3 (1.5, 2) |
| LassoSIR$^t$ | 100, 99.7 (0, 0.2) | 1, 1.2 (0.1, 0.1) | 37.2, 43 (1.1, 1.3) | 100, 100 (0, 0) | 2.5, 2.6 (0.2, 0.2) | 36.5, 40.1 (1.2, 1.4) |
| LassoSIR$^m$ | 92.8, 92.3 (1.7, 1.6) | 1.4, 1.7 (0.2, 0.2) | 46.1, 49.8 (2, 1.9) | 95.8, 95.7 (1.3, 1.4) | 2.8, 2.8 (0.2, 0.2) | 47.1, 48.9 (1.5, 1.5) |
| SEAS-SIR$^t$ | 100, 100 (0, 0) | 0, 0 (0, 0) | 3.6, 3.8 (0.2, 0.2) | 100, 100 (0, 0) | 0.9, 1.1 (0.1, 0.2) | 11.1, 12.9 (0.4, 0.4) |
| SEAS-SIR$^m$ | 94.3, 95.8 (1.5, 1.3) | 0.1, 0 (0, 0) | 14.5, 14 (2.5, 2.4) | 97, 96.8 (1.2, 1.2) | 1, 1.1 (0.1, 0.2) | 20.7, 22.2 (1.7, 1.6) |
| SEAS-PFC$^t$ | 100, 100 (0, 0) | 0.2, 0.3 (0, 0.1) | 3.3, 3.6 (0.2, 0.2) | 100, 100 (0, 0) | 0.9, 0.9 (0.2, 0.2) | 10.8, 12.3 (0.4, 0.5) |
| SEAS-PFC$^m$ | 92.3, 94.7 (1.9, 1.5) | 0.2, 0.3 (0.1, 0.1) | 14.8, 14 (2.6, 2.4) | 96.3, 95.7 (1.3, 1.5) | 0.9, 1.2 (0.2, 0.2) | 21.1, 22.7 (1.7, 1.8) |
| | | | M3 | | | |
| mixPFC | 40.7, 42.5 (1.4, 1.1) | 2.6, 4.5 (0.7, 0.9) | 80.3, 81 (0.8, 0.9) | 49.7, 49.2 (1.2, 1.1) | 2, 2.2 (0.2, 0.2) | 73.7, 74 (0.4, 0.5) |
| mixPFC-ISO | 82.8, 84.3 (2.1, 2) | 1.1, 0.9 (0.3, 0.1) | 48.8, 48.7 (2.4, 2.2) | 54, 53.6 (1.4, 1.5) | 4.6, 4.1 (0.2, 0.1) | 73.2, 73 (0.6, 0.6) |
| LassoSIR$^t$ | 98.7, 99.8 (0.6, 0.2) | 8.3, 8.3 (0.2, 0.3) | 67.6, 67.3 (0.9, 1) | 69, 67 (1.5, 1.5) | 10.1, 10.5 (0.2, 0.1) | 73.5, 73.9 (0.3, 0.3) |
| LassoSIR$^m$ | 69.5, 70 (3.2, 3.3) | 9.9, 9.3 (0.3, 0.2) | 81.6, 80.1 (1.2, 1.3) | 59, 60.8 (1.3, 1.8) | 10.5, 10.9 (0.2, 0.2) | 76.3, 76.4 (0.7, 0.7) |
| SEAS-SIR$^t$ | 94.7, 95.8 (1.4, 1.2) | 0, 0 (0, 0) | 42.9, 40.2 (1.5, 1.5) | 56.2, 56.2 (1.2, 1.1) | 1.7, 1.8 (0.3, 0.3) | 70.7, 70.6 (0.1, 0.1) |
| SEAS-SIR$^m$ | 68.3, 67.7 (2.9, 3) | 0.5, 0.3 (0.2, 0.1) | 65.4, 65.2 (1.7, 1.7) | 53.1, 53.3 (1.2, 1.2) | 2.5, 1.5 (0.3, 0.2) | 71.9, 71.6 (0.5, 0.4) |
| SEAS-PFC$^t$ | 50, 50 (0, 0) | 0, 0 (0, 0) | 70.7, 70.7 (0, 0) | 51, 52 (0.5, 0.5) | 1.4, 2.5 (0.2, 0.3) | 70.8, 70.8 (0, 0) |
| SEAS-PFC$^m$ | 43.7, 44.7 (0.9, 0.9) | 0.2, 0.5 (0.1, 0.2) | 75.3, 75.7 (0.6, 0.7) | 48.7, 49.2 (0.8, 0.9) | 1.8, 1.7 (0.3, 0.3) | 72.3, 72.4 (0.5, 0.5) |

*Table A.13.* Simulations results for scenario S3 with $n = 400, p = 1000, K = 2$. The table reports averages and standard errors of TPR, FPR, and subspace distance for each cluster.

| Method | $\boldsymbol{\Delta} = \mathrm{AR}(0.3)$ | | | $\boldsymbol{\Delta} = \mathrm{AR}(0.5)$ | | |
|---|---|---|---|---|---|---|
| | TPR(%) | FPR(%) | $\mathcal{D} \times 100$ | TPR(%) | FPR(%) | $\mathcal{D} \times 100$ |
| | | | M1 | | | |
| mixPFC | 96.7, 96.5 (1.1, 1.1) | 0.5, 0.6 (0.2, 0.2) | 35.1, 36.9 (1.6, 1.7) | 92.2, 91.3 (1.5, 1.9) | 0.4, 0.3 (0.3, 0.2) | 49.4, 49.1 (1.5, 1.5) |
| LassoSIR$^t$ | 99.7, 99.7 (0.2, 0.2) | 1.7, 2 (0.1, 0.2) | 41.6, 45.5 (1.2, 1.4) | 99, 97.3 (0.4, 0.7) | 1.2, 1.4 (0.1, 0.1) | 47.1, 53.7 (1.3, 1.2) |
| LassoSIR$^m$ | 99, 98.5 (0.6, 0.7) | 1.9, 1.9 (0.2, 0.2) | 43.9, 45.5 (1.5, 1.5) | 94.8, 94.7 (1.2, 1.4) | 1.4, 1.4 (0.2, 0.2) | 53.3, 52.3 (1.4, 1.4) |
| SEAS-SIR$^t$ | 100, 100 (0, 0) | 1, 1 (0.2, 0.2) | 15, 15 (0.4, 0.4) | 100, 100 (0, 0) | 0.2, 0.3 (0, 0) | 20, 20.9 (0.4, 0.6) |
| SEAS-SIR$^m$ | 99.8, 100 (0.2, 0) | 1.2, 1.2 (0.2, 0.2) | 16.3, 15.7 (0.6, 0.5) | 99, 99.2 (0.6, 0.8) | 0.3, 0.4 (0.1, 0.1) | 22.3, 23 (1.1, 1.1) |
| SEAS-PFC$^t$ | 100, 100 (0, 0) | 1.5, 1.9 (0.2, 0.3) | 14.7, 15.3 (0.4, 0.4) | 100, 99.8 (0, 0.2) | 0.2, 0.2 (0, 0) | 20, 20.9 (0.4, 0.6) |
| SEAS-PFC$^m$ | 100, 100 (0, 0) | 1.9, 2 (0.3, 0.3) | 16.2, 16.8 (0.7, 0.8) | 98.7, 98.7 (0.7, 1) | 0.4, 0.5 (0.2, 0.2) | 22.7, 23.7 (1.2, 1.3) |
| | | | M2 | | | |
| mixPFC | 94.2, 99 (1.5, 0.5) | 0.1, 2.3 (0.1, 0.3) | 42.3, 36.6 (1.5, 1.3) | 72.7, 94 (1.6, 1.5) | 0.1, 4.5 (0.1, 0.6) | 64.2, 52 (1.1, 1.5) |
| LassoSIR$^t$ | 99.7, 100 (0.2, 0) | 1.7, 4.9 (0.1, 0.2) | 41.6, 47.9 (1.2, 1.3) | 99.2, 86 (0.4, 1.2) | 1.3, 6.6 (0.1, 0.2) | 47.5, 70.6 (1.3, 1.2) |
| LassoSIR$^m$ | 94.8, 98 (1.2, 0.7) | 1.6, 4.9 (0.2, 0.2) | 51.3, 51.9 (1.4, 1.2) | 88.5, 85.7 (1.2, 1.4) | 1.2, 6.7 (0.1, 0.2) | 60.4, 71.5 (1.3, 1.3) |
| SEAS-SIR$^t$ | 100, 99.5 (0, 0.3) | 1, 2.3 (0.2, 0.2) | 15, 41.1 (0.4, 0.8) | 100, 66.8 (0, 1.5) | 0.2, 1.4 (0, 0.2) | 20, 73.4 (0.4, 0.5) |
| SEAS-SIR$^m$ | 98.7, 96.7 (0.7, 1.1) | 1.5, 2.3 (0.2, 0.2) | 22.5, 46.1 (1.4, 1) | 97.3, 71.3 (0.8, 1.5) | 0.1, 2 (0, 0.2) | 31.1, 72.3 (1.4, 0.5) |
| SEAS-PFC$^t$ | 100, 99.7 (0, 0.3) | 1.5, 0.9 (0.2, 0.1) | 14.7, 36 (0.4, 0.7) | 100, 73.8 (0, 1.7) | 0.2, 1 (0, 0.1) | 20, 69.7 (0.5, 0.5) |
| SEAS-PFC$^m$ | 98.8, 97 (0.6, 1.1) | 0.6, 1.4 (0.2, 0.2) | 22.1, 42.1 (1.4, 1.1) | 96.7, 71.8 (1.1, 1.5) | 0.1, 1.2 (0, 0.2) | 31.3, 70.6 (1.5, 0.5) |
| | | | M3 | | | |
| mixPFC | 50.5, 50.7 (1.2, 1.1) | 0.5, 1.3 (0.1, 0.2) | 72.6, 74 (0.4, 0.6) | 53.5, 54.3 (1.3, 1.3) | 0.3, 1 (0.2, 0.2) | 72.3, 73.6 (0.5, 0.5) |
| LassoSIR$^t$ | 78.2, 73.5 (1.6, 1.5) | 9.6, 9.5 (0.2, 0.2) | 74.2, 73.9 (0.3, 0.3) | 78.5, 77 (1.4, 1.4) | 8.7, 9 (0.2, 0.2) | 75.4, 74.1 (0.3, 0.2) |
| LassoSIR$^m$ | 71.8, 70 (1.9, 1.7) | 10, 10.2 (0.2, 0.2) | 76.5, 76.7 (0.7, 0.7) | 75.8, 75 (1.9, 1.4) | 9.2, 9.4 (0.2, 0.2) | 77.7, 76.2 (0.7, 0.6) |
| SEAS-SIR$^t$ | 91.5, 80.2 (1.3, 1.2) | 0.6, 1.3 (0.1, 0.2) | 59.5, 65.7 (1.2, 0.6) | 93, 84.2 (1, 0.8) | 0.2, 0.5 (0.1, 0.1) | 62.9, 65.7 (1.1, 0.7) |
| SEAS-SIR$^m$ | 82.6, 74.3 (1.9, 1.6) | 0.5, 1 (0.1, 0.2) | 67.9, 69.2 (0.9, 0.6) | 87.4, 81.3 (1.6, 1.4) | 0.5, 0.4 (0.1, 0.1) | 69.5, 68.4 (0.8, 0.7) |
| SEAS-PFC$^t$ | 89.7, 76.3 (1.5, 1.2) | 0.4, 0.7 (0.1, 0.2) | 53.9, 61.7 (1.4, 0.8) | 93.3, 83.2 (1, 0.6) | 0.1, 0.1 (0, 0) | 55.1, 63.2 (1.2, 0.7) |
| SEAS-PFC$^m$ | 80.3, 70.5 (2.1, 1.8) | 0.3, 0.8 (0.1, 0.2) | 64.4, 68.1 (1.2, 0.9) | 85.7, 80 (2, 1.3) | 0.1, 0.4 (0, 0.1) | 64.9, 67.5 (1.2, 0.9) |

## A.2. Real Data Analysis

For the CCLE data, we evaluate the performance of mixPFC across $K = 2, 3, \ldots, 10$. Table A.14 summarizes PMSE and the number of selected variables $\widehat{s}$ (total number of unique variables selected across all clusters) for different $K$ under mixPFC.

For mixPFC, the prediction error decreases as the number of clusters $K$ increases. However, the reduction becomes less pronounced beyond $K = 3$ for Nutlin-3 and $K = 5$ and AZD6244. For both responses, mixPFC achieves a significant reduction in prediction error compared to homogeneous methods (Table 2), suggesting heterogeneity in the data. The number of selected variables $\widehat{s}$ initially increases and then stabilizes as the number of clusters $K$ grows, indicating that different clusters select different variables. Notably, mixPFC does not select more variables than Lasso and the three homogeneous SDR methods, even for large $K$. These findings suggest that over-specification of $K$ does not significantly impact the prediction performance of mixPFC. However, over-specification of $K$ increases computational cost and introduces variability in parameter estimation. Using the tuning method described in Section B of the appendix, the average selected $K$ is 3 for both Nutlin-3 and AZD6244.

Summary plots of the response versus the reduced predictors projected onto each subspace are shown in Figure A.5 for AZD6244. These plots reveal approximately linear relationships between the response and projected predictors within each cluster for both drugs. Notably, the points in cluster 1 form a vertical band when projected onto subspaces $\widehat{\mathcal{S}}_w, w = 2, 3, 4, 5$. This pattern coincides with the example in Figure 2 (b) and suggests the first subspace is orthogonal to the remaining subspaces.

*Table A.14.* The averages of the prediction errors, the sparsity level $\widehat{s}$, and the corresponding standard errors based on 100 replicates.

|  | $K=2$ | $K=3$ | $K=4$ | $K=5$ | $K=6$ | $K=7$ | $K=8$ | $K=9$ | $K=10$ |
|---|---|---|---|---|---|---|---|---|---|
|  |  |  |  | Nutlin-3 |  |  |  |  |  |
| PMSE $\times 100$ | 11.9 (0.4) | 8.9 (0.4) | 8.3 (0.3) | 8.4 (0.3) | 7.7 (0.3) | 7.84 (0.4) | 7.36 (0.3) | 6.9 (0.3) | 6.9 (0.3) |
| $\widehat{s}$ | 16.5 (1.2) | 31.5 (2.5) | 45.2 (2.7) | 52.3 (2.7) | 53.9 (2.7) | 53.7 (2.6) | 52.6 (2.2) | 58.2 (2.4) | 59.5 (2.1) |
|  |  |  |  | AZD6244 |  |  |  |  |  |
| PMSE $\times 100$ | 66.4 (1.5) | 58.7 (2.0) | 62.2 (2.3) | 45.6 (1.7) | 41.6 (1.2) | 36.7 (1) | 33.5 (1.0) | 32.0 (1.0) | 34.7 (1.2) |
| $\widehat{s}$ | 21.6 (0.6) | 53.4 (2.5) | 62.2 (2.3) | 77.8 (3.2) | 77.4 (3.1) | 74.1 (2.8) | 65.1 (2.7) | 67.5 (2.4) | 70.6 (2.3) |

# B. Implementation Details

### B.1. Initialization and Tuning $K$

**Initialization**. To implement the proposed mixPFC algorithm, it is critical to obtain reliable initial values for $\gamma_{iw}$. Classical distance-based clustering algorithms, like $K$-means and hierarchical clustering, often produce low-quality initial values. Similarly, subspace clustering methods also fail for the high-dimensional mixture of PFC. In the Gaussian mixture model, it is recommended to initialize EM with short runs of EM, where the algorithm is stopped early rather than run to convergence (Biernacki et al., 2003). Given that existing methods are not designed for the model (3), we propose a similar initialization procedure that runs the mixPFC algorithm on transformed data with an early stopping criterion to generate initial estimates of $\gamma_{iw}$. In high-dimensional settings, we transform the original data into a lower-dimensional space by applying distance correlation (dcor) screening (Li et al., 2012) followed by principal component analysis (PCA). Specifically, we first select $u \times \lfloor n \log n \rfloor$ variables using dcor and then project the data onto the first $v$ principal components of the selected variables. Based on simulation results in Table A.15, we recommend setting $u = 2$ and $v = 10$.

**Selection of number of clusters $K$.** The gap statistic (Tibshirani et al., 2001) is a popular technique for selecting the optimal number of clusters for many clustering algorithms. Here, we adapt this approach to enhance its suitability for the mixture model 3. Let $\mathbf{Q}_w$ represent the orthogonal complement of $\boldsymbol{\beta}_w$, $G_w$ denote the indices of observations in cluster $w$ and $n_w = |G_w|$. Then define

$$D_w = \sum_{i,i' \in G_w} \|\mathbf{Q}_w^T \mathbf{X}_i - \mathbf{Q}_w^T \mathbf{X}_{i'}\|_2, \quad V_K = \sum_{w=1}^{K} \frac{1}{2n_w} D_w.$$

The gap statistic is

$$\mathrm{Gap}_n(K) = \mathrm{E}_n^*[\log(V_K)] - \log(V_K),$$

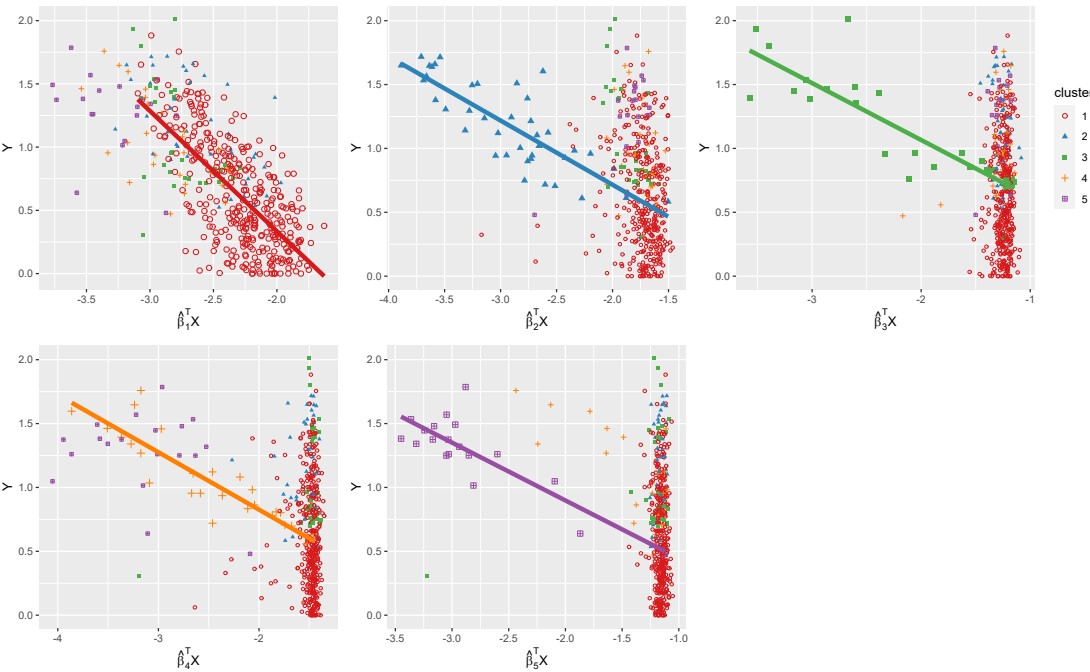

*Figure A.5.* The scatter-plot of the response $Y$ versus $\widehat{\beta}_w^T \mathbf{X}$ for the drug AZD6244. The solid line is fitted using samples in the given cluster.

*Table A.15.* Error rates averaged over 20 datasets generated under model scenario S1 using M1-M3. First, we use dcor to select $u \times \lfloor N \log(N) \rfloor$ variables and then apply PCA to project the data onto $v$ dimension space spanned by principal components.

| | M1 | | | M2 | | | M3 | | |
|---|---|---|---|---|---|---|---|---|---|
| | $v = 5$ | $v = 10$ | $v = 15$ | $v = 5$ | $v = 10$ | $v = 15$ | $v = 5$ | $v = 10$ | $v = 15$ |
| | $\boldsymbol{\Delta} = 0.1\mathbf{I}$ mixPFC | | | | | | | | |
| $u = 1$ | 5.7 | 2.2 | 6.3 | 25.2 | 27.1 | 30.0 | 8.4 | 3.0 | 7.3 |
| $u = 2$ | 13.6 | 4.0 | 4.5 | 29.2 | 31.6 | 35.0 | 11.0 | 11.9 | 13.5 |
| $u = 3$ | 15.8 | 13.2 | 4.5 | 30.1 | 38.6 | 38.1 | 7.3 | 7.5 | 10.5 |
| | $\boldsymbol{\Delta} = 0.1\mathbf{I}$ mixPFC-ISO | | | | | | | | |
| $u = 1$ | 23.8 | 25.0 | 24.9 | 6.7 | 9.0 | 11.3 | 20.6 | 24.3 | 19.2 |
| $u = 2$ | 24.7 | 22.1 | 34.5 | 11.1 | 11.3 | 17.3 | 18.7 | 19.8 | 24.5 |
| $u = 3$ | 30.5 | 28.2 | 18.7 | 9.1 | 21.9 | 9.2 | 27.3 | 22.3 | 17.2 |
| | $\boldsymbol{\Delta} = \mathbf{I}$ mixPFC | | | | | | | | |
| $u = 1$ | 14.3 | 10.7 | 10.8 | 27.3 | 37.4 | 43.5 | 9.9 | 7.3 | 9.0 |
| $u = 2$ | 20.1 | 14.4 | 11.9 | 31.6 | 38.5 | 41.3 | 13.9 | 9.9 | 8.7 |
| $u = 3$ | 16.1 | 16.5 | 18.7 | 31.8 | 42.7 | 45.9 | 11.9 | 13.8 | 20.7 |
| | $\boldsymbol{\Delta} = \mathbf{I}$ mixPFC-ISO | | | | | | | | |
| $u = 1$ | 17.3 | 13.0 | 11.4 | 9.4 | 12.0 | 15.3 | 13.9 | 19.7 | 17.6 |
| $u = 2$ | 18.8 | 15.0 | 11.5 | 15.3 | 10.2 | 12.6 | 24.8 | 22.1 | 13.3 |
| $u = 3$ | 15.0 | 15.4 | 17.6 | 12.2 | 14.4 | 13.2 | 14.7 | 19.5 | 8.3 |
| | $\boldsymbol{\Delta} = \mathrm{AR}(0.3)$ mixPFC | | | | | | | | |
| $u = 1$ | 21.0 | 12.3 | 14.4 | 27.0 | 31.2 | 38.8 | 12.9 | 12.3 | 11.4 |
| $u = 2$ | 13.9 | 15.5 | 12.5 | 23.2 | 34.3 | 41.4 | 20.5 | 19.1 | 14.9 |
| $u = 3$ | 22.5 | 18.9 | 14.3 | 29.4 | 40.9 | 36.5 | 26.3 | 28.4 | 23.5 |

where $\mathrm{E}_n^*$ denotes the expectation under a sample of size $n$ from a reference distribution. We estimate $\mathrm{E}_n^*[\log(V_K)]$ by an average of $B$ copies $\log(V_K^*)$, which is computed from samples drawn from uniform reference distribution. However, in high dimensions, the gap statistic becomes computationally expensive as it requires calculating $V_K$ for $B$ data samples. In practice, we find that bypassing the expectation calculation and directly using $V_K$ yields effective results. Let $K^*$ denote the true number of clusters. To illustrate, we use models M1 ($K^* = 2$) and M4 ($K^* = 3, 5$), with an identity covariance matrix as defined in the simulation section, to generate high-dimensional data. Notably, the two subspaces are identical in model M1. Figure A.6 presents a representative plot of the within-cluster dispersion $V_K$, calculated using the mixPFC Algorithm 1, as a function of the number of clusters. The error measure $V_K$ decreases monotonically as the number of clusters $K$ increases, but begins to rise once $K$ exceeds the true number of clusters $K^*$. Based on this finding, we propose to select the smallest $K$ such that

$$\max(V_K) - V_K \geq \rho(\max(V_K) - \min(V_K)),$$

where $\rho \in (0, 1)$ is specified to avoid overestimating $K$. Table A.16 presents the selected $K$ for models M1 and M4 across four different covariance matrices. The proposed selection method works well in most cases, with the exception of $K^* = 5$ under $\mathrm{AR}(0.3)$ and $\mathrm{AR}(0.5)$. The reason is inaccuracies in subspace estimation affect $V_K$, which depends on the orthogonal complement of central subspaces.

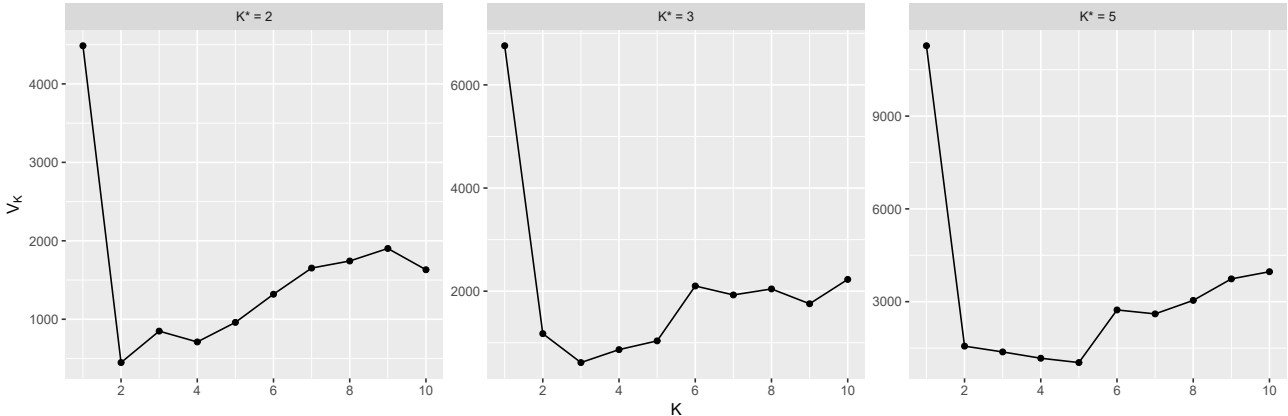

*Figure A.6.* Within-cluster dispersion $V_K$ for $K = 1, 2, \ldots 10$.

*Table A.16.* The average selected number of clusters over 100 repetitions, using statistic $V_K$. The standard errors are in parentheses and $K_{\max} = 10$.

| $K^*$ | $0.1\mathbf{I}$ | $\mathbf{I}$ | $\mathrm{AR}(0.3)$ | $\mathrm{AR}(0.5)$ |
|---|---|---|---|---|
| $K^* = 2$ | 2 (0) | 2 (0) | 2 (0) | 2.1(0) |
| $K^* = 3$ | 3.1 (0) | 3 (0) | 2.7 (0) | 2.9 (0.1) |
| $K^* = 5$ | 5 (0) | 4.8 (0) | 3.7 (0.1) | 3.1 (0.2) |

### B.2. Implementation of mixPFC

For the mixPFC algorithm 1, the tuning parameter $\lambda^{(t)}$ could either be fixed or vary across iterations. In theoretical analysis, to show statistical convergence results, we set $\lambda^{(t+1)} = \kappa\lambda^{(t)} + C_\lambda\sqrt{q^3(\log n)^2 \log p/n}$, where $0 < \kappa < 1/2$ and $C_\lambda$ is a positive constant. Note $\lambda^{(t)}$ is at the order of $\sqrt{q^3(\log n)^2 \log p/n}$ when $t$ is large. Therefore, in practice, we fix $\lambda^{(t)} = \lambda$ and tune $\lambda$ with cross-validated distance correlation (Székely et al., 2007). For fixed $\lambda^{(t)} = \lambda$, the penalized EM algorithm maximizes

$$\ell(\boldsymbol{\theta}) - \lambda \sum_{w=1}^{K} \|\mathbf{B}_w\|_{2,1}, \tag{7}$$

where $\ell(\boldsymbol{\theta})$ is the log-likelihood of $\mathbf{X}|Y$. The following lemma shows the convergence of Algorithm 1.

**Lemma B.1.** *If we set $\lambda^{(t)} = \lambda$ for all $t$, the objective function from (7) evaluated at $\widehat{\boldsymbol{\theta}}^{(t+1)}$ is guaranteed to be no less than the objective function from (7) evaluated at $\widehat{\boldsymbol{\theta}}^{(t)}$. That is, the sequence of iterates $\{\widehat{\boldsymbol{\theta}}^{(t)}\}_{t=1}^{\infty}$ generated by Algorithm 1 monotonically increase the value of the objective function from (7).*

*Proof.* Recall that the conditional log-likelihood is

$$\ell(\boldsymbol{\theta}; \mathbf{X}, Y) = \sum_{i=1}^{n} \log\Big[ \sum_{w=1}^{K} \pi_w N(\mathbf{X}_i \mid \boldsymbol{\mu}_w + \boldsymbol{\Gamma}_w \boldsymbol{\eta}_w \mathbf{f}(Y_i), \boldsymbol{\Delta}) \Big].$$

The penalized EM algorithm can be viewed as two alternating maximization steps. Consider the following function:

$$F(q, \boldsymbol{\theta}) := \mathrm{E}_q[\log(L(\boldsymbol{\theta}; \mathbf{X}, Y, W))] + H(q) - \lambda \sum_{w=1}^{K} \|\mathbf{B}_w\|_{2,1},$$

where $q = (q_1, \ldots, q_K)$ is an arbitrary probability density over the unobserved variable $W$ and $H(q) = -\sum_i q_i \log(q_i)$ is the entropy of distribution $q$. It is easy to show

$$F(q, \boldsymbol{\theta}) = \ell(\boldsymbol{\theta}; \mathbf{X}, Y) - D_{\mathrm{KL}}(q \| p_{W|\mathbf{X}, Y}(\cdot | \mathbf{X}, Y, \boldsymbol{\theta})) - \lambda \sum_{w=1}^{K} \|\mathbf{B}_w\|_{2,1},$$

where $D_{\mathrm{KL}}$ is the Kullback-Leibler (KL) divergence. The KL divergence between two distributions is non-negative and is zero when the two distributions are identical. Therefore, the E-step is to choose $q$ to maximize $F(q, \boldsymbol{\theta})$:

$$\widehat{q}^{(t+1)} = \underset{q}{\mathrm{argmax}}\, F(q, \widehat{\boldsymbol{\theta}}^{(t)}) = p_{W|\mathbf{X}, Y}(\cdot | \mathbf{X}, Y, \widehat{\boldsymbol{\theta}}^{(t)}),$$

which is given by the updating function of $\gamma_{iw}$. In M-step, we maximize $F$ over $\boldsymbol{\theta}$:

$$\begin{aligned}
\widehat{\boldsymbol{\theta}}^{(t+1)} &= \underset{\boldsymbol{\theta}}{\mathrm{argmax}}\, F(\widehat{q}^{(t)}, \boldsymbol{\theta}) \\
&= \underset{\boldsymbol{\theta}}{\mathrm{argmax}}\, \mathrm{E}_{\widehat{q}^{(t)}}[\log(L(\boldsymbol{\theta}; \mathbf{X}, Y, W))] - \lambda \sum_{w=1}^{K} \|\mathbf{B}_w\|_{2,1} \\
&= \underset{\boldsymbol{\theta}}{\mathrm{argmax}}\, Q(\boldsymbol{\theta}|\widehat{\boldsymbol{\theta}}^{(t)}) - \lambda \sum_{w=1}^{K} \|\mathbf{B}_w\|_{2,1},
\end{aligned}$$

which is the same as the penalized M-step. Through this coordinate ascent strategy, the update $\widehat{\boldsymbol{\theta}}^{(t+1)}$ makes the value of the penalized log-likelihood function converge monotonically. Combining this result with the convergence (Lemma H.1, Zeng et al. (2024)) of the group-wise coordinate descent algorithm 2, we know Algorithm 1 converges when $\lambda^{(t)} = \lambda$.

$\square$

Recall that in Algorithm 1 in the main paper. We update $\widehat{\mathbf{B}}_w^{t+1}$ by solving:

$$\underset{\mathbf{B}_w \in \mathbb{R}^{p \times q}}{\mathrm{argmin}}\, \frac{1}{2} \mathrm{tr}(\mathbf{B}_w^T \widehat{\boldsymbol{\Sigma}}_w^{(t)} \mathbf{B}_w) - \mathrm{tr}\{(\widehat{\mathbf{U}}_w^{(t)})^T \mathbf{B}_w\} + \lambda \|\mathbf{B}_w\|_{2,1}.$$

This optimization is solved by the group-wise coordinate descent algorithm proposed in Mai et al. (2019). For ease of presentation, we remove the subscript $w$ in $\mathbf{B}_w$, $\widehat{\boldsymbol{\Sigma}}_w^{(t)}$ and $\widehat{\mathbf{U}}_w^{(t)}$. The algorithm is presented in Algorithm 2.

**Tuning $\lambda$.** For two random vectors $\mathbf{T} \in \mathbb{R}^p$ and $\mathbf{Z} \in \mathbb{R}^q$, the distance correlation $\mathrm{dcor}(\mathbf{T}, \mathbf{Z}) \in [0, 1]$ measures the dependence between the two random vectors (Székely et al., 2007). In particular, $\mathrm{dcor}(\mathbf{T}, \mathbf{Z}) = 0$ if and only if $\mathbf{T}$ and $\mathbf{Z}$ are independent. Given observed samples $\widetilde{\mathbf{T}} \in \mathbb{R}^{n \times p}$ and $\widetilde{\mathbf{Z}} \in \mathbb{R}^{n \times q}$, the sample distance correlation is denoted as $\widehat{\mathrm{dcor}}(\widetilde{\mathbf{T}}, \widetilde{\mathbf{Z}})$. Under the usual SDR model, Sheng & Yin (2016) showed that the distance covariance between $Y$ and $\boldsymbol{\beta}^T \mathbf{X}$ is maximized at the central subspace over $\boldsymbol{\beta} \in \mathbb{R}^{p \times d}$ such that $\boldsymbol{\beta}^T \boldsymbol{\Sigma} \boldsymbol{\beta} = \mathbf{I}_d$. Therefore, Zeng et al. (2024) recommended selecting $\lambda$ by maximizing $\widehat{\mathrm{dcor}}(\widetilde{Y}, \widetilde{\mathbf{X}}\widehat{\boldsymbol{\beta}})$, where $\widetilde{Y} \in \mathbb{R}^n$ and $\widetilde{\mathbf{X}} \in \mathbb{R}^{n \times p}$ represent the observed response vector and predictor matrix. For the mixture SDR model, we adjust for the sample membership $\gamma_{iw}$ and select $\lambda$ by maximizing $\widehat{\mathrm{dcor}}(\mathbf{D}_w \widetilde{Y}, \mathbf{D}_w \widetilde{\mathbf{X}} \widehat{\boldsymbol{\beta}}_w)$, where $\mathbf{D}_w$ is a diagonal matrix with diagonal elements $\gamma_{iw}(\widehat{\boldsymbol{\theta}})$.

---

**Algorithm 2** Group-wise coordinate descent algorithm

---

**Input:** $\widehat{\boldsymbol{\Sigma}}^{(t)} = (\widehat{\sigma}_{ij}) \in \mathbb{R}^{p \times p}$, $\widehat{\mathbf{U}}^{(t)} = (\widehat{u}_{jl}) \in \mathbb{R}^{p \times q}$

Initialize each row of $\mathbf{B}^{(0)}$ as $\mathbf{B}_j^{(0)} = (b_{j1}^{(0)}, \ldots, b_{jq}^{(0)})^T = \mathbf{0} \in \mathbb{R}^q, j = 1, \ldots, p$ and calculate the auxiliary vector

$\mathbf{h}_1^{(0)} = (h_{11}^{(0)}, \ldots, h_{1q}^{(0)})^T \in \mathbb{R}^q$ as $h_{1l}^{(0)} = \dfrac{\widehat{u}_{1l} - \sum_{j' \neq 1} \widehat{\sigma}_{j'1} b_{j'l}}{\widehat{\sigma}_{11}}, l = 1, \ldots, q$

**repeat**

    **for** $j = 1$ **to** $p$ **do**

        Based on $\mathbf{h}_j^{(r-1)}$ in the previous step, update $\mathbf{B}_j^{(r)}$ as

$$\mathbf{B}_j^{(r)} = \mathbf{h}_j^{(r-1)} \left( 1 - \frac{\lambda}{\widehat{\sigma}_{jj} \|\mathbf{h}_j^{(r-1)}\|_2} \right)_+$$

        Based on $\mathbf{B}_1^{(r)}, \ldots, \mathbf{B}_{j-1}^{(r)}, \mathbf{B}_{j+1}^{(r-1)}, \ldots, \mathbf{B}_p^{(r-1)}$, update the auxiliary vector $\mathbf{h}_j^{(r-1)}$ as

$$h_{jl}^{(r-1)} = \frac{\widehat{u}_{jl} - \left( \sum_{j'<j} \widehat{\sigma}_{j'j} b_{j'l}^{(r)} + \sum_{j'>j} \widehat{\sigma}_{j'j} b_{j'l}^{(r-1)} \right)}{\widehat{\sigma}_{jj}}, l = 1, \ldots, q$$

    **end for**

  **until** converge

  **Output:** $\mathbf{B}^{(r)}$

---

## B.3. Implementation of mixPFC-ISO

When $\boldsymbol{\Delta} = \sigma^2 \mathbf{I}$, there is no need to compute the inverse of the $p \times p$ covariance matrix. Instead, we only need to estimate $\sigma^2$. Under this simplification, the parameters of mixture of PFC model (3) reduces to $\boldsymbol{\theta} = (\sigma^2, \pi_w, \boldsymbol{\mu}_w, \mathcal{S}_w, w = 1, \ldots, K)$. In the E-step, the estimated probability is calculated by

$$\widehat{\gamma}_{iw}(\widehat{\boldsymbol{\theta}}^{(t)}) = \frac{\widehat{\pi}_w^{(t)}}{\widehat{\pi}_w^{(t)} + \sum_{j \neq w} \widehat{\pi}_j^{(t)} \exp\{1/(\widehat{\sigma}^2)^{(t)} (\mathbf{X}_i - \frac{1}{2} [(\widehat{\boldsymbol{\Gamma}}_j^{(t)} + \widehat{\boldsymbol{\Gamma}}_w^{(t)}) \mathbf{f}_i])^T (\widehat{\boldsymbol{\Gamma}}_j^{(t)} - \widehat{\boldsymbol{\Gamma}}_w^{(t)}) \mathbf{f}_i\}}.$$

In the M-step, the most important part is the updating formula of $\boldsymbol{\Gamma}_w$. Let $C_w^{(t)} = \sum_i \gamma_{iw}(\widehat{\boldsymbol{\theta}}^{(t)})$, $\mathbf{D}_w$ be a diagonal matrix with diagonal elements $\gamma_{iw}(\widehat{\boldsymbol{\theta}}^{(t)})$, $\mathbb{X}_w \in \mathbb{R}^{n \times p}$ be the centered data matrix with rows $(\mathbf{X}_i - \widehat{\boldsymbol{\mu}}_w^{(t)})^T$, $\mathbb{F} \in \mathbb{R}^{n \times q}$ with rows $(\mathbf{f}_i - \bar{\mathbf{f}})^T$. Through straightforward calculation, the MLE of $\mathcal{S}_w$ is the span of eigenvectors of $\widehat{\boldsymbol{\Sigma}}_{\text{fit},w}^{(t)}$ corresponding to the largest $d_w$ eigenvalues, where $\widehat{\boldsymbol{\Sigma}}_{\text{fit},w}^{(t)} = \mathbb{X}_w^T \mathbf{D}_w \mathbf{P}_{\mathbb{F}} \mathbf{D}_w \mathbb{X}_w / C_w$, $\mathbf{P}_{\mathbb{F}} = \mathbb{F}(\mathbb{F}^T \mathbf{D}_w \mathbb{F})^{-1} \mathbb{F}$ and $d_w$ is the dimension of subspace $\mathcal{S}_w$. Let $\widehat{\boldsymbol{\Phi}}_w^{(t)} = (\widehat{\phi}_1^{(t)}, \ldots, \widehat{\phi}_{d_w}^{(t)})$ be the eigenvectors corresponding to the largest $d_w$ eigenvalues of $\widehat{\boldsymbol{\Sigma}}_{\text{fit},w}^{(t)}$. Then, we have

$$\widehat{\boldsymbol{\Gamma}}_w^{(t)} = \widehat{\boldsymbol{\Phi}}_w^{(t)} (\widehat{\boldsymbol{\Phi}}_w^{(t)})^T \mathbb{X}_w^T \mathbf{D}_w \mathbb{F}(\mathbb{F}^T \mathbf{D}_w \mathbb{F})^{-1}.$$

To update $\sigma^2$, we maximize $Q$ function over $\sigma^2$, while letting other parameters fixed,

$$(\widehat{\sigma}^2)^{(t)} = \frac{1}{np} \sum_{w=1}^{K} C_w^{(t)} \left[ \text{tr}(\widehat{\boldsymbol{\Sigma}}_w) - \text{tr}(\mathbf{P}_{\widehat{\boldsymbol{\Phi}}_w^{(t)}} \widehat{\boldsymbol{\Sigma}}_{\text{fit},w}) \right],$$

where $\widehat{\boldsymbol{\Sigma}}_w^{(t)} = \mathbb{X}_w^T \mathbf{D}_w \mathbb{X}_w / n$. In high dimensions, it is challenging to accurately estimate the eigenvectors of $\widehat{\boldsymbol{\Sigma}}_{\text{fit},w}^{(t)}$. To address this issue, we promote sparsity by finding the sparse eigenvectors of $\widehat{\boldsymbol{\Sigma}}_{\text{fit},w}^{(t)}$. Several algorithms have been proposed to compute sparse eigenvectors (d'Aspremont et al., 2007; Zou et al., 2006; Witten et al., 2009; Journée et al., 2010). We adopt the variable projection method proposed by Erichson et al. (2020) for its computational efficiency and robustness in high dimensions. The algorithm, mixPFC-ISO, and corresponding convergence analysis are provided in the supplement.

To obtain sparse estimates $\widehat{\boldsymbol{\Phi}}_w^{(t)}$, We adopt the variable projection method proposed by Erichson et al. (2020) for its com-

putational efficiency and robustness in high dimensions. Specifically, we solve the following problem

$$(\mathbf{A}^*, \widehat{\boldsymbol{\Phi}}_w^{(t)}) = \underset{\mathbf{A}, \boldsymbol{\Phi}_w}{\operatorname{argmin}} \frac{1}{2} \|\widehat{\boldsymbol{\Sigma}}_{\mathrm{fit},w}^{(t)} - \widehat{\boldsymbol{\Sigma}}_{\mathrm{fit},w}^{(t)} \boldsymbol{\Phi}_w \mathbf{A}^T\|_F^2 + \alpha^{(t)} \xi(\boldsymbol{\Phi}_w), \quad \text{subject to } \mathbf{A}^T \mathbf{A} = \mathbf{I},$$

where $\alpha^{(t)}$ is a tuning parameter, and $\xi$ is a sparsity-inducing penalty function such as Lasso or elastic net. For fixed $\alpha^{(t)} = \alpha$, the penalized EM algorithm 3 for mixPFC with isotonic errors maximizes

$$\ell(\boldsymbol{\theta}) - \alpha \sum_{w=1}^{K} \xi(\boldsymbol{\Phi}_w). \tag{8}$$

The following result guarantees the convergence of Algorithm 3.

**Lemma B.2.** *If we set $\alpha^{(t)} = \alpha$ for all $t$, the objective function from (8) evaluated at $\widehat{\boldsymbol{\theta}}^{(t+1)}$ is guaranteed to be no less than the objective function from (8) evaluated at $\widehat{\boldsymbol{\theta}}^{(t)}$. That is, the sequence of iterates $\{\widehat{\boldsymbol{\theta}}^{(t)}\}_{i=1}^{\infty}$ generated by Algorithm 3 monotonically increase the value of the objective function from (8).*

*Proof.* The conditional log-likelihood is

$$\ell(\boldsymbol{\theta}; \mathbf{X}, Y) = \sum_{i=1}^{n} \log\left[\sum_{w=1}^{K} \pi_w N(\mathbf{X}_i \mid \boldsymbol{\mu}_w + \boldsymbol{\Gamma}_w \boldsymbol{\eta}_w \mathbf{f}(Y_i), \sigma^2 \mathbf{I})\right].$$

The penalized EM algorithm can be viewed as two alternating maximization steps. Consider the following function:

$$F(q, \boldsymbol{\theta}) := \mathrm{E}_q[\log(L(\boldsymbol{\theta}; \mathbf{X}, Y, W))] + H(q) - \alpha \sum_{w=1}^{K} \xi(\boldsymbol{\Psi}_w),$$

where $q = (q_1, \ldots, q_K)$ is an arbitrary probability density over the unobserved variable $W$. It is easy to show

$$F(q, \boldsymbol{\theta}) = \ell(\boldsymbol{\theta}; \mathbf{X}, Y) - D_{\mathrm{KL}}(q \| p_{W|\mathbf{X}, Y}(\cdot|\mathbf{X}, Y, \boldsymbol{\theta})) - \alpha \sum_{w=1}^{K} \xi(\boldsymbol{\Psi}_w),$$

where $D_{\mathrm{KL}}$ is the Kullback-Leibler (KL) divergence. The KL divergence between two distributions is non-negative and is zero when the two distributions are identical. Therefore, the E-step is to choose $q$ to maximize $F(q, \boldsymbol{\theta})$:

$$\widehat{q}^{(t+1)} = \underset{q}{\operatorname{argmax}} F(q, \widehat{\boldsymbol{\theta}}^{(t)}) = p_{W|\mathbf{X}, Y}(\cdot|\mathbf{X}, Y, \widehat{\boldsymbol{\theta}}^{(t)}),$$

which is given by the updating function of $\gamma_{iw}$. In M-step, we maximize $F$ over $\boldsymbol{\theta}$:

$$\begin{aligned}
\widehat{\boldsymbol{\theta}}^{(t+1)} &= \underset{\boldsymbol{\theta}}{\operatorname{argmax}} F(\widehat{q}^{(t)}, \boldsymbol{\theta}) \\
&= \underset{\boldsymbol{\theta}}{\operatorname{argmax}} \mathrm{E}_{\widehat{q}^{(t)}}[\log(L(\boldsymbol{\theta}; \mathbf{X}, Y, W))] - \alpha \sum_{w=1}^{K} \xi(\boldsymbol{\Psi}_w) \\
&= \underset{\boldsymbol{\theta}}{\operatorname{argmax}} Q(\boldsymbol{\theta}|\widehat{\boldsymbol{\theta}}^{(t)}) - \alpha \sum_{w=1}^{K} \xi(\boldsymbol{\Psi}_w).
\end{aligned}$$

Through this coordinate ascent strategy, the update $\widehat{\boldsymbol{\theta}}^{(t+1)}$ makes the value of the penalized log-likelihood function converge monotonically. Combining this result with the convergence of the variable projection algorithm solving the penalized $Q$-function (Erichson et al., 2020), we know Algorithm 3 converges when $\alpha^{(t)} = \alpha$.

$\square$

---

**Algorithm 3** Penalized EM algorithm for mixture PFC with isotonic errors

---

**Input:** Data $\{(\mathbf{X}_i, Y_i)\}_{i=1}^n$, fitting function $\mathbf{f}$

Initialize $\widehat{\gamma}_{iw}(\boldsymbol{\theta}^0)$

**repeat**

E-Step: $\widehat{\gamma}_{iw}^{(t)} = \dfrac{\widehat{\pi}_w^{(t)}}{\widehat{\pi}_w^{(t)} + \sum_{j \neq w} \widehat{\pi}_j^{(t)} \exp\{1/(\widehat{\sigma}^2)^{(t)}(\mathbf{X}_i - \frac{1}{2}[(\widehat{\boldsymbol{\Gamma}}_j^{(t)} + \widehat{\boldsymbol{\Gamma}}_w^{(t)})\mathbf{f}_i])^T(\widehat{\boldsymbol{\Gamma}}_j^{(t)} - \widehat{\boldsymbol{\Gamma}}_w^{(t)})\mathbf{f}_i\}}$

M-Step:

$\widehat{\boldsymbol{\Sigma}}_{\text{fit},w}^{(t)} = \mathbb{X}_w^T \mathbf{D}_w \mathbf{P}_{\mathbb{F}} \mathbf{D}_w \mathbb{X}_w / n, \ \mathbf{P}_{\mathbb{F}} = \mathbb{F}(\mathbb{F}^T \mathbf{D}_w \mathbb{F})^{-1} \mathbb{F}$

$(\mathbf{A}^*, \widehat{\boldsymbol{\Phi}}_w^{(t)}) = \underset{\mathbf{A}, \boldsymbol{\Phi}_w}{\operatorname{argmax}} -\dfrac{1}{2}\|\widehat{\boldsymbol{\Sigma}}_{\text{fit},w}^{(t)} - \widehat{\boldsymbol{\Sigma}}_{\text{fit},w}^{(t)} \boldsymbol{\Phi}_w \mathbf{A}^T\|_F^2 - \alpha^{(t)} \xi(\boldsymbol{\Phi}_w), \text{ subject to } \mathbf{A}^T \mathbf{A} = \mathbf{I}$

$\widehat{\boldsymbol{\Gamma}}_w^{(t+1)} = \widehat{\boldsymbol{\Phi}}_w^{(t)}(\widehat{\boldsymbol{\Phi}}_w^{(t)})^T \mathbb{X}_w^T \mathbf{D}_w \mathbb{F}(\mathbb{F}^T \mathbf{D}_w \mathbb{F})^{-1}$

$(\widehat{\sigma}^2)^{(t+1)} = \dfrac{1}{np} \sum_{w=1}^K C_w^{(t)} \left[\operatorname{tr}(\widehat{\boldsymbol{\Sigma}}_w) - \operatorname{tr}(\mathbf{P}_{\widehat{\boldsymbol{\Phi}}_w^{(t)}} \widehat{\boldsymbol{\Sigma}}_{\text{fit},w})\right]$

$\widehat{\pi}_w^{(t+1)} = \dfrac{1}{n} \sum_{i=1}^n \widehat{\gamma}_{iw}^{(t)}$

$\widehat{\boldsymbol{\mu}}_w^{(t+1)} = \dfrac{1}{\sum_i \gamma_{iw}(\widehat{\boldsymbol{\theta}}^{(t)})} \sum \gamma_{iw}(\widehat{\boldsymbol{\theta}}^{(t)})\mathbf{X}_i$

**until** converge

**Output:** $\widehat{\pi}_w, \widehat{\boldsymbol{\Gamma}}_w$

---

# C. Proof of Theorem 1

From this section onward, we focus on the theoretical properties of a simplified version (Algorithm 4) of the mixPFC algorithm presented in the main paper.

Recall the five conditions are:

(C1) The singular values of $\widehat{\boldsymbol{\Sigma}}_{\mathbf{f}} = \frac{1}{n}\sum_{i=1}^n \mathbf{f}_i \mathbf{f}_i^T$ satisfy that $M_1 \leq \sigma_{\min}(\widehat{\boldsymbol{\Sigma}}_{\mathbf{f}}) \leq \sigma_{\max}(\widehat{\boldsymbol{\Sigma}}_{\mathbf{f}}) \leq M_2$, and $M_3 \leq \min_{1 \leq i \leq n}\|\mathbf{f}_i\|_2 \leq \max_{1 \leq i \leq n}\|\mathbf{f}_i\|_2 \leq M_4$.

(C2) The initialization $\boldsymbol{\theta}^{(0)}$ satisfies that $d_F(\boldsymbol{\theta}^{(0)}, \boldsymbol{\theta}^*) \vee \|\mathbf{B}_1^{(0)} - \mathbf{B}_1^*\|_F \vee \|\mathbf{B}_2^{(0)} - \mathbf{B}_2^*\|_F < r\Omega$, and $\operatorname{vec}(\boldsymbol{\Gamma}_w^{(0)} - \boldsymbol{\Gamma}_w^*) \in \mathcal{L}(s)$, with $r < |c_0 - c_\pi|/\Omega \wedge C_b \wedge \frac{1}{a}(\sqrt{C_d - 1/(4\sqrt{M_1}) + \frac{b^2}{4a^2}} - \frac{b}{2a})$, $a^2 = \frac{2M_2^{3/2}}{\sqrt{M_1}}$, $b = 2\sqrt{M_2} + \frac{2M_2 + \sqrt{M_2}}{2\sqrt{M_1}}$.

(C3) There exists a sufficiently large constant $M_5 > 0$, which does not depend on $n, p, s$, such that $\sigma_d(\mathbf{B}_w^*) \geq M_5 \geq \sqrt{sq^3(\log n)^2 \log p/n}$.

(C4) $\Omega = \sqrt{\operatorname{tr}[(\boldsymbol{\Gamma}_2^* - \boldsymbol{\Gamma}_1^*)\widehat{\boldsymbol{\Sigma}}_{\mathbf{f}}(\boldsymbol{\Gamma}_2^* - \boldsymbol{\Gamma}_1^*)^T]} \geq C(c_0, C_b, M_b, M_i; i = 1, \ldots, 4)$, which is a constant that is only depends on $c_0, M_b, C_b$, and $M_i, i = 1, \ldots, 4$.

(C5) $n > C_3 sq^3 \log(p)$ for a sufficiently large constant $C_3$.

Before the proof, we review some definitions and present properties about $\boldsymbol{\Sigma}_w^*$ and $\mathbf{U}_w^*$. The parameter of interest is $\boldsymbol{\theta} = \{\pi_1, \boldsymbol{\Gamma}_1, \boldsymbol{\Gamma}_2\}$. Let $\boldsymbol{\theta}^*$ be the true value of $\boldsymbol{\theta}$, and $\widehat{\boldsymbol{\theta}}^{(t)}$ be the estimate of $\boldsymbol{\theta}$ at the $t$-th iteration. The true parameter space we consider is

$$\boldsymbol{\Theta}^* = \{\boldsymbol{\theta}^* : \pi_1^* \in (c_\pi, 1 - c_\pi), \|\operatorname{vec}(\boldsymbol{\Gamma}_w^*)\|_0 \leq sq, \|\mathbf{B}_w^*\|_F \leq M_a, \|\boldsymbol{\Gamma}_w^*\|_F \leq M_b, w = 1, 2\},$$

and the constriction basin $\mathcal{B}_{\text{con}}(\boldsymbol{\theta}^*)$ is

$$\mathcal{B}_{\text{con}}(\boldsymbol{\theta}^*) = \{\boldsymbol{\theta} : \pi_w \in (c_0, 1 - c_0), \|\boldsymbol{\Gamma}_w - \boldsymbol{\Gamma}_w^*\|_F \leq C_b\Omega,$$
$$(1 - C_d)\Omega^2 \leq |\operatorname{tr}(\boldsymbol{\delta}_w(\boldsymbol{\Gamma})\widehat{\boldsymbol{\Sigma}}_{\mathbf{f}}(\boldsymbol{\Gamma}_2 - \boldsymbol{\Gamma}_1)^T)| \leq (1 + C_d)\Omega^2,$$
$$\operatorname{vec}(\boldsymbol{\Gamma}_w - \boldsymbol{\Gamma}_w^*) \in \mathcal{L}(s), w = 1, 2\},$$

where $\Omega$ is the signal strength, $c_0 \leq c_\pi$, and $\boldsymbol{\delta}_w(\boldsymbol{\Gamma}) = \boldsymbol{\Gamma}_w^* - (\boldsymbol{\Gamma}_2 + \boldsymbol{\Gamma}_1)/2$. We define

$$\mathcal{L}(s) = \{\mathbf{u} \in \mathbb{R}^{pq} : \|\mathbf{u}_{\widetilde{\mathcal{S}}_1^c}\|_1 \leq (\sqrt{sq} + 2q\sqrt{3s})\|\mathbf{u}_{\widetilde{\mathcal{S}}_1}\|_2 + \sqrt{sq}\|\mathbf{u}\|_2,$$
$$\text{for some } \widetilde{\mathcal{S}}_1 \subset [pq] \text{ with } |\widetilde{\mathcal{S}}_1| = 3sq.\}$$

---

**Algorithm 4** Penalized EM algorithm for mixture PFC (simplified version for theoretical analysis)

---

**Input:** Data $(\mathbf{X}_i, \mathbf{Y}_i)$, fitting function $\mathbf{f}$, $d_1, d_2$

Initial value $\widehat{\pi}_w^{(0)}, \widehat{\mathbf{\Gamma}}_w^{(0)}, \mathbf{\Sigma}_w^{(0)}, \mathbf{U}_w^{(0)}$, and $\widehat{\mathbf{B}}_w^{(0)} = \underset{\mathbf{B}_w}{\operatorname{argmin}} \frac{1}{2} \operatorname{tr}(\mathbf{B}_w^T \widehat{\mathbf{\Sigma}}_w^{(0)} \mathbf{B}_w) - \operatorname{tr}\{(\widehat{\mathbf{U}}_w^{(0)})^T \mathbf{B}_w\} + \lambda^{(0)} \|\mathbf{B}_w\|_{2,1}$, where the initial tuning parameter is

$$\lambda^{(0)} = C_1 \frac{d_F(\widehat{\boldsymbol{\theta}}^{(0)}, \boldsymbol{\theta}^*) \vee \|\widehat{\mathbf{B}}_1^{(0)} - \mathbf{B}_1^*\|_F \vee \|\widehat{\mathbf{B}}_2^{(0)} - \mathbf{B}_2^*\|_F}{\sqrt{s}} + C_\lambda \sqrt{\frac{q^3 (\log n)^2 \log p}{n}}$$

**repeat**

   E-Step:

$$\widehat{\gamma}_{iw}(\widehat{\boldsymbol{\theta}}^{(t)}) = \frac{\widehat{\pi}_w^{(t)}}{\widehat{\pi}_w^{(t)} + \sum_{w' \neq w} \widehat{\pi}_j^{(t)} \exp\{[(\widehat{\sigma}^2)^{(t)}]^{-1}(\mathbf{X}_i - \frac{1}{2}[(\widehat{\mathbf{\Gamma}}_{w'}^{(t)} + \widehat{\mathbf{\Gamma}}_w^{(t)})\mathbf{f}(Y_i)])^T (\widehat{\mathbf{\Gamma}}_{w'}^{(t)} - \widehat{\mathbf{\Gamma}}_w^{(t)})\mathbf{f}(Y_i)\}}$$

   M-Step:

$$\widehat{\pi}_w^{(t+1)} = \frac{1}{n} \sum_{i=1}^n \widehat{\gamma}_{iw}(\widehat{\boldsymbol{\theta}}^{(t)}), \quad \widehat{\mathbf{\Sigma}}_w^{(t+1)} = \frac{1}{n} \sum_{i=1}^n \gamma_{iw}(\widehat{\boldsymbol{\theta}}^{(t)}) \mathbf{X}_i \mathbf{X}_i, \quad \widehat{\mathbf{U}}_w^{(t+1)} = \frac{1}{n} \sum_{i=1}^n \gamma_{iw}(\widehat{\boldsymbol{\theta}}^{(t)}) \mathbf{X}_i \mathbf{f}(Y_i)^T$$

$$\widehat{\mathbf{B}}_w^{(t+1)} = \underset{\mathbf{B}_w}{\operatorname{argmin}} \frac{1}{2} \operatorname{tr}(\mathbf{B}_w^T \widehat{\mathbf{\Sigma}}_w^{(t+1)} \mathbf{B}_w) - \operatorname{tr}\{(\widehat{\mathbf{U}}_w^{(t+1)})^T \mathbf{B}_w\} + \lambda^{(t+1)} \|\mathbf{B}_w\|_{2,1}, \lambda^{(t+1)} = \kappa \lambda^{(t)} + C_\lambda \sqrt{\frac{q^3 (\log n)^2 \log p}{n}}$$

Compute $\widehat{\boldsymbol{\beta}}_w^{(t+1)}$, the top-$d$ left singular vectors of $\widehat{\mathbf{B}}_w^{(t+1)}$ and then update according to

$$\widehat{\mathbf{\Gamma}}_w^{(t+1)} = \mathbf{P}_{\widehat{\boldsymbol{\beta}}_w^{(t+1)}} \widehat{\mathbf{U}}_w^{(t+1)} [\widehat{\pi}_w^{(t+1)} \widehat{\mathbf{\Sigma}}_{\mathbf{f}}]^{-1}$$

$$(\widehat{\sigma}^2)^{(t+1)} = \frac{1}{np} \sum_{w=1}^2 (\sum_{i=1}^n \widehat{\gamma}_{iw})(\widehat{\boldsymbol{\theta}}^{(t)}) [\operatorname{tr}(\widehat{\mathbf{\Sigma}}_w^{(t+1)}) - \operatorname{tr}(\mathbf{P}_{\widehat{\mathbf{\Gamma}}_w^{(t+1)}} \widehat{\mathbf{\Sigma}}_{\text{fit},w})]$$

   **until** converge
   **Output:** $\widehat{\pi}_w, \widehat{\mathbf{\Gamma}}_w, \widehat{\boldsymbol{\eta}}_w$

---

Let $M(\boldsymbol{\theta}) = \{\pi_w(\boldsymbol{\theta}), \mathbf{U}_w(\boldsymbol{\theta}), \mathbf{\Sigma}_w(\boldsymbol{\theta}), w = 1, 2\}$. Note that $\|\operatorname{vec}(\mathbf{\Gamma}^*)\|_0 \leq sq$ implies $\|\operatorname{vec}(\mathbf{B}^*)\|_0 \leq sq$. Define

$$\widehat{\pi}_w(\boldsymbol{\theta}) = \frac{1}{n} \sum_{i=1}^n \gamma_{iw}(\boldsymbol{\theta}), \quad \pi_w(\boldsymbol{\theta}) = \mathrm{E}[\widehat{\pi}_w(\boldsymbol{\theta})]$$

$$\widehat{\mathbf{U}}_w(\boldsymbol{\theta}) = \frac{1}{n} \sum_{i=1}^n \gamma_{iw}(\boldsymbol{\theta}) \mathbf{X}_i \mathbf{f}_i^T, \quad \mathbf{U}_w(\boldsymbol{\theta}) = \mathrm{E}[\widehat{\mathbf{U}}_w(\boldsymbol{\theta})]$$

$$\widehat{\mathbf{\Sigma}}_w(\boldsymbol{\theta}) = \frac{1}{n} \sum_{i=1}^n \gamma_{iw}(\boldsymbol{\theta}) \mathbf{X}_i \mathbf{X}_i^T, \quad \mathbf{\Sigma}_w(\boldsymbol{\theta}) = \mathrm{E}[\widehat{\mathbf{\Sigma}}_w(\boldsymbol{\theta})].$$

By definition,

$$\widehat{\pi}_w^{(t+1)} = \widehat{\pi}_w(\widehat{\boldsymbol{\theta}}^{(t)}), \quad \pi_w^{(t+1)} = \mathrm{E}[\widehat{\pi}_w^{(t+1)}] = \mathrm{E}[\widehat{\pi}_w(\widehat{\boldsymbol{\theta}}^{(t)})] = \pi_w(\widehat{\boldsymbol{\theta}}^{(t)})$$

$$\widehat{\mathbf{U}}_w^{(t+1)} = \widehat{\mathbf{U}}_w(\widehat{\boldsymbol{\theta}}^{(t)}), \quad \mathbf{U}_w^{(t+1)} = \mathrm{E}[\widehat{\mathbf{U}}_w^{(t+1)}] = \mathrm{E}[\widehat{\mathbf{U}}_w(\widehat{\boldsymbol{\theta}}^{(t)})] = \mathbf{U}_w(\widehat{\boldsymbol{\theta}}^{(t)})$$

$$\widehat{\mathbf{\Sigma}}_w^{(t+1)} = \widehat{\mathbf{\Sigma}}_w(\widehat{\boldsymbol{\theta}}^{(t)}), \quad \mathbf{\Sigma}_w^{(t+1)} = \mathrm{E}[\widehat{\mathbf{\Sigma}}_w^{(t+1)}] = \mathrm{E}[\widehat{\mathbf{\Sigma}}_w(\widehat{\boldsymbol{\theta}}^{(t)})] = \mathbf{\Sigma}_w(\widehat{\boldsymbol{\theta}}^{(t)}).$$

Note that

$$\pi_w(\boldsymbol{\theta}^*) = \mathrm{E}[\frac{1}{n} \sum_{i=1}^n \gamma_{iw}(\boldsymbol{\theta}^*)] = \mathrm{E}[\frac{1}{n} \sum_{i=1}^n P(W_i = w | \mathbf{X}_i, Y_i)] = P(W_i = w | \mathbf{X}_i, Y_i) = \pi_w^*.$$

We have

$$
\mathbf{U}_w^* = \mathbf{U}_1(\boldsymbol{\theta}^*) = \frac{1}{n}\sum_{i=1}^{n} \mathrm{E}[\gamma_{iw}(\boldsymbol{\theta}^*)\mathbf{X}_i]\mathbf{f}_i^T
$$

$$
= \frac{1}{n}\sum_{i=1}^{n}\mathrm{E}[\mathrm{E}_{W|\mathbf{X}}[I(W_i = w)\mathbf{X}_i]]\mathbf{f}_i^T = \frac{1}{n}\sum_{i=1}^{n}\mathrm{E}_{W,\mathbf{X}}[I(W_i = w)\mathbf{X}_i]\mathbf{f}_i^T
$$

$$
= \frac{1}{n}\sum_{i=1}^{n}\mathrm{E}_W[I(W_i = w)\mathrm{E}_{\mathbf{X}|W=w}[\mathbf{X}_i]]\mathbf{f}_i^T = \frac{1}{n}\sum_{i=1}^{n}\pi_w^*\boldsymbol{\Gamma}_w^*\mathbf{f}_i\mathbf{f}_i^T = \pi_w^*\boldsymbol{\Gamma}_w^*\widehat{\boldsymbol{\Sigma}}_\mathbf{f},
$$

where $\widehat{\boldsymbol{\Sigma}}_\mathbf{f} = \frac{1}{n}\sum_{i=1}^{n}\mathbf{f}_i\mathbf{f}_i^T$. Similarly,

$$
\boldsymbol{\Sigma}_w^* = \boldsymbol{\Sigma}_w(\boldsymbol{\theta}^*) = \frac{1}{n}\sum_{i=1}^{n}\mathrm{E}[\gamma_{iw}(\boldsymbol{\theta}^*)\mathbf{X}_i\mathbf{X}_i^T]
$$

$$
= \frac{1}{n}\sum_{i=1}^{n}\mathrm{E}_W[I(W_i = w)\,\mathrm{E}_{\mathbf{X}|W=w}[\mathbf{X}_i\mathbf{X}_i^T]] = \frac{1}{n}\sum_{i=1}^{n}\mathrm{E}_W[I(W_i = w)(\mathbf{I}_p + \boldsymbol{\Gamma}_w^*\mathbf{f}_i\mathbf{f}_i^T(\boldsymbol{\Gamma}_w^*)^T)]
$$

$$
= \frac{1}{n}\sum_{i=1}^{n}(\pi_w^*\mathbf{I}_p) + \pi_w^*\boldsymbol{\Gamma}_w^*\widehat{\boldsymbol{\Sigma}}_\mathbf{f}(\boldsymbol{\Gamma}_w^*)^T = \pi_w^*\mathbf{I}_p + \pi_w^*\boldsymbol{\Gamma}_w^*\widehat{\boldsymbol{\Sigma}}_\mathbf{f}(\boldsymbol{\Gamma}_w^*)^T.
$$

Let $\mathbf{B}_w^* = (\boldsymbol{\Sigma}_w^*)^{-1}\mathbf{U}_w^*$, $\boldsymbol{\beta}_w^*$ be the top-$d$ left singular vectors. Then $\mathrm{span}(\boldsymbol{\beta}_w^*) = \mathrm{span}(\boldsymbol{\Gamma}_w^*)$. We further have $\boldsymbol{\Gamma}_w^* = \mathbf{P}_{\boldsymbol{\Gamma}_w^*}\mathbf{U}_w^*(\pi_w^*\widehat{\boldsymbol{\Sigma}}_\mathbf{f})^{-1} = \mathbf{P}_{\boldsymbol{\beta}_w^*}\mathbf{U}_w^*(\pi_w^*\widehat{\boldsymbol{\Sigma}}_\mathbf{f})^{-1}$.

Since $\sigma_{\min}(\boldsymbol{\Gamma}_w^*) \geq 0$, $\sigma_{\min}(\mathbf{I}_p + \boldsymbol{\Gamma}_w^*\widehat{\boldsymbol{\Sigma}}_\mathbf{f}(\boldsymbol{\Gamma}_w^*)^T) \geq 1$, we can bound the 2-norm as following

$$
\begin{aligned}
\|\mathbf{B}_w^*\|_2 &= \|[\pi_w^*\mathbf{I}_p + \pi_w^*\boldsymbol{\Gamma}_w^*\widehat{\boldsymbol{\Sigma}}_\mathbf{f}(\boldsymbol{\Gamma}_w^*)^T]^{-1}\pi_w^*\boldsymbol{\Gamma}_w^*\widehat{\boldsymbol{\Sigma}}_\mathbf{f}\|_2 \\
&= \|[\mathbf{I}_p + \boldsymbol{\Gamma}_w^*\widehat{\boldsymbol{\Sigma}}_\mathbf{f}(\boldsymbol{\Gamma}_w^*)^T]^{-1}\boldsymbol{\Gamma}_w^*\widehat{\boldsymbol{\Sigma}}_\mathbf{f}\|_2 \\
&\leq \|[\mathbf{I}_p + \boldsymbol{\Gamma}_w^*\widehat{\boldsymbol{\Sigma}}_\mathbf{f}(\boldsymbol{\Gamma}_w^*)^T]^{-1}\|_2 \cdot \|\boldsymbol{\Gamma}_w^*\widehat{\boldsymbol{\Sigma}}_\mathbf{f}\|_2 \\
&\leq \|\boldsymbol{\Gamma}_w^*\|_F \cdot \|\widehat{\boldsymbol{\Sigma}}_\mathbf{f}\|_2 \leq M_b M_2.
\end{aligned}
$$

Define $d_{F,s}(M(\boldsymbol{\theta}_1), M(\boldsymbol{\theta}_2))$ and $d_2(M(\boldsymbol{\theta}_1), M(\boldsymbol{\theta}_2))$ as

$$
\max_{w=1,2}\{|\pi_w(\boldsymbol{\theta}_1) - \pi_w(\boldsymbol{\theta}_2)| \vee \|\mathbf{U}_w(\boldsymbol{\theta}_1) - \mathbf{U}_w(\boldsymbol{\theta}_2)\|_{F,s} \vee \|(\boldsymbol{\Sigma}_w(\boldsymbol{\theta}_1) - \boldsymbol{\Sigma}_w(\boldsymbol{\theta}_2))\mathbf{B}_w^*\|_{F,s}\},
$$

$$
\max_{w=1,2}\{|\pi_w(\boldsymbol{\theta}_1) - \pi_w(\boldsymbol{\theta}_2)| \vee \|\mathbf{U}_w(\boldsymbol{\theta}_1) - \mathbf{U}_w(\boldsymbol{\theta}_2)\|_F \vee \|(\boldsymbol{\Sigma}_w(\boldsymbol{\theta}_1) - \boldsymbol{\Sigma}_w(\boldsymbol{\theta}_2))\mathbf{B}_w^*\|_F\},
$$

where $\|A\|_{F,s} = \sup_{\mathbf{u}\in\mathbb{R}^{p\times q}, \|\mathbf{u}\|_F=1, \mathrm{vec}(\mathbf{u})\in\mathcal{L}(s)}\langle A, \mathbf{u}\rangle_F$. The proof of Theorem 4.1 is based on the following two lemmas.

**Lemma C.1.** *Under conditions (C1) and (C4), if* $\boldsymbol{\theta}\in\mathcal{B}_{con}(\boldsymbol{\theta}^*)$*, for some* $0 < \kappa_0 < \frac{1}{2\vee(256C_0 q/\tau_0)\vee 8C_0}$*,*

$$
d_F(M(\boldsymbol{\theta}), M(\boldsymbol{\theta}^*)) \leq \kappa_0(d_F(\boldsymbol{\theta}, \boldsymbol{\theta}^*) \vee \|\mathbf{B}_1 - \mathbf{B}_1^*\|_F \vee \|\mathbf{B}_2 - \mathbf{B}_2^*\|_F),
$$

*where* $d_F(\boldsymbol{\theta}, \boldsymbol{\theta}^*) = \{|\pi_1 - \pi_1^*| \vee \|\boldsymbol{\Gamma}_1 - \boldsymbol{\Gamma}_1^*\|_F \vee \|\boldsymbol{\Gamma}_2 - \boldsymbol{\Gamma}_2^*\|_F\}$

**Lemma C.2.** *Suppose* $\boldsymbol{\theta}^* \in \boldsymbol{\Theta}^*$*, under condition (C1) and (C5), there exists a constant* $C_{con} > 0$*, such that with probability at least* $1 - o(1)$*,*

$$
\sup_{\boldsymbol{\theta}\in\mathcal{B}_{con}(\boldsymbol{\theta}^*)} d_{F,s}(M_n(\boldsymbol{\theta}), M(\boldsymbol{\theta})) \leq C_{con}\sqrt{\frac{sq^3(\log n)^2\log p}{n}}.
$$

The proof of these two lemmas is quite involved and is presented in Sections D and E. We first establish a concentration result for the estimator $\widehat{\mathbf{B}}_w$ in Section C.1 before proving Theorem 4.1 in Section C.2.

## C.1. Concentration of the estimator $\widehat{\mathbf{B}}_w$

**Lemma C.3.** *Suppose that $\boldsymbol{\theta}^* \in \boldsymbol{\Theta}^*$ and $\mathrm{vec}(\widehat{\mathbf{B}}_w^{(0)}) \in \mathcal{B}_{con}(\boldsymbol{\theta}^*)$. Let*

$$\lambda^{(t+1)} \geq 4C_{con}\sqrt{\frac{q^3(\log n)^2 \log p}{n}} + 4\kappa_0\Big(\frac{d_F(\widehat{\boldsymbol{\theta}}^{(t)}, \boldsymbol{\theta}^*) \vee \|\widehat{\mathbf{B}}_1^{(t)} - \mathbf{B}_1^*\|_F \vee \|\widehat{\mathbf{B}}_1^{(t)} - \mathbf{B}_2^*\|_F}{\sqrt{s}}\Big),$$

*for $\kappa_0$ defined before and some constant $C_{con}$ and $\widehat{\mathbf{B}}_w^{(t+1)}$ solved by*

$$\widehat{\mathbf{B}}_w^{(t+1)} = \underset{\mathbf{B}_w}{\arg\min}\,\frac{1}{2}\,\mathrm{tr}(\mathbf{B}_w^T\widehat{\boldsymbol{\Sigma}}_w^{(t+1)}\mathbf{B}_w) - \mathrm{tr}\{(\widehat{\mathbf{U}}_w^{(t+1)})^T\mathbf{B}_w\} + \lambda^{(t+1)}\|\mathbf{B}_w\|_{2,1}.$$

*With probability at least $1 - o(1)$, we have*

$$\mathrm{vec}(\widehat{\mathbf{B}}_w^{(t+1)} - \mathbf{B}_w^*) \in \mathcal{L}(s),$$

*and*

$$\|\widehat{\mathbf{B}}_w^{(t+1)} - \mathbf{B}_w^*\|_F \leq \frac{4}{\tau_0}d_{F,s}(M_n(\widehat{\boldsymbol{\theta}}^{(t)}), M(\boldsymbol{\theta}^*)) + \frac{2}{\tau_0}\lambda^{(t+1)}(\sqrt{3sq} + 2\sqrt{sq} + 2\sqrt{3sq^2}).$$

*Proof.* Recall that
$$\mathcal{L}(s) = \{\mathbf{u} \in \mathbb{R}^{pq} : \|\mathbf{u}_{\widetilde{\mathcal{S}}_1^c}\|_1 \leq (\sqrt{sq} + 2q\sqrt{3s})\|\mathbf{u}_{\widetilde{\mathcal{S}}_1}\|_2 + \sqrt{sq}\|\mathbf{u}\|_2,$$
$$\text{for some } \widetilde{\mathcal{S}}_1 \subset [pq] \text{ with } |\widetilde{\mathcal{S}}_1| = 3sq.\}$$

Consider
$$\widehat{\mathbf{B}}_w^{(t+1)} = \underset{\mathbf{B}_w}{\arg\min}\,\frac{1}{2}\,\mathrm{tr}(\mathbf{B}_w^T\widehat{\boldsymbol{\Sigma}}_w^{(t+1)}\mathbf{B}_w) - \mathrm{tr}\{(\widehat{\mathbf{U}}_w^{(t+1)})^T\mathbf{B}_w\} + \lambda^{(t+1)}\|\mathbf{B}_w\|_{2,1}.$$

For simplicity, we let $w = 1$ in the following. To show
$$\mathrm{vec}(\widehat{\mathbf{B}}_1^{(t+1)} - \mathbf{B}_1^*) \in \mathcal{L}(s),$$

we note that

$$\lambda^{(t+1)}(\|\widehat{\mathbf{B}}_1^{(t+1)}\|_{2,1} - \|\mathbf{B}_1^*\|_{2,1})$$
$$\leq \frac{1}{2}\,\mathrm{tr}((\mathbf{B}_1^*)^T\widehat{\boldsymbol{\Sigma}}_1^{(t+1)}\mathbf{B}_1^*) - \frac{1}{2}\,\mathrm{tr}((\widehat{\mathbf{B}}_1^{(t+1)})^T\widehat{\boldsymbol{\Sigma}}_1^{(t+1)}\widehat{\mathbf{B}}_1^{(t+1)}) - \mathrm{tr}((\widehat{\mathbf{U}}_1^{(t+1)})^T\mathbf{B}_1^*) + \mathrm{tr}((\widehat{\mathbf{U}}_1^{(t+1)})^T\widehat{\mathbf{B}}_1^{(t+1)}))$$
$$\leq \mathrm{tr}((\mathbf{B}_1^* - \widehat{\mathbf{B}}_1^{(t+1)})^T\widehat{\boldsymbol{\Sigma}}_1^{(t+1)}\mathbf{B}_1^*) - \frac{1}{2}\,\mathrm{tr}((\mathbf{B}_1^* - \widehat{\mathbf{B}}_1^{(t+1)})^T\widehat{\boldsymbol{\Sigma}}_1^{(t+1)}(\mathbf{B}_1^* - \widehat{\mathbf{B}}_1^{(t+1)})) -$$
$$\mathrm{tr}((\widehat{\mathbf{U}}_1^{(t+1)})^T(\mathbf{B}_1^* - \widehat{\mathbf{B}}_1^{(t+1)})),$$

where we use

$$\frac{1}{2}\,\mathrm{tr}((\mathbf{B}_1^* - \widehat{\mathbf{B}}_1^{(t+1)})^T\widehat{\boldsymbol{\Sigma}}_1^{(t+1)}(\mathbf{B}_1^* - \widehat{\mathbf{B}}_1^{(t+1)}))$$
$$= \frac{1}{2}\,\mathrm{tr}((\mathbf{B}_1^*)^T\widehat{\boldsymbol{\Sigma}}_1^{(t+1)}\mathbf{B}_1^*) + \frac{1}{2}\,\mathrm{tr}((\widehat{\mathbf{B}}_1^{(t+1)})^T\widehat{\boldsymbol{\Sigma}}_1^{(t+1)}\widehat{\mathbf{B}}_1^{(t+1)}) - \mathrm{tr}((\mathbf{B}_1^*)^T\widehat{\boldsymbol{\Sigma}}_1^{(t+1)}\widehat{\mathbf{B}}_1^{(t+1)}).$$

Since $\widehat{\boldsymbol{\Sigma}}_1^{(t+1)}$ is symmetric semi-positive definite, we have

$$\mathrm{tr}((\mathbf{B}_1^* - \widehat{\mathbf{B}}_1^{(t+1)})^T\widehat{\boldsymbol{\Sigma}}_1^{(t+1)}(\mathbf{B}_1^* - \widehat{\mathbf{B}}_1^{(t+1)}))$$
$$= \mathrm{vec}(\mathbf{B}_1^* - \widehat{\mathbf{B}}_1^{(t+1)})^T\,\mathrm{vec}(\widehat{\boldsymbol{\Sigma}}_1^{(t+1)}(\mathbf{B}_1^* - \widehat{\mathbf{B}}_1^{(t+1)}))$$
$$= \mathrm{vec}(\mathbf{B}_1^* - \widehat{\mathbf{B}}_1^{(t+1)})^T(\mathbf{I}_q \otimes \widehat{\boldsymbol{\Sigma}}_1^{(t+1)})\,\mathrm{vec}(\mathbf{B}_1^* - \widehat{\mathbf{B}}_1^{(t+1)}) \geq 0.$$

Using the fact that $\mathbf{\Sigma}_1^* \mathbf{B}_1^* - \mathbf{U}_1^* = 0$, we have

$$
\begin{aligned}
&\lambda^{(t+1)}(\|\widehat{\mathbf{B}}_1^{(t+1)}\|_{2,1} - \|\mathbf{B}_1^*\|_{2,1}) \\
&\leq \operatorname{tr}((\mathbf{B}_1^* - \widehat{\mathbf{B}}_1^{(t+1)})^T \widehat{\mathbf{\Sigma}}_1^{(t+1)} \mathbf{B}_1^*) - \operatorname{tr}((\widehat{\mathbf{U}}_1^{(t+1)})^T (\mathbf{B}_1^* - \widehat{\mathbf{B}}_1^{(t+1)})) \\
&= \operatorname{tr}((\mathbf{B}_1^* - \widehat{\mathbf{B}}_1^{(t+1)})^T (\widehat{\mathbf{\Sigma}}_1^{(t+1)} \mathbf{B}_1^* - \widehat{\mathbf{U}}_1^{(t+1)})) \\
&= \langle \mathbf{B}_1^* - \widehat{\mathbf{B}}_1^{(t+1)}, \widehat{\mathbf{\Sigma}}_1^{(t+1)} \mathbf{B}_1^* - \widehat{\mathbf{U}}_1^{(t+1)} \rangle_F \\
&= \langle \mathbf{B}_1^* - \widehat{\mathbf{B}}_1^{(t+1)}, (\widehat{\mathbf{\Sigma}}_1^{(t+1)} - \mathbf{\Sigma}_1^{(t+1)}) \mathbf{B}_1^* + \mathbf{\Sigma}_1^{(t+1)} \mathbf{B}_1^* - \mathbf{U}_1^{(t+1)} - (\widehat{\mathbf{U}}_1^{(t+1)} - \mathbf{U}_1^{(t+1)}) \rangle_F \\
&= \langle \mathbf{B}_1^* - \widehat{\mathbf{B}}_1^{(t+1)}, (\widehat{\mathbf{\Sigma}}_1^{(t+1)} - \mathbf{\Sigma}_1^{(t+1)}) \mathbf{B}_1^* - (\widehat{\mathbf{U}}_1^{(t+1)} - \mathbf{U}_1^{(t+1)}) \rangle_F + \\
&\quad \langle \mathbf{B}_1^* - \widehat{\mathbf{B}}_1^{(t+1)}, \mathbf{\Sigma}_1^{(t+1)} \mathbf{B}_1^* - \mathbf{U}_1^{(t+1)} \rangle_F \\
&= \langle \mathbf{B}_1^* - \widehat{\mathbf{B}}_1^{(t+1)}, (\widehat{\mathbf{\Sigma}}_1^{(t+1)} - \mathbf{\Sigma}_1^{(t+1)}) \mathbf{B}_1^* - (\widehat{\mathbf{U}}_1^{(t+1)} - \mathbf{U}_1^{(t+1)}) \rangle_F + \\
&\quad \langle \mathbf{B}_1^* - \widehat{\mathbf{B}}_1^{(t+1)}, (\mathbf{\Sigma}_1^{(t+1)} - \mathbf{\Sigma}_1^*) \mathbf{B}_1^* + \mathbf{\Sigma}_1^* \mathbf{B}_1^* - \mathbf{U}_1^* - (\mathbf{U}_1^{(t+1)} - \mathbf{U}_1^*) \rangle_F \\
&= \underbrace{\langle \mathbf{B}_1^* - \widehat{\mathbf{B}}_1^{(t+1)}, (\widehat{\mathbf{\Sigma}}_1^{(t+1)} - \mathbf{\Sigma}_1^{(t+1)}) \mathbf{B}_1^* - (\widehat{\mathbf{U}}_1^{(t+1)} - \mathbf{U}_1^{(t+1)}) \rangle_F}_{(i)} + \\
&\quad \underbrace{\langle \mathbf{B}_1^* - \widehat{\mathbf{B}}_1^{(t+1)}, (\mathbf{\Sigma}_1^{(t+1)} - \mathbf{\Sigma}_1^*) \mathbf{B}_1^* - (\mathbf{U}_1^{(t+1)} - \mathbf{U}_1^*) \rangle_F}_{(ii)} + \\
&\quad \underbrace{\langle \mathbf{B}_1^* - \widehat{\mathbf{B}}_1^{(t+1)}, \mathbf{\Sigma}_1^* \mathbf{B}_1^* - \mathbf{U}_1^* \rangle_F}_{=0}.
\end{aligned}
$$

Let $\mathbf{u}^{(t+1)} = \widehat{\mathbf{B}}_1^{(t+1)} - \mathbf{B}_1^*$. For a matrix $\mathbf{A} \in \mathbb{R}^{p \times q}$ and a set $\mathcal{S} \in [p]$, $\mathbf{A}_{\mathcal{S}} \in \mathbb{R}^{|\mathcal{S}| \times q}$ is the submatrix where rows are in set $\mathcal{S}$. Occasionally, we use the same notation $\mathbf{A}_{\mathcal{S}}$ to represent its 0-extended version $\mathbf{A}' \in \mathbb{R}^{p \times q}$ such that $\mathbf{A}'_{\mathcal{S}^c} = 0$ and $\mathbf{A}'_{\mathcal{S}} = \mathbf{A}_{\mathcal{S}}$. Then

$$
\begin{aligned}
\|\widehat{\mathbf{B}}_1^{(t+1)}\|_{2,1} - \|\mathbf{B}_1^*\|_{2,1} &= \|\mathbf{B}_1^* + \mathbf{u}^{(t+1)}\|_{2,1} - \|\mathbf{B}_1^*\|_{2,1} \\
&= \|\mathbf{B}_1^* + \mathbf{u}_{\mathcal{S}}^{(t+1)} + \mathbf{u}_{\mathcal{S}^c}^{(t+1)}\|_{2,1} - \|\mathbf{B}_1^*\|_{2,1} \\
&\geq \|\mathbf{B}_1^* + \mathbf{u}_{\mathcal{S}^c}^{(t+1)}\|_{2,1} - \|\mathbf{u}_{\mathcal{S}}^{(t+1)}\|_{2,1} - \|\mathbf{B}_1^*\|_{2,1} \\
&= \|\mathbf{B}_1^*\|_{2,1} + \|\mathbf{u}_{\mathcal{S}^c}^{(t+1)}\|_{2,1} - \|\mathbf{u}_{\mathcal{S}}^{(t+1)}\|_{2,1} - \|\mathbf{B}_1^*\|_{2,1} \\
&= \|\mathbf{u}_{\mathcal{S}^c}^{(t+1)}\|_{2,1} - \|\mathbf{u}_{\mathcal{S}}^{(t+1)}\|_{2,1}.
\end{aligned}
$$

Therefore

$$
\lambda^{(t+1)}(\|\mathbf{u}_{\mathcal{S}^c}^{(t+1)}\|_{2,1} - \|\mathbf{u}_{\mathcal{S}}^{(t+1)}\|_{2,1}) \leq (i) + (ii) + (iii).
$$

For a vector $\mathbf{x} \in \mathbb{R}^q$, we have $\|\mathbf{x}\|_2 \leq \|\mathbf{x}\|_1 \leq \sqrt{q}\|\mathbf{x}\|_2$. Then,

$$
\sum_{i=1}^p \sqrt{\sum_{j=1}^q u_{ij}^2} \leq \sum_{i=1}^p \sum_{j=1}^q |u_{ij}| \leq \sqrt{q} \sum_{i=1}^p \sqrt{\sum_{j=1}^q u_{ij}^2}.
$$

Let $\mathcal{S}_1 = \{\mathcal{S}, \mathcal{S} + p, \ldots, \mathcal{S} + qp\}$, where $\mathcal{S} + p$ means the set of indices in $\mathcal{S}$ adds $p$. Then

$$
\|\mathbf{u}_{\mathcal{S}^c}^{(t+1)}\|_{2,1} = \sum_{i \in \mathcal{S}^c} \sqrt{\sum_{j=1}^q (u_{ij}^{(t+1)})^2} \geq \frac{1}{\sqrt{q}} \sum_{i \in \mathcal{S}^c} \sum_{j=1}^q |u_{ij}^{(t+1)}| = \frac{1}{\sqrt{q}} \| \operatorname{vec}(\mathbf{u}^{(t+1)})_{\mathcal{S}_1^c} \|_1.
$$

And

$$
\|\mathbf{u}_{\mathcal{S}}^{(t+1)}\|_{2,1} = \sum_{i \in \mathcal{S}} \sqrt{\sum_{j=1}^q (u_{ij}^{(t+1)})^2} \leq \sum_{i \in \mathcal{S}} \sum_{j=1}^q |u_{ij}^{(t+1)}| = \| \operatorname{vec}(\mathbf{u}^{(t+1)})_{\mathcal{S}_1} \|_1.
$$

Thus, we have

$$
\lambda^{(t+1)}(\frac{1}{\sqrt{q}} \| \operatorname{vec}(\mathbf{u}^{(t+1)})_{\mathcal{S}_1^c} \|_1 - \| \operatorname{vec}(\mathbf{u}^{(t+1)})_{\mathcal{S}_1} \|_1) \leq \lambda^{(t+1)}(\|\mathbf{u}_{\mathcal{S}^c}^{(t+1)}\|_{2,1} - \|\mathbf{u}_{\mathcal{S}}^{(t+1)}\|_{2,1}) \leq (i) + (ii) + (iii).
$$

Recall that

$$(i) = \langle \mathbf{B}_1^* - \widehat{\mathbf{B}}_1^{(t+1)}, (\widehat{\boldsymbol{\Sigma}}_1^{(t+1)} - \boldsymbol{\Sigma}_1^{(t+1)})\mathbf{B}_1^* - (\widehat{\mathbf{U}}_1^{(t+1)} - \mathbf{U}_1^{(t+1)})\rangle_F$$
$$= \mathrm{vec}(\mathbf{B}_1^* - \widehat{\mathbf{B}}_1^{(t+1)})^T \mathrm{vec}((\widehat{\boldsymbol{\Sigma}}_1^{(t+1)} - \boldsymbol{\Sigma}_1^{(t+1)})\mathbf{B}_1^*) - \mathrm{vec}(\mathbf{B}_1^* - \widehat{\mathbf{B}}_1^{(t+1)})^T \mathrm{vec}(\widehat{\mathbf{U}}_1^{(t+1)} - \mathbf{U}_1^{(t+1)}).$$

We want to bound $\ell_2$ norm of $\mathrm{vec}((\widehat{\boldsymbol{\Sigma}}_1^{(t+1)} - \boldsymbol{\Sigma}_1^{(t+1)})\mathbf{B}_1^*)$ and $\mathrm{vec}(\widehat{\mathbf{U}}_1^{(t+1)} - \mathbf{U}_1^{(t+1)})$. Let $\widetilde{\mathcal{S}}_1$ be a set of size $3sq$, which contains $\mathcal{S}_1$ and the largest $sq$ coefficients of $\widehat{\mathbf{U}}_1^{(t+1)} - \mathbf{U}_1^{(t+1)}$ and the largest $sq$ coefficients of $(\widehat{\boldsymbol{\Sigma}}_1^{(t+1)} - \boldsymbol{\Sigma}_1^{(t+1)})\mathbf{B}_1^*$. We have

$$| \mathrm{vec}(\mathbf{B}_1^* - \widehat{\mathbf{B}}_1^{(t+1)})^T \mathrm{vec}(\widehat{\mathbf{U}}_1^{(t+1)} - \mathbf{U}_1^{(t+1)})|$$
$$\leq | \mathrm{vec}((\widehat{\mathbf{U}}_1^{(t+1)} - \mathbf{U}_1^{(t+1)})_{\widetilde{\mathcal{S}}_1})^T \mathrm{vec}((\mathbf{B}_1^* - \widehat{\mathbf{B}}_1^{(t+1)})_{\widetilde{\mathcal{S}}_1})|+$$
$$| \mathrm{vec}((\widehat{\mathbf{U}}_1^{(t+1)} - \mathbf{U}_1^{(t+1)})_{\widetilde{\mathcal{S}}_1^c})^T \mathrm{vec}((\mathbf{B}_1^* - \widehat{\mathbf{B}}_1^{(t+1)})_{\widetilde{\mathcal{S}}_1^c})|$$
$$\leq \| \mathrm{vec}((\widehat{\mathbf{U}}_1^{(t+1)} - \mathbf{U}_1^{(t+1)})_{\widetilde{\mathcal{S}}_1})\|_2 \cdot \| \mathrm{vec}((\mathbf{B}_1^* - \widehat{\mathbf{B}}_1^{(t+1)})_{\widetilde{\mathcal{S}}_1})\|_2+$$
$$\| \mathrm{vec}((\widehat{\mathbf{U}}_1^{(t+1)} - \mathbf{U}_1^{(t+1)})_{\widetilde{\mathcal{S}}_1^c})\|_\infty \cdot \| \mathrm{vec}((\mathbf{B}_1^* - \widehat{\mathbf{B}}_1^{(t+1)})_{\widetilde{\mathcal{S}}_1^c})\|_1.$$

By the definition, we have

$$\|\mathbf{A}\|_{F,s} = \sup_{\substack{\boldsymbol{\mu} \in \mathbb{R}^{p \times q} \\ \mathrm{vec}(\boldsymbol{\mu}) \in \mathcal{L}(s) \cap \mathbb{S}^{pq-1}}} \langle \mathrm{vec}(A), \mathrm{vec}(\boldsymbol{\mu})\rangle.$$

Let $\mathbf{v} \in \mathbb{R}^{pq}$ such that $\mathbf{v}_{\widetilde{\mathcal{S}}_1} = \mathrm{vec}(\mathbf{A})_{\widetilde{\mathcal{S}}_1}$ and $\mathbf{v}_{\widetilde{\mathcal{S}}_1^c} = 0$. To define $\mathcal{L}(s)$, we want vectors $\mathbf{u}$ that $\|\mathbf{u}_{\widetilde{\mathcal{S}}_1^c}\|_1$ is bounded. Clearly $\mathbf{v} \in \mathcal{L}(s)$ since $\|\mathbf{v}_{\widetilde{\mathcal{S}}_1^c}\|_1 = 0$. Then

$$\|\mathbf{A}\|_{F,s} \geq \frac{1}{\|\mathbf{v}\|_2}\langle \mathrm{vec}(\mathbf{A}), \mathbf{v}\rangle = \frac{1}{\|\mathbf{v}\|_2}\|\mathbf{v}\|_2^2 = \|\mathbf{v}\|_2 = \| \mathrm{vec}(\mathbf{A})_{\widetilde{\mathcal{S}}_1}\|_2.$$

Thus, $\| \mathrm{vec}((\widehat{\mathbf{U}}_1^{(t+1)} - \mathbf{U}_1^{(t+1)})_{\widetilde{\mathcal{S}}_1})\|_2 \leq \|\widehat{\mathbf{U}}_1^{(t+1)} - \mathbf{U}_1^{(t+1)})_{\widetilde{\mathcal{S}}_1}\|_{F,s}$. By the definition of $\widetilde{\mathcal{S}}_1$, $\| \mathrm{vec}((\widehat{\mathbf{U}}_1^{(t+1)} - \mathbf{U}_1^{(t+1)})_{\widetilde{\mathcal{S}}_1^c})\|_\infty \leq \| \mathrm{vec}((\widehat{\mathbf{U}}_1^{(t+1)} - \mathbf{U}_1^{(t+1)})_{\widetilde{\mathcal{S}}_1})\|_2/\sqrt{sq} \leq \|\widehat{\mathbf{U}}_1^{(t+1)} - \mathbf{U}_1^{(t+1)})_{\widetilde{\mathcal{S}}_1}\|_{F,s}/\sqrt{sq}$. Therefore,

$$| \mathrm{vec}(\mathbf{B}_1^* - \widehat{\mathbf{B}}_1^{(t+1)})^T \mathrm{vec}(\widehat{\mathbf{U}}_1^{(t+1)} - \mathbf{U}_1^{(t+1)})|$$
$$\leq \|\widehat{\mathbf{U}}_1^{(t+1)} - \mathbf{U}_1^{(t+1)}\|_{F,s} \cdot \| \mathrm{vec}(\mathbf{u}^{(t+1)})_{\widetilde{\mathcal{S}}_1}\|_2 + \frac{1}{\sqrt{sq}}\|\widehat{\mathbf{U}}_1^{(t+1)} - \mathbf{U}_1^{(t+1)}\|_{F,s} \cdot \| \mathrm{vec}(\mathbf{u}^{(t+1)})_{\widetilde{\mathcal{S}}_1^c}\|_1.$$

Similarly, we have

$$\mathrm{vec}(\mathbf{B}_1^* - \widehat{\mathbf{B}}_1^{(t+1)})^T \mathrm{vec}((\widehat{\boldsymbol{\Sigma}}_1^{(t+1)} - \boldsymbol{\Sigma}_1^{(t+1)})\mathbf{B}_1^*)$$
$$\leq \| \mathrm{vec}((\widehat{\boldsymbol{\Sigma}}_1^{(t+1)} - \boldsymbol{\Sigma}_1^{(t+1)})\mathbf{B}_1^*)_{\widetilde{\mathcal{S}}_1}\|_2 \cdot \| \mathrm{vec}((\mathbf{B}_1^* - \widehat{\mathbf{B}}_1^{(t+1)})_{\widetilde{\mathcal{S}}_1})\|_2+$$
$$\| \mathrm{vec}((\widehat{\boldsymbol{\Sigma}}_1^{(t+1)} - \boldsymbol{\Sigma}_1^{(t+1)})\mathbf{B}_1^*)_{\widetilde{\mathcal{S}}_1^c}\|_\infty \cdot \| \mathrm{vec}((\mathbf{B}_1^* - \widehat{\mathbf{B}}_1^{(t+1)})_{\widetilde{\mathcal{S}}_1^c})\|_1.$$
$$\leq \|(\widehat{\boldsymbol{\Sigma}}_1^{(t+1)} - \boldsymbol{\Sigma}_1^{(t+1)})\mathbf{B}_1^*\|_{F,s} \cdot \| \mathrm{vec}(\mathbf{u}^{(t+1)})_{\widetilde{\mathcal{S}}_1}\|_2 + \frac{1}{\sqrt{sq}}\|(\widehat{\boldsymbol{\Sigma}}_1^{(t+1)} - \boldsymbol{\Sigma}_1^{(t+1)})\mathbf{B}_1^*\|_{F,s} \cdot \| \mathrm{vec}(\mathbf{u}^{(t+1)})_{\widetilde{\mathcal{S}}_1^c}\|_1.$$

Therefore,

$$|(i)| \leq 2C_{\mathrm{con}}\sqrt{\frac{sq^3(\log n)^2\log p}{n}}\| \mathrm{vec}(\mathbf{u}^{(t+1)})_{\widetilde{\mathcal{S}}_1}\|_2 + 2C_{\mathrm{con}}\sqrt{\frac{sq^3(\log n)^2\log p}{n}}\frac{1}{\sqrt{sq}}\| \mathrm{vec}(\mathbf{u}^{(t+1)})_{\widetilde{\mathcal{S}}_1^c}\|_1.$$

Using Lemma C.1, we have

$$|(ii)| \leq (\|(\boldsymbol{\Sigma}_1^{(t+1)} - \boldsymbol{\Sigma}_1^*)\mathbf{B}_1^*\|_F + \|\mathbf{U}_1^{(t+1)} - \mathbf{U}_1^*\|_F)\|\mathbf{B}_1^* - \widehat{\mathbf{B}}_1^{(t+1)}\|_F$$
$$= (\|(\boldsymbol{\Sigma}_1(\widehat{\boldsymbol{\theta}}^{(t)}) - \boldsymbol{\Sigma}_1^*)\mathbf{B}_1^*\|_F + \|\mathbf{U}_1(\widehat{\boldsymbol{\theta}}^{(t)}) - \mathbf{U}_1^*\|_F)\|\mathbf{B}_1^* - \widehat{\mathbf{B}}_1^{(t+1)}\|_F$$
$$\leq 2\kappa_0 \frac{d_F(\widehat{\boldsymbol{\theta}}^{(t)}, \boldsymbol{\theta}^*) \vee \|\widehat{\mathbf{B}}_1^{(t)} - \mathbf{B}_1^*\|_F}{\sqrt{sq}}\sqrt{sq}\|\mathbf{B}_1^* - \widehat{\mathbf{B}}_1^{(t+1)}\|_F$$
$$= 2\kappa_0 \frac{d_F(\widehat{\boldsymbol{\theta}}^{(t)}, \boldsymbol{\theta}^*) \vee \|\widehat{\mathbf{B}}_1^{(t)} - \mathbf{B}_1^*\|_F}{\sqrt{sq}}\sqrt{sq}\| \mathrm{vec}(\mathbf{u}^{(t+1)})\|_2.$$

Combine the bound for terms $(i)$ and $(ii)$, we have

$$\lambda^{(t+1)}(\frac{1}{\sqrt{q}}\|\operatorname{vec}(\mathbf{u}^{(t+1)})_{\mathcal{S}_1^c}\|_1 - \|\operatorname{vec}(\mathbf{u}^{(t+1)})_{\mathcal{S}_1}\|_1)$$

$$\leq 2C_{\mathrm{con}}\sqrt{\frac{sq^3(\log n)^2\log p}{n}}\frac{1}{\sqrt{sq}}\sqrt{sq}\|\operatorname{vec}(\mathbf{u}^{(t+1)})_{\widetilde{\mathcal{S}}_1}\|_2+$$

$$2C_{\mathrm{con}}\sqrt{\frac{sq^3(\log n)^2\log p}{n}}\frac{1}{\sqrt{sq}}\|\operatorname{vec}(\mathbf{u}^{(t+1)})_{\widetilde{\mathcal{S}}_1^c}\|_1+$$

$$2\kappa_0\frac{d_F(\widehat{\boldsymbol{\theta}}^{(t)},\boldsymbol{\theta}^*)\vee\|\widehat{\mathbf{B}}_1^{(t)}-\mathbf{B}_1^*\|_F}{\sqrt{sq}}\sqrt{sq}\|\operatorname{vec}(\mathbf{u}^{(t+1)})\|_2.$$

Let

$$\lambda^{(t+1)}\geq 2\sqrt{q}\left(2C_{\mathrm{con}}\sqrt{\frac{q^2(\log n)^2\log p}{n}}+2\kappa_0\frac{d_F(\widehat{\boldsymbol{\theta}}^{(t)},\boldsymbol{\theta}^*)\vee\|\widehat{\mathbf{B}}_1^{(t)}-\mathbf{B}_1^*\|_F}{\sqrt{sq}}\right).$$

Then

$$\frac{1}{\sqrt{q}}\|\operatorname{vec}(\mathbf{u}^{(t+1)})_{\mathcal{S}_1^c}\|_1 - \|\operatorname{vec}(\mathbf{u}^{(t+1)})_{\mathcal{S}_1}\|_1$$

$$\leq \frac{\sqrt{sq}}{2\sqrt{q}}\|\operatorname{vec}(\mathbf{u}^{(t+1)})_{\widetilde{\mathcal{S}}_1}\|_2 + \frac{1}{2\sqrt{q}}\|\operatorname{vec}(\mathbf{u}^{(t+1)})_{\widetilde{\mathcal{S}}_1^c}\|_1 + \frac{\sqrt{sq}}{2\sqrt{q}}\|\operatorname{vec}(\mathbf{u}^{(t+1)})\|_2$$

$$= \frac{\sqrt{s}}{2}\|\operatorname{vec}(\mathbf{u}^{(t+1)})_{\widetilde{\mathcal{S}}_1}\|_2 + \frac{1}{2\sqrt{q}}\|\operatorname{vec}(\mathbf{u}^{(t+1)})_{\widetilde{\mathcal{S}}_1^c}\|_1 + \frac{\sqrt{s}}{2}\|\operatorname{vec}(\mathbf{u}^{(t+1)})\|_2,$$

which implies

$$\frac{1}{2\sqrt{q}}\|\operatorname{vec}(\mathbf{u}^{(t+1)})_{\widetilde{\mathcal{S}}_1^c}\|_1 \leq \frac{\sqrt{s}}{2}\|\operatorname{vec}(\mathbf{u}^{(t+1)})_{\widetilde{\mathcal{S}}_1}\|_2 + \frac{\sqrt{s}}{2}\|\operatorname{vec}(\mathbf{u}^{(t+1)})\|_2 + \|\operatorname{vec}(\mathbf{u}^{(t+1)})_{\widetilde{\mathcal{S}}_1}\|_1,$$

where we use $\mathcal{S}_1\subset\widetilde{\mathcal{S}}_1$. Using $\|\operatorname{vec}(\mathbf{u}^{(t+1)})_{\widetilde{\mathcal{S}}_1}\|_1 \leq \sqrt{3sq}\|\operatorname{vec}(\mathbf{u}^{(t+1)})_{\widetilde{\mathcal{S}}_1}\|_2$, we have

$$\|\operatorname{vec}(\mathbf{u}^{(t+1)})_{\widetilde{\mathcal{S}}_1^c}\|_1 \leq (\sqrt{sq}+2q\sqrt{3s})\|\operatorname{vec}(\mathbf{u}^{(t+1)})_{\widetilde{\mathcal{S}}_1}\|_2 + \sqrt{sq}\|\operatorname{vec}(\mathbf{u}^{(t+1)})\|_2.$$

Now, we focus on the second result. Let $w=1$. Recall that

$$\lambda^{(t+1)}(\|\widehat{\mathbf{B}}_1^{(t+1)}\|_{2,1} - \|\mathbf{B}_1^*\|_{2,1})$$

$$\leq \operatorname{tr}((\mathbf{B}_1^* - \widehat{\mathbf{B}}_1^{(t+1)})^T\widehat{\boldsymbol{\Sigma}}_1^{(t+1)}\mathbf{B}_1^*) - \frac{1}{2}\operatorname{tr}((\mathbf{B}_1^* - \widehat{\mathbf{B}}_1^{(t+1)})^T\widehat{\boldsymbol{\Sigma}}_1^{(t+1)}(\mathbf{B}_1^* - \widehat{\mathbf{B}}_1^{(t+1)})) -$$

$$\operatorname{tr}((\widehat{\mathbf{U}}_1^{(t+1)})^T(\mathbf{B}_1^* - \widehat{\mathbf{B}}_1^{(t+1)}))$$

$$= \langle\mathbf{B}_1^* - \widehat{\mathbf{B}}_1^{(t+1)}, \widehat{\boldsymbol{\Sigma}}_1^{(t+1)}\mathbf{B}_1^*\rangle_F - \frac{1}{2}\langle\mathbf{B}_1^* - \widehat{\mathbf{B}}_1^{(t+1)}, \widehat{\boldsymbol{\Sigma}}_1^{(t+1)}(\mathbf{B}_1^* - \widehat{\mathbf{B}}_1^{(t+1)})\rangle_F - \langle\widehat{\mathbf{U}}_1^{(t+1)}, \mathbf{B}_1^* - \widehat{\mathbf{B}}_1^{(t+1)}\rangle_F.$$

It follows that

$$|\langle\mathbf{B}_1^* - \widehat{\mathbf{B}}_1^{(t+1)}, \widehat{\boldsymbol{\Sigma}}_1^{(t+1)}(\mathbf{B}_1^* - \widehat{\mathbf{B}}_1^{(t+1)})\rangle_F| \leq 2\underbrace{|\langle\mathbf{B}_1^* - \widehat{\mathbf{B}}_1^{(t+1)}, \widehat{\boldsymbol{\Sigma}}_1^{(t+1)}\mathbf{B}_1^* - \widehat{\mathbf{U}}_1^{(t+1)}\rangle_F|}_{(I)} +$$

$$2\lambda^{(t+1)}(\underbrace{\left|\|\widehat{\mathbf{B}}_1^{(t+1)}\|_{2,1} - \|\mathbf{B}_1^*\|_{2,1}\right|}_{(II)}). \tag{9}$$

Recall that $\boldsymbol{\Sigma}_1^* \mathbf{B}_1^* - \mathbf{U}_1^* = 0$. For term $(I)$, since $\mathrm{vec}(\mathbf{B}_1^* - \widehat{\mathbf{B}}_1^{(t+1)})/\|\mathrm{vec}(\mathbf{B}_1^* - \widehat{\mathbf{B}}_1^{(t+1)})\|_2 \in \mathcal{L}(s) \cap \mathbb{S}^{pq-1}$, we have

$$
\begin{aligned}
&|\langle \mathbf{B}_1^* - \widehat{\mathbf{B}}_1^{(t+1)}, \widehat{\boldsymbol{\Sigma}}_1^{(t+1)}\mathbf{B}_1^* - \widehat{\mathbf{U}}_1^{(t+1)}\rangle_F| \\
&= |\langle \mathbf{B}_1^* - \widehat{\mathbf{B}}_1^{(t+1)}, \widehat{\boldsymbol{\Sigma}}_1^{(t+1)}\mathbf{B}_1^* - \boldsymbol{\Sigma}_1^*\mathbf{B}_1^* + \boldsymbol{\Sigma}_1^*\mathbf{B}_1^* - \mathbf{U}_1^* + \mathbf{U}_1^* - \widehat{\mathbf{U}}_1^{(t+1)}\rangle_F| \\
&= |\langle \mathbf{B}_1^* - \widehat{\mathbf{B}}_1^{(t+1)}, \widehat{\boldsymbol{\Sigma}}_1^{(t+1)}\mathbf{B}_1^* - \boldsymbol{\Sigma}_1^*\mathbf{B}_1^*\rangle_F + \langle \mathbf{B}_1^* - \widehat{\mathbf{B}}_1^{(t+1)}, \mathbf{U}_1^* - \widehat{\mathbf{U}}_1^{(t+1)}\rangle_F + \langle \mathbf{B}_1^* - \widehat{\mathbf{B}}_1^{(t+1)}, \boldsymbol{\Sigma}_1^*\mathbf{B}_1^* - \mathbf{U}_1^*\rangle_F| \\
&\leq \|\mathbf{B}_1^* - \widehat{\mathbf{B}}_1^{(t+1)}\|_F(\|\widehat{\boldsymbol{\Sigma}}_1^{(t+1)}\mathbf{B}_1^* - \boldsymbol{\Sigma}_1^*\mathbf{B}_1^*\|_{F,s} + \|\widehat{\mathbf{U}}_1^{(t+1)} - \mathbf{U}_1^*\|_{F,s}) \\
&\leq 2\|\mathbf{B}_1^* - \widehat{\mathbf{B}}_1^{(t+1)}\|_F d_{F,s}(M_n(\widehat{\boldsymbol{\theta}}^{(t)}), M(\boldsymbol{\theta}^*)),
\end{aligned}
$$

where we use the definitions of $\|\cdot\|_{F,s}$ norm and $d_{F,s}$. In the last inequality, we use the fact

$$
M_n(\widehat{\boldsymbol{\theta}}^{(t)}) = \{\widehat{\pi}_w(\widehat{\boldsymbol{\theta}}^{(t)}), \widehat{\mathbf{U}}_w(\widehat{\boldsymbol{\theta}}^{(t)}), \widehat{\boldsymbol{\Sigma}}_w(\widehat{\boldsymbol{\theta}}^{(t)}), w = 1, 2\} = \{\widehat{\pi}_w^{(t+1)}, \widehat{\mathbf{U}}_w^{(t+1)}, \widehat{\boldsymbol{\Sigma}}_w^{(t+1)}, w = 1, 2\}.
$$

For term $(II)$, using reverse triangle inequality,

$$
\begin{aligned}
\left|\|\widehat{\mathbf{B}}_1^{(t+1)}\|_{2,1} - \|\mathbf{B}_1^*\|_{2,1}\right| &= \left|\sum_{j=1}^p \sqrt{\sum_{k=1}^q (\widehat{\mathbf{B}}_{1,jk}^{(t+1)})^2} - \sum_{j=1}^p \sqrt{\sum_{k=1}^q (\mathbf{B}_{1,jk}^*)^2}\right| \\
&\leq \sum_{j=1}^p \sqrt{\sum_{k=1}^q (\widehat{\mathbf{B}}_{1,jk}^{(t+1)} - \mathbf{B}_{1,jk}^*)^2} = \sum_{j=1}^p \sum_{k=1}^q |\widehat{\mathbf{B}}_{1,jk}^{(t+1)} - \mathbf{B}_{1,jk}^*| \\
&\leq \|\mathrm{vec}(\widehat{\mathbf{B}}_1^{(t+1)} - \mathbf{B}_1^*)\|_1 \\
&= \|\mathrm{vec}(\widehat{\mathbf{B}}_1^{(t+1)} - \mathbf{B}_1^*)_{\widetilde{\mathcal{S}}_1}\|_1 + \|\mathrm{vec}(\widehat{\mathbf{B}}_1^{(t+1)} - \mathbf{B}_1^*)_{\widetilde{\mathcal{S}}_1^c}\|_1 \\
&\leq \sqrt{3sq}\|\mathrm{vec}(\widehat{\mathbf{B}}_1^{(t+1)} - \mathbf{B}_1^*)_{\widetilde{\mathcal{S}}_1}\|_2 + (\sqrt{sq} + 2\sqrt{3sq^2})\|\mathrm{vec}(\widehat{\mathbf{B}}_1^{(t+1)} - \mathbf{B}_1^*)_{\widetilde{\mathcal{S}}_1}\|_2 + \\
&\quad \sqrt{sq}\|\mathrm{vec}(\widehat{\mathbf{B}}_1^{(t+1)} - \mathbf{B}_1^*)\|_2 \\
&\leq (\sqrt{3sq} + 2\sqrt{sq} + 2\sqrt{3sq^2})\|\widehat{\mathbf{B}}_1^{(t+1)} - \mathbf{B}_1^*\|_F.
\end{aligned}
$$

For the right hand side of (9), we have

$$
\begin{aligned}
|\langle \mathbf{B}_1^* - \widehat{\mathbf{B}}_1^{(t+1)}, \widehat{\boldsymbol{\Sigma}}_1^{(t+1)}(\mathbf{B}_1^* - \widehat{\mathbf{B}}_1^{(t+1)})\rangle_F| &= |\mathrm{vec}(\mathbf{B}_1^* - \widehat{\mathbf{B}}_1^{(t+1)})^T \mathrm{vec}(\widehat{\boldsymbol{\Sigma}}_1^{(t+1)}(\mathbf{B}_1^* - \widehat{\mathbf{B}}_1^{(t+1)}))| \\
&= |\mathrm{vec}(\mathbf{B}_1^* - \widehat{\mathbf{B}}_1^{(t+1)})^T(\mathbf{I}_q \otimes \widehat{\boldsymbol{\Sigma}}_1^{(t+1)})\mathrm{vec}(\mathbf{B}_1^* - \widehat{\mathbf{B}}_1^{(t+1)})| \\
&= \left|\sum_{k=1}^q (\mathbf{B}_{1,k}^* - \widehat{\mathbf{B}}_{1,k}^{(t+1)})^T \widehat{\boldsymbol{\Sigma}}_1^{(t+1)}(\mathbf{B}_{1,k}^* - \widehat{\mathbf{B}}_{1,k}^{(t+1)})\right|,
\end{aligned}
$$

where $\mathbf{B}_{1,k}^*$ and $\widehat{\mathbf{B}}_{1,k}^{(t+1)}$ represent the $k$-th column of $\mathbf{B}_1^*$ and $\widehat{\mathbf{B}}_1^{(t+1)}$. Recall that

$$
\widehat{\boldsymbol{\Sigma}}_1^{(t+1)} = \frac{1}{n}\sum_{i=1}^n \gamma_{1,\widehat{\boldsymbol{\theta}}^{(t)}}(\mathbf{X}_i, Y_i)\mathbf{X}_i\mathbf{X}_i^T.
$$

By Lemma C.2, we know that

$$
|\frac{1}{n}\sum_{i=1}^n \gamma_{1,\widehat{\boldsymbol{\theta}}^{(t)}}(\mathbf{X}_i, Y_i) - \mathrm{E}[\widehat{\pi}_1^{(t+1)}]| = O(\sqrt{\frac{sq^3(\log n)^2 \log p}{n}}),
$$

with probability at least $1 - o(1)$. Thus, $\frac{1}{n}\sum_{i=1}^n \gamma_{1,\widehat{\boldsymbol{\theta}}^{(t)}}(\mathbf{X}_i, Y_i) > \tau_1$ for some positive constant $\tau_1$ with probability at

least $1 - o(1)$. Define the set $\mathcal{N} = \{i : \gamma_{1,\widehat{\boldsymbol{\theta}}^{(t)}}(\mathbf{X}_i, Y_i) > \tau_1/2\}$. Then by Lemma F.2,

$$(\mathbf{B}_{1,k}^* - \widehat{\mathbf{B}}_{1,k}^{(t+1)})^T \widehat{\boldsymbol{\Sigma}}_1^{(t+1)} (\mathbf{B}_{1,k}^* - \widehat{\mathbf{B}}_{1,k}^{(t+1)})$$

$$\geq \|\mathbf{B}_{1,k}^* - \widehat{\mathbf{B}}_{1,k}^{(t+1)}\|_2^2 \inf_{\mathbf{u} \in \mathcal{L}_p(s) \cap \mathbb{S}^{p-1}} \mathbf{u}^T \frac{1}{n} \sum_{i=1}^n \gamma_{1,\widehat{\boldsymbol{\theta}}^{(t)}}(\mathbf{X}_i, Y_i) \mathbf{X}_i \mathbf{X}_i^T \mathbf{u}$$

$$\geq \|\mathbf{B}_{1,k}^* - \widehat{\mathbf{B}}_{1,k}^{(t+1)}\|_2^2 \inf_{\mathbf{u} \in \mathcal{L}_p(s) \cap \mathbb{S}^{p-1}} \mathbf{u}^T \frac{1}{n} \sum_{i \in \mathcal{N}} \gamma_{1,\widehat{\boldsymbol{\theta}}^{(t)}}(\mathbf{X}_i, Y_i) \mathbf{X}_i \mathbf{X}_i^T \mathbf{u}$$

$$\geq \|\mathbf{B}_{1,k}^* - \widehat{\mathbf{B}}_{1,k}^{(t+1)}\|_2^2 \cdot \tau_1/2 \cdot \tau.$$

Then

$$|\langle \mathbf{B}_1^* - \widehat{\mathbf{B}}_1^{(t+1)}, \widehat{\boldsymbol{\Sigma}}_1^{(t+1)}(\mathbf{B}_1^* - \widehat{\mathbf{B}}_1^{(t+1)})\rangle_F| \geq \sum_{k=1}^q \|\operatorname{vec}(\mathbf{B}_{1,k}^* - \widehat{\mathbf{B}}_{1,k}^{(t+1)})\|_2^2 \cdot \tau_1/2 \cdot \tau = \|\operatorname{vec}(\widehat{\mathbf{B}}_1^{(t+1)} - \mathbf{B}_1^*)\|_2^2 \tau_0,$$

where $\tau_0 = \tau_1/2 \cdot \tau$. Combing the above results, we have

$$\tau_0 \|\widehat{\mathbf{B}}_1^{(t+1)} - \mathbf{B}_1^*\|_F^2 \leq 4\|\widehat{\mathbf{B}}_1^{(t+1)} - \mathbf{B}_1^*\|_F d_{F,s}(M_n(\boldsymbol{\theta}^{(t)}), M(\boldsymbol{\theta}^*)) + \\ 2\lambda^{(t+1)}(\sqrt{3sq} + 2\sqrt{sq} + 2\sqrt{3sq^2})\|\widehat{\mathbf{B}}_1^{(t+1)} - \mathbf{B}_1^*\|_F.$$

Hence,

$$\|\widehat{\mathbf{B}}_1^{(t+1)} - \mathbf{B}_1^*\|_F \leq \frac{4}{\tau_0} d_{F,s}(M_n(\widehat{\boldsymbol{\theta}}^{(t)}), M(\boldsymbol{\theta}^*)) + \frac{2}{\tau_0} \lambda^{(t+1)}(\sqrt{3sq} + 2\sqrt{sq} + 2\sqrt{3sq^2}).$$

$\square$

## C.2. Proof of Theorem

**Theorem C.4.** *Under conditions (C1)-(C5), there exists a constant $0 < \kappa < 1/2$, such that $\widehat{\mathbf{B}}_w^{(t)}$ satisfies, with probability at least $1 - o(1)$,*

$$\|\widehat{\mathbf{B}}_w^{(t)} - \mathbf{B}_w^*\|_F = O\left(\kappa^t(d_F(\widehat{\boldsymbol{\theta}}^{(0)}, \boldsymbol{\theta}^*) \vee \|\widehat{\mathbf{B}}_1^{(0)} - \mathbf{B}_1^*\|_F \vee \|\widehat{\mathbf{B}}_2^{(0)} - \mathbf{B}_2^*\|_F) + \sqrt{\frac{sq^3(\log n)^2 \log p}{n}}\right).$$

*Consequently, for $t \geq (-\log \kappa)^{-1}\log\{n(d_F(\widehat{\boldsymbol{\theta}}^{(0)}, \boldsymbol{\theta}^*) \vee \|\widehat{\mathbf{B}}_1^{(0)} - \mathbf{B}_1^*\|_F \vee \|\widehat{\mathbf{B}}_2^{(0)} - \mathbf{B}_2^*\|_F)\}$,*

$$\|\widehat{\mathbf{B}}_w^{(t)} - \mathbf{B}_w^*\|_F, \|\mathcal{D}(\mathcal{S}_{\widehat{\boldsymbol{\beta}}_w^{(t)}}, \mathcal{S}_{\boldsymbol{\beta}_w^*})\|_F = O\left(\sqrt{\frac{sq^3(\log n)^2 \log p}{n}}\right).$$

*Proof.* We update $\lambda^{(t)}$ by

$$\lambda^{(t)} = \kappa \lambda^{(t-1)} + C_\lambda \sqrt{\frac{q^3(\log n)^2 \log p}{n}},$$

$$\lambda^{(0)} = C_1 \frac{d_F(\widehat{\boldsymbol{\theta}}^{(0)}, \boldsymbol{\theta}^*) \vee \|\widehat{\mathbf{B}}_1^{(0)} - \mathbf{B}_1^*\|_F \vee \|\widehat{\mathbf{B}}_2^{(0)} - \mathbf{B}_2^*\|_F}{\sqrt{s}} + C_\lambda \sqrt{\frac{q^3(\log n)^2 \log p}{n}},$$

where $C_1 = \tau_0/(32C_0 q)$. Thus, we have

$$\lambda^{(t)} = \kappa^t C_1 \frac{d_F(\widehat{\boldsymbol{\theta}}^{(0)}, \boldsymbol{\theta}^*) \vee \|\widehat{\mathbf{B}}_1^{(0)} - \mathbf{B}_1^*\|_F \vee \|\widehat{\mathbf{B}}_2^{(0)} - \mathbf{B}_2^*\|_F}{\sqrt{s}} + \frac{1 - \kappa^{t+1}}{1 - \kappa} C_\lambda \sqrt{\frac{q^3(\log n)^2 \log p}{n}}.$$

Let $\kappa = (1 \vee \frac{128C_0 q}{\tau_0} \vee 4C_0)\kappa_0$, since $0 < \kappa_0 < \frac{1}{2\vee(256C_0 q/\tau_0)\vee 8C_0}$, we have $0 < \kappa < 1/2$. Then define

$$C^* = \left[\left(\frac{2\kappa^2 - 4\kappa + 2}{2\kappa^2 - 5\kappa + 2} \cdot (C_0 + \frac{4C_0 q}{\tau_0} + \frac{64C_0 q}{\tau_0(1 - \kappa)})\right) \vee \frac{1 - \kappa}{1 - 2\kappa}\right] C_{\text{con}},$$

$$C_\lambda = 4C_{\text{con}} + \frac{4\kappa_0}{1 - \kappa} C^*.$$

We claim

(i) $\kappa \geq \kappa_0$, $C_1\kappa \geq 4\kappa_0$, and $(\frac{4C_0q}{\tau_0} + C_0)\kappa_0 + \frac{16C_0q}{\tau_0}\kappa C_1 \leq \kappa$.

(ii) $\frac{\kappa_0}{1-\kappa}C^* + C_{\text{con}} \leq C^*$, $(\frac{4C_0q}{\tau_0} + C_0)C_{\text{con}} + \frac{16C_0q}{\tau_0(1-\kappa)}C_\lambda \leq C^*$.

The first two inequalities in $(I)$ can be seen from the definition of $\kappa$ and $C_1$. For the third, we have

$$(\frac{4C_0q}{\tau_0} + C_0)\kappa_0 + \frac{16C_0q}{\tau_0}\kappa C_1 = (\frac{4C_0q}{\tau_0} + C_0)\kappa_0 + \frac{1}{2}\kappa \leq \frac{1}{32}\kappa + \frac{1}{4}\kappa + \frac{1}{2}\kappa \leq \kappa.$$

For the first inequality in $(ii)$, it is equivalent to $C_{\text{con}} \leq \frac{1-\kappa-\kappa_0}{1-\kappa}C^*$. Since $\kappa_0 \leq \kappa$,

$$\frac{1-\kappa-\kappa_0}{1-\kappa}C^* \geq \frac{1-2\kappa}{1-\kappa}C^* \geq C_{\text{con}},$$

where in the last inequality we use the definition of $C^*$. For the second inequality in $(ii)$, we have

$$(\frac{4C_0q}{\tau_0} + C_0)C_{\text{con}} + \frac{16C_0q}{\tau_0(1-\kappa)}C_\lambda = (\frac{4C_0q}{\tau_0} + C_0)C_{\text{con}} + \frac{16C_0q}{\tau_0(1-\kappa)}[4C_{\text{con}} + \frac{4\kappa_0}{1-\kappa}C^*]$$

$$= \left[\frac{4C_0q}{\tau_0} + C_0 + \frac{64C_0q}{\tau_0(1-\kappa)}\right]C_{\text{con}} + \frac{64C_0q\kappa_0}{\tau_0(1-\kappa)^2}C^*.$$

Use the second inequality in $(i)$,

$$\frac{64C_0q\kappa_0}{\tau_0} = \frac{2\kappa_0}{C_1} \leq \frac{\kappa}{2}.$$

Then

$$(\frac{4C_0q}{\tau_0} + C_0)C_{\text{con}} + \frac{16C_0q}{\tau_0(1-\kappa)}C_\lambda \leq \left[\frac{4C_0q}{\tau_0} + C_0 + \frac{64C_0q}{\tau_0(1-\kappa)}\right]C_{\text{con}} + \frac{\kappa}{2(1-\kappa)^2}C^*$$

$$\leq \frac{2\kappa^2 - 5\kappa + 2}{2\kappa^2 - 4\kappa + 2}C^* + \frac{\kappa}{2(1-\kappa)^2}C^* = C^*.$$

Next, we use induction to show the following results

$$\lambda^{(t+1)} \geq 4C_{\text{con}}\sqrt{\frac{q^3(\log n)^2 \log p}{n}} + 4\kappa_0(\frac{d_F(\widehat{\boldsymbol{\theta}}^{(t)}, \boldsymbol{\theta}^*) \vee \|\widehat{\mathbf{B}}_1^{(t)} - \mathbf{B}_1^*\|_F \vee \|\widehat{\mathbf{B}}_1^{(t)} - \mathbf{B}_2^*\|_F}{\sqrt{s}}),$$

$$d_F(\widehat{\boldsymbol{\theta}}^{(t+1)}, \boldsymbol{\theta}^*) \vee \|\widehat{\mathbf{B}}_1^{(t+1)} - \mathbf{B}_1^*\|_F \vee \|\widehat{\mathbf{B}}_1^{(t+1)} - \mathbf{B}_2^*\|_F \leq$$

$$\kappa^{t+1}(d_F(\widehat{\boldsymbol{\theta}}^{(0)}, \boldsymbol{\theta}^*) \vee \|\widehat{\mathbf{B}}_1^{(0)} - \mathbf{B}_1^*\|_F \vee \|\widehat{\mathbf{B}}_1^{(0)} - \mathbf{B}_2^*\|_F) + \frac{1-\kappa^{t+2}}{1-\kappa}C^*\sqrt{\frac{sq^3(\log n)^2 \log p}{n}},$$

$$d_F(\widehat{\boldsymbol{\theta}}^{(t+1)}, \boldsymbol{\theta}^*) \vee \|\widehat{\mathbf{B}}_1^{(t+1)} - \mathbf{B}_1^*\|_F \vee \|\widehat{\mathbf{B}}_1^{(t+1)} - \mathbf{B}_2^*\|_F \leq r\Omega, \text{vec}(\widehat{\boldsymbol{\Gamma}}_w^{(t+1)} - \boldsymbol{\Gamma}_w^*) \in \mathcal{L}(s).$$

It is easy to verify that $d_{F,s}$ satisfies the triangle inequality. Then using Lemma C.1 and C.2

$$d_{F,s}(M_n(\widehat{\boldsymbol{\theta}}^{(0)}), M(\boldsymbol{\theta}^*)) \leq d_{F,s}(M(\widehat{\boldsymbol{\theta}}^{(0)}), M(\boldsymbol{\theta}^*)) + d_{F,s}(M_n(\widehat{\boldsymbol{\theta}}^{(0)}), M(\widehat{\boldsymbol{\theta}}^{(0)}))$$

$$\leq \kappa_0(d_F(\widehat{\boldsymbol{\theta}}^{(0)}, \boldsymbol{\theta}^*) \vee \|\widehat{\mathbf{B}}_1^{(0)} - \mathbf{B}_1^*\|_F \vee \|\widehat{\mathbf{B}}_1^{(0)} - \mathbf{B}_2^*\|_F) +$$

$$C_{\text{con}}\sqrt{\frac{sq^3(\log n)^2 \log p}{n}}$$

$$\leq \kappa(d_F(\widehat{\boldsymbol{\theta}}^{(0)}, \boldsymbol{\theta}^*) \vee \|\widehat{\mathbf{B}}_1^{(0)} - \mathbf{B}_1^*\|_F \vee \|\widehat{\mathbf{B}}_1^{(0)} - \mathbf{B}_2^*\|_F) +$$

$$\frac{1-\kappa^2}{1-\kappa}C^*\sqrt{\frac{sq^3(\log n)^2 \log p}{n}},$$

where we use $\kappa_0 \leq \kappa$ and $C_{\text{con}} \leq \frac{\kappa_0}{1-\kappa}C^* + C_{\text{con}} \leq C^* \leq (1+\kappa)C^*$. For $\lambda^{(1)}$, we have

$$
\begin{aligned}
\lambda^{(1)} &= \kappa\lambda^{(0)} + C_\lambda\sqrt{\frac{q^3(\log n)^2\log p}{n}} \\
&= \kappa C_1\frac{d_F(\widehat{\boldsymbol{\theta}}^{(0)},\boldsymbol{\theta}^*)\vee\|\widehat{\mathbf{B}}_1^{(0)}-\mathbf{B}_1^*\|_F\vee\|\widehat{\mathbf{B}}_2^{(0)}-\mathbf{B}_2^*\|_F}{\sqrt{s}} + (1+\kappa)C_\lambda\sqrt{\frac{q^3(\log n)^2\log p}{n}} \\
&\geq 4C_{\text{con}}\sqrt{\frac{q^3(\log n)^2\log p}{n}} + 4\kappa_0\frac{d_F(\widehat{\boldsymbol{\theta}}^{(0)},\boldsymbol{\theta}^*)\vee\|\widehat{\mathbf{B}}_1^{(0)}-\mathbf{B}_1^*\|_F\vee\|\widehat{\mathbf{B}}_2^{(0)}-\mathbf{B}_2^*\|_F}{\sqrt{s}},
\end{aligned}
$$

since $(1+\kappa)C_\lambda \geq 4C_{\text{con}}$ and $C_1\kappa \geq 4\kappa_0$.

Note that $\sqrt{3sq} + 2\sqrt{sq} + 2\sqrt{3sq^2} \leq 8\sqrt{sq^2}$. By Lemma C.3, we have

$$
\begin{aligned}
&\|\widehat{\mathbf{B}}_1^{(1)} - \mathbf{B}_1^*\|_F \\
&\leq \frac{4}{\tau_0}d_{F,s}(M_n(\widehat{\boldsymbol{\theta}}^{(0)}), M(\boldsymbol{\theta}^*)) + \frac{2}{\tau_0}\lambda^{(1)}(\sqrt{3sq}+2\sqrt{sq}+2\sqrt{3sq^2}) \\
&\leq \frac{4}{\tau_0}\left[\kappa_0(d_F(\widehat{\boldsymbol{\theta}}^{(0)},\boldsymbol{\theta}^*)\vee\|\widehat{\mathbf{B}}_1^{(0)}-\mathbf{B}_1^*\|_F\vee\|\widehat{\mathbf{B}}_1^{(0)}-\mathbf{B}_2^*\|_F) + C_{\text{con}}\sqrt{\frac{sq^3(\log n)^2\log p}{n}}\right] + \\
&\quad \frac{16}{\tau_0}\sqrt{sq^2}\left[\kappa C_1\frac{d_F(\widehat{\boldsymbol{\theta}}^{(0)},\boldsymbol{\theta}^*)\vee\|\widehat{\mathbf{B}}_1^{(0)}-\mathbf{B}_1^*\|_F\vee\|\widehat{\mathbf{B}}_2^{(0)}-\mathbf{B}_2^*\|_F}{\sqrt{s}} + (1+\kappa)C_\lambda\sqrt{\frac{q^3(\log n)^2\log p}{n}}\right] \\
&\leq \left[\frac{4}{\tau_0}\kappa_0 + \frac{16}{\tau_0}q\kappa C_1\right](d_F(\widehat{\boldsymbol{\theta}}^{(0)},\boldsymbol{\theta}^*)\vee\|\widehat{\mathbf{B}}_1^{(0)}-\mathbf{B}_1^*\|_F\vee\|\widehat{\mathbf{B}}_2^{(0)}-\mathbf{B}_2^*\|_F) + \\
&\quad \left[\frac{4}{\tau_0}C_{\text{con}} + \frac{16}{\tau_0}q(1+\kappa)C_\lambda\right]\sqrt{\frac{sq^3(\log n)^2\log p}{n}} \\
&\leq \kappa(d_F(\widehat{\boldsymbol{\theta}}^{(0)},\boldsymbol{\theta}^*)\vee\|\widehat{\mathbf{B}}_1^{(0)}-\mathbf{B}_1^*\|_F\vee\|\widehat{\mathbf{B}}_2^{(0)}-\mathbf{B}_2^*\|_F) + \frac{1-\kappa^2}{1-\kappa}C^*\sqrt{\frac{sq^3(\log n)^2\log p}{n}}.
\end{aligned}
$$

By results in $(i)$, $\frac{4q}{\tau_0}\kappa_0 + \frac{16q}{\tau_0}\kappa C_1 \leq \kappa$. To show $\frac{4q}{\tau_0}C_{\text{con}} + \frac{16q}{\tau_0}(1+\kappa)C_\lambda \leq (1+\kappa)C^*$, it is equivalent to show $\frac{4q}{\tau_0}C_{\text{con}}/(1+\kappa) + \frac{16q}{\tau_0}C_\lambda \leq C^*$. Since $1/(1+\kappa) < 1 < 1/(1-\kappa)$, the result holds by applying second inequality in $(ii)$.

Let $\widehat{\boldsymbol{\beta}}_1^{(t)}$ be the top-$d$ left singular vectors of $\widehat{\mathbf{B}}_1^{(t)}$. For matrix $\mathbf{A} \in \mathbb{R}^{p\times q}$,

$$
\begin{aligned}
\|\mathbf{P}_{\boldsymbol{\beta}_1^*}\mathbf{A}\|_{F,s} &= \sup_{\substack{\mathbf{u}\in\mathbb{R}^{p\times q} \\ \text{vec}(\mathbf{u})\in\mathcal{L}(s)\cap\mathbb{S}^{pq-1}}} \langle\mathbf{P}_{\boldsymbol{\beta}_1^*}\mathbf{A}, \mathbf{u}\rangle_F = \sup_{\substack{\mathbf{u}\in\mathbb{R}^{p\times q} \\ \text{vec}(\mathbf{u})\in\mathcal{L}(s)\cap\mathbb{S}^{pq-1}}} \langle\mathbf{P}_{\boldsymbol{\beta}_1^*}\mathbf{u}, \mathbf{A}\rangle_F \\
&\leq \|\mathbf{P}_{\boldsymbol{\beta}_1^*}\mathbf{u}\|_F\|\mathbf{A}\|_{F,s} \leq \sqrt{d}\|\mathbf{A}\|_{F,s},
\end{aligned}
$$

where we use the fact that $\text{vec}(\mathbf{P}_{\boldsymbol{\beta}_1^*}\mathbf{A}) \in \mathcal{L}(s)$. If $\text{vec}(\mathbf{A}) \in \mathcal{L}(s)$,

$$
\|\mathbf{A}\|_{F,s} = \sup_{\substack{\mathbf{u}\in\mathbb{R}^{p\times q} \\ \text{vec}(\mathbf{u})\in\mathcal{L}(s)\cap\mathbb{S}^{pq-1}}} \langle\mathbf{A}, \mathbf{u}\rangle_F \geq \langle\mathbf{A}, \mathbf{A}/\|\mathbf{A}\|_F\rangle_F = \|\mathbf{A}\|_F.
$$

Then, we have

$$
\begin{aligned}
\|\widehat{\boldsymbol{\Gamma}}_1^{(1)} - \boldsymbol{\Gamma}_1^*\|_F &= \|\mathbf{P}_{\widehat{\boldsymbol{\beta}}_1^{(1)}}\widehat{\mathbf{U}}_1(\widehat{\boldsymbol{\theta}}^{(0)})[\widehat{\pi}_1(\widehat{\boldsymbol{\theta}}^{(0)})\widehat{\boldsymbol{\Sigma}}_{\mathbf{f}}]^{-1} - \mathbf{P}_{\boldsymbol{\beta}_1^*}\mathbf{U}_1^*[\pi_1^*\widehat{\boldsymbol{\Sigma}}_{\mathbf{f}}]^{-1}\|_F \\
&\leq \|\mathbf{P}_{\widehat{\boldsymbol{\beta}}_1^{(1)}}\widehat{\mathbf{U}}_1(\widehat{\boldsymbol{\theta}}^{(0)})[\widehat{\pi}_1(\widehat{\boldsymbol{\theta}}^{(0)})\widehat{\boldsymbol{\Sigma}}_{\mathbf{f}}]^{-1} - \mathbf{P}_{\boldsymbol{\beta}_1^*}\widehat{\mathbf{U}}_1(\widehat{\boldsymbol{\theta}}^{(0)})[\widehat{\pi}_1(\widehat{\boldsymbol{\theta}}^{(0)})\widehat{\boldsymbol{\Sigma}}_{\mathbf{f}}]^{-1}\|_{F,s} + \\
&\quad \|\mathbf{P}_{\boldsymbol{\beta}_1^*}\widehat{\mathbf{U}}_1(\widehat{\boldsymbol{\theta}}^{(0)})[\widehat{\pi}_1(\widehat{\boldsymbol{\theta}}^{(0)})\widehat{\boldsymbol{\Sigma}}_{\mathbf{f}}]^{-1} - \mathbf{P}_{\boldsymbol{\beta}_1^*}\mathbf{U}_1^*[\pi_1^*\widehat{\boldsymbol{\Sigma}}_{\mathbf{f}}]^{-1}\|_{F,s} \\
&\leq \|\mathbf{P}_{\widehat{\boldsymbol{\beta}}_1^{(1)}} - \mathbf{P}_{\boldsymbol{\beta}_1^*}\|_F \cdot \|\widehat{\mathbf{U}}_1(\widehat{\boldsymbol{\theta}}^{(0)})\frac{1}{\widehat{\pi}_1(\widehat{\boldsymbol{\theta}}^{(0)})}\widehat{\boldsymbol{\Sigma}}_{\mathbf{f}}^{-1}\|_{F,s} + \\
&\quad \sqrt{d}\|\widehat{\mathbf{U}}_1(\widehat{\boldsymbol{\theta}}^{(0)})[\widehat{\pi}_1(\widehat{\boldsymbol{\theta}}^{(0)})\widehat{\boldsymbol{\Sigma}}_{\mathbf{f}}]^{-1} - \mathbf{U}_1^*[\pi_1^*\widehat{\boldsymbol{\Sigma}}_{\mathbf{f}}]^{-1}\|_{F,s}.
\end{aligned}
$$

By condition (C3), we have $\|\widehat{\mathbf{B}}_1^{(1)} - \mathbf{B}_1^*\|_2 \leq \|\widehat{\mathbf{B}}_1^{(1)} - \mathbf{B}_1^*\|_F \leq \kappa(d_F(\widehat{\boldsymbol{\theta}}^{(0)}, \boldsymbol{\theta}^*) \vee \|\widehat{\mathbf{B}}_1^{(0)} - \mathbf{B}_1^*\|_F \vee \|\widehat{\mathbf{B}}_2^{(0)} - \mathbf{B}_2^*\|_F) + \frac{1}{(1-\kappa)M_5}C^*\sigma_d(\mathbf{B}_1^*) := C'$. For the first term in the last equality, by Lemma F.4, we have

$$\|\mathbf{P}_{\widehat{\boldsymbol{\beta}}_1^{(1)}} - \mathbf{P}_{\boldsymbol{\beta}_1^*}\|_F \leq C_{\boldsymbol{\beta}}\|\widehat{\mathbf{B}}_1^{(1)} - \mathbf{B}_1^*\|_F,$$

where $C_{\boldsymbol{\beta}} = \sqrt{2d}(4\sigma_1(\mathbf{B}^*) + 2C')/\sigma_d^2(\mathbf{B}_1^*)$. Note that $\widehat{\mathbf{U}}_1(\widehat{\boldsymbol{\theta}}^{(0)}) = \frac{1}{n}\sum_{i=1}^n \gamma_{1,\widehat{\boldsymbol{\theta}}^{(0)}}(\mathbf{X}_i, Y_i)\mathbf{X}_i\mathbf{f}_i^T$ and $0 \leq \gamma_{1,\widehat{\boldsymbol{\theta}}^{(0)}}(\mathbf{X}_i, Y_i) \leq 1$. According to Lemma F.3,

$$\|\widehat{\mathbf{U}}_1(\widehat{\boldsymbol{\theta}}^{(0)})\widehat{\boldsymbol{\Sigma}}_{\mathbf{f}}^{-1}\|_{F,s} \leq \frac{1}{M_1}\|\widehat{\mathbf{U}}_1(\widehat{\boldsymbol{\theta}}^{(0)})\|_{F,s} \leq M/M_1,$$

with probability at least $1 - o(1)$. For the second inequality,

$$\|\widehat{\mathbf{U}}_1(\widehat{\boldsymbol{\theta}}^{(0)})[\widehat{\pi}_1(\widehat{\boldsymbol{\theta}}^{(0)})\widehat{\boldsymbol{\Sigma}}_{\mathbf{f}}]^{-1} - \mathbf{U}_1^*[\pi_1^*\widehat{\boldsymbol{\Sigma}}_{\mathbf{f}}]^{-1}\|_{F,s}$$

$$\leq \frac{1}{M_1}\|\widehat{\mathbf{U}}_1(\widehat{\boldsymbol{\theta}}^{(0)})[\widehat{\pi}_1(\widehat{\boldsymbol{\theta}}^{(0)})]^{-1} - \mathbf{U}_1^*(\pi_1^*)^{-1}\|_{F,s}$$

$$\leq \frac{1}{M_1}\|\widehat{\mathbf{U}}_1(\widehat{\boldsymbol{\theta}}^{(0)}) - \mathbf{U}_1^*\|_{F,s}[\widehat{\pi}_1(\widehat{\boldsymbol{\theta}}^{(0)})]^{-1} + \frac{1}{M_1}\|\mathbf{U}_1^*\|_F \cdot |[\widehat{\pi}_1(\widehat{\boldsymbol{\theta}}^{(0)})]^{-1} - (\pi_1^*)^{-1}|$$

$$\leq \frac{1}{M_1}\|\widehat{\mathbf{U}}_1(\widehat{\boldsymbol{\theta}}^{(0)}) - \mathbf{U}_1^*\|_{F,s}[\widehat{\pi}_1(\widehat{\boldsymbol{\theta}}^{(0)})]^{-1} + \frac{M_b M_2}{M_1} \cdot |[\widehat{\pi}_1(\widehat{\boldsymbol{\theta}}^{(0)})]^{-1} - (\pi_1^*)^{-1}|.$$

Then, there exists a positive constant $C_0$, such that

$$\|\widehat{\boldsymbol{\Gamma}}_1^{(1)} - \boldsymbol{\Gamma}_1^*\|_F \leq C_0[\|\widehat{\mathbf{B}}_1^{(1)} - \mathbf{B}_1^*\|_F + d_{F,s}(M_n(\widehat{\boldsymbol{\theta}}^{(0)}), M(\boldsymbol{\theta}^*))].$$

Without loss of generality, we assume $C_0 \geq 1$. Therefore, we have

$$d_F(\widehat{\boldsymbol{\theta}}^{(1)}, \boldsymbol{\theta}^*) \vee \|\widehat{\mathbf{B}}_1^{(1)} - \mathbf{B}_1^*\|_F \vee \|\widehat{\mathbf{B}}_2^{(1)} - \mathbf{B}_2^*\|_F$$

$$\leq (C_0\frac{4}{\tau_0} + C_0)d_{F,s}(M_n(\widehat{\boldsymbol{\theta}}^{(0)}), M(\boldsymbol{\theta}^*)) + C_0\frac{2}{\tau_0}\lambda^{(1)}(\sqrt{3sq} + 2\sqrt{sq} + 2\sqrt{3sq^2})$$

$$\leq C_0(\frac{4}{\tau_0} + 1)\left[\kappa_0(d_F(\widehat{\boldsymbol{\theta}}^{(0)}, \boldsymbol{\theta}^*) \vee \|\widehat{\mathbf{B}}_1^{(0)} - \mathbf{B}_1^*\|_F \vee \|\widehat{\mathbf{B}}_1^{(0)} - \mathbf{B}_2^*\|_F) + C_{\text{con}}\sqrt{\frac{sq^3(\log n)^2 \log p}{n}}\right] +$$

$$\frac{16C_0}{\tau_0}\sqrt{sq^2}\left[\kappa C_1\frac{d_F(\widehat{\boldsymbol{\theta}}^{(0)}, \boldsymbol{\theta}^*) \vee \|\widehat{\mathbf{B}}_1^{(0)} - \mathbf{B}_1^*\|_F \vee \|\widehat{\mathbf{B}}_2^{(0)} - \mathbf{B}_2^*\|_F}{\sqrt{s}} + (1+\kappa)C_\lambda\sqrt{\frac{q^3(\log n)^2 \log p}{n}}\right]$$

$$\leq [C_0(\frac{4}{\tau_0} + 1)\kappa_0 + \frac{16C_0}{\tau_0}q\kappa C_1](d_F(\widehat{\boldsymbol{\theta}}^{(0)}, \boldsymbol{\theta}^*) \vee \|\widehat{\mathbf{B}}_1^{(0)} - \mathbf{B}_1^*\|_F \vee \|\widehat{\mathbf{B}}_2^{(0)} - \mathbf{B}_2^*\|_F) +$$

$$[C_0(\frac{4}{\tau_0} + 1)C_{\text{con}} + \frac{16C_0}{\tau_0}q(1+\kappa)C_\lambda]\sqrt{\frac{q^3(\log n)^2 \log p}{n}}$$

$$\leq \kappa(d_F(\widehat{\boldsymbol{\theta}}^{(0)}, \boldsymbol{\theta}^*) \vee \|\widehat{\mathbf{B}}_1^{(0)} - \mathbf{B}_1^*\|_F \vee \|\widehat{\mathbf{B}}_2^{(0)} - \mathbf{B}_2^*\|_F) + \frac{1-\kappa^2}{1-\kappa}C^*\sqrt{\frac{q^3(\log n)^2 \log p}{n}}.$$

By results in $(i)$, $(\frac{4C_0q}{\tau_0} + C_0)\kappa_0 + \frac{16C_0q}{\tau_0}\kappa C_1 \leq \kappa$. To show $(\frac{4C_0q}{\tau_0} + C_0)C_{\text{con}} + \frac{16C_0q}{\tau_0}(1+\kappa)C_\lambda \leq (1+\kappa)C^*$, it is equivalent to show $(\frac{4C_0q}{\tau_0} + C_0)C_{\text{con}}/(1+\kappa) + \frac{16C_0q}{\tau_0}C_\lambda \leq C^*$. Since $1/(1+\kappa) < 1 < 1/(1-\kappa)$, the result holds by applying second inequality in $(ii)$.

In addition, since $d_F(\boldsymbol{\theta}^{(0)}, \boldsymbol{\theta}^*) \vee \|\mathbf{B}_1^{(0)} - \mathbf{B}_1^*\|_F \vee \|\mathbf{B}_2^{(0)} - \mathbf{B}_2^*\|_F < r\Omega$,

$$d_F(\widehat{\boldsymbol{\theta}}^{(1)}, \boldsymbol{\theta}^*) \vee \|\widehat{\mathbf{B}}_1^{(1)} - \mathbf{B}_1^*\|_F \vee \|\widehat{\mathbf{B}}_2^{(1)} - \mathbf{B}_2^*\|_F$$

$$\leq \kappa(d_F(\widehat{\boldsymbol{\theta}}^{(0)}, \boldsymbol{\theta}^*) \vee \|\widehat{\mathbf{B}}_1^{(0)} - \mathbf{B}_1^*\|_F \vee \|\widehat{\mathbf{B}}_2^{(0)} - \mathbf{B}_2^*\|_F) + \frac{1-\kappa^2}{1-\kappa}C^*\sqrt{\frac{q^3(\log n)^2 \log p}{n}}$$

$$\leq \kappa r\Omega + (1+\kappa)C^*\sqrt{\frac{q^3(\log n)^2 \log p}{n}} \leq r\Omega,$$

since $r\Omega > \frac{1+\kappa}{1-\kappa}C^*\sqrt{\frac{q^3(\log n)^2\log p}{n}}$ when $n$ is sufficiently large. Then by Lemma F.5, $\widehat{\boldsymbol{\theta}}^{(1)} \in \mathcal{B}_{\mathrm{con}}(\boldsymbol{\theta}^*)$.

Next, we assume the following holds for $t$-th step,

$$\lambda^{(t)} \geq 4C_{\mathrm{con}}\sqrt{\frac{q^3(\log n)^2\log p}{n}} + 4\kappa_0\Big(\frac{d_F(\widehat{\boldsymbol{\theta}}^{(t-1)},\boldsymbol{\theta}^*)\vee\|\widehat{\mathbf{B}}_1^{(t-1)}-\mathbf{B}_1^*\|_F\vee\|\widehat{\mathbf{B}}_1^{(t-1)}-\mathbf{B}_2^*\|_F}{\sqrt{s}}\Big),$$

$$d_F(\widehat{\boldsymbol{\theta}}^{(t)},\boldsymbol{\theta}^*)\vee\|\widehat{\mathbf{B}}_1^{(t)}-\mathbf{B}_1^*\|_F\vee\|\widehat{\mathbf{B}}_2^{(t)}-\mathbf{B}_2^*\|_F \leq \kappa^t(d_F(\widehat{\boldsymbol{\theta}}^{(0)},\boldsymbol{\theta}^*)\vee\|\widehat{\mathbf{B}}_1^{(0)}-\mathbf{B}_1^*\|_F\vee\|\widehat{\mathbf{B}}_2^{(0)}-\mathbf{B}_2^*\|_F)+$$
$$\frac{1-\kappa^{t+1}}{1-\kappa}C^*\sqrt{\frac{sq^3(\log n)^2\log p}{n}},$$
$$d_F(\widehat{\boldsymbol{\theta}}^{(t)},\boldsymbol{\theta}^*)\vee\|\widehat{\mathbf{B}}_1^{(t)}-\mathbf{B}_1^*\|_F\vee\|\widehat{\mathbf{B}}_2^{(t)}-\mathbf{B}_2^*\|_F \leq r\Omega, \mathrm{vec}(\widehat{\boldsymbol{\Gamma}}_w^{(t)}-\boldsymbol{\Gamma}_w^*)\in\mathcal{L}(s).$$

By Lemma F.5, $\widehat{\boldsymbol{\theta}}^{(t)} \in \mathcal{B}_{\mathrm{con}}(\boldsymbol{\theta}^*)$. Then

$$4C_{\mathrm{con}}\sqrt{\frac{q^3(\log n)^2\log p}{n}} + 4\kappa_0\Big(\frac{d_F(\widehat{\boldsymbol{\theta}}^{(t)},\boldsymbol{\theta}^*)\vee\|\widehat{\mathbf{B}}_1^{(t)}-\mathbf{B}_1^*\|_F\vee\|\widehat{\mathbf{B}}_1^{(t)}-\mathbf{B}_2^*\|_F}{\sqrt{s}}\Big)$$

$$\leq 4C_{\mathrm{con}}\sqrt{\frac{q^3(\log n)^2\log p}{n}} + \frac{4\kappa_0}{\sqrt{s}}\Big(\kappa^t(d_F(\widehat{\boldsymbol{\theta}}^{(0)},\boldsymbol{\theta}^*)\vee\|\widehat{\mathbf{B}}_1^{(0)}-\mathbf{B}_1^*\|_F\vee\|\widehat{\mathbf{B}}_2^{(0)}-\mathbf{B}_2^*\|_F)+$$

$$\frac{1-\kappa^{t+1}}{1-\kappa}C^*\sqrt{\frac{sq^3(\log n)^2\log p}{n}}\Big)$$

$$\leq 4\kappa_0\kappa^t\frac{(d_F(\widehat{\boldsymbol{\theta}}^{(0)},\boldsymbol{\theta}^*)\vee\|\widehat{\mathbf{B}}_1^{(0)}-\mathbf{B}_1^*\|_F\vee\|\widehat{\mathbf{B}}_2^{(0)}-\mathbf{B}_2^*\|_F)}{\sqrt{s}}+$$

$$(4C_{\mathrm{con}}+4\kappa_0\frac{1-\kappa^{t+1}}{1-\kappa}C^*)\sqrt{\frac{q^3(\log n)^2\log p}{n}}$$

$$\leq \kappa^{t+1}C_1\frac{(d_F(\widehat{\boldsymbol{\theta}}^{(0)},\boldsymbol{\theta}^*)\vee\|\widehat{\mathbf{B}}_1^{(0)}-\mathbf{B}_1^*\|_F\vee\|\widehat{\mathbf{B}}_2^{(0)}-\mathbf{B}_2^*\|_F)}{\sqrt{s}} + \frac{1-\kappa^{t+2}}{1-\kappa}C_\lambda\sqrt{\frac{q^3(\log n)^2\log p}{n}}$$

$$= \lambda^{(t+1)}.$$

Use (i), $4\kappa_0 \leq C_1\kappa$. By the definition of $C_\lambda$,

$$\frac{1-\kappa^{t+2}}{1-\kappa}C_\lambda = \underbrace{\frac{1-\kappa^{t+2}}{1-\kappa}}_{>1}(4C_{\mathrm{con}}+\frac{4\kappa_0}{1-\kappa}C^*) \geq 4C_{\mathrm{con}} + \frac{4\kappa_0}{1-\kappa}\frac{1-\kappa^{t+1}}{1-\kappa}C^*$$

$$\geq 4C_{\mathrm{con}} + 4\kappa_0\frac{1-\kappa^{t+1}}{1-\kappa}C^*.$$

Then note that

$$d_{F,s}(M_n(\widehat{\boldsymbol{\theta}}^{(t)}),M(\boldsymbol{\theta}^*))$$
$$\leq d_{F,s}(M(\widehat{\boldsymbol{\theta}}^{(t)}),M(\boldsymbol{\theta}^*)) + d_{F,s}(M_n(\widehat{\boldsymbol{\theta}}^{(t)}),M(\widehat{\boldsymbol{\theta}}^{(t)}))$$
$$\leq \kappa_0(d_F(\widehat{\boldsymbol{\theta}}^{(t)},\boldsymbol{\theta}^*)\vee\|\widehat{\mathbf{B}}_1^{(t)}-\mathbf{B}_1^*\|_F\vee\|\widehat{\mathbf{B}}_1^{(t)}-\mathbf{B}_2^*\|_F) + C_{\mathrm{con}}\sqrt{\frac{sq^3(\log n)^2\log p}{n}}$$
$$\leq \kappa_0\Big[\kappa^t(d_F(\widehat{\boldsymbol{\theta}}^{(0)},\boldsymbol{\theta}^*)\vee\|\widehat{\mathbf{B}}_1^{(0)}-\mathbf{B}_1^*\|_F\vee\|\widehat{\mathbf{B}}_2^{(0)}-\mathbf{B}_2^*\|_F) + \frac{1-\kappa^{t+1}}{1-\kappa}C^*\sqrt{\frac{sq^3(\log n)^2\log p}{n}}\Big]+$$
$$C_{\mathrm{con}}\sqrt{\frac{sq^3(\log n)^2\log p}{n}}$$
$$\leq \kappa^{t+1}(d_F(\widehat{\boldsymbol{\theta}}^{(0)},\boldsymbol{\theta}^*)\vee\|\widehat{\mathbf{B}}_1^{(0)}-\mathbf{B}_1^*\|_F\vee\|\widehat{\mathbf{B}}_2^{(0)}-\mathbf{B}_2^*\|_F) + \frac{1-\kappa^{t+2}}{1-\kappa}C^*\sqrt{\frac{sq^3(\log n)^2\log p}{n}},$$

since $\kappa_0 \leq \kappa$ and $\kappa_0 \frac{1-\kappa^{t+1}}{1-\kappa} C^* + C_{\text{con}} \leq \frac{1-\kappa^{t+2}}{1-\kappa} C^*$. Then by Lemma C.3,

$$
\begin{aligned}
&\|\widehat{\mathbf{B}}_1^{(t+1)} - \mathbf{B}_1^*\|_F \\
&\leq \frac{4}{\tau_0} d_{F,s}(M_n(\widehat{\boldsymbol{\theta}}^{(t)}), M(\boldsymbol{\theta}^*)) + \frac{16q}{\tau_0}\sqrt{s}\lambda^{(t+1)} \\
&\leq \frac{4}{\tau_0}\left\{\kappa_0\left[\kappa^t(d_F(\widehat{\boldsymbol{\theta}}^{(0)},\boldsymbol{\theta}^*) \vee \|\widehat{\mathbf{B}}_1^{(0)}-\mathbf{B}_1^*\|_F \vee \|\widehat{\mathbf{B}}_2^{(0)}-\mathbf{B}_2^*\|_F) + \frac{1-\kappa^{t+1}}{1-\kappa}C^*\sqrt{\frac{sq^3(\log n)^2\log p}{n}}\right] + \right. \\
&\qquad \left. C_{\text{con}}\sqrt{\frac{sq^3(\log n)^2\log p}{n}}\right\} + \frac{16q}{\tau_0}\sqrt{s}\left[\kappa^{t+1}C_1\frac{(d_F(\widehat{\boldsymbol{\theta}}^{(0)},\boldsymbol{\theta}^*)\vee\|\widehat{\mathbf{B}}_1^{(0)}-\mathbf{B}_1^*\|_F \vee \|\widehat{\mathbf{B}}_2^{(0)}-\mathbf{B}_2^*\|_F)}{\sqrt{s}} + \right. \\
&\qquad \left. \frac{1-\kappa^{t+2}}{1-\kappa}C_\lambda\sqrt{\frac{q^3(\log n)^2\log p}{n}}\right] \\
&\leq [\frac{4}{\tau_0}\kappa_0 + \frac{16}{\tau_0}q\kappa C_1]\kappa^t(d_F(\widehat{\boldsymbol{\theta}}^{(0)},\boldsymbol{\theta}^*)\vee\|\widehat{\mathbf{B}}_1^{(0)}-\mathbf{B}_1^*\|_F \vee \|\widehat{\mathbf{B}}_2^{(0)}-\mathbf{B}_2^*\|_F)+ \\
&\qquad [\frac{4}{\tau_0}C_{\text{con}} + \frac{4}{\tau_0}\kappa_0\frac{1-\kappa^{t+1}}{1-\kappa}C^* + \frac{16q}{\tau_0}\frac{1-\kappa^{t+2}}{1-\kappa}C_\lambda]\sqrt{\frac{sq^3(\log n)^2\log p}{n}} \\
&\leq \kappa^{t+1}(d_F(\widehat{\boldsymbol{\theta}}^{(0)},\boldsymbol{\theta}^*)\vee\|\widehat{\mathbf{B}}_1^{(0)}-\mathbf{B}_1^*\|_F \vee \|\widehat{\mathbf{B}}_2^{(0)}-\mathbf{B}_2^*\|_F) + \frac{1-\kappa^{t+2}}{1-\kappa}C^*\sqrt{\frac{sq^3(\log n)^2\log p}{n}}.
\end{aligned}
$$

In the last inequality, we use $(\frac{4}{\tau_0}C_0 q + C_0)\kappa_0 + \frac{16}{\tau_0}C_0 q\kappa C_1 \leq \kappa$, and

$$
\begin{aligned}
&(\frac{4C_0 q}{\tau_0} + C_0)C_{\text{con}} + (\frac{4C_0 q}{\tau_0} + C_0)\kappa_0\frac{1-\kappa^{t+1}}{1-\kappa}C^* + \frac{16C_0 q}{\tau_0}\frac{1-\kappa^{t+2}}{1-\kappa}C_\lambda \leq \frac{1-\kappa^{t+2}}{1-\kappa}C^* \\
&\iff (\frac{4C_0 q}{\tau_0} + C_0)C_{\text{con}} + (\frac{1}{32}\kappa + \frac{1}{4}\kappa)\frac{1-\kappa^{t+1}}{1-\kappa}C^* + \frac{16C_0 q}{\tau_0}\frac{1-\kappa^{t+2}}{1-\kappa}C_\lambda \leq \frac{1-\kappa^{t+2}}{1-\kappa}C^* \\
&\iff (\frac{4C_0 q}{\tau_0} + C_0)C_{\text{con}} + \frac{16C_0 q}{\tau_0}\frac{1-\kappa^{t+2}}{1-\kappa}C_\lambda \leq \frac{1-\kappa^{t+2}}{1-\kappa}C^* - \frac{9}{32}\kappa\frac{1-\kappa^{t+1}}{1-\kappa}C^*,
\end{aligned}
$$

where the right-hand side is greater than $C^*$ and thus the inequality holds due to (ii). Using the same argument,

$$
\|\widehat{\boldsymbol{\Gamma}}_1^{(t+1)} - \boldsymbol{\Gamma}_1^*\|_F \leq C_0[\|\widehat{\mathbf{B}}_1^{(t+1)} - \mathbf{B}_1^*\|_F + d_{F,s}(M_n(\widehat{\boldsymbol{\theta}}^{(t)}), M(\boldsymbol{\theta}^*))].
$$

Therefore,

$$
\begin{aligned}
&d_F(\widehat{\boldsymbol{\theta}}^{(t+1)},\boldsymbol{\theta}^*) \vee \|\widehat{\mathbf{B}}_1^{(t+1)} - \mathbf{B}_1^*\|_F \vee \|\widehat{\mathbf{B}}_2^{(t+1)} - \mathbf{B}_2^*\|_F \\
&\leq (\frac{4C_0}{\tau_0} + C_0)d_{F,s}(M_n(\widehat{\boldsymbol{\theta}}^{(t)}), M(\boldsymbol{\theta}^*)) + \frac{16C_0 q}{\tau_0}\sqrt{s}\lambda^{(t+1)} \\
&\leq (\frac{4C_0}{\tau_0} + C_0)\left\{\kappa_0\left[\kappa^t(d_F(\widehat{\boldsymbol{\theta}}^{(0)},\boldsymbol{\theta}^*)\vee\|\widehat{\mathbf{B}}_1^{(0)}-\mathbf{B}_1^*\|_F \vee \|\widehat{\mathbf{B}}_2^{(0)}-\mathbf{B}_2^*\|_F)+ \right.\right. \\
&\qquad \left. \frac{1-\kappa^{t+1}}{1-\kappa}C^*\sqrt{\frac{sq^3(\log n)^2\log p}{n}}\right] + C_{\text{con}}\sqrt{\frac{sq^3(\log n)^2\log p}{n}}\right\} + \\
&\qquad \frac{16C_0 q}{\tau_0}\sqrt{s}\left[\kappa^{t+1}C_1\frac{(d_F(\widehat{\boldsymbol{\theta}}^{(0)},\boldsymbol{\theta}^*)\vee\|\widehat{\mathbf{B}}_1^{(0)}-\mathbf{B}_1^*\|_F \vee \|\widehat{\mathbf{B}}_2^{(0)}-\mathbf{B}_2^*\|_F)}{\sqrt{s}} + \frac{1-\kappa^{t+2}}{1-\kappa}C_\lambda\sqrt{\frac{q^3(\log n)^2\log p}{n}}\right] \\
&\leq [(\frac{4C_0}{\tau_0} + C_0)\kappa_0 + \frac{16C_0 q}{\tau_0}\kappa C_1]\kappa^t(d_F(\widehat{\boldsymbol{\theta}}^{(0)},\boldsymbol{\theta}^*)\vee\|\widehat{\mathbf{B}}_1^{(0)}-\mathbf{B}_1^*\|_F \vee \|\widehat{\mathbf{B}}_2^{(0)}-\mathbf{B}_2^*\|_F)+ \\
&\qquad [(\frac{4C_0}{\tau_0} + C_0)C_{\text{con}} + (\frac{4C_0}{\tau_0} + C_0)\kappa_0\frac{1-\kappa^{t+1}}{1-\kappa}C^* + \frac{16C_0 q}{\tau_0}\frac{1-\kappa^{t+2}}{1-\kappa}C_\lambda]\sqrt{\frac{sq^3(\log n)^2\log p}{n}} \\
&\leq \kappa^{t+1}(d_F(\widehat{\boldsymbol{\theta}}^{(0)},\boldsymbol{\theta}^*)\vee\|\widehat{\mathbf{B}}_1^{(0)}-\mathbf{B}_1^*\|_F \vee \|\widehat{\mathbf{B}}_2^{(0)}-\mathbf{B}_2^*\|_F) + \frac{1-\kappa^{t+2}}{1-\kappa}C^*\sqrt{\frac{sq^3(\log n)^2\log p}{n}}.
\end{aligned}
$$

Further, since $\frac{1-\kappa^{t+2}}{1+\kappa} \leq 1 - \kappa^{t+1}$ and $r\Omega > \frac{1+\kappa}{1-\kappa}C^*\sqrt{\frac{q^3(\log n)^2 \log p}{n}}$ when $n$ is sufficiently large,

$$
\begin{aligned}
& d_F(\widehat{\boldsymbol{\theta}}^{(t+1)}, \boldsymbol{\theta}^*) \vee \|\widehat{\mathbf{B}}_1^{(t+1)} - \mathbf{B}_1^*\|_F \vee \|\widehat{\mathbf{B}}_2^{(t+1)} - \mathbf{B}_2^*\|_F \\
& \leq \kappa^{t+1}(d_F(\widehat{\boldsymbol{\theta}}^{(0)}, \boldsymbol{\theta}^*) \vee \|\widehat{\mathbf{B}}_1^{(0)} - \mathbf{B}_1^*\|_F \vee \|\widehat{\mathbf{B}}_2^{(0)} - \mathbf{B}_2^*\|_F) + \frac{1 - \kappa^{t+2}}{1-\kappa}C^*\sqrt{\frac{sq^3(\log n)^2 \log p}{n}} \\
& \leq \kappa^{t+1}r\Omega + \frac{1 - \kappa^{t+2}}{1-\kappa}C^*\sqrt{\frac{sq^3(\log n)^2 \log p}{n}} \leq \kappa^{t+1}r\Omega + \frac{1 - \kappa^{t+2}}{1+\kappa}r\Omega \leq r\Omega.
\end{aligned}
$$

When $t \geq (-\log \kappa)^{-1}\log\{n(d_F(\widehat{\boldsymbol{\theta}}^{(0)}, \boldsymbol{\theta}^*) \vee \|\widehat{\mathbf{B}}_1^{(0)} - \mathbf{B}_1^*\|_F \vee \|\widehat{\mathbf{B}}_2^{(0)} - \mathbf{B}_2^*\|_F)\}$,

$$
\begin{aligned}
\kappa^t & \leq \kappa^{-\log_\kappa\{n(d_F(\widehat{\boldsymbol{\theta}}^{(0)}, \boldsymbol{\theta}^*) \vee \|\widehat{\mathbf{B}}_1^{(0)} - \mathbf{B}_1^*\|_F \vee \|\widehat{\mathbf{B}}_2^{(0)} - \mathbf{B}_2^*\|_F)\}} \\
& \leq \frac{1}{n(d_F(\widehat{\boldsymbol{\theta}}^{(0)}, \boldsymbol{\theta}^*) \vee \|\widehat{\mathbf{B}}_1^{(0)} - \mathbf{B}_1^*\|_F \vee \|\widehat{\mathbf{B}}_2^{(0)} - \mathbf{B}_2^*\|_F)}.
\end{aligned}
$$

which implies

$$
\|\widehat{\mathbf{B}}_w^{(t)} - \mathbf{B}_w^*\|_F = O\left(\sqrt{\frac{sq^3(\log n)^2 \log p}{n}}\right).
$$

With Lemma F.4,

$$
\mathcal{D}(\mathcal{S}_{\widehat{\boldsymbol{\beta}}_w^{(t)}}, \mathcal{S}_{\boldsymbol{\beta}_w^*}) = \frac{\|\mathbf{P}_{\widehat{\boldsymbol{\beta}}_w^{(t)}} - \mathbf{P}_{\boldsymbol{\beta}_w^*}\|_F}{\sqrt{2d}} \leq C_{\boldsymbol{\beta}}/\sqrt{2d}\|\widehat{\mathbf{B}}_w^{(t)} - \mathbf{B}_w^*\|_F = O\left(\sqrt{\frac{sq^3(\log n)^2 \log p}{n}}\right).
$$

$\square$

## D. Proof of Lemma C.1

### D.1. Contraction of weights

In this section, we show $|\pi_1(\boldsymbol{\theta}) - \pi_1(\boldsymbol{\theta}^*)| \leq \kappa_0(d_F(\boldsymbol{\theta}, \boldsymbol{\theta}^*) \vee \|\mathbf{B}_1 - \mathbf{B}_1^*\|_F \vee \|\mathbf{B}_2 - \mathbf{B}_2^*\|_F)$. By definition,

$$
\pi_1(\boldsymbol{\theta}) - \pi_1(\boldsymbol{\theta}^*) = \mathrm{E}[\frac{1}{n}\sum_{i=1}^n (\gamma_{1,\boldsymbol{\theta}}(\mathbf{X}_i, Y_i) - \gamma_{1,\boldsymbol{\theta}^*}(\mathbf{X}_i, Y_i))].
$$

When $Y_i$ is fixed, we can not further simplify the above. Thus, for given $i$, we bound $|\mathrm{E}[\gamma_{1,\boldsymbol{\theta}}(\mathbf{X}_i, Y_i) - \gamma_{1,\boldsymbol{\theta}^*}(\mathbf{X}_i, Y_i)]|$. Let $\boldsymbol{\xi}^T = (\pi_1, \mathrm{vec}(\boldsymbol{\Gamma}_2 - \boldsymbol{\Gamma}_1)^T, \mathrm{vec}(\boldsymbol{\Gamma}_2 + \boldsymbol{\Gamma}_1)^T), \Delta_{\boldsymbol{\xi}} = \boldsymbol{\xi} - \boldsymbol{\xi}^*, \boldsymbol{\xi}_u = \boldsymbol{\xi}^* + u\Delta_{\boldsymbol{\xi}}$. Then $\boldsymbol{\xi}_0 = \boldsymbol{\xi}^*, \boldsymbol{\xi}_1 = \boldsymbol{\xi}$. Then

$$
\begin{aligned}
& \left| \mathrm{E}[\gamma_{1,\boldsymbol{\theta}}(\mathbf{X}_i, Y_i) - \gamma_{1,\boldsymbol{\theta}^*}(\mathbf{X}_i, Y_i)] \right| \\
& = \left| \mathrm{E}\left\{ \int_0^1 \langle \frac{\partial \gamma_{1,\boldsymbol{\xi}}(\mathbf{X}_i, Y_i)}{\partial \boldsymbol{\xi}}\Big|_{\boldsymbol{\xi}=\boldsymbol{\xi}_u}, \frac{\partial \boldsymbol{\xi}_u}{\partial u}\rangle du \right\} \right| \\
& \leq \left| \mathrm{E}\left\{ \int_0^1 \langle \frac{\partial \gamma_{1,\boldsymbol{\xi}}(\mathbf{X}_i, Y_i)}{\partial \pi_1}\Big|_{\boldsymbol{\xi}=\boldsymbol{\xi}_u}, \Delta_{\pi_1}\rangle du \right\} \right| + \left| \mathrm{E}\left\{ \int_0^1 \langle \frac{\partial \gamma_{1,\boldsymbol{\xi}}(\mathbf{X}_i, Y_i)}{\partial \mathrm{vec}(\boldsymbol{\Gamma}_2 - \boldsymbol{\Gamma}_1)}\Big|_{\boldsymbol{\xi}=\boldsymbol{\xi}_u}, \Delta_{\boldsymbol{\Gamma}_2-\boldsymbol{\Gamma}_1}\rangle du \right\} \right| + \\
& \quad \left| \mathrm{E}\left\{ \int_0^1 \langle \frac{\partial \gamma_{1,\boldsymbol{\xi}}(\mathbf{X}_i, Y_i)}{\partial \mathrm{vec}(\boldsymbol{\Gamma}_2 + \boldsymbol{\Gamma}_1)}\Big|_{\boldsymbol{\xi}=\boldsymbol{\xi}_u}, \Delta_{\boldsymbol{\Gamma}_2+\boldsymbol{\Gamma}_1}\rangle du \right\} \right| \\
& \leq \int_0^1 \langle \mathrm{E}[\frac{\partial \gamma_{1,\boldsymbol{\xi}}(\mathbf{X}_i, Y_i)}{\partial \pi_1}]\Big|_{\boldsymbol{\xi}=\boldsymbol{\xi}_u}, (\pi_1 - \pi_1^*)\rangle du + \int_0^1 \langle \mathrm{E}[\frac{\partial \gamma_{1,\boldsymbol{\xi}}(\mathbf{X}_i, Y_i)}{\partial \mathrm{vec}(\boldsymbol{\Gamma}_2 - \boldsymbol{\Gamma}_1)}]\Big|_{\boldsymbol{\xi}=\boldsymbol{\xi}_u}, \Delta_{\boldsymbol{\Gamma}_2-\boldsymbol{\Gamma}_1}\rangle du + \\
& \quad \int_0^1 \langle \mathrm{E}[\frac{\partial \gamma_{1,\boldsymbol{\xi}}(\mathbf{X}_i, Y_i)}{\partial \mathrm{vec}(\boldsymbol{\Gamma}_2 + \boldsymbol{\Gamma}_1)}]\Big|_{\boldsymbol{\xi}=\boldsymbol{\xi}_u}, \Delta_{\boldsymbol{\Gamma}_2+\boldsymbol{\Gamma}_1}\rangle du \\
& \leq \underbrace{\sup_{\boldsymbol{\xi}\in\mathcal{B}_{\mathrm{con}}(\boldsymbol{\theta}^*)} |\mathrm{E}[\frac{\partial \gamma_{1,\boldsymbol{\xi}}(\mathbf{X}_i, Y_i)}{\partial \pi_1}]| \cdot |\pi_1 - \pi_1^*|}_{(I)} + \\
& \quad \underbrace{\sup_{\boldsymbol{\xi}\in\mathcal{B}_{\mathrm{con}}(\boldsymbol{\theta}^*)} \|\mathrm{E}[\frac{\partial \gamma_{1,\boldsymbol{\xi}}(\mathbf{X}_i, Y_i)}{\partial \mathrm{vec}(\boldsymbol{\Gamma}_2 - \boldsymbol{\Gamma}_1)}]\|_2 \cdot \|\boldsymbol{\Gamma}_2 - \boldsymbol{\Gamma}_1 - \boldsymbol{\Gamma}_2^* + \boldsymbol{\Gamma}_1^*\|_F}_{(II)} + \\
& \quad \underbrace{\sup_{\boldsymbol{\xi}\in\mathcal{B}_{\mathrm{con}}(\boldsymbol{\theta}^*)} \|\mathrm{E}[\frac{\partial \gamma_{1,\boldsymbol{\xi}}(\mathbf{X}_i, Y_i)}{\partial \mathrm{vec}(\boldsymbol{\Gamma}_2 + \boldsymbol{\Gamma}_1)}]\|_2 \cdot \|\boldsymbol{\Gamma}_2 + \boldsymbol{\Gamma}_1 - \boldsymbol{\Gamma}_2^* - \boldsymbol{\Gamma}_1^*\|_F}_{(III)} .
\end{aligned}
$$

Thus, we bound the three terms in the last inequality. Recall that

$$
\gamma_{1,\boldsymbol{\xi}}(\mathbf{X}_i, Y_i) = \frac{\pi_1}{\pi_1 + (1 - \pi_1)\exp\{[\mathbf{X}_i - \frac{1}{2}(\boldsymbol{\Gamma}_2 + \boldsymbol{\Gamma}_1)\mathbf{f}_i]^T(\boldsymbol{\Gamma}_2 - \boldsymbol{\Gamma}_1)\mathbf{f}_i\}} .
$$

We can decompose $\mathbf{X}_i$ as the sum of two independent random variables, $\mathbf{X}_i \sim_d \mathbf{Z} + \boldsymbol{\psi}\mathbf{f}_i$, where $\mathbf{Z} \sim N(0, \mathbf{I}_p)$ and is independent of $\mathbf{f}_i$ and $W_i$, $P(\boldsymbol{\psi} = \boldsymbol{\Gamma}_1^*) = \pi_1^*$ and $P(\boldsymbol{\psi} = \boldsymbol{\Gamma}_2^*) = 1 - \pi_1^*$. Let $\boldsymbol{\delta}(\boldsymbol{\Gamma}) = \boldsymbol{\psi} - (\boldsymbol{\Gamma}_2 + \boldsymbol{\Gamma}_1)/2$. Then

$$
\mathbf{X}_i - \frac{1}{2}(\boldsymbol{\Gamma}_2 + \boldsymbol{\Gamma}_1)\mathbf{f}_i \sim_d \mathbf{Z} + \boldsymbol{\psi}\mathbf{f}_i - \frac{1}{2}(\boldsymbol{\Gamma}_2 + \boldsymbol{\Gamma}_1)\mathbf{f}_i \sim_d \mathbf{Z} + \boldsymbol{\delta}(\boldsymbol{\Gamma})\mathbf{f}_i.
$$

Therefore, we can write

$$
\gamma_{1,\boldsymbol{\xi}}(\mathbf{X}_i, Y_i) = \frac{\pi_1}{\pi_1 + (1 - \pi_1)\exp\{(\mathbf{Z} + \boldsymbol{\delta}(\boldsymbol{\Gamma})\mathbf{f}_i)^T(\boldsymbol{\Gamma}_2 - \boldsymbol{\Gamma}_1)\mathbf{f}_i\}} .
$$

By calculation, we have

$$
\begin{aligned}
\frac{\partial \gamma_{1,\boldsymbol{\xi}}(\mathbf{X}_i, Y_i)}{\partial \pi_1} &= \frac{\exp\{(\mathbf{Z} + \boldsymbol{\delta}(\boldsymbol{\Gamma})\mathbf{f}_i)^T(\boldsymbol{\Gamma}_2 - \boldsymbol{\Gamma}_1)\mathbf{f}_i\}}{(\pi_1 + (1 - \pi_1)\exp\{(\mathbf{Z} + \boldsymbol{\delta}(\boldsymbol{\Gamma})\mathbf{f}_i)^T(\boldsymbol{\Gamma}_2 - \boldsymbol{\Gamma}_1)\mathbf{f}_i\})^2} \\
\frac{\partial \gamma_{1,\boldsymbol{\xi}}(\mathbf{X}_i, Y_i)}{\partial \mathrm{vec}(\boldsymbol{\Gamma}_2 - \boldsymbol{\Gamma}_1)} &= -\pi_1(1 - \pi_1)\frac{\exp\{(\mathbf{Z} + \boldsymbol{\delta}(\boldsymbol{\Gamma})\mathbf{f}_i)^T(\boldsymbol{\Gamma}_2 - \boldsymbol{\Gamma}_1)\mathbf{f}_i\} \cdot \mathbf{f}_i \otimes (\mathbf{Z} + \boldsymbol{\delta}(\boldsymbol{\Gamma})\mathbf{f}_i)}{(\pi_1 + (1 - \pi_1)\exp\{(\mathbf{Z} + \boldsymbol{\delta}(\boldsymbol{\Gamma})\mathbf{f}_i)^T(\boldsymbol{\Gamma}_2 - \boldsymbol{\Gamma}_1)\mathbf{f}_i\})^2} \\
\frac{\partial \gamma_{1,\boldsymbol{\xi}}(\mathbf{X}_i, Y_i)}{\partial \mathrm{vec}(\boldsymbol{\Gamma}_2 + \boldsymbol{\Gamma}_1)} &= \pi_1(1 - \pi_1)\frac{\exp\{(\mathbf{Z} + \boldsymbol{\delta}(\boldsymbol{\Gamma})\mathbf{f}_i)^T(\boldsymbol{\Gamma}_2 - \boldsymbol{\Gamma}_1)\mathbf{f}_i\} \cdot \frac{1}{2}\mathbf{f}_i \otimes (\boldsymbol{\Gamma}_2 - \boldsymbol{\Gamma}_1)\mathbf{f}_i}{(\pi_1 + (1 - \pi_1)\exp\{(\mathbf{Z} + \boldsymbol{\delta}(\boldsymbol{\Gamma})\mathbf{f}_i)^T(\boldsymbol{\Gamma}_2 - \boldsymbol{\Gamma}_1)\mathbf{f}_i\})^2} .
\end{aligned}
$$

Let $\mathbf{T}_1 = (\boldsymbol{\Gamma}_2 - \boldsymbol{\Gamma}_1)\mathbf{f}_i$ and $\mathbf{T}_2 = \boldsymbol{\delta}(\boldsymbol{\Gamma})\mathbf{f}_i$. Let $\mathbf{H}_i$ be an orthonormal matrix whose first row is $\mathbf{T}_1^T/\|\mathbf{T}_1\|_2$. Then $\mathbf{H}_i\mathbf{T}_1 = \|\mathbf{T}_1\|_2\mathbf{e}_1$, where $\mathbf{e}_1$ is the basis vector in the Euclidean space whose first entry is 1 and zero otherwise. Then

$$
\mathbf{f}_i^T(\boldsymbol{\Gamma}_2 - \boldsymbol{\Gamma}_1)^T\mathbf{Z} = \mathbf{T}_1^T\mathbf{Z} = \mathbf{T}_1^T\mathbf{H}_i^T\mathbf{H}_i\mathbf{Z} = \mathbf{T}_1^T\mathbf{H}_i^T\mathbf{V} = V_1\|\mathbf{T}_1\|_2, \tag{10}
$$

where $V_1$ is the first coordinate of $\mathbf{V} = \mathbf{H}_i\mathbf{Z} \sim N(0, \mathbf{I}_p)$ and is a standard normal distribution. Then

$$\mathrm{E}[\frac{\partial\gamma_{1,\boldsymbol{\xi}}(\mathbf{X}_i, Y_i)}{\partial\pi_1}] = \mathrm{E}[\frac{\exp(\mathbf{T}_1^T(\mathbf{Z} + \mathbf{T}_2)}{(\pi_1 + (1 - \pi_1)\exp\{\mathbf{T}_1^T(\mathbf{Z} + \mathbf{T}_2)\})^2}]$$

$$= \mathrm{E}[\frac{\exp(\|\mathbf{T}_1\|_2 Z_1 + \mathbf{T}_1^T\mathbf{T}_2))}{(\pi_1 + (1 - \pi_1)\exp\{\|\mathbf{T}_1\|_2 Z_1 + \mathbf{T}_1^T\mathbf{T}_2)\})^2}],$$

where $Z_1$ is a standard normal distribution. Note that

$$|\mathbf{T}_1^T\mathbf{T}_2| = |\mathbf{f}_i^T(\boldsymbol{\Gamma}_2 - \boldsymbol{\Gamma}_1)^T\boldsymbol{\delta}(\boldsymbol{\Gamma})\mathbf{f}_i| \geq c|\operatorname{tr}(\boldsymbol{\delta}(\boldsymbol{\Gamma})\widehat{\boldsymbol{\Sigma}}_{\mathbf{f}}(\boldsymbol{\Gamma}_2 - \boldsymbol{\Gamma}_1)^T)| \geq c_1\Omega^2,$$

where $c_1 = c(1 - C_d)$. Similarly, we can show $|\mathbf{T}_1^T\mathbf{T}_2| \leq c_2\Omega^2$.

Recall that $\Omega = \sqrt{\operatorname{tr}[(\boldsymbol{\Gamma}_2^* - \boldsymbol{\Gamma}_1^*)\widehat{\boldsymbol{\Sigma}}_{\mathbf{f}}(\boldsymbol{\Gamma}_2^* - \boldsymbol{\Gamma}_1^*)^T]} = \|\widehat{\boldsymbol{\Sigma}}_{\mathbf{f}}^{1/2}(\boldsymbol{\Gamma}_2^* - \boldsymbol{\Gamma}_1^*)^T\|_F$. We have

$$\Omega^2 = \operatorname{vec}((\boldsymbol{\Gamma}_2^* - \boldsymbol{\Gamma}_1^*)^T)^T \operatorname{vec}(\widehat{\boldsymbol{\Sigma}}_{\mathbf{f}}(\boldsymbol{\Gamma}_2^* - \boldsymbol{\Gamma}_1^*)^T)$$

$$= \operatorname{vec}((\boldsymbol{\Gamma}_2^* - \boldsymbol{\Gamma}_1^*)^T)^T(\mathbf{I}_p \otimes \widehat{\boldsymbol{\Sigma}}_{\mathbf{f}})\operatorname{vec}((\boldsymbol{\Gamma}_2^* - \boldsymbol{\Gamma}_1^*)^T),$$

which implies

$$\Omega^2/M_2 \leq \|\boldsymbol{\Gamma}_2^* - \boldsymbol{\Gamma}_1^*\|_F^2 \leq \Omega^2/M_1.$$

Then, when $2\sqrt{M_2}C_b < 1$,

$$\sqrt{\mathbf{T}_1^T\mathbf{T}_1} = \sqrt{\mathbf{f}_i^T(\boldsymbol{\Gamma}_2 - \boldsymbol{\Gamma}_1)^T(\boldsymbol{\Gamma}_2 - \boldsymbol{\Gamma}_1)\mathbf{f}_i} = \sqrt{\operatorname{tr}[(\boldsymbol{\Gamma}_2 - \boldsymbol{\Gamma}_1)\mathbf{f}_i\mathbf{f}_i^T(\boldsymbol{\Gamma}_2 - \boldsymbol{\Gamma}_1)^T]}$$

$$\geq c\sqrt{\operatorname{tr}[(\boldsymbol{\Gamma}_2 - \boldsymbol{\Gamma}_1)\widehat{\boldsymbol{\Sigma}}_{\mathbf{f}}(\boldsymbol{\Gamma}_2 - \boldsymbol{\Gamma}_1)^T]} = c\|\widehat{\boldsymbol{\Sigma}}_{\mathbf{f}}^{1/2}(\boldsymbol{\Gamma}_2 - \boldsymbol{\Gamma}_1)^T\|_F$$

$$\geq c\Big|\|\widehat{\boldsymbol{\Sigma}}_{\mathbf{f}}^{1/2}(\boldsymbol{\Gamma}_2^* - \boldsymbol{\Gamma}_1^*)^T\|_F - \|\widehat{\boldsymbol{\Sigma}}_{\mathbf{f}}^{1/2}(\boldsymbol{\Gamma}_2 - \boldsymbol{\Gamma}_2^* - \boldsymbol{\Gamma}_1 + \boldsymbol{\Gamma}_1^*)\|_F\Big|$$

$$\geq c(\Omega - 2\sqrt{M_2}C_b\Omega) \geq c_3\Omega,$$

for some constant $c_3$ that depends on $M_2$ and $C_b$. Similarly, we can show $\sqrt{\mathbf{T}_1^T\mathbf{T}_1} \leq c_4\Omega$. Define events

$$\mathcal{E}_i = \{|\|\mathbf{T}_1\|_2 Z_1| \leq \frac{c_1}{2}\Omega^2\}.$$

On the event $\mathcal{E}_i$, $|\|\mathbf{T}_1\|_2 Z_1 + \mathbf{T}_1^T\mathbf{T}_2| \geq |\mathbf{T}_1^T\mathbf{T}_2| - |\|\mathbf{T}_1\|_2 Z_1| \geq c_1\Omega^2/2$. Using the tail probability of normal distribution, we obtain

$$P(\mathcal{E}_i^c) \leq 2\exp(-\frac{c_1^2\Omega^4}{8\|\mathbf{T}_1\|_2^2}) \leq 2\exp(-\frac{c_1^2\Omega^2}{8c_4^2}).$$

Then

$$\mathrm{E}[\frac{\partial\gamma_{1,\boldsymbol{\xi}}(\mathbf{X}_i, Y_i)}{\partial\pi_1}]$$

$$= \mathrm{E}[\frac{\exp(\|\mathbf{T}_1\|_2 Z_1 + \mathbf{T}_1^T\mathbf{T}_2)}{(\pi_1 + (1 - \pi_1)\exp\{\|\mathbf{T}_1\|_2 Z_1 + \mathbf{T}_1^T\mathbf{T}_2)\})^2}]$$

$$= \mathrm{E}[\frac{\exp(\|\mathbf{T}_1\|_2 Z_1 + \mathbf{T}_1^T\mathbf{T}_2)}{(\pi_1 + (1 - \pi_1)\exp\{\|\mathbf{T}_1\|_2 Z_1 + \mathbf{T}_1^T\mathbf{T}_2)\})^2}\Big|\mathcal{E}_i]P(\mathcal{E}_i) +$$

$$\mathrm{E}\frac{\exp(\|\mathbf{T}_1\|_2 Z_1 + \mathbf{T}_1^T\mathbf{T}_2)}{(\pi_1 + (1 - \pi_1)\exp\{\|\mathbf{T}_1\|_2 Z_1 + \mathbf{T}_1^T\mathbf{T}_2)\})^2}\Big|\mathcal{E}_i^c]P(\mathcal{E}_i^c)$$

$$\leq \frac{1}{\min\{\pi_1^2, (1 - \pi_1)^2\}}\exp(-\frac{c_1\Omega^2}{2}) + \frac{1}{2\min\{\pi_1^2, (1 - \pi_1)^2\}}\exp(-\frac{c_1^2\Omega^2}{8c_4^2})$$

$$\leq \frac{1}{c_0^2}\exp(-\frac{c_1\Omega^2}{2}) + \frac{1}{2c_0^2}\exp(-\frac{c_1^2\Omega^2}{8c_4^2})$$

$$\leq \frac{2}{c_0^2}\exp(-(\frac{c_1}{2} \wedge \frac{c_1^2}{8c_4^2})\Omega^2). \tag{11}$$

We proceed to bound $(II)$. Note that

$$-\frac{1}{\pi_1(1-\pi_1)}\frac{\partial\gamma_{1,\boldsymbol{\xi}}(\mathbf{X}_i,Y_i)}{\partial\,\mathrm{vec}(\boldsymbol{\Gamma}_2-\boldsymbol{\Gamma}_1)}$$

$$=\frac{\exp\{(\mathbf{Z}+\boldsymbol{\delta}(\boldsymbol{\Gamma})\mathbf{f}_i)^T(\boldsymbol{\Gamma}_2-\boldsymbol{\Gamma}_1)\mathbf{f}_i\}}{(\pi_1+(1-\pi_1)\exp\{(\mathbf{Z}+\boldsymbol{\delta}(\boldsymbol{\Gamma})\mathbf{f}_i)^T(\boldsymbol{\Gamma}_2-\boldsymbol{\Gamma}_1)\mathbf{f}_i\})^2}\mathbf{f}_i\otimes(\mathbf{Z}+\boldsymbol{\delta}(\boldsymbol{\Gamma})\mathbf{f}_i)$$

$$=\underbrace{\frac{\partial\gamma_{1,\boldsymbol{\xi}}(\mathbf{X}_i,Y_i)}{\partial\pi_1}\mathbf{f}_i\otimes\mathbf{Z}}_{(II.i)}+\underbrace{\frac{\partial\gamma_{1,\boldsymbol{\xi}}(\mathbf{X}_i,Y_i)}{\partial\pi_1}\mathbf{f}_i\otimes\boldsymbol{\delta}(\boldsymbol{\Gamma})\mathbf{f}_i}_{(II.ii)}.$$

By definition of $\mathbf{H}_i$,

$$\mathbf{f}_i\otimes\mathbf{Z}=\mathbf{f}_i\mathbf{I}_1\otimes\mathbf{H}_i^T\mathbf{H}_i\mathbf{Z}=(\mathbf{f}_i\otimes\mathbf{H}_i^T)(\mathbf{H}_i\mathbf{Z})=(\mathbf{f}_i\otimes\mathbf{H}_i^T)\mathbf{V}.$$

For the first term, we have

$$\mathrm{E}\left[\frac{\partial\gamma_{1,\boldsymbol{\xi}}(\mathbf{X}_i,Y_i)}{\partial\pi_1}\mathbf{f}_i\otimes\mathbf{Z}\right]=\mathrm{E}\left[(\mathbf{f}_i\otimes\mathbf{H}_i^T)\frac{\exp(\|\mathbf{T}_1\|_2V_1+\mathbf{T}_1^T\mathbf{T}_2))}{(\pi_1+(1-\pi_1)\exp\{\|\mathbf{T}_1\|_2V_1+\mathbf{T}_1^T\mathbf{T}_2)\})^2}\mathbf{V}\right]$$

$$=(\mathbf{f}_i\otimes\mathbf{H}_i^T)\,\mathrm{E}\left[\frac{\exp(\|\mathbf{T}_1\|_2V_1+\mathbf{T}_1^T\mathbf{T}_2))}{(\pi_1+(1-\pi_1)\exp\{\|\mathbf{T}_1\|_2V_1+\mathbf{T}_1^T\mathbf{T}_2)\})^2}V_1\mathbf{e}_1\right],$$

where the last inequality uses the fact that $V_1$ and $V_j$ are independent for any $1<j\le p$ and $\mathrm{E}[V_j]=0$. Then

$$\left|\mathrm{E}\left[\frac{\exp(\|\mathbf{T}_1\|_2V_1+\mathbf{T}_1^T\mathbf{T}_2))}{(\pi_1+(1-\pi_1)\exp\{\|\mathbf{T}_1\|_2V_1+\mathbf{T}_1^T\mathbf{T}_2)\})^2}V_1\right]\right|$$

$$=\left|\mathrm{E}\left[\frac{\exp(\|\mathbf{T}_1\|_2Z_1+\mathbf{T}_1^T\mathbf{T}_2))}{(\pi_1+(1-\pi_1)\exp\{\|\mathbf{T}_1\|_2Z_1+\mathbf{T}_1^T\mathbf{T}_2)\})^2}(\|\mathbf{T}_1\|_2Z_1+\mathbf{T}_1^T\mathbf{T}_2-\mathbf{T}_1^T\mathbf{T}_2)\right]\frac{1}{\|\mathbf{T}_1\|_2}\right|$$

$$\le\left|\mathrm{E}\left[\frac{\exp(\|\mathbf{T}_1\|_2Z_1+\mathbf{T}_1^T\mathbf{T}_2))}{(\pi_1+(1-\pi_1)\exp\{\|\mathbf{T}_1\|_2Z_1+\mathbf{T}_1^T\mathbf{T}_2)\})^2}(\|\mathbf{T}_1\|_2Z_1+\mathbf{T}_1^T\mathbf{T}_2)\Big|\mathcal{E}_i\right]P(\mathcal{E}_i)\frac{1}{\|\mathbf{T}_1\|_2}\right|+$$

$$\left|\mathrm{E}\left[\frac{\exp(\|\mathbf{T}_1\|_2Z_1+\mathbf{T}_1^T\mathbf{T}_2))}{(\pi_1+(1-\pi_1)\exp\{\|\mathbf{T}_1\|_2Z_1+\mathbf{T}_1^T\mathbf{T}_2)\})^2}(\|\mathbf{T}_1\|_2Z_1+\mathbf{T}_1^T\mathbf{T}_2)\Big|\mathcal{E}_i^c\right]P(\mathcal{E}_i^c)\frac{1}{\|\mathbf{T}_1\|_2}\right|+$$

$$\left|\mathrm{E}\left[\frac{\exp(\|\mathbf{T}_1\|_2Z_1+\mathbf{T}_1^T\mathbf{T}_2))}{(\pi_1+(1-\pi_1)\exp\{\|\mathbf{T}_1\|_2Z_1+\mathbf{T}_1^T\mathbf{T}_2)\})^2}\mathbf{T}_1^T\mathbf{T}_2\right]\frac{1}{\|\mathbf{T}_1\|_2}\right|$$

For the first term in the last equality, using the fact that on the event $\mathcal{E}_i$, $|\|\mathbf{T}_1\|_2Z_1+\mathbf{T}_1^T\mathbf{T}_2|\ge c_1\Omega^2/2$, it is bounded by

$$\frac{2}{\min\{\pi_1^2,(1-\pi_1)^2\}}\exp(-\frac{3c_1\Omega^2}{8})\frac{1}{c_3\Omega}.$$

The second term is bounded by

$$\frac{1}{\min\{\pi_1^2,(1-\pi_1)^2\}}\exp(-\frac{c_1^2\Omega^2}{8c_4^2})\frac{1}{c_3\Omega}.$$

For the third term, we have

$$\left|\mathrm{E}\left[\frac{\exp(\|\mathbf{T}_1\|_2Z_1+\mathbf{T}_1^T\mathbf{T}_2))}{(\pi_1+(1-\pi_1)\exp\{\|\mathbf{T}_1\|_2Z_1+\mathbf{T}_1^T\mathbf{T}_2)\})^2}\mathbf{T}_1^T\mathbf{T}_2\right]\frac{1}{\|\mathbf{T}_1\|_2}\right|$$

$$\le\sum_{w=1}^2\pi_w^*\,\mathrm{E}\left[\frac{\exp(\|\mathbf{T}_1\|_2Z_1+\mathbf{T}_1^T\mathbf{T}_2))}{(\pi_1+(1-\pi_1)\exp\{\|\mathbf{T}_1\|_2Z_1+\mathbf{T}_1^T\mathbf{T}_2)\})^2}\Big|W_i=w\right]\frac{|\mathbf{T}_1^T\mathbf{T}_2|}{\|\mathbf{T}_1\|_2}$$

$$\le\frac{2}{c_0^2}\exp(-(\frac{c_1}{2}\wedge\frac{c_1^2}{8c_4^2})\Omega^2)c_2\Omega/c_3.$$

Since $(\mathbf{f}_i \otimes \mathbf{H}_i^T)\mathbf{e}_1 = \text{vec}(\mathbf{H}_i^T \mathbf{e}_1 \mathbf{f}_i^T) = \text{vec}(\mathbf{T}_1/\|\mathbf{T}_1\|_2 \mathbf{f}_i^T)$, it follows that,

$$
\begin{aligned}
\| \text{E}[(II.i)]\|_2 &\leq \|(\mathbf{f}_i \otimes \mathbf{H}_i^T)\mathbf{e}_1\|_2 \left[ \frac{2}{\min\{\pi_1^2, (1-\pi_1)^2\}} \exp(-\frac{3c_1\Omega^2}{8})\frac{1}{c_3\Omega} + \right. \\
&\qquad \left. \frac{1}{\min\{\pi_1^2, (1-\pi_1)^2\}} \exp(-\frac{c_1^2\Omega^2}{8c_4^2})\frac{1}{c_3\Omega} + \frac{2}{c_0^2}\exp(-(\frac{c_1}{2} \wedge \frac{c_1^2}{8c_4^2})\Omega^2)c_2\Omega/c_3 \right] \\
&\leq \| \text{vec}(\mathbf{T}_1/\|\mathbf{T}_1\|_2 \mathbf{f}_i^T)\|_2 (\frac{2}{c_0^2 c_3\Omega} \vee \frac{2c_2\Omega}{c_0^2 c_3}) \exp(-(\frac{3c_1}{8} \wedge \frac{c_1^2}{8c_4^2})\Omega^2) \\
&\leq \|\mathbf{f}_i\|_2 (\frac{2}{c_0^2 c_3\Omega} \vee \frac{2c_2\Omega}{c_0^2 c_3}) \exp(-(\frac{3c_1}{8} \wedge \frac{c_1^2}{8c_4^2})\Omega^2) \\
&\leq M_4 (\frac{2}{c_0^2 c_3\Omega} \vee \frac{2c_2\Omega}{c_0^2 c_3}) \exp(-(\frac{3c_1}{8} \wedge \frac{c_1^2}{8c_4^2})\Omega^2).
\end{aligned} \tag{12}
$$

We proceed to bound $(II.ii)$. Note that

$$
\begin{aligned}
\|\boldsymbol{\Gamma}_1^* - \frac{\boldsymbol{\Gamma}_2 + \boldsymbol{\Gamma}_1}{2}\|_F &= \|\frac{\boldsymbol{\Gamma}_1^*}{2} - \frac{\boldsymbol{\Gamma}_1}{2} + \frac{\boldsymbol{\Gamma}_1^*}{2} - \frac{\boldsymbol{\Gamma}_2}{2}\|_F \leq \frac{1}{2}\|\boldsymbol{\Gamma}_1^* - \boldsymbol{\Gamma}_1\|_F + \frac{1}{2}\|\boldsymbol{\Gamma}_1^* - \boldsymbol{\Gamma}_2\|_F \\
&\leq \frac{1}{2}C_b\Omega + \frac{1}{2}\|\boldsymbol{\Gamma}_2^* - \boldsymbol{\Gamma}_2 + \boldsymbol{\Gamma}_1^* - \boldsymbol{\Gamma}_2^*\|_F \\
&\leq \frac{1}{2}C_b\Omega + \frac{1}{2}C_b\Omega + \frac{1}{2\sqrt{M_1}}\Omega = (C_b + \frac{1}{2\sqrt{M_1}})\Omega,
\end{aligned}
$$

and similarly $\|\boldsymbol{\Gamma}_2^* - \frac{\boldsymbol{\Gamma}_2+\boldsymbol{\Gamma}_1}{2}\|_F \leq (C_b + \frac{1}{2\sqrt{M_1}})\Omega$. Therefore $\|\boldsymbol{\delta}(\boldsymbol{\Gamma})\mathbf{f}_i\|_2^2 \big| W_i \leq M_4^2(C_b + \frac{1}{2\sqrt{M_1}})^2\Omega^2$. By the definition of Kronecker product,

$$
\|\mathbf{f}_i \otimes \boldsymbol{\delta}(\boldsymbol{\Gamma})\mathbf{f}_i\|_2^2 \big| W_i = \sum_{j=1}^q \mathbf{f}_{ij}^2 \|\boldsymbol{\delta}(\boldsymbol{\Gamma})\mathbf{f}_i\|_2^2 \big| W_i \leq M_4^4 (C_b + \frac{1}{2\sqrt{M_1}})^2\Omega^2,
$$

where $\mathbf{f}_{ij}$ is the $j$-th element of $\mathbf{f}_i$. Then

$$
\begin{aligned}
\| \text{E}[(II.ii)]\|_2 &= \| \text{E}[\frac{\partial \gamma_{1,\boldsymbol{\xi}}(\mathbf{X}_i, Y_i)}{\partial \pi_1}\mathbf{f}_i \otimes \boldsymbol{\delta}(\boldsymbol{\Gamma})\mathbf{f}_i]\|_2 \\
&\leq \frac{2}{c_0^2}\exp(-(\frac{c_1}{2} \wedge \frac{c_1^2}{8c_4^2})\Omega^2)M_4^2(C_b + \frac{1}{2\sqrt{M_1}})\Omega.
\end{aligned}
$$

Therefore,

$$
\begin{aligned}
\| \text{E}[\frac{\partial \gamma_{1,\boldsymbol{\xi}}(\mathbf{X}_i, Y_i)}{\partial \text{vec}(\boldsymbol{\Gamma}_2 - \boldsymbol{\Gamma}_1)}]\|_2 &\leq \pi_1(1-\pi_1)(\| \text{E}[(II.i)]\|_2 + \| \text{E}[(II.ii)]\|_2) \\
&\leq \frac{M_4}{4}(\frac{2}{c_0^2 c_3\Omega} \vee \frac{2c_2\Omega}{c_0^2 c_3})\exp(-(\frac{3c_1}{8} \wedge \frac{c_1^2}{8c_4^2})\Omega^2)+ \\
&\qquad \frac{1}{2c_0^2}M_4^2(C_b + \frac{1}{2\sqrt{M_1}})\Omega \exp(-(\frac{c_1}{2} \wedge \frac{c_1^2}{8c_4^2})\Omega^2) \\
&\leq c_5 \exp(-(\frac{3c_1}{8} \wedge \frac{c_1^2}{8c_4^2})\Omega^2),
\end{aligned} \tag{13}
$$

where $c_5 = \max\{\frac{M_4}{4}(\frac{2}{c_0^2 c_3\Omega} \vee \frac{2c_2\Omega}{c_0^2 c_3}), \frac{1}{2c_0^2}M_4^2(C_b + \frac{1}{2\sqrt{M_1}})\Omega\}$.

For the term $(III)$, note that

$$
\|\mathbf{f}_i \otimes \mathbf{T}_1\|_2^2 = \sum_{j=1}^q \mathbf{f}_{ij}^2 \|\mathbf{T}_1\|_2^2 = \|\mathbf{f}_i\|_2^2 \cdot \|\mathbf{T}_1\|_2^2 \leq M_4^2 c_4^2\Omega^2.
$$

Then, we have

$$
\begin{aligned}
&\| \operatorname{E}[\frac{\partial \gamma_{1,\boldsymbol{\xi}}(\mathbf{X}_i, Y_i)}{\partial \operatorname{vec}(\boldsymbol{\Gamma}_2 + \boldsymbol{\Gamma}_1)}]\|_2 \\
&= \| \operatorname{E}[\pi_1(1-\pi_1)\frac{\exp\{(\mathbf{Z}+\boldsymbol{\delta}(\boldsymbol{\Gamma})\mathbf{f}_i)^T(\boldsymbol{\Gamma}_2 - \boldsymbol{\Gamma}_1)\mathbf{f}_i\}\cdot\frac{1}{2}\mathbf{f}_i\otimes(\boldsymbol{\Gamma}_2-\boldsymbol{\Gamma}_1)\mathbf{f}_i}{(\pi_1+(1-\pi_1)\exp\{(\mathbf{Z}+\boldsymbol{\delta}(\boldsymbol{\Gamma})\mathbf{f}_i)^T(\boldsymbol{\Gamma}_2-\boldsymbol{\Gamma}_1)\mathbf{f}_i\})^2}]\|_2 \\
&= \frac{\pi_1(1-\pi_1)}{2}\operatorname{E}[\frac{\partial \gamma_{1,\boldsymbol{\xi}}(\mathbf{X}_i, Y_i)}{\partial \pi_1}]\|\mathbf{f}_i\otimes\mathbf{T}_1\|_2 \\
&\leq \frac{1}{8}\frac{2}{c_0^2}\exp(-(\frac{c_1}{2}\wedge\frac{c_1^2}{8c_4^2})\Omega^2)M_4 c_4 \Omega \\
&= \frac{M_4 c_4 \Omega}{4c_0^2}\exp(-(\frac{c_1}{2}\wedge\frac{c_1^2}{8c_4^2})\Omega^2).
\end{aligned}
\tag{14}
$$

Combing results in (11), (13) and (14), we have

$$
\left| \operatorname{E}[\gamma_{1,\boldsymbol{\theta}}(\mathbf{X}_i, Y_i) - \gamma_{1,\boldsymbol{\theta}^*}(\mathbf{X}_i, Y_i)] \right| \leq \kappa_\pi (d_F(\boldsymbol{\theta}, \boldsymbol{\theta}^*) \vee \|\mathbf{B}_1 - \mathbf{B}_1^*\|_F \vee \|\mathbf{B}_2 - \mathbf{B}_2^*\|_F),
\tag{15}
$$

where $\kappa_\pi = (\frac{2}{c_0^2} + c_5 + \frac{M_4 c_4 \Omega}{4c_0^2})\exp(-(\frac{3c_1}{8}\wedge\frac{c_1^2}{8c_4^2})\Omega^2)$.

## D.2. Contraction of matrices $\mathbf{U}_w$

We aim to show

$$
\|\mathbf{U}_1(\boldsymbol{\theta}) - \mathbf{U}_1(\boldsymbol{\theta}^*)\|_F \leq \kappa_{\mathbf{U}}(d_F(\boldsymbol{\theta}, \boldsymbol{\theta}^*) \vee \|\mathbf{B}_1 - \mathbf{B}_1^*\|_F \vee \|\mathbf{B}_2 - \mathbf{B}_2^*\|_F).
$$

By definition,

$$
\mathbf{U}_w(\boldsymbol{\theta}) - \mathbf{U}_w(\boldsymbol{\theta}) = \operatorname{E}(\frac{1}{n}\sum_{i=1}^{n}\{[\gamma_{1,\boldsymbol{\xi}}(\mathbf{X}_i, Y_i) - \gamma_{1,\boldsymbol{\xi}^*}(\mathbf{X}_i, Y_i)]\mathbf{X}_i\mathbf{f}_i^T\}).
$$

When $Y_i$ is fixed, $\mathbf{X}_i$ are not identically distributed. Thus, we bound the expectation of given $i$. Let $\boldsymbol{\xi} = (\pi_1, \operatorname{vec}(\boldsymbol{\Gamma}_2 - \boldsymbol{\Gamma}_1), \operatorname{vec}(\boldsymbol{\Gamma}_2 + \boldsymbol{\Gamma}_1)), \Delta_{\boldsymbol{\xi}} = \boldsymbol{\xi} - \boldsymbol{\xi}^*, \boldsymbol{\xi}_u = \boldsymbol{\xi}^* + u\Delta_{\boldsymbol{\xi}}$. Then $\boldsymbol{\xi}_0 = \boldsymbol{\xi}^*, \boldsymbol{\xi}_1 = \boldsymbol{\xi}$. Define the Jacobian matrix $\mathbf{J} = \partial \mathbf{f}/\partial \mathbf{x} \in \mathbb{R}^{m \times n}$ as $\mathbf{J}_{ij} = \partial f_i/\partial x_j$, where $\mathbf{x} = (x_1, \ldots, x_n)^T$ and $\mathbf{f} = (f_1, \ldots, f_m)^T$. Then

$$
\begin{aligned}
&\operatorname{vec}(\operatorname{E}\{[\gamma_{1,\boldsymbol{\xi}}(\mathbf{X}_i, Y_i) - \gamma_{1,\boldsymbol{\xi}^*}(\mathbf{X}_i, Y_i)]\mathbf{X}_i\mathbf{f}_i^T\}) \\
&= \operatorname{E}\left\{\int_0^1 \frac{\partial \operatorname{vec}(\gamma_{1,\boldsymbol{\xi}}(\mathbf{X}_i, Y_i)\mathbf{X}_i\mathbf{f}_i^T)}{\partial \operatorname{vec}(\boldsymbol{\xi})}\Big|_{\boldsymbol{\xi}=\boldsymbol{\xi}_u}\frac{\partial \operatorname{vec}(\boldsymbol{\xi}_u)}{\partial u}du\right\} \\
&= \operatorname{E}\left\{\int_0^1 \frac{\partial \operatorname{vec}(\gamma_{1,\boldsymbol{\xi}}(\mathbf{X}_i, Y_i)\mathbf{X}_i\mathbf{f}_i^T)}{\partial \pi_1}\Big|_{\boldsymbol{\xi}=\boldsymbol{\xi}_u}\Delta_{\pi_1}du\right\} + \\
&\quad \operatorname{E}\left\{\int_0^1 \frac{\partial \operatorname{vec}(\gamma_{1,\boldsymbol{\xi}}(\mathbf{X}_i, Y_i)\mathbf{X}_i\mathbf{f}_i^T)}{\partial \operatorname{vec}(\boldsymbol{\Gamma}_2 - \boldsymbol{\Gamma}_1)}\Big|_{\boldsymbol{\xi}=\boldsymbol{\xi}_u}\Delta_{\boldsymbol{\Gamma}_2 - \boldsymbol{\Gamma}_1}du\right\} + \\
&\quad \operatorname{E}\left\{\int_0^1 \frac{\partial \operatorname{vec}(\gamma_{1,\boldsymbol{\xi}}(\mathbf{X}_i, Y_i)\mathbf{X}_i\mathbf{f}_i^T)}{\partial \operatorname{vec}(\boldsymbol{\Gamma}_2 + \boldsymbol{\Gamma}_1)}\Big|_{\boldsymbol{\xi}=\boldsymbol{\xi}_u}\Delta_{\boldsymbol{\Gamma}_2 + \boldsymbol{\Gamma}_1}du\right\}.
\end{aligned}
$$

Therefore,

$$
\begin{aligned}
&\| \mathrm{E}\{[\gamma_{1,\boldsymbol{\xi}}(\mathbf{X}_i, Y_i) - \gamma_{1,\boldsymbol{\xi}^*}(\mathbf{X}_i, Y_i)]\mathbf{X}_i\mathbf{f}_i^T\}\|_F \\
&\leq \sqrt{q}\| \mathrm{E}\{[\gamma_{1,\boldsymbol{\xi}}(\mathbf{X}_i, Y_i) - \gamma_{1,\boldsymbol{\xi}^*}(\mathbf{X}_i, Y_i)]\mathbf{X}_i\mathbf{f}_i^T\}\|_2 \\
&\leq \sqrt{q}\| \sup_{\boldsymbol{\xi}\in\mathcal{B}_{\mathrm{con}}(\boldsymbol{\theta}^*)} \mathrm{E}\left(\frac{\partial \mathrm{vec}(\gamma_{1,\boldsymbol{\xi}}(\mathbf{X}_i, Y_i)\mathbf{X}_i\mathbf{f}_i^T)}{\partial \pi_1}\right)\Delta_{\pi_1}\|_2 + \\
&\quad \sqrt{q}\| \sup_{\boldsymbol{\xi}\in\mathcal{B}_{\mathrm{con}}(\boldsymbol{\theta}^*)} \mathrm{E}\left(\frac{\partial \mathrm{vec}(\gamma_{1,\boldsymbol{\xi}}(\mathbf{X}_i, Y_i)\mathbf{X}_i\mathbf{f}_i^T)}{\partial \mathrm{vec}(\boldsymbol{\Gamma}_2 - \boldsymbol{\Gamma}_1)}\right)\Delta_{\boldsymbol{\Gamma}_2 - \boldsymbol{\Gamma}_1}\|_2 + \\
&\quad \sqrt{q}\| \sup_{\boldsymbol{\xi}\in\mathcal{B}_{\mathrm{con}}(\boldsymbol{\theta}^*)} \mathrm{E}\left(\frac{\partial \mathrm{vec}(\gamma_{1,\boldsymbol{\xi}}(\mathbf{X}_i, Y_i)\mathbf{X}_i\mathbf{f}_i^T)}{\partial \mathrm{vec}(\boldsymbol{\Gamma}_2 + \boldsymbol{\Gamma}_1)}\right)\Delta_{\boldsymbol{\Gamma}_2 + \boldsymbol{\Gamma}_1}\|_2 \\
&\leq \sqrt{q} \underbrace{\sup_{\boldsymbol{\xi}\in\mathcal{B}_{\mathrm{con}}(\boldsymbol{\theta}^*)} \| \mathrm{E}\left(\frac{\partial \gamma_{1,\boldsymbol{\xi}}(\mathbf{X}_i, Y_i)}{\partial \pi_1} \mathrm{vec}^T(\mathbf{X}_i\mathbf{f}_i^T)\right)\|_2 |\pi_1 - \pi_1^*|}_{(I)} + \\
&\quad \sqrt{q} \underbrace{\sup_{\boldsymbol{\xi}\in\mathcal{B}_{\mathrm{con}}(\boldsymbol{\theta}^*)} \| \mathrm{E}\left(\frac{\partial \gamma_{1,\boldsymbol{\xi}}(\mathbf{X}_i, Y_i)}{\partial \mathrm{vec}(\boldsymbol{\Gamma}_2 - \boldsymbol{\Gamma}_1)} \mathrm{vec}^T(\mathbf{X}_i\mathbf{f}_i^T)\right)\|_2 \|\boldsymbol{\Gamma}_1 - \boldsymbol{\Gamma}_1^* - \boldsymbol{\Gamma}_2 + \boldsymbol{\Gamma}_2^*\|_F}_{(II)} + \\
&\quad \sqrt{q} \underbrace{\sup_{\boldsymbol{\xi}\in\mathcal{B}_{\mathrm{con}}(\boldsymbol{\theta}^*)} \| \mathrm{E}\left(\frac{\partial \gamma_{1,\boldsymbol{\xi}}(\mathbf{X}_i, Y_i)}{\partial \mathrm{vec}(\boldsymbol{\Gamma}_2 + \boldsymbol{\Gamma}_1)} \mathrm{vec}^T(\mathbf{X}_i\mathbf{f}_i^T)\right)\|_2 \|\boldsymbol{\Gamma}_1 - \boldsymbol{\Gamma}_1^* + \boldsymbol{\Gamma}_2 - \boldsymbol{\Gamma}_2^*\|_F}_{(III)}.
\end{aligned}
$$

Note that $\mathrm{vec}(\mathbf{X}_i\mathbf{f}_i^T) = \mathrm{vec}[(\mathbf{Z} + \boldsymbol{\psi}\mathbf{f}_i)\mathbf{f}_i^T] = (\mathbf{f}_i \otimes \mathbf{I}_p)(\mathbf{Z} + \boldsymbol{\psi}\mathbf{f}_i)$. For the second term (II), we have

$$
\begin{aligned}
&-\frac{1}{\pi_1(1-\pi_1)}\frac{\partial \gamma_{1,\boldsymbol{\xi}}(\mathbf{X}_i, Y_i)}{\partial \mathrm{vec}(\boldsymbol{\Gamma}_2 - \boldsymbol{\Gamma}_1)} \mathrm{vec}^T(\mathbf{X}_i\mathbf{f}_i^T) \\
&= \frac{\exp((\mathbf{Z} + \boldsymbol{\delta}(\boldsymbol{\Gamma})\mathbf{f}_i)^T((\boldsymbol{\Gamma}_2 - \boldsymbol{\Gamma}_1)\mathbf{f}_i)}{\{\pi_1 + (1-\pi_1)\exp((\mathbf{Z} + \boldsymbol{\delta}(\boldsymbol{\Gamma})\mathbf{f}_i)^T((\boldsymbol{\Gamma}_2 - \boldsymbol{\Gamma}_1)\mathbf{f}_i))\}^2} \cdot (\mathbf{f}_i \otimes \mathbf{I}_p)(\mathbf{Z} + \boldsymbol{\delta}(\boldsymbol{\Gamma})\mathbf{f}_i)\mathrm{vec}^T(\mathbf{X}_i\mathbf{f}_i^T) \\
&= \frac{\partial \gamma_{1,\boldsymbol{\xi}}(\mathbf{X}_i, Y_i)}{\partial \pi_1} \cdot (\mathbf{f}_i \otimes \mathbf{I}_p)(\mathbf{Z} + \boldsymbol{\delta}(\boldsymbol{\Gamma})\mathbf{f}_i)(\mathbf{Z} + \boldsymbol{\psi}\mathbf{f}_i)^T (\mathbf{f}_i^T \otimes \mathbf{I}_p) \\
&= \underbrace{\frac{\partial \gamma_{1,\boldsymbol{\xi}}(\mathbf{X}_i, Y_i)}{\partial \pi_1} \cdot (\mathbf{f}_i \otimes \mathbf{I}_p)\mathbf{Z}\mathbf{Z}^T(\mathbf{f}_i^T \otimes \mathbf{I}_p)}_{(II.i)} + \underbrace{\frac{\partial \gamma_{1,\boldsymbol{\xi}}(\mathbf{X}_i, Y_i)}{\partial \pi_1} \cdot (\mathbf{f}_i \otimes \mathbf{I}_p)\mathbf{Z}\mathbf{f}_i^T\boldsymbol{\psi}^T(\mathbf{f}_i^T \otimes \mathbf{I}_p)}_{(II.ii)} + \\
&\quad \underbrace{\frac{\partial \gamma_{1,\boldsymbol{\xi}}(\mathbf{X}_i, Y_i)}{\partial \pi_1} \cdot (\mathbf{f}_i \otimes \mathbf{I}_p)\boldsymbol{\delta}(\boldsymbol{\Gamma})\mathbf{f}_i\mathbf{Z}^T(\mathbf{f}_i^T \otimes \mathbf{I}_p)}_{(II.iii)} + \underbrace{\frac{\partial \gamma_{1,\boldsymbol{\xi}}(\mathbf{X}_i, Y_i)}{\partial \pi_1} \cdot (\mathbf{f}_i \otimes \mathbf{I}_p)\boldsymbol{\delta}(\boldsymbol{\Gamma})\mathbf{f}_i\mathbf{f}_i^T\boldsymbol{\psi}^T(\mathbf{f}_i^T \otimes \mathbf{I}_p)}_{(II.iv)}.
\end{aligned}
$$

Recall that $\mathbf{T}_1 = (\boldsymbol{\Gamma}_2 - \boldsymbol{\Gamma}_1)\mathbf{f}_i$, $\mathbf{T}_2 = \boldsymbol{\delta}(\boldsymbol{\Gamma})\mathbf{f}_i$, $\mathbf{H}_i$ is an orthonormal matrix whose first row is $\mathbf{T}_1^T/\|\mathbf{T}_1\|_2$, $\mathbf{V} = \mathbf{H}_i\mathbf{Z} \sim$

$N(0, \mathbf{I}_p)$. For the first term, we have

$$
\begin{aligned}
&\mathrm{E}[(II.i)] \\
&= (\mathbf{f}_i \otimes \mathbf{I}_p)\, \mathrm{E}[\frac{\partial \gamma_{1,\boldsymbol{\xi}}(\mathbf{X}_i, Y_i)}{\partial \pi_1} \mathbf{Z}\mathbf{Z}^T](\mathbf{f}_i^T \otimes \mathbf{I}_p) \\
&= (\mathbf{f}_i \otimes \mathbf{I}_p)\mathbf{H}_i^T\, \mathrm{E}[\frac{\partial \gamma_{1,\boldsymbol{\xi}}(\mathbf{X}_i, Y_i)}{\partial \pi_1} \mathbf{H}_i\mathbf{Z}\mathbf{Z}^T\mathbf{H}_i^T](\mathbf{f}_i^T \otimes \mathbf{I}_p)\mathbf{H}_i \\
&= (\mathbf{f}_i \otimes \mathbf{I}_p)\mathbf{H}_i^T\, \mathrm{E}[\frac{\partial \gamma_{1,\boldsymbol{\xi}}(\mathbf{X}_i, Y_i)}{\partial \pi_1} \mathbf{V}\mathbf{V}^T](\mathbf{f}_i^T \otimes \mathbf{I}_p)\mathbf{H}_i \\
&= (\mathbf{f}_i \otimes \mathbf{I}_p)\mathbf{H}_i^T\, \mathrm{E}[\frac{\exp(\|\mathbf{T}_1\|_2 V_1 + \mathbf{T}_1^T\mathbf{T}_2))}{(\pi_1 + (1-\pi_1)\exp\{\|\mathbf{T}_1\|_2 V_1 + \mathbf{T}_1^T\mathbf{T}_2)\})^2} \mathbf{V}\mathbf{V}^T](\mathbf{f}_i^T \otimes \mathbf{I}_p)\mathbf{H}_i \ (\text{by}(10)) \\
&= (\mathbf{f}_i \otimes \mathbf{I}_p)\mathbf{H}_i^T\, \underbrace{\mathrm{E}[\frac{\exp(\|\mathbf{T}_1\|_2 V_1 + \mathbf{T}_1^T\mathbf{T}_2))}{(\pi_1 + (1-\pi_1)\exp\{\|\mathbf{T}_1\|_2 V_1 + \mathbf{T}_1^T\mathbf{T}_2)\})^2} (V_1^2 - 1)]}_{(II.i.a)} \mathbf{e}_1\mathbf{e}_1^T(\mathbf{f}_i^T \otimes \mathbf{I}_p)\mathbf{H}_i + \\
&\quad (\mathbf{f}_i \otimes \mathbf{I}_p)\mathbf{H}_i^T\, \underbrace{\mathrm{E}[\frac{\exp(\|\mathbf{T}_1\|_2 V_1 + \mathbf{T}_1^T\mathbf{T}_2))}{(\pi_1 + (1-\pi_1)\exp\{\|\mathbf{T}_1\|_2 V_1 + \mathbf{T}_1^T\mathbf{T}_2)\})^2}]}_{(II.i.b)} \mathbf{I}_p(\mathbf{f}_i^T \otimes \mathbf{I}_p)\mathbf{H}_i.
\end{aligned}
$$

Recall that the event $\mathcal{E}_i = \{|\|\mathbf{T}_1\|_2 Z_1| \leq \frac{c_1}{2}\Omega^2\}$ and on the event $\mathcal{E}_i$, $|\|\mathbf{T}_1\|_2 Z_1 + \mathbf{T}_1^T\mathbf{T}_2| \geq |\mathbf{T}_1^T\mathbf{T}_2| - |\mathbf{T}_1\|_2 Z_1| \geq c_1\Omega^2/2$. Let $t = \|\mathbf{T}_1\|_2 Z_1 + \mathbf{T}_1^T\mathbf{T}_2$, $g(t) = \frac{\exp(t)}{(\pi_1 + (1-\pi_1)\exp\{t\})^2}$. Then

$$
\begin{aligned}
&|\mathrm{E}[(II.i.a)]| \\
&= \left| \mathrm{E}[\frac{\exp(\|\mathbf{T}_1\|_2 Z_1 + \mathbf{T}_1^T\mathbf{T}_2)}{(\pi_1 + (1-\pi_1)\exp\{\|\mathbf{T}_1\|_2 Z_1 + \mathbf{T}_1^T\mathbf{T}_2)\})^2}(Z_1^2 - 1)] \right| \\
&\leq \left| \mathrm{E}[g(t)(Z_1^2 - 1)|\mathcal{E}_i]P(\mathcal{E}_i) \right| + \left| \mathrm{E}[g(Z_1)(Z_1^2 - 1)|\mathcal{E}_i^c]P(\mathcal{E}_i^c) \right| \\
&= \left| \mathrm{E}[g(t)(\frac{(\|\mathbf{T}_1\|_2 Z_1 + \mathbf{T}_1^T\mathbf{T}_2)^2 - 2\mathbf{T}_1^T\mathbf{T}_2(\|\mathbf{T}_1\|_2 Z_1 + \mathbf{T}_1^T\mathbf{T}_2) + (\mathbf{T}_1^T\mathbf{T}_2)^2 - \|\mathbf{T}_1\|_2^2}{\|\mathbf{T}_1\|_2^2})|\mathcal{E}_i]P(\mathcal{E}_i) \right| + \\
&\quad \left| \mathrm{E}[g(t)(\frac{(\|\mathbf{T}_1\|_2 Z_1 + \mathbf{T}_1^T\mathbf{T}_2)^2 - 2\mathbf{T}_1^T\mathbf{T}_2(\|\mathbf{T}_1\|_2 Z_1 + \mathbf{T}_1^T\mathbf{T}_2) + (\mathbf{T}_1^T\mathbf{T}_2)^2 - \|\mathbf{T}_1\|_2^2}{\|\mathbf{T}_1\|_2^2})|\mathcal{E}_i^c]P(\mathcal{E}_i^c) \right| \\
&\leq \frac{1}{\|\mathbf{T}_1\|_2^2}\left\{ \left| \mathrm{E}[g(t)t^2|\mathcal{E}_i] \right| + \left| \mathrm{E}[g(t)(\|\mathbf{T}_1\|_2^2 - (\mathbf{T}_1^T\mathbf{T}_2)^2)|\mathcal{E}_i] \right| + \left| \mathrm{E}[g(t) \cdot 2\mathbf{T}_1^T\mathbf{T}_2 t|\mathcal{E}_i] \right| \right\} + \\
&\quad \frac{1}{\|\mathbf{T}_1\|_2^2}\left\{ \left| \mathrm{E}[g(t)t^2|\mathcal{E}_i^c] \right| + \left| \mathrm{E}[g(t)(\|\mathbf{T}_1\|_2^2 - (\mathbf{T}_1^T\mathbf{T}_2)^2)|\mathcal{E}_i] \right| + \left| \mathrm{E}[g(t) \cdot 2\mathbf{T}_1^T\mathbf{T}_2 t|\mathcal{E}_i] \right| \right\} P(\mathcal{E}_i^c) + \\
&\leq \frac{1}{c_3^2\Omega^2}\left\{ \frac{4}{\min\{\pi_1^2, (1-\pi_1)^2\}}\exp(-\frac{c_1}{4}\Omega^2) + (c_4^2 + c_2^2)\Omega^2\frac{2}{c_0^2}\exp(-(\frac{c_1}{2} \wedge \frac{c_1^2}{8c_4^2})\Omega^2) + \right. \\
&\quad \left. 2c_2\Omega^2\frac{2}{\min\{\pi_1^2, (1-\pi_1)^2\}}\exp(-\frac{3c_1}{8}\Omega^2) \right\} + \frac{1}{c_3^2\Omega^2}\left\{ \frac{4}{\min\{\pi_1^2, (1-\pi_1)^2\}} + \frac{(c_4^2 + c_2^2)\Omega^2}{4\pi_1(1-\pi_1)} + \right. \\
&\quad \left. \frac{c_2\Omega^2}{\min\{\pi_1^2, (1-\pi_1)^2\}} \right\} 2\exp(-\frac{c_1^2\Omega^2}{8c_4^2}) \\
&\leq \frac{4 + 2(c_2^2 + c_4^2)\Omega^2 + 4c_2\Omega^2}{c_3^2 c_0^2\Omega^2}\exp(-(\frac{c_1}{4} \wedge \frac{c_1^2}{8c_4^2})\Omega^2) + \frac{8 + (c_2^2 + c_4^2)\Omega^2 + 2c_2\Omega^2}{c_3^2 c_0^2\Omega^2}\exp(-\frac{c_1^2\Omega^2}{8c_4^2}).
\end{aligned}
$$

Let $c_5 = \frac{4 + 2(c_2^2 + c_4^2)\Omega^2 + 4c_2\Omega^2}{c_3^2 c_0^2\Omega^2} + \frac{8 + (c_2^2 + c_4^2)\Omega^2 + 2c_2\Omega^2}{c_3^2 c_0^2\Omega^2}$. We have

$$
|\mathrm{E}[(II.i.a)]| \leq c_5\exp(-(\frac{c_1}{4} \wedge \frac{c_1^2}{8c_4^2})\Omega^2). \tag{16}
$$

The term $(II.i.b)$ can be bounded same as (11),

$$| \mathrm{E}[(II.i.b)]| \leq \frac{2}{c_0^2} \exp(-(\frac{c_1}{2} \wedge \frac{c_1^2}{8c_4^2})\Omega^2). \tag{17}$$

Then we bound the spectral norm of $\mathrm{E}[(II.i)]$,

$$
\begin{aligned}
&\| \mathrm{E}[(II.i)]\|_2 \\
&\leq \|(\mathbf{f}_i \otimes \mathbf{I}_p)\mathbf{H}_i^T\mathbf{e}_1\mathbf{e}_1^T(\mathbf{f}_i^T \otimes \mathbf{I}_p)\mathbf{H}_i\|_2 \cdot | \mathrm{E}[(II.i.a)]|+ \\
&\quad \|(\mathbf{f}_i \otimes \mathbf{I}_p)\mathbf{H}_i^T\mathbf{I}_p(\mathbf{f}_i^T \otimes \mathbf{I}_p)\mathbf{H}_i\|_2 \cdot | \mathrm{E}[(II.i.b)]| \\
&\leq M_4^2| \mathrm{E}[(II.i.a)]| + M_4^2| \mathrm{E}[(II.i.a)]| \\
&\leq M_4^2 c_5 \exp(-(\frac{c_1}{4} \wedge \frac{c_1^2}{8c_4^2})\Omega^2) + M_4^2 \frac{2}{c_0^2} \exp(-(\frac{c_1}{2} \wedge \frac{c_1^2}{8c_4^2})\Omega^2) \\
&\leq M_4^2 (c_5 + \frac{2}{c_0^2}) \exp(-(\frac{c_1}{4} \wedge \frac{c_1^2}{8c_4^2})\Omega^2).
\end{aligned} \tag{18}
$$

The other three terms are easier to bound. Note that

$$
\begin{aligned}
\mathrm{E}[(II.ii)] &= (\mathbf{f}_i \otimes \mathbf{I}_p) \mathrm{E}[\frac{\partial \gamma_{1,\boldsymbol{\xi}}(\mathbf{X}_i, Y_i)}{\partial \pi_1}\mathbf{Z}\mathbf{f}_i^T\boldsymbol{\psi}^T](\mathbf{f}_i^T \otimes \mathbf{I}_p) \\
&= (\mathbf{f}_i \otimes \mathbf{I}_p)\mathbf{H}_i^T \mathrm{E}[\frac{\partial \gamma_{1,\boldsymbol{\xi}}(\mathbf{X}_i, Y_i)}{\partial \pi_1}V_1\mathbf{e}_1\mathbf{f}_i^T\boldsymbol{\psi}^T](\mathbf{f}_i^T \otimes \mathbf{I}_p).
\end{aligned}
$$

Using the similar technique in (12), we have

$$
\begin{aligned}
\|E[\frac{\partial \gamma_{1,\boldsymbol{\xi}}(\mathbf{X}_i, Y_i)}{\partial \pi_1}V_1\mathbf{e}_1\mathbf{f}_i^T\boldsymbol{\psi}^T]\|_2 &\leq |E[\frac{\partial \gamma_{1,\boldsymbol{\xi}}(\mathbf{X}_i, Y_i)}{\partial \pi_1}V_1| M_4 M_b \\
&\leq M_4 M_b (\frac{2}{c_0^2 c_3 \Omega} \vee \frac{2c_2\Omega}{c_0^2 c_3}) \exp(-(\frac{3c_1}{8} \wedge \frac{c_1^2}{8c_4^2})).
\end{aligned}
$$

Therefore,

$$
\begin{aligned}
&\| \mathrm{E}[(II.ii)]\|_2 \\
&\leq \|(\mathbf{f}_i \otimes \mathbf{I}_p)\mathbf{H}_i^T\|_2 \cdot \|E[\frac{\partial \gamma_{1,\boldsymbol{\xi}}(\mathbf{X}_i, Y_i)}{\partial \pi_1}V_1\mathbf{e}_1\mathbf{f}_i^T\boldsymbol{\psi}^T]\|_2 \cdot \|(\mathbf{f}_i^T \otimes \mathbf{I}_p)\|_2 \\
&\leq M_4^3 M_b (\frac{2}{c_0^2 c_3 \Omega} \vee \frac{2c_2\Omega}{c_0^2 c_3}) \exp(-(\frac{3c_1}{8} \wedge \frac{c_1^2}{8c_4^2})).
\end{aligned} \tag{19}
$$

We proceed to bound $(II.iii)$,

$$
\begin{aligned}
\mathrm{E}[(II.iii)] &= (\mathbf{f}_i \otimes \mathbf{I}_p) \mathrm{E}[\frac{\partial \gamma_{1,\boldsymbol{\xi}}(\mathbf{X}_i, Y_i)}{\partial \pi_1}\boldsymbol{\delta}(\boldsymbol{\Gamma})\mathbf{f}_i\mathbf{Z}^T](\mathbf{f}_i^T \otimes \mathbf{I}_p) \\
&= (\mathbf{f}_i \otimes \mathbf{I}_p) \mathrm{E}[\frac{\partial \gamma_{1,\boldsymbol{\xi}}(\mathbf{X}_i, Y_i)}{\partial \pi_1}\boldsymbol{\delta}(\boldsymbol{\Gamma})\mathbf{f}_i\mathbf{e}_1^T V_1]\mathbf{H}_i(\mathbf{f}_i^T \otimes \mathbf{I}_p).
\end{aligned}
$$

Using the similar technique in (12) and $\|\boldsymbol{\delta}(\boldsymbol{\Gamma})\mathbf{f}_i\|_2^2 \leq M_4^2(C_b + \frac{1}{2\sqrt{M_1}})^2\Omega^2$, we have

$$\| \mathrm{E}[\frac{\partial \gamma_{1,\boldsymbol{\xi}}(\mathbf{X}_i, Y_i)}{\partial \pi_1}\boldsymbol{\delta}(\boldsymbol{\Gamma})\mathbf{f}_i\mathbf{e}_1^T V_1]\|_2 \leq M_4(C_b + \frac{1}{2\sqrt{M_1}})(\frac{2}{c_0^2 c_3} \vee \frac{2c_2\Omega^2}{c_0^2 c_3}) \exp(-(\frac{3c_1}{8} \wedge \frac{c_1^2}{8c_4^2})).$$

Therefore,

$$
\begin{aligned}
&\| \mathrm{E}[(II.iii)]\|_2 \\
&\leq \|(\mathbf{f}_i \otimes \mathbf{I}_p)\|_2 \cdot \| \mathrm{E}[\frac{\partial \gamma_{1,\boldsymbol{\xi}}(\mathbf{X}_i, Y_i)}{\partial \pi_1}\boldsymbol{\delta}(\boldsymbol{\Gamma})\mathbf{f}_i\mathbf{e}_1^T V_1]\|_2 \cdot \|\mathbf{H}_i(\mathbf{f}_i^T \otimes \mathbf{I}_p)\|_2 \\
&\leq M_4^3(C_b + \frac{1}{2\sqrt{M_1}})(\frac{2}{c_0^2 c_3} \vee \frac{2c_2\Omega^2}{c_0^2 c_3}) \exp(-(\frac{3c_1}{8} \wedge \frac{c_1^2}{8c_4^2})).
\end{aligned} \tag{20}
$$

Finally, we bound $(II.iv)$,

$$\| \operatorname{E}[(II.iv)]\|_2$$
$$= \|(\mathbf{f}_i \otimes \mathbf{I}_p) \operatorname{E}[\frac{\partial \gamma_{1,\boldsymbol{\xi}}(\mathbf{X}_i, Y_i)}{\partial \pi_1} \cdot \boldsymbol{\delta}(\boldsymbol{\Gamma})\mathbf{f}_i\mathbf{f}_i^T\boldsymbol{\psi}^T](\mathbf{f}_i^T \otimes \mathbf{I}_p)\|_2$$
$$\leq M_4^4 M_b(C_b + \frac{1}{2\sqrt{M_1}})\Omega\frac{2}{c_0^2}\exp(-(\frac{c_1}{2} \wedge \frac{c_1^2}{8c_4^2})\Omega^2). \tag{21}$$

Combing results (18), (19), (20), and (21), we have

$$\| \operatorname{E}\left[\frac{\partial \gamma_{1,\boldsymbol{\xi}}(\mathbf{X}_i, Y_i)}{\partial \operatorname{vec}(\boldsymbol{\Gamma}_2 - \boldsymbol{\Gamma}_1)}\operatorname{vec}^T(\mathbf{X}_i\mathbf{f}_i^T)\right]\|_2$$
$$\leq \pi_1(1 - \pi_1)(\| \operatorname{E}[(II.i)]\|_2 + \| \operatorname{E}[(II.ii)]\|_2 + \| \operatorname{E}[(II.iii)]\|_2 + \| \operatorname{E}[(II.iv)]\|_2)$$
$$\leq \frac{1}{4}\left\{ M_4^2(c_5 + \frac{2}{c_0^2})\exp(-(\frac{c_1}{4} \wedge \frac{c_1^2}{8c_4^2})\Omega^2) + M_4^3 M_b(\frac{2}{c_0^2c_3\Omega} \vee \frac{2c_2\Omega}{c_0^2c_3})\exp(-(\frac{3c_1}{8} \wedge \frac{c_1^2}{8c_4^2}) + \right.$$
$$M_4^3(C_b + \frac{1}{2\sqrt{M_1}})(\frac{2}{c_0^2c_3} \vee \frac{2c_2\Omega^2}{c_0^2c_3})\exp(-(\frac{3c_1}{8} \wedge \frac{c_1^2}{8c_4^2})+$$
$$\left. M_4^4 M_b(C_b + \frac{1}{2\sqrt{M_1}})\Omega\frac{2}{c_0^2}\exp(-(\frac{c_1}{2} \wedge \frac{c_1^2}{8c_4^2})\Omega^2)\right\}$$
$$\leq c_6\exp(-(\frac{c_1}{4} \wedge \frac{c_1^2}{8c_4^2})\Omega^2), \tag{22}$$

where $c_6 = \frac{1}{4}\left\{ M_4^2(c_5 + \frac{2}{c_0^2}) + M_4^3 M_b(\frac{2}{c_0^2c_3\Omega} \vee \frac{2c_2\Omega}{c_0^2c_3}) + M_4^3(C_b + \frac{1}{2\sqrt{M_1}})(\frac{2}{c_0^2c_3} \vee \frac{2c_2\Omega^2}{c_0^2c_3})+ \right.$
$\left. M_4^4 M_b(C_b + \frac{1}{2\sqrt{M_1}})\Omega\frac{2}{c_0^2}\right\}$.

Next, we bound the first term $(I)$,

$$\| \operatorname{E}\left[\frac{\partial \gamma_{1,\boldsymbol{\xi}}(\mathbf{X}_i, Y_i)}{\partial \pi_1}\operatorname{vec}^T(\mathbf{X}_i\mathbf{f}_i^T)\right]\|_2$$
$$= \| \operatorname{E}\left[\frac{\partial \gamma_{1,\boldsymbol{\xi}}(\mathbf{X}_i, Y_i)}{\partial \pi_1}(\mathbf{f}_i \otimes \mathbf{I}_p)(\mathbf{Z} + \boldsymbol{\psi}\mathbf{f}_i)\right]\|_2$$
$$\leq \underbrace{\|(\mathbf{f}_i \otimes \mathbf{I}_p)\operatorname{E}\left[\frac{\partial \gamma_{1,\boldsymbol{\xi}}(\mathbf{X}_i, Y_i)}{\partial \pi_1}\mathbf{Z}\right]\|_2}_{(I.i)} + \underbrace{\|(\mathbf{f}_i \otimes \mathbf{I}_p)\operatorname{E}\left[\frac{\partial \gamma_{1,\boldsymbol{\xi}}(\mathbf{X}_i, Y_i)}{\partial \pi_1}\boldsymbol{\psi}\mathbf{f}_i\right]\|_2}_{(I.ii)}.$$

Similarly to (12), we have

$$(I.i) = \|(\mathbf{f}_i \otimes \mathbf{I}_p)\mathbf{H}_i^T \operatorname{E}\left[\frac{\partial \gamma_{1,\boldsymbol{\xi}}(\mathbf{X}_i, Y_i)}{\partial \pi_1}\mathbf{V}\right]\|_2$$
$$= \|(\mathbf{f}_i \otimes \mathbf{I}_p)\mathbf{H}_i^T\mathbf{e}_1 \operatorname{E}\left[\frac{\exp(\|\mathbf{T}_1\|_2 V_1 + \mathbf{T}_1^T\mathbf{T}_2))}{(\pi_1 + (1 - \pi_1)\exp\{\|\mathbf{T}_1\|_2 V_1 + \mathbf{T}_1^T\mathbf{T}_2)\})^2}V_1\right]\|_2$$
$$\leq M_4(\frac{2}{c_0^2c_3\Omega} \vee \frac{2c_2\Omega}{c_0^2c_3})\exp(-(\frac{3c_1}{8} \wedge \frac{c_1^2}{8c_4^2})\Omega^2). \tag{23}$$

According to (11),

$$(I.ii) \leq M_4^2 M_b\frac{2}{c_0^2}\exp(-(\frac{c_1}{2} \wedge \frac{c_1^2}{8c_4^2})\Omega^2). \tag{24}$$

Therefore,

$$\| \mathrm{E}\left[\frac{\partial\gamma_{1,\boldsymbol{\xi}}(\mathbf{X}_i,Y_i)}{\partial\pi_1}\mathrm{vec}^T(\mathbf{X}_i\mathbf{f}_i^T)\right]\|_2$$
$$\leq M_4(\frac{2}{c_0^2 c_3 \Omega}\vee\frac{2c_2\Omega}{c_0^2 c_3})\exp(-(\frac{3c_1}{8}\wedge\frac{c_1^2}{8c_4^2})\Omega^2)+M_4^2 M_b\frac{2}{c_0^2}\exp(-(\frac{c_1}{2}\wedge\frac{c_1^2}{8c_4^2})\Omega^2)$$
$$\leq c_7\exp(-(\frac{3c_1}{8}\wedge\frac{c_1^2}{8c_4^2})\Omega^2),\tag{25}$$

where $c_7 = M_4(\frac{2}{c_0^2 c_3 \Omega}\vee\frac{2c_2\Omega}{c_0^2 c_3})+M_4^2 M_b\frac{2}{c_0^2}$.

For the third term, using (25) and the fact that $\|\mathbf{f}_i\otimes(\boldsymbol{\Gamma}_2-\boldsymbol{\Gamma}_1)\mathbf{f}_i\|_2\leq M_4 c_4\Omega$,

$$\| \mathrm{E}\left[\frac{\partial\gamma_{1,\boldsymbol{\xi}}(\mathbf{X}_i,Y_i)}{\partial\mathrm{vec}(\boldsymbol{\Gamma}_2+\boldsymbol{\Gamma}_1)}\mathrm{vec}^T(\mathbf{X}_i\mathbf{f}_i^T)\right]\|_2$$
$$=\|\frac{\pi_1(1-\pi_1)}{2}\mathbf{f}_i\otimes(\boldsymbol{\Gamma}_2-\boldsymbol{\Gamma}_1)\mathbf{f}_i\,\mathrm{E}[\frac{\partial\gamma_{1,\boldsymbol{\xi}}(\mathbf{X}_i,Y_i)}{\partial\pi_1}\mathrm{vec}^T(\mathbf{X}_i\mathbf{f}_i^T)]\|_2$$
$$\leq\frac{1}{8}\|\mathbf{f}_i\otimes(\boldsymbol{\Gamma}_2-\boldsymbol{\Gamma}_1)\mathbf{f}_i\|_2\cdot\| \mathrm{E}[\frac{\partial\gamma_{1,\boldsymbol{\xi}}(\mathbf{X}_i,Y_i)}{\partial\pi_1}\mathrm{vec}^T(\mathbf{X}_i\mathbf{f}_i^T)]\|_2$$
$$\leq\frac{M_4 c_4\Omega}{8}c_7\exp(-(\frac{3c_1}{8}\wedge\frac{c_1^2}{8c_4^2})\Omega^2).\tag{26}$$

Combining results (22), (25) and (26), we have

$$\| \mathrm{E}\{[\gamma_{1,\boldsymbol{\xi}}(\mathbf{X}_i,Y_i)-\gamma_{1,\boldsymbol{\xi}^*}(\mathbf{X}_i,Y_i)]\mathbf{X}_i\mathbf{f}_i^T\}\|_F$$
$$\leq\sqrt{q}d_F(\boldsymbol{\theta},\boldsymbol{\theta}^*)\left\{c_6\exp(-(\frac{c_1}{4}\wedge\frac{c_1^2}{8c_4^2})\Omega^2)+c_7\exp(-(\frac{3c_1}{8}\wedge\frac{c_1^2}{8c_4^2})\Omega^2)+\right.$$
$$\left.\frac{M_4 c_4\Omega}{8}c_7\exp(-(\frac{3c_1}{8}\wedge\frac{c_1^2}{8c_4^2})\Omega^2)\right\}$$
$$\leq\sqrt{q}d_F(\boldsymbol{\theta},\boldsymbol{\theta}^*)[c_6+c_7+\frac{M_4 c_4\Omega}{8}c_7]\exp(-(\frac{3c_1}{8}\wedge\frac{c_1^2}{8c_4^2})\Omega^2)$$
$$\leq\kappa_{\mathbf{U}}(d_F(\boldsymbol{\theta},\boldsymbol{\theta}^*)\vee\|\mathbf{B}_1-\mathbf{B}_1^*\|_F\vee\|\mathbf{B}_2-\mathbf{B}_2^*\|_F).\tag{27}$$

where $\kappa_{\mathbf{U}}=\sqrt{q}[c_6+c_7+\frac{M_4 c_4\Omega}{8}c_7]\exp[-(\frac{3c_1}{8}\wedge\frac{c_1^2}{8c_4^2})\Omega^2]$.

### D.3. Contraction of covariance matrices

We aim to show

$$\|[\boldsymbol{\Sigma}_1(\boldsymbol{\theta})-\boldsymbol{\Sigma}_1(\boldsymbol{\theta}^*)]\mathbf{B}_1^*\|_F\leq\kappa_{\boldsymbol{\Sigma}}(d_F(\boldsymbol{\theta},\boldsymbol{\theta}^*)\vee\|\mathbf{B}_1-\mathbf{B}_1^*\|_F\vee\|\mathbf{B}_2-\mathbf{B}_2^*\|_F).$$

By definition,

$$[\boldsymbol{\Sigma}_1(\boldsymbol{\theta})-\boldsymbol{\Sigma}_1(\boldsymbol{\theta}^*)]\mathbf{B}_1^*=\mathrm{E}(\frac{1}{n}\sum_{i=1}^{n}\{[\gamma_{1,\boldsymbol{\xi}}(\mathbf{X}_i,Y_i)-\gamma_{1,\boldsymbol{\xi}^*}(\mathbf{X}_i,Y_i)]\mathbf{X}_i\mathbf{X}_i^T\mathbf{B}_1^*\}).$$

Using the same argument as before, we obtain

$$
\begin{aligned}
& \| \mathrm{E}\{[\gamma_{1,\boldsymbol{\xi}}(\mathbf{X}_i, Y_i) - \gamma_{1,\boldsymbol{\xi}^*}(\mathbf{X}_i, Y_i)]\mathbf{X}_i\mathbf{X}_i^T\mathbf{B}_1^*\}\|_F \\
& \leq \sqrt{q}\| \mathrm{E}\{[\gamma_{1,\boldsymbol{\xi}}(\mathbf{X}_i, Y_i) - \gamma_{1,\boldsymbol{\xi}^*}(\mathbf{X}_i, Y_i)]\mathbf{X}_i\mathbf{X}_i^T\mathbf{B}_1^*\}\|_2 \\
& \leq \sqrt{q}\| \sup_{\boldsymbol{\xi}\in\mathcal{B}_{\mathrm{con}}(\boldsymbol{\theta}^*)} \mathrm{E}\left( \frac{\partial \operatorname{vec}(\gamma_{1,\boldsymbol{\xi}}(\mathbf{X}_i, Y_i)\mathbf{X}_i\mathbf{X}_i^T\mathbf{B}_1^*)}{\partial \pi_1} \right) \Delta_{\pi_1}\|_2 + \\
& \quad \sqrt{q}\| \sup_{\boldsymbol{\xi}\in\mathcal{B}_{\mathrm{con}}(\boldsymbol{\theta}^*)} \mathrm{E}\left( \frac{\partial \operatorname{vec}(\gamma_{1,\boldsymbol{\xi}}(\mathbf{X}_i, Y_i)\mathbf{X}_i\mathbf{X}_i^T\mathbf{B}_1^*)}{\partial \operatorname{vec}(\boldsymbol{\Gamma}_2 - \boldsymbol{\Gamma}_1)} \right) \Delta_{\boldsymbol{\Gamma}_2-\boldsymbol{\Gamma}_1}\|_2 + \\
& \quad \sqrt{q}\| \sup_{\boldsymbol{\xi}\in\mathcal{B}_{\mathrm{con}}(\boldsymbol{\theta}^*)} \mathrm{E}\left( \frac{\partial \operatorname{vec}(\gamma_{1,\boldsymbol{\xi}}(\mathbf{X}_i, Y_i)\mathbf{X}_i\mathbf{X}_i^T\mathbf{B}_1^*)}{\partial \operatorname{vec}(\boldsymbol{\Gamma}_2 + \boldsymbol{\Gamma}_1)} \right) \Delta_{\boldsymbol{\Gamma}_2+\boldsymbol{\Gamma}_1}\|_2 \\
& \leq \sqrt{q} \underbrace{\sup_{\boldsymbol{\xi}\in\mathcal{B}_{\mathrm{con}}(\boldsymbol{\theta}^*)} \| \mathrm{E}\left( \frac{\partial \gamma_{1,\boldsymbol{\xi}}(\mathbf{X}_i, Y_i)}{\partial \pi_1} \operatorname{vec}^T(\mathbf{X}_i\mathbf{X}_i^T\mathbf{B}_1^*) \right)\|_2|\pi_1 - \pi_1^*|}_{(I)} + \\
& \quad \sqrt{q} \underbrace{\sup_{\boldsymbol{\xi}\in\mathcal{B}_{\mathrm{con}}(\boldsymbol{\theta}^*)} \| \mathrm{E}\left( \frac{\partial \gamma_{1,\boldsymbol{\xi}}(\mathbf{X}_i, Y_i)}{\partial \operatorname{vec}(\boldsymbol{\Gamma}_2 - \boldsymbol{\Gamma}_1)} \operatorname{vec}^T(\mathbf{X}_i\mathbf{X}_i^T\mathbf{B}_1^*) \right)\|_2\|\boldsymbol{\Gamma}_1 - \boldsymbol{\Gamma}_1^* - \boldsymbol{\Gamma}_2 + \boldsymbol{\Gamma}_2^*\|_F}_{(II)} + \\
& \quad \sqrt{q} \underbrace{\sup_{\boldsymbol{\xi}\in\mathcal{B}_{\mathrm{con}}(\boldsymbol{\theta}^*)} \| \mathrm{E}\left( \frac{\partial \gamma_{1,\boldsymbol{\xi}}(\mathbf{X}_i, Y_i)}{\partial \operatorname{vec}(\boldsymbol{\Gamma}_2 + \boldsymbol{\Gamma}_1)} \operatorname{vec}^T(\mathbf{X}_i\mathbf{X}_i^T\mathbf{B}_1^*) \right)\|_2\|\boldsymbol{\Gamma}_1 - \boldsymbol{\Gamma}_1^* + \boldsymbol{\Gamma}_2 - \boldsymbol{\Gamma}_2^*\|_F}_{(III)} .
\end{aligned}
$$

We focus on the second term (II). Note that $\operatorname{vec}(\mathbf{X}_i\mathbf{X}_i^T\mathbf{B}_1^*) = [(\mathbf{X}_i^T\mathbf{B}_1^*)^T \otimes \mathbf{I}_p]\mathbf{X}_i$. Then,

$$
\begin{aligned}
& -\frac{1}{\pi_1(1-\pi_1)}\frac{\partial\gamma_{1,\boldsymbol{\xi}}(\mathbf{X}_i, Y_i)}{\partial \operatorname{vec}(\boldsymbol{\Gamma}_2 - \boldsymbol{\Gamma}_1)} \operatorname{vec}^T(\mathbf{X}_i\mathbf{X}_i^T\mathbf{B}_1^*) \\
& = \frac{\exp((\mathbf{Z} + \boldsymbol{\delta}(\boldsymbol{\Gamma})\mathbf{f}_i)^T((\boldsymbol{\Gamma}_2 - \boldsymbol{\Gamma}_1)\mathbf{f}_i))}{\{\pi_1 + (1-\pi_1)\exp((\mathbf{Z} + \boldsymbol{\delta}(\boldsymbol{\Gamma})\mathbf{f}_i)^T((\boldsymbol{\Gamma}_2 - \boldsymbol{\Gamma}_1)\mathbf{f}_i)\}^2} \cdot (\mathbf{f}_i \otimes \mathbf{I}_p)(\mathbf{Z} + \boldsymbol{\delta}(\boldsymbol{\Gamma})\mathbf{f}_i)\operatorname{vec}^T(\mathbf{X}_i\mathbf{X}_i^T\mathbf{B}_1^*) \\
& = \frac{\partial\gamma_{1,\boldsymbol{\xi}}(\mathbf{X}_i, Y_i)}{\partial\pi_1} \cdot (\mathbf{f}_i \otimes \mathbf{I}_p)(\mathbf{Z} + \boldsymbol{\delta}(\boldsymbol{\Gamma})\mathbf{f}_i)(\mathbf{Z} + \boldsymbol{\psi}\mathbf{f}_i)^T((\mathbf{Z} + \boldsymbol{\psi}\mathbf{f}_i)^T\mathbf{B}_1^* \otimes \mathbf{I}_p). \quad (28)
\end{aligned}
$$

We can decompose the last line into a sum of 8 terms. The term involves three $\mathbf{Z}$ is the most complicated one. We consider

$$
\frac{\partial\gamma_{1,\boldsymbol{\xi}}(\mathbf{X}_i, Y_i)}{\partial\pi_1} \cdot (\mathbf{f}_i \otimes \mathbf{I}_p)\mathbf{Z}\mathbf{Z}^T(\mathbf{Z}^T\mathbf{B}_1^* \otimes \mathbf{I}_p).
$$

Recall that $\mathbf{H}_i$ is an orthonormal matrix whose first row is $\mathbf{T}_1^T/\|\mathbf{T}_1\|_2$. We further require that $\mathbf{B}_{1,1:}^* \in \operatorname{span}(\mathbf{H}_{i,1:}, \mathbf{H}_{i,2:})$, $\mathbf{B}_{1,2:}^* \in \operatorname{span}(\mathbf{H}_{i,1:}, \mathbf{H}_{i,2:}, \mathbf{H}_{i,3:}), \ldots, \mathbf{B}_{1,q:}^* \in \operatorname{span}(\mathbf{H}_{i,1:}, \ldots, \mathbf{H}_{i,(q+1):})$, where $\mathbf{A}_{j:}$ is the $j$-th row of matrix $\mathbf{A}$. Thus there exists a matrix $\boldsymbol{\Lambda} \in \mathbb{R}^{p\times q}$ such that $\mathbf{B}_1^* = \mathbf{H}_i^T\boldsymbol{\Lambda}$. Then

$$
\begin{aligned}
\mathbf{Z}^T\mathbf{B}_1^* &= (\mathbf{H}_i^T\mathbf{H}_i\mathbf{Z})^T\mathbf{B}_1^* = \mathbf{Z}^T\mathbf{H}_i^T\mathbf{H}_i\mathbf{H}_i^T\boldsymbol{\Lambda} = \mathbf{V}^T\boldsymbol{\Lambda} \\
&= (\sum_{j=1}^{2}\lambda_{j1}V_j, \sum_{j=1}^{3}\lambda_{j2}V_j, \ldots, \sum_{j=1}^{q+1}\lambda_{jq}V_j) := \mathbf{M}^T,
\end{aligned}
$$

where $\lambda_{jk}$ is the $(j,k)$-th element of $\mathbf{\Lambda}$ and $V_j$ is the $j$-th element of $\mathbf{V}$. When $j > k+1$, $\lambda_{jk} = 0$ and $\|\mathbf{\Lambda}\|_F = \|\mathbf{B}_1^*\|_F \leq \sqrt{d}\|\mathbf{B}_1^*\|_2 = \sqrt{d}M_b M_2$. Therefore,

$$\frac{\partial \gamma_{1,\boldsymbol{\xi}}(\mathbf{X}_i, Y_i)}{\partial \pi_1} \cdot (\mathbf{f}_i \otimes \mathbf{I}_p)\mathbf{Z}\mathbf{Z}^T(\mathbf{Z}^T\mathbf{B}_1^* \otimes \mathbf{I}_p)$$

$$= \frac{\partial \gamma_{1,\boldsymbol{\xi}}(\mathbf{X}_i, Y_i)}{\partial \pi_1} \cdot (\mathbf{f}_i \otimes \mathbf{I}_p)\mathbf{H}_i^T\mathbf{V}\mathbf{V}^T\mathbf{H}_i(\mathbf{M}^T \otimes \mathbf{I}_p)$$

$$= \frac{\partial \gamma_{1,\boldsymbol{\xi}}(\mathbf{X}_i, Y_i)}{\partial \pi_1} \cdot (\mathbf{f}_i \otimes \mathbf{I}_p)\mathbf{H}_i^T\mathbf{V}\mathbf{V}^T(\mathbf{M}^T \otimes \mathbf{H}_i\mathbf{I}_p)$$

$$= \frac{\partial \gamma_{1,\boldsymbol{\xi}}(\mathbf{X}_i, Y_i)}{\partial \pi_1} \cdot (\mathbf{f}_i \otimes \mathbf{I}_p)\mathbf{H}_i^T\mathbf{V}\mathbf{V}^T(M_1\mathbf{H}_i, M_2\mathbf{H}_i, \ldots, M_q\mathbf{H}_i),$$

where $M_j$ is the $j$-th element of $\mathbf{M}$. Therefore, we have

$$\mathrm{E}[\frac{\partial \gamma_{1,\boldsymbol{\xi}}(\mathbf{X}_i, Y_i)}{\partial \pi_1} \cdot (\mathbf{f}_i \otimes \mathbf{I}_p)\mathbf{Z}\mathbf{Z}^T(\mathbf{Z}^T\mathbf{B}_1^* \otimes \mathbf{I}_p)]$$

$$= (\mathbf{f}_i \otimes \mathbf{I}_p)\mathbf{H}_i^T \, \mathrm{E}[\frac{\partial \gamma_{1,\boldsymbol{\xi}}(\mathbf{X}_i, Y_i)}{\partial \pi_1}\mathbf{V}\mathbf{V}^T(M_1\mathbf{H}_i, M_2\mathbf{H}_i, \ldots, M_q\mathbf{H}_i)]$$

$$= (\mathbf{f}_i \otimes \mathbf{I}_p)\mathbf{H}_i^T \, \mathrm{E}[\underbrace{\frac{\exp(\|\mathbf{T}_1\|_2 V_1 + \mathbf{T}_1^T\mathbf{T}_2))}{(\pi_1 + (1 - \pi_1)\exp\{\|\mathbf{T}_1\|_2 V_1 + \mathbf{T}_1^T\mathbf{T}_2)\})^2}}_{g(V_1)}\mathbf{V}\mathbf{V}^T(M_1\mathbf{H}_i, M_2\mathbf{H}_i, \ldots, M_q\mathbf{H}_i)].$$

Note that

$$\mathrm{E}[g(V_1)\mathbf{V}\mathbf{V}^T M_k]$$

$$= \mathrm{E}(g(V_1)\begin{bmatrix} \lambda_{1k}V_1^3 & & & \\ & \lambda_{1k}V_1 V_2^2 & & \\ & & \ddots & \\ & & & \lambda_{1k}V_1 V_p^2 \end{bmatrix} +$$

$$\begin{bmatrix} & \lambda_{2k}V_1 V_2^2 & \lambda_{3k}V_1 V_3^2 & \cdots & \lambda_{(k+1)k}V_1 V_{k+1}^2 & 0 \\ \lambda_{2k}V_1 V_2^2 & \lambda_{1k}V_2^3 & & & & \\ \lambda_{3k}V_1 V_3^2 & & \lambda_{1k}V_3^3 & & & \\ \vdots & & & \ddots & & \\ \lambda_{(k+1)k}V_1 V_{k+1}^2 & & & & \lambda_{(k+1)k}V_{k+1}^3 & \\ 0 & & & & & 0 \end{bmatrix}).$$

Therefore, for each $k$, the matrix $\mathrm{E}[g(V_1)\mathbf{V}\mathbf{V}^T M_k]$ can be written as the sum of a diagonal matrix and a matrix with $2k$ non-zero elements. Then, the $p \times pq$ matrix $\mathrm{E}[g(V_1)\mathbf{V}\mathbf{V}^T(M_1\mathbf{H}_i,$
$M_2\mathbf{H}_i, \ldots, M_q\mathbf{H}_i)]$ can be written as the sum of two block matrices $(\mathrm{E}[\mathbf{J}_1] + \mathrm{E}[\mathbf{J}_2])$, where each block has size $p \times p$. In the first matrix $\mathrm{E}[\mathbf{J}_1]$, the $k$-th block is $\mathbf{D}_k\mathbf{H}_i$, where $\mathbf{D}_k$ is a diagonal matrix. And in the second matrix $\mathrm{E}[\mathbf{J}_2]$, the $k$-th block is $\mathbf{A}_k\mathbf{H}_i$, where $\mathbf{A}_k$ is a matrix that only has $2k$ non-zero elements. Since $\|\mathrm{E}[\mathbf{J}_2]\|_2 \leq \|\mathrm{E}[\mathbf{J}_2]\|_F = \|\mathrm{E}[(\mathbf{A}_1\mathbf{H}_i, \ldots, \mathbf{A}_q\mathbf{H}_i)]\|_F = \|\mathrm{E}[(\mathbf{A}_1, \ldots, \mathbf{A}_q)]\|_F$, each element of $\mathrm{E}[\mathbf{A}_k]$ only involve $\mathrm{E}[g(V_1)V_1]$, which can be bounded with the same argument used in (12). The total number of non-zeros elements in $\mathrm{E}[(\mathbf{A}_1, \ldots, \mathbf{A}_q)]$ is $q(1 + q)$. Then we have

$$\|\mathrm{E}[\mathbf{J}_2]\|_2 \leq c_8 q(1 + q)\exp(-(\frac{3c_1}{8} \wedge \frac{c_1^2}{8c_4^2})\Omega^2), \tag{29}$$

where $c_8$ is some positive constant.

Next, we bound $\|\mathrm{E}[\mathbf{J}_1]\|_2$. Since each block of $\mathbf{J}_1$ is a diagonal matrix $\mathbf{D}_1$ times the orthonormal matrix $\mathbf{H}_i$, rows of $\mathbf{J}_1$ are orthogonal. Then we can construct its SVD $\mathrm{E}[\mathbf{J}_1] = \mathbf{F}\mathbf{D}\mathbf{K}^T$ in the following way. Let $\mathbf{F} \in \mathbb{R}^{p \times p}$ be the identity matrix, $\mathbf{D}$ is a diagonal matrix that elements are $\ell_2$ norm of rows of $\mathbf{J}_1$, and $\mathbf{K}^T$ is normalized $\mathrm{E}[\mathbf{J}_1]$ where each row is divided by its $\ell_2$ norm. Therefore, $\|\mathrm{E}[\mathbf{J}_1]\|_2$ equals to the largest $\ell_2$ norm of rows $\mathrm{E}[\mathbf{J}_1]$. It is easy to see rows of $\mathrm{E}[\mathbf{J}_1]$ and rows of $\mathrm{E}[(\mathbf{D}_1, \ldots, \mathbf{D}_2)]$ have same $\ell_2$ norm. Thus, we only need to bound the largest $\ell_2$ norm of $\mathrm{E}[(\mathbf{D}_1, \ldots, \mathbf{D}_2)]$. Each row of $\mathrm{E}[(\mathbf{D}_1, \ldots, \mathbf{D}_2)]$ has $q$ non-zero elements, which contain either $\mathrm{E}[g(V_1)V_1]$ or $\mathrm{E}[g(V_1)V_1^3]$. Since $\mathrm{E}[g(V_1)V_1]$ can be

bounded similar to (12). We focus on bounding $\mathrm{E}[g(V_1)V_1^3]$. Recall that the event $\mathcal{E}_i = \{\|\mathbf{T}_1\|_2 Z_1 \leq \frac{c_1}{2}\Omega^2\}$ and on the event $\mathcal{E}_i$, $\|\mathbf{T}_1\|_2 Z_1 + \mathbf{T}_1^T\mathbf{T}_2\| \geq |\mathbf{T}_1^T\mathbf{T}_2| - |\mathbf{T}_1\|_2 Z_1| \geq c_1\Omega^2/2$. Let $t = \|\mathbf{T}_1\|_2 Z_1 + \mathbf{T}_1^T\mathbf{T}_2$, $h(t) = \frac{\exp(t)}{(\pi_1 + (1-\pi_1)\exp\{t\})^2}$. Then

$$
\begin{aligned}
&|\mathrm{E}[g(V_1)V_1^3]| \\
&= \Big| \mathrm{E}\Big[\frac{\exp(\|\mathbf{T}_1\|_2 Z_1 + \mathbf{T}_1^T\mathbf{T}_2)}{(\pi_1 + (1-\pi_1)\exp\{\|\mathbf{T}_1\|_2 Z_1 + \mathbf{T}_1^T\mathbf{T}_2)\})^2} Z_1^3\Big] \Big| \\
&= \Big| \mathrm{E}[h(t)\frac{(\|\mathbf{T}_1\|_2 Z_1 + \mathbf{T}_1^T\mathbf{T}_2)^3 - 3(\mathbf{T}_1^T\mathbf{T}_2)^2\|\mathbf{T}_1\|_2 Z_1 - 3\|\mathbf{T}_1\|_2^2\mathbf{T}_1^T\mathbf{T}_2\mathbf{Z}_1^2 - (\mathbf{T}_1^T\mathbf{T}_2)^3}{\|\mathbf{T}_1\|_2^3}] \Big| \\
&\leq \frac{1}{\|\mathbf{T}_1\|_2^3}\Big| \mathrm{E}[h(t)(\|\mathbf{T}_1\|_2 Z_1 + \mathbf{T}_1^T\mathbf{T}_2)^3]\Big| + \frac{3}{\|\mathbf{T}_1\|_2^2}\Big| \mathrm{E}[h(t)(\mathbf{T}_1^T\mathbf{T}_2)^2 Z_1]\Big| \\
&\quad \frac{3}{\|\mathbf{T}_1\|_2}\Big| \mathrm{E}[h(t)\mathbf{T}_1^T\mathbf{T}_2\mathbf{Z}_1^2]\Big| + \frac{1}{\|\mathbf{T}_1\|_2^3}\Big| \mathrm{E}[h(t)(\mathbf{T}_1^T\mathbf{T}_2)^3].
\end{aligned}
$$

The second, third, and fourth terms can be bounded similarly as (12), (16) and (11). For some positive constant $c_9$, we have

$$
\begin{aligned}
&\frac{3}{\|\mathbf{T}_1\|_2^2}\Big| \mathrm{E}[h(t)(\mathbf{T}_1^T\mathbf{T}_2)^2 Z_1]\Big| + \frac{3}{\|\mathbf{T}_1\|_2}\Big| \mathrm{E}[h(t)\mathbf{T}_1^T\mathbf{T}_2\mathbf{Z}_1^2]\Big| + \frac{1}{\|\mathbf{T}_1\|_2^3}\Big| \mathrm{E}[h(t)(\mathbf{T}_1^T\mathbf{T}_2)^3] \\
&\leq c_9 \exp(-(\frac{c_1}{4} \wedge \frac{c_1^2}{8c_4^2})\Omega^2).
\end{aligned}
$$

Using Lemma F.1,

$$
\begin{aligned}
&\frac{1}{\|\mathbf{T}_1\|_2^3}\Big| \mathrm{E}[h(t)(\|\mathbf{T}_1\|_2 Z_1 + \mathbf{T}_1^T\mathbf{T}_2)^3]\Big| \\
&\leq \frac{1}{c_3^3\Omega^3}\Big\{ \Big| \mathrm{E}[\frac{\exp(t)}{(\pi_1 + (1-\pi_1)\exp\{t\})^2}t^3|\mathcal{E}_i]P(\mathcal{E}_i)\Big| + \Big| \mathrm{E}[\frac{\exp(t)}{(\pi_1 + (1-\pi_1)\exp\{t\})^2}t^3|\mathcal{E}_i^c]P(\mathcal{E}_i^c)\Big| \Big\} \\
&\leq \frac{1}{c_3^3\Omega^3}\Big\{ \frac{8}{\min\{\pi_1^2, (1-\pi_1)^2\}}\exp(-\frac{c_1\Omega^2}{8}) + \frac{4}{\min\{\pi_1^2, (1-\pi_1)^2\}}\exp(-\frac{c_1^2\Omega^2}{8c_4^2}) \Big\} \\
&\leq \frac{8}{c_0^2 c_3^3\Omega^3}\exp(-\frac{c_1\Omega^2}{8}) + \frac{4}{c_0^2 c_3^3\Omega^3}\exp(-\frac{c_1^2\Omega^2}{8c_4^2}) \leq \frac{8}{c_0^2 c_3^3\Omega^3}\exp(-(\frac{c_1}{8} \wedge \frac{c_1^2}{8c_4^2})\Omega^2).
\end{aligned}
$$

Combining the above results, we have

$$
|\mathrm{E}[g(V_1)V_1^3]| \leq (c_9 + \frac{8}{c_0^2 c_3^3\Omega^3})\exp(-(\frac{c_1}{8} \wedge \frac{c_1^2}{8c_4^2})\Omega^2).
$$

Therefore, the 2-norm of $\mathrm{E}[\mathbf{J}_1]$ (the largest $\ell_2$ norm of rows of $\mathbf{J}_1$) is bounded by

$$
\| \mathrm{E}[\mathbf{J}_1]\|_2 \leq c_{10}q \exp(-(\frac{c_1}{8} \wedge \frac{c_1^2}{8c_4^2})\Omega^2). \tag{30}
$$

Combing results (29) and (30),

$$
\| \mathrm{E}[\frac{\partial \gamma_{1,\boldsymbol{\xi}}(\mathbf{X}_i, Y_i)}{\partial \pi_1} \cdot (\mathbf{f}_i \otimes \mathbf{I}_p)\mathbf{Z}\mathbf{Z}^T(\mathbf{Z}^T\mathbf{B}_1^* \otimes \mathbf{I}_p)]\|_2 \leq M_4[c_8 q(1+q) + c_{10}q]\exp(-(\frac{c_1}{8} \wedge \frac{c_1^2}{8c_4^2})\Omega^2).
$$

The other 7 terms in (28) involve $\mathbf{Z}$ at most twice and therefore can be bounded with the same technique in (11), (12), and (16). Therefore, we have

$$
\| \mathrm{E}\left(\frac{\partial \gamma_{1,\boldsymbol{\xi}}(\mathbf{X}_i, Y_i)}{\partial \mathrm{vec}(\boldsymbol{\Gamma}_2 - \boldsymbol{\Gamma}_1)} \mathrm{vec}^T(\mathbf{X}_i \mathbf{X}_i^T \mathbf{B}_1^*)\right)\|_2 \leq c_{11}q^2 \exp(-(\frac{c_1}{8} \wedge \frac{c_1^2}{8c_4^2})\Omega^2), \tag{31}
$$

for some constant $c_{11}$.

For the first term (I),

$$\frac{\partial \gamma_{1,\boldsymbol{\xi}}(\mathbf{X}_i, Y_i)}{\partial \pi_1} \operatorname{vec}^T(\mathbf{X}_i \mathbf{X}_i^T \mathbf{B}_1^*) = \frac{\partial \gamma_{1,\boldsymbol{\xi}}(\mathbf{X}_i, Y_i)}{\partial \pi_1} \mathbf{X}_i^T [(\mathbf{X}_i^T \mathbf{B}_1^*) \otimes \mathbf{I}_p]$$

$$= \frac{\partial \gamma_{1,\boldsymbol{\xi}}(\mathbf{X}_i, Y_i)}{\partial \pi_1} (\mathbf{Z} + \boldsymbol{\psi} \mathbf{f}_i)^T ((\mathbf{Z} + \boldsymbol{\psi} \mathbf{f}_i)^T \mathbf{B}_1^* \otimes \mathbf{I}_p),$$

and the third term (III)

$$\frac{\partial \gamma_{1,\boldsymbol{\xi}}(\mathbf{X}_i, Y_i)}{\partial \operatorname{vec}(\boldsymbol{\Gamma}_2 + \boldsymbol{\Gamma}_1)} \operatorname{vec}^T(\mathbf{X}_i \mathbf{X}_i^T \mathbf{B}_1^*)$$

$$= \frac{\pi_1(1 - \pi_1)}{2} \frac{\partial \gamma_{1,\boldsymbol{\xi}}(\mathbf{X}_i, Y_i)}{\partial \pi_1} (\mathbf{f}_i \otimes (\boldsymbol{\Gamma}_2 - \boldsymbol{\Gamma}_1) \mathbf{f}_i) \operatorname{vec}^T(\mathbf{X}_i \mathbf{X}_i^T \mathbf{B}_1^*)$$

$$= \frac{\pi_1(1 - \pi_1)}{2} \frac{\partial \gamma_{1,\boldsymbol{\xi}}(\mathbf{X}_i, Y_i)}{\partial \pi_1} (\mathbf{f}_i \otimes (\boldsymbol{\Gamma}_2 - \boldsymbol{\Gamma}_1) \mathbf{f}_i) (\mathbf{Z} + \boldsymbol{\psi} \mathbf{f}_i)^T ((\mathbf{Z} + \boldsymbol{\psi} \mathbf{f}_i)^T \mathbf{B}_1^* \otimes \mathbf{I}_p).$$

We see that both terms only have $\mathbf{Z}\mathbf{Z}^T$ and thus can be bounded similarly to (11), (12), and (16). Finally, for some positive constant $c_{12}$ we obtain

$$\| \operatorname{E}\{[\gamma_{1,\boldsymbol{\xi}}(\mathbf{X}_i, Y_i) - \gamma_{1,\boldsymbol{\xi}^*}(\mathbf{X}_i, Y_i)] \mathbf{X}_i \mathbf{X}_i^T \mathbf{B}_1^*\}\|_F \le \sqrt{q} q^2 c_{12} \exp(-(\frac{c_1}{8} \wedge \frac{c_1^2}{8c_4^2})\Omega^2) d_F(\boldsymbol{\theta}, \boldsymbol{\theta}^*)$$

$$\le \kappa_{\boldsymbol{\Sigma}}(d_F(\boldsymbol{\theta}, \boldsymbol{\theta}^*) \vee \|\mathbf{B}_1 - \mathbf{B}_1^*\|_F \vee \|\mathbf{B}_2 - \mathbf{B}_2^*\|_F), \tag{32}$$

where $\kappa_{\boldsymbol{\Sigma}} = \sqrt{q} q^2 c_{12} \exp(-(\frac{c_1}{8} \wedge \frac{c_1^2}{8c_4^2})\Omega^2)$.

## E. Proof of Lemma C.2

### E.1. Covering number of $\mathcal{L}(s)$

We state two lemmas that are used later.

**Lemma E.1** (Rudelson & Zhou (2012), Lemma 11)**.** *Let* $\mathbf{u}_1, \dots, \mathbf{u}_M \in \mathbb{R}^{pq}$. *Let* $\mathbf{y} \in \operatorname{conv}(\mathbf{u}_1, \dots, \mathbf{u}_M)$. *There exists a set* $L \in [M]$ *such that*

$$|L| \le m = \frac{4 \max_{j \in [M]} \|\mathbf{u}_j\|_2^2}{\varepsilon^2}$$

*and a vector* $\mathbf{y}' \in \operatorname{conv}(\mathbf{u}_j, j \in L)$ *such that*

$$\|\mathbf{y}' - \mathbf{y}\|_2 \le \varepsilon.$$

**Lemma E.2** (Rudelson & Zhou (2012), Lemma 21)**.** *Let* $\mathbf{u}, \boldsymbol{\theta}, \mathbf{x} \in \mathbb{R}^{pq}$ *be vectors such that* $\|\boldsymbol{\theta}\|_2 = 1$, $\langle \mathbf{x}, \boldsymbol{\theta} \rangle \ne 0$, *and* $\mathbf{u}$ *is not parallel to* $\mathbf{x}$. *Define* $\phi : \mathbb{R} \to \mathbb{R}$ *by:*

$$\phi(\lambda) = \frac{\langle \mathbf{x} + \lambda \mathbf{u}, \boldsymbol{\theta} \rangle}{\|\mathbf{x} + \lambda \mathbf{u}\|_2}.$$

*Assume* $\phi(\lambda)$ *has a local maximum at 0, then*

$$\frac{\langle \mathbf{x} + \mathbf{u}, \boldsymbol{\theta} \rangle}{\langle \mathbf{x}, \boldsymbol{\theta} \rangle} \ge 1 - \frac{\|\mathbf{u}\|_2}{\|\mathbf{x}\|_2}.$$

Next, we show the following lemma

**Lemma E.3.** *Let* $0 < s < 1/3p$, *and* $d = 26883sq^3$, *then*

$$\mathcal{L}(s) \cap \mathbb{S}^{pq-1} \subset 2 \operatorname{conv}(\bigcup_{|J| \le d} E_J(pq) \cap \mathbb{S}^{pq-1}), \tag{33}$$

*where* $\operatorname{conv}$ *denotes the convex hull and* $E_J(pq) = \operatorname{span}(\mathbf{e}_j : j \in J)$.

*Proof.* The proof is analogous to that for Lemma 13 of Rudelson & Zhou (2012) with some modifications. We assume $d < pq$, otherwise the lemma is trivially true. For a vector $\mathbf{u} \in \mathbb{R}^{pq}$, let $\mathcal{T}$ denote the indices of the $3sq$ largest absolute coefficients of $\mathbf{u}$. Then $\|\mathbf{u}_{\mathcal{T}^c}\|_1 \leq \|\mathbf{u}_{\widetilde{\mathcal{S}}_1^c}\|_1$. We decompose a vector $\mathbf{u} \in \mathcal{L}(s) \cap \mathbb{S}^{pq-1}$ as

$$\mathbf{u} = \mathbf{u}_{\mathcal{T}} + \mathbf{u}_{\mathcal{T}^c} \in \mathbf{u}_{\mathcal{T}} + [(\sqrt{sq} + 2q\sqrt{3s})\|\mathbf{u}_{\widetilde{\mathcal{S}}_1}\|_2 + \sqrt{sq}\|\mathbf{u}\|_2] \cdot \mathrm{absconv}(e_j : j \in \mathcal{T}^c),$$

where $\mathrm{absconv}$ denotes the absolutely convex hull. Since

$$
\begin{aligned}
\|\mathbf{u}_{\mathcal{T}^c}\|_2^2 &\leq \|\mathbf{u}_{\mathcal{T}^c}\|_1 \|\mathbf{u}_{\mathcal{T}^c}\|_\infty \leq \|\mathbf{u}_{\widetilde{\mathcal{S}}_1^c}\|_1 \frac{\|\mathbf{u}_{\mathcal{T}}\|_1}{3sq} \\
&\leq [(\sqrt{sq} + 2q\sqrt{3s})\|\mathbf{u}_{\widetilde{\mathcal{S}}_1}\|_2 + \sqrt{sq}\|\mathbf{u}\|_2] \frac{\|\mathbf{u}_{\mathcal{T}}\|_2}{\sqrt{3sq}} \\
&\leq [(\sqrt{sq} + 2q\sqrt{3s})\|\mathbf{u}_{\mathcal{T}}\|_2 + \sqrt{sq}\|\mathbf{u}\|_2] \frac{\|\mathbf{u}_{\mathcal{T}}\|_2}{\sqrt{3sq}} \\
&= [(\sqrt{3}/3 + 2\sqrt{q})\|\mathbf{u}_{\mathcal{T}}\|_2 + \sqrt{3}/3\|\mathbf{u}\|_2] \|\mathbf{u}_{\mathcal{T}}\|_2,
\end{aligned}
$$

we have

$$1 = \|\mathbf{u}\|_2^2 = \|\mathbf{u}_{\mathcal{T}}\|_2^2 + \|\mathbf{u}_{\mathcal{T}^c}\|_2^2 \leq \underbrace{(1 + \sqrt{3}/3 + 2\sqrt{q})}_{a^2}\|\mathbf{u}_{\mathcal{T}}\|_2^2 + \underbrace{\sqrt{3}/3}_{b}\|\mathbf{u}_{\mathcal{T}}\|_2.$$

Let $\|\mathbf{u}_{\mathcal{T}}\|_2 = x$. We are interested in finding a range of $x$ that satisfies $a^2 x^2 + bx \geq 1$, which is equivalent to $(ax + b/(2a))^2 \geq b^2/(4a^2) + 1$. Then we have $\|\mathbf{u}_{\mathcal{T}}\|_2 \geq \frac{\sqrt{4a^2 + b^2} - b}{2a^2}$

Define

$$\mathcal{V} = \{\mathbf{u}_{\mathcal{T}} + [(\sqrt{sq} + 2q\sqrt{3s})\|\mathbf{u}_{\widetilde{\mathcal{S}}_1}\|_2 + \sqrt{sq}\|\mathbf{u}\|_2] \cdot \mathrm{absconv}(e_j : j \in \mathcal{T}^c) : \mathbf{u} \in \mathcal{L}(s) \cap \mathbb{S}^{pq-1}\}.$$

We have $\mathcal{L}(s) \cap \mathbb{S}^{pq-1} \subset \mathcal{V} \subset \mathcal{L}(s)$ and $\mathcal{V}$ is compact. Therefore, $\mathcal{V}$ contains a base of $\mathcal{L}(s)$, that is, for any $\mathbf{y} \in \mathcal{L}(s)/\{0\}$, there exists $\lambda > 0$ such that $\lambda \mathbf{y} \in \mathcal{V}$. For any nonzero vector $\mathbf{v} \in \mathbb{R}^{pq}$, we define

$$F(\mathbf{v}) = \frac{\mathbf{v}}{\|\mathbf{v}\|_2}.$$

Then function $F$ is continuous on $\mathcal{L}(s)/\{0\}$ and $\mathcal{V}$. Thus,

$$\mathcal{L}(s) \cap \mathbb{S}^{pq-1} = F(\mathcal{L}(s)/\{0\}) = F(\mathcal{V}).$$

By duality, inclusion (33) can be derived by showing the supremum of any linear functional over the left side of (33) does not exceed the supremum over the right side of it. Since $\mathcal{L}(s) \cap \mathbb{S}^{pq-1} = F(\mathcal{V})$, it is enough to show that for any $\boldsymbol{\theta} \in \mathbb{S}^{pq-1}$, there exists $\mathbf{z}' \in \mathbb{R}^{pq}/\{0\}$ such that $\mathrm{supp}(\mathbf{z}') \leq d$ and $F(\mathbf{z}')$ satisfies that

$$\max_{\mathbf{v} \in \mathcal{V}} \langle F(\mathbf{v}), \boldsymbol{\theta} \rangle \leq 2 \langle F(\mathbf{z}'), \boldsymbol{\theta} \rangle. \tag{34}$$

For a given $\boldsymbol{\theta}$, we construct a $d$-sparse vector $\mathbf{z}'$ that satisfies (34). Let $\mathbf{z} = \mathrm{argmax}_{\mathbf{v} \in \mathcal{V}} \langle F(\mathbf{v}), \boldsymbol{\theta} \rangle$. By the definition of $\mathcal{V}$, there exists a set $\mathcal{I} \in [pq]$ such that $|\mathcal{I}| = 3sq$ and for some $\eta_j \in \{1, -1\}$,

$$\mathbf{z} = \mathbf{z}_{\mathcal{I}} + [(\sqrt{sq} + 2q\sqrt{3s})\|\mathbf{z}_{\widetilde{\mathcal{S}}_1}\|_2 + \sqrt{sq}\|\mathbf{z}\|_2] \sum_{j \in \mathcal{I}^c} \alpha_j \eta_j e_j,$$

where $\alpha_j \in [0,1]$, $\sum_{j \in \mathcal{I}^c} \alpha_j \leq 1$ and $\|\mathbf{z}_{\mathcal{I}}\|_2 \geq \frac{\sqrt{4a^2 + b^2} - b}{2a^2}$. If $\alpha_i = 1$ for some $i \in \mathcal{I}^c$, then $\alpha_j = 0$ for $j \in \mathcal{I}^c/\{i\}$ and $\mathbf{z}$ is a sparse vector with $\mathrm{supp}(\mathbf{z}) \leq 3sq + 1$. Let $\mathbf{z}' = \mathbf{z}$. Clearly, (33) holds with $d = 3sq + 1$. In the following, we assume $\alpha_i \in [0,1), \forall i \in \mathcal{I}^c$. To use lemma E.1, denote $\mathbf{e}_{pq+1} = 0$, $\eta_{pq+1} = 1$ and set

$$\alpha_{pq+1} = 1 - \sum_{j \in \mathcal{I}^c} \alpha_j.$$

Then $\mathbf{y} := \mathbf{z}_{\mathcal{I}^c} \in \mathrm{conv}(\mathbf{u}_1, \ldots, \mathbf{u}_{|\mathcal{M}|})$, where $\mathbf{u}_j = [(\sqrt{sq} + 2q\sqrt{3s})\|\mathbf{z}_{\widetilde{\mathcal{S}}_1}\|_2 + \sqrt{sq}\|\mathbf{z}\|_2] \cdot \eta_j \mathbf{e}_j$, $j \in \mathcal{M} = \{j \in \mathcal{I}^c \cup \{pq + 1\} : \alpha_j > 0\}$. We define $\alpha_{pq+1}$ because the sum of coefficients must equal 1 in convex combinations. According to lemma E.1, there exists a set $\mathcal{J}' \subset \mathcal{M}$ such that

$$
\begin{aligned}
|\mathcal{J}'| \leq m &:= \frac{4\max_{j\in\mathcal{M}}\|\mathbf{u}_j\|_2^2}{\varepsilon^2} \\
&= \frac{4\max_{j\in\mathcal{M}}[(\sqrt{sq} + 2q\sqrt{3s})\|\mathbf{z}_{\widetilde{\mathcal{S}}_1}\|_2 + \sqrt{sq}\|\mathbf{z}\|_2]^2\|\mathbf{e}_j\|_2^2}{\varepsilon^2} \\
&\leq 4\frac{sq + 12sq^2 + 4sq\sqrt{3q} + sq + 2sq + 4sq\sqrt{3q}}{\varepsilon^2} \\
&= 4\frac{4sq + 8sq\sqrt{3q} + 12sq^2}{\varepsilon^2} = \frac{16sq + 32sq\sqrt{3q} + 48sq^2}{\varepsilon^2} \\
&\leq \frac{120sq^2}{\varepsilon^2},
\end{aligned}
$$

and a vector $\mathbf{y}' \in \mathrm{conv}(\mathbf{u}_j, j \in \mathcal{J}')$

$$
\mathbf{y}' = [(\sqrt{sq} + 2q\sqrt{3s})\|\mathbf{z}_{\widetilde{\mathcal{S}}_1}\|_2 + \sqrt{sq}\|\mathbf{z}\|_2] \sum_{j\in\mathcal{J}'} \beta_j \eta_j \mathbf{e}_j
$$

such that $\sum_{j\in\mathcal{J}'} \beta_j = 1$ and $\|\mathbf{y} - \mathbf{y}'\|_2 \leq \varepsilon$. Set $\mathbf{z}' = \mathbf{z}_{\mathcal{I}} + \mathbf{y}'$. Then $\mathbf{z}' \in \mathrm{E}_{\mathcal{J}}$, where $\mathcal{J} = (\mathcal{I} \cup \mathcal{J}') \cap [pq]$ and $|\mathcal{J}| \leq |\mathcal{I}| + |\mathcal{J}'| \leq 3sq + m$. We have

$$
\|\mathbf{z} - \mathbf{z}'\|_2 = \|\mathbf{z} - \mathbf{z}_{\mathcal{I}} - \mathbf{y}'\|_2 = \|\mathbf{z}_{\mathcal{I}^c} - \mathbf{y}'\|_2 = \|\mathbf{y} - \mathbf{y}'\|_2 \leq \varepsilon.
$$

For $\{\beta_j : j \in \mathcal{J}'\}$ as above, we extend it to $\{\beta_j : j \in \mathcal{I}^c \cup \{pq + 1\}\}$ by setting $\beta_j = 0$ if $j \in \mathcal{I}^c \cup \{pq + 1\}/\{\mathcal{J}'\}$ and write

$$
\mathbf{z}' = \mathbf{z}_{\mathcal{I}} + [(\sqrt{sq} + 2q\sqrt{3s})\|\mathbf{z}_{\widetilde{\mathcal{S}}_1}\|_2 + \sqrt{sq}\|\mathbf{z}\|_2] \sum_{j\in\mathcal{I}^c\cup\{pq+1\}} \beta_j \eta_j \mathbf{e}_j,
$$

where $\beta_j \in [0, 1]$ and $\sum_{j\in\mathcal{I}^c\cup\{pq+1\}} \beta_j = 1$. If $\mathbf{z} = \mathbf{z}'$, then (34) holds naturally

$$
\max_{v\in\mathcal{V}}\langle F(\mathbf{v}), \boldsymbol{\theta} \rangle = \langle F(\mathbf{z}), \boldsymbol{\theta} \rangle \leq 2\langle F(\mathbf{z}'), \boldsymbol{\theta} \rangle = 2\langle F(\mathbf{z}), \boldsymbol{\theta} \rangle,
$$

and $d = 3sq + m$. Otherwise, for some $\lambda$ to be specified, consider the vector

$$
\mathbf{z} + \lambda(\mathbf{z}' - \mathbf{z}) = \mathbf{z}_{\mathcal{I}} + [(\sqrt{sq} + 2q\sqrt{3s})\|\mathbf{z}_{\widetilde{\mathcal{S}}_1}\|_2 + \sqrt{sq}\|\mathbf{z}\|_2] \sum_{j\in\mathcal{I}^c\cup\{pq+1\}} [(1-\lambda)\alpha_j + \lambda\beta_j]\eta_j\mathbf{e}_j.
$$

We have $\sum_{j\in\mathcal{I}^c\cup\{pq+1\}}[(1-\lambda)\alpha_j + \lambda\beta_j] = 1$. There exists $\delta_0 > 0$ such that $\forall j \in \mathcal{I}^c \cup \{pq+1\}$, $(1-\lambda)\alpha_j + \lambda\beta_j \in [0, 1]$ if $|\lambda| < \delta_0$ since

- This condition holds by continuity for all $j$ such that $\alpha_j \in (0, 1)$.

- If $\alpha_j = 0$ for some $j$, then $\beta_j = 0$ by definition of $\mathcal{M}$.

Therefore, we have $\sum_{j\in\mathcal{I}^c}[(1-\lambda)\alpha_j + \lambda\beta_j] \leq 1$, which implies $\mathbf{z} + \lambda(\mathbf{z}' - \mathbf{z}) \in \mathcal{V}$. Now consider function $\phi : (-\delta_0, \delta_0) \to \mathbb{R}$,

$$
\phi(\lambda) = \langle F(\mathbf{z} + \lambda(\mathbf{z}' - \mathbf{z})), \boldsymbol{\theta} \rangle = \frac{\langle \mathbf{z} + \lambda(\mathbf{z}' - \mathbf{z}), \boldsymbol{\theta} \rangle}{\|\mathbf{z} + \lambda(\mathbf{z}' - \mathbf{z})\|_2}.
$$

Since $\mathbf{z} = \mathrm{argmax}_{\mathbf{v}\in\mathcal{V}}\langle F(\mathbf{v}), \boldsymbol{\theta} \rangle$, $\phi(\lambda)$ attains a local maximum at 0. According to lemma E.2,

$$
\frac{\langle \mathbf{z}', \boldsymbol{\theta} \rangle}{\langle \mathbf{z}, \boldsymbol{\theta} \rangle} = \frac{\langle \mathbf{z} + (\mathbf{z}' - \mathbf{z}), \boldsymbol{\theta} \rangle}{\langle \mathbf{z}, \boldsymbol{\theta} \rangle} \geq 1 - \frac{\|\mathbf{z}' - \mathbf{z}\|_2}{\|\mathbf{z}\|_2} = \frac{\|\mathbf{z}\|_2 - \|\mathbf{z}' - \mathbf{z}\|_2}{\|\mathbf{z}\|_2}.
$$

It follows that

$$
\begin{aligned}
\frac{\langle F(\mathbf{z}'), \boldsymbol{\theta} \rangle}{\langle F(\mathbf{z}), \boldsymbol{\theta} \rangle} &= \frac{\langle \mathbf{z}'/\|\mathbf{z}'\|_2, \boldsymbol{\theta} \rangle}{\langle \mathbf{z}/\|\mathbf{z}\|_2, \boldsymbol{\theta} \rangle} = \frac{\|\mathbf{z}\|_2}{\|\mathbf{z}'\|_2} \cdot \frac{\langle \mathbf{z}', \boldsymbol{\theta} \rangle}{\langle \mathbf{z}, \boldsymbol{\theta} \rangle} \geq \frac{\|\mathbf{z}\|_2}{\|\mathbf{z}'\|_2} \cdot \frac{\|\mathbf{z}\|_2 - \|\mathbf{z}' - \mathbf{z}\|_2}{\|\mathbf{z}\|_2} \\
&\geq \frac{\|\mathbf{z}\|_2}{\|\mathbf{z}\|_2 + \|\mathbf{z}' - \mathbf{z}\|_2} \cdot \frac{\|\mathbf{z}\|_2 - \|\mathbf{z}' - \mathbf{z}\|_2}{\|\mathbf{z}\|_2} = \frac{\|\mathbf{z}\|_2 - \|\mathbf{z}' - \mathbf{z}\|_2}{\|\mathbf{z}\|_2 + \|\mathbf{z}' - \mathbf{z}\|_2} \\
&\geq \frac{\|\mathbf{z}\|_2 - \varepsilon}{\|\mathbf{z}\|_2 + \varepsilon} = 1 - \frac{2\varepsilon}{\|\mathbf{z}\|_2 + \varepsilon}.
\end{aligned}
$$

We know $\|\mathbf{z}\|_2 \geq \|\mathbf{z}_\mathcal{I}\|_2 \geq \frac{\sqrt{4a^2 + b^2} - b}{2a^2}$, where $a^2 = 1 + \sqrt{3}/3 + 2\sqrt{q}$ and $b = \sqrt{3}/3$. Let $\varepsilon = \frac{\sqrt{4a^2 + b^2} - b}{6a^2}$, we have

$$
\frac{\langle F(\mathbf{z}'), \boldsymbol{\theta} \rangle}{\langle F(\mathbf{z}), \boldsymbol{\theta} \rangle} \geq \frac{1}{2}.
$$

Therefore, we construct a sparse vector $\mathbf{z}'$ such that (34) holds. To derive $d$, we have

$$
m \leq \frac{120sq^2}{\varepsilon^2}.
$$

Note that

$$
\begin{aligned}
\varepsilon^2 &= \frac{4a^2 + 2b^2 - 2b\sqrt{4a^2 + b^2}}{36a^4} \\
&= \frac{4(1 + \sqrt{3}/3 + 2\sqrt{q}) + 2/3 - 2\sqrt{3}/3\sqrt{4(1 + \sqrt{3}/3 + 2\sqrt{q}) + 1/3}}{36(1 + \sqrt{3}/3 + 2\sqrt{q})^2} \\
&= \frac{x + 2/3 - 2\sqrt{3}/3\sqrt{x + 1/3}}{9x^2},
\end{aligned}
$$

where $x = 4(1 + \sqrt{3}/3 + 2\sqrt{q})$. Since $q \geq 1$, $x \geq 4(3 + \sqrt{3}/3)$. By the derivative

$$
\frac{d}{dx}(x + 2/3 - 2\sqrt{3}/3\sqrt{x + 1/3}) = 1 - \frac{1}{\sqrt{3x + 1}} > 0, \text{ when } x > 4(3 + \sqrt{3}/3).
$$

Substitute $x = 14$ into the numerator

$$
\varepsilon^2 > \frac{10}{9x^2} > \frac{1}{x^2} = \frac{1}{16(1 + \sqrt{3}/3 + 2\sqrt{q})^2} > \frac{1}{16 \times 14q} = \frac{1}{224q}.
$$

Then, we have

$$
m \leq \frac{120sq^2}{\varepsilon^2} < 26880sq^3.
$$

$\square$

Let $M_{\text{net}}$ be the cardinality of a $1/2$-net of $\mathcal{L}(s) \cap \mathbb{S}^{pq-1}$. We want to bound $M_{\text{net}}$, which will be used in the later proof. With lemma E.3 and using the same argument in Rudelson & Zhou (2012)[Section H.1], we have

$$
M_{\text{net}} \leq (1 + 2/(\frac{1}{2}))^d \cdot \binom{pq}{d} \leq 5^d (\frac{epq}{d})^d = \exp(d \log(\frac{5epq}{d})),
$$

Let $C_d = 26883$,

$$
\log(M_{\text{net}}) \leq d \log(\frac{5epq}{d}) = C_d sq^3 \log(\frac{5epq}{C_d sq^3}) = C_d sq^3 [\log(\frac{p}{sq^2}) + \log(\frac{5e}{C_d})] \leq C_{\text{net}} sq^3 \log(\frac{p}{sq^2}),
$$

for some $C_{\text{net}} > 0$, when $p > csq^2$ for sufficiently large $c$. If we want to eliminate $\log 5e/C_d$, $\log(\frac{p}{sq^2}) \geq \log 5e/C_d$. That is the reason we require $p > csq^2$.

## E.2. Concentration of the matrices $\mathbf{U}_w$

Recall that $\widehat{\mathbf{U}}_1(\boldsymbol{\theta}) = \frac{1}{n} \sum_{i=1}^n \gamma_{1,\boldsymbol{\theta}}(\mathbf{X}_i, Y_i) \mathbf{X}_i \mathbf{f}_i^T$, $\mathbf{U}_1(\boldsymbol{\theta}) = \frac{1}{n} \sum_{i=1}^n \mathrm{E}[\gamma_{1,\boldsymbol{\theta}}(\mathbf{X}_i, Y_i) \mathbf{X}_i] \mathbf{f}_i^T$. We want to bound

$$W^{\mathbf{U}} = \sup_{\boldsymbol{\theta} \in \mathcal{B}_{\mathrm{con}}(\boldsymbol{\theta}^*)} \|\widehat{\mathbf{U}}_1(\boldsymbol{\theta}) - \mathbf{U}_1(\boldsymbol{\theta})\|_{F,s}.$$

By definition, we have

$$W^{\mathbf{U}} = \sup_{\mathrm{vec}(\mathbf{u}) \in \mathbf{L}(s) \cap \mathcal{S}^{pq-1}} \sup_{\boldsymbol{\theta} \in \mathcal{B}_{\mathrm{con}}(\boldsymbol{\theta}^*)} \langle \frac{1}{n} \sum_{i=1}^n \gamma_{1,\boldsymbol{\theta}}(\mathbf{X}_i, Y_i) \mathbf{X}_i \mathbf{f}_i^T - \frac{1}{n} \sum_{i=1}^n \mathrm{E}[\gamma_{1,\boldsymbol{\theta}}(\mathbf{X}_i, Y_i) \mathbf{X}_i] \mathbf{f}_i^T, \mathbf{u} \rangle_F$$

$$= \sup_{\mathrm{vec}(\mathbf{u}) \in \mathbf{L}(s) \cap \mathcal{S}^{pq-1}} \sup_{\boldsymbol{\theta} \in \mathcal{B}_{\mathrm{con}}(\boldsymbol{\theta}^*)} \langle \frac{1}{n} \sum_{i=1}^n (\gamma_{1,\boldsymbol{\theta}}(\mathbf{X}_i, Y_i) \mathbf{X}_i - \mathrm{E}[\gamma_{1,\boldsymbol{\theta}}(\mathbf{X}_i, Y_i) \mathbf{X}_i]) \mathbf{f}_i^T, \mathbf{u} \rangle_F.$$

Define

$$W_{\mathbf{u}}^{\mathbf{U}} = \sup_{\boldsymbol{\theta} \in \mathcal{B}_{\mathrm{con}}(\boldsymbol{\theta}^*)} \langle \frac{1}{n} \sum_{i=1}^n (\gamma_{1,\boldsymbol{\theta}}(\mathbf{X}_i, Y_i) \mathbf{X}_i - \mathrm{E}[\gamma_{1,\boldsymbol{\theta}}(\mathbf{X}_i, Y_i) \mathbf{X}_i]) \mathbf{f}_i^T, \mathbf{u} \rangle_F.$$

Then $W^{\mathbf{U}} = \sup_{\mathrm{vec}(\mathbf{u}) \in \mathbf{L}(s) \cap \mathcal{S}^{pq-1}} W_{\mathbf{u}}^{\mathbf{U}}$. We use an $\varepsilon$-net argument. The first step is approximation. Let $\mathrm{vec}(\mathbf{u}_1), \dots, \mathrm{vec}(\mathbf{u}_{M_{\mathrm{net}}})$ be a $1/2$-net of $\mathcal{L}(s) \cap \mathcal{S}^{pq-1}$. This means that for any $\mathbf{v} \in \mathcal{L}(s) \cap \mathcal{S}^{pq-1}$, there is some index $j \in [M_{\mathrm{net}}]$ such that $\|\mathbf{v} - \mathbf{u}_j\|_F \le 1/2$. We have

$$
\begin{aligned}
W_{\mathbf{v}}^{\mathbf{U}} &\le W_{\mathbf{u}_j}^{\mathbf{U}} + |W_{\mathbf{u}_j}^{\mathbf{U}} - W_{\mathbf{v}}^{\mathbf{U}}| \\
&\le \max_{j \in [M_{\mathrm{net}}]} W_{\mathbf{u}_j}^{\mathbf{U}} + W^{\mathbf{U}} \|\mathbf{u}_j - \mathbf{v}\|_F \\
&\le \max_{j \in [M_{\mathrm{net}}]} W_{\mathbf{u}_j}^{\mathbf{U}} + \frac{1}{2} W^{\mathbf{U}}.
\end{aligned}
$$

Then $W^{\mathbf{U}} = \sup_{\mathbf{v}} W_{\mathbf{v}}^{\mathbf{U}} \le \max_{j \in [M_{\mathrm{net}}]} W_{\mathbf{u}_j}^{\mathbf{U}} + 1/2 W^{\mathbf{U}}$, which implies $W^{\mathbf{U}} \le 2 \max_{j \in [M_{\mathrm{net}}]} W_{\mathbf{u}_j}^{\mathbf{U}}$. Then in the second step, we bound the tail of $W_{\mathbf{u}_j}^{\mathbf{U}}$ for fixed $j$. And the third step is union bound, where use the covering number of $\mathbf{L}(s) \cap \mathcal{S}^{pq-1}$. Let $\{\epsilon_i\}_{i=1}^n$ be i.i.d. Rademacher variables. Recall that

$$\gamma_{1,\boldsymbol{\theta}}(\mathbf{X}_i, Y_i) = \frac{\pi_1}{\pi_1 + (1-\pi_1) \exp\{\underbrace{(\mathbf{X}_i - 1/2(\boldsymbol{\Gamma}_2 + \boldsymbol{\Gamma}_1)\mathbf{f}_i)^T (\boldsymbol{\Gamma}_2 - \boldsymbol{\Gamma}_1)\mathbf{f}_i}_{C_{\boldsymbol{\theta},Y}(\mathbf{X}_i)}\}}$$

$$= \frac{\pi_1}{\pi_1 + (1-\pi_1) \exp(C_{\boldsymbol{\theta},Y}(\mathbf{X}_i))}.$$

Then according to Lemma S.5 in Wang et al. (2024) and Hölder's inequality, we have

$$
E[\exp(\lambda W_{\mathbf{u}_j}^{\mathbf{U}})]
$$

$$
= E[\exp(\lambda \sup_{\boldsymbol{\theta} \in \mathcal{B}_{\mathrm{con}}(\boldsymbol{\theta}^*)} \langle \frac{1}{n} \sum_{i=1}^{n} (\gamma_{1,\boldsymbol{\theta}}(\mathbf{X}_i, Y_i)\mathbf{X}_i - E[\gamma_{1,\boldsymbol{\theta}}(\mathbf{X}_i, Y_i)\mathbf{X}_i]) \, \mathbf{f}_i^T, \mathbf{u}_j \rangle_F)]
$$

$$
= E[\exp(\frac{\lambda}{n} \sup_{\boldsymbol{\theta} \in \mathcal{B}_{\mathrm{con}}(\boldsymbol{\theta}^*)} \sum_{i=1}^{n} \langle (\gamma_{1,\boldsymbol{\theta}}(\mathbf{X}_i, Y_i)\mathbf{X}_i - E[\gamma_{1,\boldsymbol{\theta}}(\mathbf{X}_i, Y_i)\mathbf{X}_i]) \, \mathbf{f}_i^T, \mathbf{u}_j \rangle_F)]
$$

$$
\leq E[\exp(\frac{2\lambda}{n} \Big| \sup_{\boldsymbol{\theta} \in \mathcal{B}_{\mathrm{con}}(\boldsymbol{\theta}^*)} \sum_{i=1}^{n} \epsilon_i \langle \gamma_{1,\boldsymbol{\theta}}(\mathbf{X}_i, Y_i)\mathbf{X}_i \mathbf{f}_i^T, \mathbf{u}_j \rangle_F \Big|)]
$$

$$
= E[\exp(\frac{2\lambda}{n} \Big| \sup_{\boldsymbol{\theta} \in \mathcal{B}_{\mathrm{con}}(\boldsymbol{\theta}^*)} \sum_{i=1}^{n} \Big\{ \epsilon_i (\frac{\pi_1}{\pi_1 + (1-\pi_1)\exp(C_{\boldsymbol{\theta},Y}(\mathbf{X}_i))} - \pi_1) \langle \mathbf{X}_i \mathbf{f}_i^T, \mathbf{u}_j \rangle_F + \epsilon_i \pi_1 \langle \mathbf{X}_i \mathbf{f}_i^T, \mathbf{u}_j \rangle_F \Big\} \Big|)]
$$

$$
\leq \underbrace{E[\exp(\frac{4\lambda}{n} \Big| \sup_{\boldsymbol{\theta} \in \mathcal{B}_{\mathrm{con}}(\boldsymbol{\theta}^*)} \sum_{i=1}^{n} \epsilon_i (\frac{\pi_1}{\pi_1 + (1-\pi_1)\exp(C_{\boldsymbol{\theta},Y}(\mathbf{X}_i))} - \pi_1) \langle \mathbf{X}_i \mathbf{f}_i^T, \mathbf{u}_j \rangle_F \Big|)]^{1/2}}_{(I)} \cdot
$$

$$
\underbrace{E[\exp(\frac{4\lambda}{n} \Big| \sup_{\boldsymbol{\theta} \in \mathcal{B}_{\mathrm{con}}(\boldsymbol{\theta}^*)} \sum_{i=1}^{n} \epsilon_i \pi_1 \langle \mathbf{X}_i \mathbf{f}_i^T, \mathbf{u}_j \rangle_F \Big|)]^{1/2}}_{(II)} \cdot
$$

To bound $(I)$, we use lemma C.1 in Cai et al. (2019). The function $\psi(x) = \frac{\pi_1}{\pi_1 + (1-\pi_1)\exp(x)} - \pi_1$ is Lipschitz with constant $\frac{1-\pi_1}{\pi_1} \leq \frac{1-c_0}{c_0}$ and $\psi(0) = 0$. Since $Y_i, \mathbf{f}_i$ are fixed, $C_{\boldsymbol{\theta},Y}(\mathbf{X}_i)$ is a function defined for random variable $\mathbf{X}_i$. Let $\boldsymbol{\Gamma}^* = \pi_1^* \boldsymbol{\Gamma}_1^* + (1-\pi_1^*) \boldsymbol{\Gamma}_2^*$, and $\mathbf{Z}_i = \mathbf{X}_i - \boldsymbol{\Gamma}^* \mathbf{f}_i$ be the centered random variable. Note that $\mathbf{Z}_i \sim \pi_1^* N(\boldsymbol{\Gamma}_1^* \mathbf{f}_i - \boldsymbol{\Gamma}^* \mathbf{f}_i, \mathbf{I}_p) + (1-\pi_1^*) N(\boldsymbol{\Gamma}_2^* \mathbf{f}_i - \boldsymbol{\Gamma}^* \mathbf{f}_i, \mathbf{I}_p)$.

Therefore

$(I)^2$

$$\leq \mathrm{E}[\exp(\frac{8\lambda}{n}\frac{1-c_0}{c_0}\Big|\sup_{\boldsymbol{\theta}\in\mathcal{B}_{\mathrm{con}}(\boldsymbol{\theta}^*)}\sum_{i=1}^n \epsilon_i C_{\boldsymbol{\theta},Y}(\mathbf{X}_i)\langle\mathbf{X}_i\mathbf{f}_i^T,\mathbf{u}_j\rangle_F\Big|)]$$

$$= \mathrm{E}[\exp(\frac{8\lambda}{n}\frac{1-c_0}{c_0}\Big|\sup_{\boldsymbol{\theta}\in\mathcal{B}_{\mathrm{con}}(\boldsymbol{\theta}^*)}\sum_{i=1}^n \epsilon_i[(\mathbf{X}_i-1/2(\boldsymbol{\Gamma}_2+\boldsymbol{\Gamma}_1)\mathbf{f}_i)^T(\boldsymbol{\Gamma}_2-\boldsymbol{\Gamma}_1)\mathbf{f}_i]\langle\mathbf{X}_i\mathbf{f}_i^T,\mathbf{u}_j\rangle_F\Big|)]$$

$$= \mathrm{E}[\exp(\frac{8\lambda}{n}\frac{1-c_0}{c_0}\Big|\sup_{\boldsymbol{\theta}\in\mathcal{B}_{\mathrm{con}}(\boldsymbol{\theta}^*)}\sum_{i=1}^n \epsilon_i\langle\mathbf{X}_i-1/2(\boldsymbol{\Gamma}_2+\boldsymbol{\Gamma}_1)\mathbf{f}_i,(\boldsymbol{\Gamma}_2-\boldsymbol{\Gamma}_1)\mathbf{f}_i\rangle\langle\mathbf{X}_i\mathbf{f}_i^T,\mathbf{u}_j\rangle_F\Big|)]$$

$$= \mathrm{E}[\exp(\frac{8\lambda}{n}\frac{1-c_0}{c_0}\Big|\sup_{\boldsymbol{\theta}\in\mathcal{B}_{\mathrm{con}}(\boldsymbol{\theta}^*)}\sum_{i=1}^n \epsilon_i\langle\mathbf{Z}_i+\boldsymbol{\Gamma}^*\mathbf{f}_i-1/2(\boldsymbol{\Gamma}_2+\boldsymbol{\Gamma}_1)\mathbf{f}_i,(\boldsymbol{\Gamma}_2-\boldsymbol{\Gamma}_1)\mathbf{f}_i\rangle\langle(\mathbf{Z}_i+\boldsymbol{\Gamma}^*\mathbf{f}_i)\mathbf{f}_i^T,\mathbf{u}_j\rangle_F\Big|)]$$

$$\leq \underbrace{\mathrm{E}[\exp(\frac{32\lambda}{n}\frac{1-c_0}{c_0}\Big|\sup_{\boldsymbol{\theta}\in\mathcal{B}_{\mathrm{con}}(\boldsymbol{\theta}^*)}\sum_{i=1}^n \epsilon_i\langle\mathbf{Z}_i,(\boldsymbol{\Gamma}_2-\boldsymbol{\Gamma}_1)\mathbf{f}_i\rangle\langle\mathbf{Z}_i\mathbf{f}_i^T,\mathbf{u}_j\rangle_F\Big|)]^{1/4}}_{(iv)}\cdot$$

$$\underbrace{\mathrm{E}[\exp(\frac{32\lambda}{n}\frac{1-c_0}{c_0}\Big|\sup_{\boldsymbol{\theta}\in\mathcal{B}_{\mathrm{con}}(\boldsymbol{\theta}^*)}\sum_{i=1}^n \epsilon_i\langle\boldsymbol{\Gamma}^*\mathbf{f}_i-1/2(\boldsymbol{\Gamma}_2+\boldsymbol{\Gamma}_1)\mathbf{f}_i,(\boldsymbol{\Gamma}_2-\boldsymbol{\Gamma}_1)\mathbf{f}_i\rangle\langle\mathbf{Z}_i\mathbf{f}_i^T,\mathbf{u}_j\rangle_F\Big|)]^{1/4}}_{(i)}\cdot$$

$$\underbrace{\mathrm{E}[\exp(\frac{32\lambda}{n}\frac{1-c_0}{c_0}\Big|\sup_{\boldsymbol{\theta}\in\mathcal{B}_{\mathrm{con}}(\boldsymbol{\theta}^*)}\sum_{i=1}^n \epsilon_i\langle\mathbf{Z}_i,(\boldsymbol{\Gamma}_2-\boldsymbol{\Gamma}_1)\mathbf{f}_i\rangle\langle\boldsymbol{\Gamma}^*\mathbf{f}_i\mathbf{f}_i^T,\mathbf{u}_j\rangle_F\Big|)]^{1/4}}_{(ii)}\cdot$$

$$\underbrace{\mathrm{E}[\exp(\frac{32\lambda}{n}\frac{1-c_0}{c_0}\Big|\sup_{\boldsymbol{\theta}\in\mathcal{B}_{\mathrm{con}}(\boldsymbol{\theta}^*)}\sum_{i=1}^n \epsilon_i\langle\boldsymbol{\Gamma}^*\mathbf{f}_i-1/2(\boldsymbol{\Gamma}_2+\boldsymbol{\Gamma}_1)\mathbf{f}_i,(\boldsymbol{\Gamma}_2-\boldsymbol{\Gamma}_1)\mathbf{f}_i\rangle\langle\boldsymbol{\Gamma}^*\mathbf{f}_i\mathbf{f}_i^T,\mathbf{u}_j\rangle_F\Big|)]^{1/4}}_{(iii)}\cdot$$

Note that

$$\begin{aligned}
\|\boldsymbol{\Gamma}_2-\boldsymbol{\Gamma}_1\|_F &= \|\boldsymbol{\Gamma}_2-\boldsymbol{\Gamma}_2^*-(\boldsymbol{\Gamma}_1-\boldsymbol{\Gamma}_1^*)+\boldsymbol{\Gamma}_2^*-\boldsymbol{\Gamma}_1^*\|_F \\
&\leq \|\boldsymbol{\Gamma}_2-\boldsymbol{\Gamma}_2^*\|_F+\|\boldsymbol{\Gamma}_1-\boldsymbol{\Gamma}_1^*\|_F+\|\boldsymbol{\Gamma}_2^*\|_F+\|\boldsymbol{\Gamma}_1^*\|_F \\
&\leq 2C_b\Omega+2M_b.
\end{aligned} \tag{35}$$

We can bound $\|\boldsymbol{\Gamma}_2+\boldsymbol{\Gamma}_1\|_F$ with same quantity. Therefore, $\|(\boldsymbol{\Gamma}_1^*)^T(\boldsymbol{\Gamma}_1-\boldsymbol{\Gamma}_2)\|_2 \leq M_b(2C_b\Omega+2M_b) = 2C_bM_b\Omega+2M_b^2$.
We like to bound

$$\begin{aligned}
\|(\boldsymbol{\Gamma}_1^*-\frac{\boldsymbol{\Gamma}_2+\boldsymbol{\Gamma}_1}{2})^T(\boldsymbol{\Gamma}_2-\boldsymbol{\Gamma}_1)\|_2 &\leq \|\boldsymbol{\Gamma}_1^*-\frac{\boldsymbol{\Gamma}_2+\boldsymbol{\Gamma}_1}{2}\|_F\|\boldsymbol{\Gamma}_2-\boldsymbol{\Gamma}_1\|_2 \\
&\leq (\|\boldsymbol{\Gamma}_1^*\|_F+\|\frac{\boldsymbol{\Gamma}_2+\boldsymbol{\Gamma}_1}{2}\|_F)(2C_b\Omega+2M_b) \\
&\leq 2(C_b\Omega+2M_b)(C_b\Omega+M_b) \\
&= 2C_b^2\Omega^2+6C_bM_b\Omega+4M_b^2.
\end{aligned}$$

We know that $\|\mathbf{A}\|_2 = \max_{\mathbf{x},\mathbf{y}}\mathbf{x}^T\mathbf{A}\mathbf{y}/(\|\mathbf{x}\|_2\|\mathbf{y}\|_2)$ For $(i)$, we have

$$\begin{aligned}
\Big|\sup_{\boldsymbol{\theta}\in\mathcal{B}_{\mathrm{con}}(\boldsymbol{\theta}^*)}&\langle\boldsymbol{\Gamma}^*\mathbf{f}_i-1/2(\boldsymbol{\Gamma}_2+\boldsymbol{\Gamma}_1)\mathbf{f}_i,(\boldsymbol{\Gamma}_2-\boldsymbol{\Gamma}_1)\mathbf{f}_i\rangle\Big| \\
&\leq \sup_{\boldsymbol{\theta}\in\mathcal{B}_{\mathrm{con}}(\boldsymbol{\theta}^*)}\Big|\pi_1^*\langle(\boldsymbol{\Gamma}_1^*-1/2(\boldsymbol{\Gamma}_2+\boldsymbol{\Gamma}_1))\mathbf{f}_i,(\boldsymbol{\Gamma}_2-\boldsymbol{\Gamma}_1)\mathbf{f}_i\rangle\Big|+ \\
&\quad \sup_{\boldsymbol{\theta}\in\mathcal{B}_{\mathrm{con}}(\boldsymbol{\theta}^*)}\Big|(1-\pi_1^*)\langle(\boldsymbol{\Gamma}_2^*-1/2(\boldsymbol{\Gamma}_2+\boldsymbol{\Gamma}_1))\mathbf{f}_i,(\boldsymbol{\Gamma}_2-\boldsymbol{\Gamma}_1)\mathbf{f}_i\rangle\Big| \\
&\leq \sup_{\boldsymbol{\theta}\in\mathcal{B}_{\mathrm{con}}(\boldsymbol{\theta}^*)}\Big|\|\mathbf{f}_i\|_2^2(2C_b^2\Omega^2+6C_bM_b\Omega+4M_b^2)\Big|.
\end{aligned}$$

Let $\widetilde{C}_{11} = (2C_b^2\Omega^2 + 6C_b M_b \Omega + 4M_b^2)\max_i \|\mathbf{f}_i\|_2^2$. Therefore,

$$(i)^4$$

$$= \mathrm{E}[\exp(\frac{32\lambda}{n}\frac{1-c_0}{c_0}\Big|\sup_{\boldsymbol{\theta}\in\mathcal{B}_{\mathrm{con}}(\boldsymbol{\theta}^*)}\sum_{i=1}^n \epsilon_i\langle\boldsymbol{\Gamma}^*\mathbf{f}_i - 1/2(\boldsymbol{\Gamma}_2+\boldsymbol{\Gamma}_1)\mathbf{f}_i, (\boldsymbol{\Gamma}_2-\boldsymbol{\Gamma}_1)\mathbf{f}_i\rangle\langle\mathbf{Z}_i\mathbf{f}_i^T,\mathbf{u}_j\rangle_F\Big|)]$$

$$\leq \mathrm{E}[\exp(\frac{32\lambda}{n}\frac{1-c_0}{c_0}\underbrace{\Big|\sup_{\boldsymbol{\theta}\in\mathcal{B}_{\mathrm{con}}(\boldsymbol{\theta}^*)}\langle\boldsymbol{\Gamma}^*\mathbf{f}_i - 1/2(\boldsymbol{\Gamma}_2+\boldsymbol{\Gamma}_1)\mathbf{f}_i, (\boldsymbol{\Gamma}_2-\boldsymbol{\Gamma}_1)\mathbf{f}_i\rangle\Big|}_{\leq\widetilde{C}_{11}}\cdot\Big|\sup_{\boldsymbol{\theta}\in\mathcal{B}_{\mathrm{con}}(\boldsymbol{\theta}^*)}\sum_{i=1}^n \epsilon_i\langle\mathbf{Z}_i\mathbf{f}_i^T,\mathbf{u}_j\rangle_F\Big|)]$$

$$\leq \mathrm{E}[\exp(\frac{32\lambda}{n}\frac{1-c_0}{c_0}\widetilde{C}_{11}\cdot\Big|\sup_{\boldsymbol{\theta}\in\mathcal{B}_{\mathrm{con}}(\boldsymbol{\theta}^*)}\sum_{i=1}^n \epsilon_i\langle\mathbf{Z}_i\mathbf{f}_i^T,\mathbf{u}_j\rangle_F\Big|)].$$

Let that

$$\widetilde{\mathbf{Z}}_i|(W_i = w) = \langle\mathbf{Z}_i\mathbf{f}_i^T,\mathbf{u}_j\rangle_F|(W_i = w) = \mathrm{vec}(\mathbf{u}_j)^T(\mathbf{f}_i\otimes\mathbf{I}_p)\mathbf{Z}_i|(W_i = w).$$

Since $\mathbf{Z}_i|(W_i = w) \sim N(\boldsymbol{\Gamma}_w^*\mathbf{f}_i - \boldsymbol{\Gamma}^*\mathbf{f}_i, \mathbf{I}_p)$, $\mathrm{var}(\widetilde{\mathbf{Z}}_i|(W_i = w)) = \|\mathrm{vec}(\mathbf{u}_j)^T(\mathbf{f}_i\otimes\mathbf{I}_p)\|_2^2 \leq M_4^2$. Therefore $\|\epsilon_i\widetilde{\mathbf{Z}}_i|(W_i = w)\|_{\psi_2} = \|\widetilde{\mathbf{Z}}_i|(W_i = w)\|_{\psi_2} \leq CM_4$. Since $\epsilon_i$ is independent of $\mathbf{Z}_i$, $\mathrm{E}[\epsilon_i\widetilde{\mathbf{Z}}_i|(W_i = w)] = 0$. Then, by equation (5.12) in Vershynin (2010), the moment generating function of $\epsilon_i\widetilde{Z}_i|(W_i = w)$ is

$$\mathrm{E}[\exp(t\epsilon_i\widetilde{\mathbf{Z}}_i|(W_i = w))] \leq \exp(CM_4^2t^2).$$

Then, we have

$$\mathrm{E}[\exp(\frac{32\lambda}{n}\frac{1-c_0}{c_0}\widetilde{C}_{11}\cdot\Big|\sum_{i=1}^n \epsilon_i\langle\mathbf{Z}_i\mathbf{f}_i^T,\mathbf{u}_j\rangle_F\Big|)]$$

$$= \mathrm{E}[\max\left\{\exp(\frac{32\lambda}{n}\frac{1-c_0}{c_0}\widetilde{C}_{11}\cdot\sum_{i=1}^n \epsilon_i\langle\mathbf{Z}_i\mathbf{f}_i^T,\mathbf{u}_j\rangle_F, \exp(\frac{32\lambda}{n}\frac{1-c_0}{c_0}\widetilde{C}_{11}\cdot-\sum_{i=1}^n \epsilon_i\langle\mathbf{Z}_i\mathbf{f}_i^T,\mathbf{u}_j\rangle_F\right\}]$$

$$= \mathrm{E}[\max\left\{\exp(\frac{32\lambda}{n}\frac{1-c_0}{c_0}\widetilde{C}_{11}\cdot\sum_{i=1}^n \epsilon_i\widetilde{\mathbf{Z}}_i), \exp(-\frac{32\lambda}{n}\frac{1-c_0}{c_0}\widetilde{C}_{11}\cdot\sum_{i=1}^n \epsilon_i\widetilde{\mathbf{Z}}_i)\right\}]$$

$$\leq \mathrm{E}[\exp(\frac{32\lambda}{n}\frac{1-c_0}{c_0}\widetilde{C}_{11}\cdot\sum_{i=1}^n \epsilon_i\widetilde{\mathbf{Z}}_i)] + \mathrm{E}[\exp(-\frac{32\lambda}{n}\frac{1-c_0}{c_0}\widetilde{C}_{11}\cdot\sum_{i=1}^n \epsilon_i\widetilde{\mathbf{Z}}_i)]$$

$$\leq 2\exp(\frac{\lambda^2}{n}\frac{32^2(1-c_0)^2}{c_0^2}\widetilde{C}_{11}^2 CM_4^2) = 2\exp(\frac{\lambda^2}{n}4C_{11}),$$

where $C_{11} = 32^2/4CM_4^2\frac{(1-c_0)^2}{c_0^2}\widetilde{C}_{11}^2$. Then

$$(i) \leq \mathrm{E}[\exp(\frac{32\lambda}{n}\frac{1-c_0}{c_0}\widetilde{C}_{11}\cdot\Big|\sup_{\boldsymbol{\theta}\in\mathcal{B}_{\mathrm{con}}(\boldsymbol{\theta}^*)}\sum_{i=1}^n \epsilon_i\langle\mathbf{Z}_i\mathbf{f}_i^T,\mathbf{u}_j\rangle_F\Big|)]^{1/4} \leq 2^{1/4}\exp(\frac{\lambda^2}{n}C_{11}). \tag{36}$$

When $\boldsymbol{\theta}\in\mathcal{B}_{\mathrm{con}}(\boldsymbol{\theta}^*)$, $\mathrm{vec}(\boldsymbol{\Gamma}_w-\boldsymbol{\Gamma}_w^*)\in\mathcal{L}(s)$, which implies $\mathrm{vec}(\boldsymbol{\Gamma}_w) = \mathrm{vec}(\boldsymbol{\Gamma}_w^*) + \mathrm{vec}(\mathbf{u}_w)$, $\mathrm{vec}(\mathbf{u}_w)\in\mathcal{L}(s)$. Next, we bound the second term $(ii)$. Note that

$$|\langle\boldsymbol{\Gamma}^*\mathbf{f}_i\mathbf{f}_i^T,\mathbf{u}_j\rangle_F| = |\mathrm{vec}(\mathbf{u}_j)^T\mathrm{vec}(\boldsymbol{\Gamma}^*\mathbf{f}_i\mathbf{f}_i)| \leq \|\mathrm{vec}(\mathbf{u}_j)\|_2 \cdot \|\mathrm{vec}(\boldsymbol{\Gamma}^*\mathbf{f}_i\mathbf{f}_i^T)\|_2$$

$$= \|(\mathbf{f}_i\mathbf{f}_i^T\otimes\mathbf{I}_p)\mathrm{vec}(\boldsymbol{\Gamma}^*)\|_2 \leq \|\mathbf{f}_i\mathbf{f}_i^T\|_2 \cdot \|\mathrm{vec}(\boldsymbol{\Gamma}^*)\|_2$$

$$\leq M_b\|\mathbf{f}_i\|_2^2 \leq M_b\max_i\|\mathbf{f}_i\|_2^2.$$

Then

$$(ii)^4 = \mathrm{E}[\exp(\frac{32\lambda}{n}\frac{1-c_0}{c_0}\Big|\sup_{\boldsymbol{\theta}\in\mathcal{B}_{\mathrm{con}}(\boldsymbol{\theta}^*)}\sum_{i=1}^{n}\epsilon_i\langle\mathbf{Z}_i,(\boldsymbol{\Gamma}_2-\boldsymbol{\Gamma}_1)\mathbf{f}_i\rangle\langle\boldsymbol{\Gamma}^*\mathbf{f}_i\mathbf{f}_i^T,\mathbf{u}_j\rangle_F\Big|)]$$

$$\leq \mathrm{E}[\exp(\frac{32\lambda}{n}\frac{1-c_0}{c_0}M_b\max_i\|\mathbf{f}_i\|_2^2\cdot\sup_{\boldsymbol{\theta}\in\mathcal{B}_{\mathrm{con}}(\boldsymbol{\theta}^*)}\Big|\sum_{i=1}^{n}\epsilon_i\langle\mathbf{Z}_i,(\boldsymbol{\Gamma}_2-\boldsymbol{\Gamma}_1)\mathbf{f}_i\rangle\Big|)]$$

$$\leq \mathrm{E}[\exp(\frac{64\lambda}{n}\frac{1-c_0}{c_0}M_b\max_i\|\mathbf{f}_i\|_2^2\cdot\sup_{\boldsymbol{\theta}\in\mathcal{B}_{\mathrm{con}}(\boldsymbol{\theta}^*)}\Big|\sum_{i=1}^{n}\epsilon_i\langle\mathbf{Z}_i,(\boldsymbol{\Gamma}_2^*+\mathbf{u}_2)\mathbf{f}_i\rangle\Big|)]^{1/2}\cdot$$

$$\mathrm{E}[\exp(\frac{64\lambda}{n}\frac{1-c_0}{c_0}M_b\max_i\|\mathbf{f}_i\|_2^2\cdot\sup_{\boldsymbol{\theta}\in\mathcal{B}_{\mathrm{con}}(\boldsymbol{\theta}^*)}\Big|\sum_{i=1}^{n}\epsilon_i\langle\mathbf{Z}_i,(\boldsymbol{\Gamma}_1^*+\mathbf{u}_1)\mathbf{f}_i\rangle\Big|)]^{1/2}\cdot$$

$$\leq \underbrace{\mathrm{E}[\exp(\frac{128\lambda}{n}\frac{1-c_0}{c_0}M_b\max_i\|\mathbf{f}_i\|_2^2\cdot\Big|\sum_{i=1}^{n}\epsilon_i\langle\mathbf{Z}_i,\boldsymbol{\Gamma}_2^*\mathbf{f}_i\rangle\Big|)]^{1/4}}_{(ii.1)}\cdot$$

$$\underbrace{\mathrm{E}[\exp(\frac{128\lambda}{n}\frac{1-c_0}{c_0}M_b\max_i\|\mathbf{f}_i\|_2^2\cdot\sup_{\mathrm{vec}(\mathbf{u}_2)\in\mathcal{L}(s)}\Big|\sum_{i=1}^{n}\epsilon_i\langle\mathbf{Z}_i,\mathbf{u}_2\mathbf{f}_i\rangle\Big|)]^{1/4}}_{(ii.2)}\cdot$$

$$\underbrace{\mathrm{E}[\exp(\frac{128\lambda}{n}\frac{1-c_0}{c_0}M_b\max_i\|\mathbf{f}_i\|_2^2\cdot\Big|\sum_{i=1}^{n}\epsilon_i\langle\mathbf{Z}_i,\boldsymbol{\Gamma}_1^*\mathbf{f}_i\rangle\Big|)]^{1/4}}_{(ii.3)}\cdot$$

$$\underbrace{\mathrm{E}[\exp(\frac{128\lambda}{n}\frac{1-c_0}{c_0}M_b\max_i\|\mathbf{f}_i\|_2^2\cdot\sup_{\mathrm{vec}(\mathbf{u}_1)\in\mathcal{L}(s)}\Big|\sum_{i=1}^{n}\epsilon_i\langle\mathbf{Z}_i,\mathbf{u}_1\mathbf{f}_i\rangle\Big|)]^{1/4}}_{(ii.4)}\cdot$$

We first focus on the term $(ii.4)$

$$\mathrm{E}[\exp(\frac{128\lambda}{n}\frac{1-c_0}{c_0}M_b\max_i\|\mathbf{f}_i\|_2^2\cdot\sup_{\mathrm{vec}(\mathbf{u}_1)\in\mathcal{L}(s)}\Big|\sum_{i=1}^{n}\epsilon_i\langle\mathbf{Z}_i,\mathbf{u}_1\mathbf{f}_i\rangle\Big|)].$$

Again, we use the $\epsilon$-net argument. Let $\mathrm{vec}(\widetilde{\mathbf{u}}_1),\ldots,\mathrm{vec}(\widetilde{\mathbf{u}}_{M_{\mathrm{net}}})$ be a $1/2$-net of $\mathcal{L}(s)\cap\mathcal{S}^{pq-1}$.

$$\sup_{\mathrm{vec}(\mathbf{u}_1)\in\mathcal{L}(s)}\Big|\sum_{i=1}^{n}\epsilon_i\langle\mathbf{Z}_i,\mathbf{u}_1\mathbf{f}_i\rangle\Big| = \sup_{\mathrm{vec}(\mathbf{u}_1)\in\mathcal{L}(s)}\|\mathbf{u}_1\|_F\Big|\sum_{i=1}^{n}\epsilon_i\langle\mathbf{Z}_i,\mathbf{u}_1/\|\mathbf{u}_1\|_2\mathbf{f}_i\rangle\Big|$$

$$\leq \sup_{\mathrm{vec}(\mathbf{u}_1)\in\mathcal{L}(s)}\|\mathbf{u}_1\|_F\sup_{\mathrm{vec}(\widetilde{\mathbf{u}})\in\mathcal{L}(s)\cap\mathcal{S}^{pq-1}}\Big|\sum_{i=1}^{n}\epsilon_i\langle\mathbf{Z}_i,\widetilde{\mathbf{u}}\mathbf{f}_i\rangle\Big|$$

$$\leq 2\sup_{\mathrm{vec}(\mathbf{u}_1)\in\mathcal{L}(s)}\|\mathbf{u}_1\|_F\max_{j\in[M_{\mathrm{net}}]}\Big|\sum_{i=1}^{n}\epsilon_i\langle\mathbf{Z}_i,\widetilde{\mathbf{u}}_j\mathbf{f}_i\rangle\Big|$$

$$\leq 2C_b\Omega\max_{j\in[M_{\mathrm{net}}]}\Big|\sum_{i=1}^{n}\epsilon_i\langle\mathbf{Z}_i,\widetilde{\mathbf{u}}_j\mathbf{f}_i\rangle\Big|,$$

where in the last equality we use $\text{vec}(\mathbf{u}_1) = \text{vec}(\mathbf{\Gamma}_1) - \text{vec}(\mathbf{\Gamma}_1^*)$ and $\|\mathbf{\Gamma}_1 - \mathbf{\Gamma}_1^*\|_F \le C_b\Omega$. Then

$$\mathrm{E}[\exp(\frac{128\lambda}{n}\frac{1-c_0}{c_0}M_b\max_i\|\mathbf{f}_i\|_2^2 \cdot \sup_{\text{vec}(\mathbf{u}_1)\in\mathcal{L}(s)}\Big|\sum_{i=1}^n\epsilon_i\langle\mathbf{Z}_i,\mathbf{u}_1\mathbf{f}_i\rangle\Big|)]$$

$$\le \mathrm{E}[\exp(\frac{256\lambda}{n}\frac{1-c_0}{c_0}M_b\max_i\|\mathbf{f}_i\|_2^2 C_b\Omega \cdot \max_{j\in[M_\text{net}]}\Big|\sum_{i=1}^n\epsilon_i\langle\mathbf{Z}_i,\widetilde{\mathbf{u}}_j\mathbf{f}_i\rangle\Big|)]$$

$$\le \sum_{j\in[M_\text{net}]}\mathrm{E}[\exp(\frac{256\lambda}{n}\frac{1-c_0}{c_0}M_b\max_i\|\mathbf{f}_i\|_2^2 C_b\Omega\cdot\Big|\sum_{i=1}^n\epsilon_i\langle\mathbf{Z}_i,\widetilde{\mathbf{u}}_j\mathbf{f}_i\rangle\Big|)]$$

$$= \sum_{j\in[M_\text{net}]}\mathrm{E}[\exp(\frac{\lambda}{n}\underbrace{256\frac{1-c_0}{c_0}M_b\max_i\|\mathbf{f}_i\|_2^2 C_b\Omega}_{\widetilde{C}_{12}}\cdot\Big|\sum_{i=1}^n\epsilon_i\widetilde{Z}_i\Big|)],$$

where $\widetilde{Z}_i = \langle\mathbf{Z}_i,\widetilde{\mathbf{u}}_j\mathbf{f}_i\rangle$. Since $\mathrm{E}[\epsilon_i\widetilde{Z}_i|(W_i = w)] = 0$ and $\text{var}[\widetilde{Z}_i|(W_i = w)] \le \|\mathbf{f}_i\|_2^2$, $\mathrm{E}[\exp(t\epsilon_i\widetilde{\mathbf{Z}}_i|(W_i = w))] \le \exp(CM_4^2t^2)$. Therefore, we have

$$(ii.4)^4 \le \sum_{j\in[M_\text{net}]}\mathrm{E}[\exp(\frac{\lambda}{n}\widetilde{C}_{12}\cdot\Big|\sum_{i=1}^n\epsilon_i\widetilde{Z}_i\Big|)]$$

$$= \sum_{j\in[M_\text{net}]}\mathrm{E}[\max\left\{\exp(\frac{\lambda}{n}\widetilde{C}_{12}\cdot\sum_{i=1}^n\widetilde{Z}_i),\exp(\frac{\lambda}{n}\widetilde{C}_{12}\cdot-\sum_{i=1}^n\epsilon_i\widetilde{Z}_i)\right\}]$$

$$\le \sum_{j\in[M_\text{net}]}\mathrm{E}[\exp(\frac{\lambda}{n}\widetilde{C}_{12}\cdot\sum_{i=1}^n\epsilon_i\widetilde{Z}_i)] + \sum_{j\in[M_\text{net}]}\mathrm{E}[\exp(\frac{\lambda}{n}\widetilde{C}_{12}\cdot-\sum_{i=1}^n\epsilon_i\widetilde{Z}_i)]$$

$$\le 2M_\text{net}\prod_{i=1}^n\exp(\frac{\lambda^2}{n^2}\widetilde{C}_{12}^2 CM_4^2) = 2M_\text{net}\exp(\underbrace{\widetilde{C}_{12}^2 CM_4^2}_{4\tilde{C}_{12}}\frac{\lambda^2}{n})$$

$$\le 2\exp(4\bar{C}_{12}\frac{\lambda^2}{n} + C_\text{net}sq^3\log(\frac{p}{sq^2})).$$

Let $\widetilde{Z}_i = \langle\mathbf{Z}_i,\mathbf{\Gamma}_1^*\mathbf{f}_i\rangle$. Then $\mathrm{E}[\epsilon_i\widetilde{Z}_i|(W_i = w)] = 0$, $\text{var}[\widetilde{Z}_i|(W_i = w)] \le M_b^2\|\mathbf{f}_i\|_2^2$. With same the argument,

$$(ii.3)^4 = \mathrm{E}[\exp(\frac{128\lambda}{n}\frac{1-c_0}{c_0}M_b\max_i\|\mathbf{f}_i\|_2^2\cdot\Big|\sum_{i=1}^n\epsilon_i\langle\mathbf{Z}_i,\mathbf{\Gamma}_1^*\mathbf{f}_i\rangle\Big|)]$$

$$\le \mathrm{E}[\exp(\frac{128\lambda}{n}\frac{1-c_0}{c_0}M_b\max_i\|\mathbf{f}_i\|_2^2\cdot\sum_{i=1}^n\epsilon_i\widetilde{Z}_i)] + \mathrm{E}[\exp(\frac{128\lambda}{n}\frac{1-c_0}{c_0}M_b\max_i\|\mathbf{f}_i\|_2^2\cdot-\sum_{i=1}^n\epsilon_i\widetilde{Z}_i)]$$

$$\le 2\exp(\frac{\lambda^2}{n}\underbrace{\frac{128^2(1-c_0)^2}{c_0^2}CM_b^4M_4^4}_{4\widehat{C}_{12}}).$$

Then we have

$$(ii.3)\cdot(ii.4) \le \sqrt{2}\exp((\bar{C}_{12}+\widehat{C}_{12})\frac{\lambda^2}{n} + C_\text{net}/4sq^3\log(\frac{p}{sq^2})).$$

With the same argument, we can derive a similar bound for $(ii.1)\cdot(ii.2)$. Then, for some positive constant $C_{12}$,

$$(ii) \le 2^{1/4}\exp(C_{12}\frac{\lambda^2}{n} + C_\text{net}/8sq^3\log(\frac{p}{sq^2})). \tag{37}$$

Recall that for a bounded random variable $X \in [a,b]$ almost surely with $\mathrm{E}[X] = 0$, $\mathrm{E}[\exp(tX)] \le \exp(t^2(b-a)^2/8)$ for any $t \in \mathbb{R}$. From earlier derivation, we have $|\langle\mathbf{\Gamma}^*\mathbf{f}_i\mathbf{f}_i^T,\mathbf{u}_j\rangle_F| \le M_b\max_i\|\mathbf{f}_i\|_2^2$. Thus, $\mathrm{E}[\epsilon_i\langle\mathbf{\Gamma}^*\mathbf{f}_i\mathbf{f}_i^T,\mathbf{u}_j\rangle_F] = 0$ and

$\epsilon_i \langle \mathbf{\Gamma}^* \mathbf{f}_i \mathbf{f}_i^T, \mathbf{u}_j \rangle_F \in [-M_b \max_i \|\mathbf{f}_i\|_2^2, M_b \max_i \|\mathbf{f}_i\|_2^2]$, which implies

$$\mathrm{E}[\exp(t\epsilon_i \langle \mathbf{\Gamma}^* \mathbf{f}_i \mathbf{f}_i^T, \mathbf{u}_j \rangle_F)] \leq \exp(\frac{t^2 M_b^2 (\max_i \|\mathbf{f}_i\|_2^2)^2}{2}), \quad \forall t \in \mathbb{R}.$$

For $(iii)$, we have

$$(iii)^4$$
$$= \mathrm{E}[\exp(\frac{32\lambda}{n} \frac{1-c_0}{c_0} \Big| \sup_{\boldsymbol{\theta} \in \mathcal{B}_{\mathrm{con}}(\boldsymbol{\theta}^*)} \sum_{i=1}^n \epsilon_i \langle \mathbf{\Gamma}^* \mathbf{f}_i - 1/2(\mathbf{\Gamma}_2 + \mathbf{\Gamma}_1)\mathbf{f}_i, (\mathbf{\Gamma}_2 - \mathbf{\Gamma}_1)\mathbf{f}_i \rangle \langle \mathbf{\Gamma}^* \mathbf{f}_i \mathbf{f}_i^T, \mathbf{u}_j \rangle_F \Big|)]$$
$$\leq \mathrm{E}[\exp(\frac{32\lambda}{n} \frac{1-c_0}{c_0} \widetilde{C}_{11} \cdot \sup_{\boldsymbol{\theta} \in \mathcal{B}_{\mathrm{con}}(\boldsymbol{\theta}^*)} \Big| \sum_{i=1}^n \epsilon_i \langle \mathbf{\Gamma}^* \mathbf{f}_i \mathbf{f}_i^T, \mathbf{u}_j \rangle_F \Big|)]$$
$$\leq \mathrm{E}[\exp(\frac{32\lambda}{n} \frac{1-c_0}{c_0} \widetilde{C}_{11} \cdot \sum_{i=1}^n \epsilon_i \langle \mathbf{\Gamma}^* \mathbf{f}_i \mathbf{f}_i^T, \mathbf{u}_j \rangle_F)] + \mathrm{E}[\exp(\frac{32\lambda}{n} \frac{1-c_0}{c_0} \widetilde{C}_{11} \cdot - \sum_{i=1}^n \epsilon_i \langle \mathbf{\Gamma}^* \mathbf{f}_i \mathbf{f}_i^T, \mathbf{u}_j \rangle_F)]$$
$$\leq 2\exp(\frac{\lambda^2}{n} \underbrace{\frac{512(1-c_0)^2}{c_0^2} \widetilde{C}_{11}^2 M_b^2 M_4^4}_{4C_{13}}).$$

Thus

$$(iii) \leq 2^{1/4} \exp(\frac{\lambda^2}{n} C_{13}). \tag{38}$$

Recall that $\mathrm{vec}(\mathbf{\Gamma}_w - \mathbf{\Gamma}_w^*) \in \mathcal{L}(s)$. There exist $\mathbf{u}_w \in \mathbb{R}^{p \times q}$ such that $\mathrm{vec}(\mathbf{\Gamma}_w) = \mathrm{vec}(\mathbf{\Gamma}_w^*) + \mathrm{vec}(\mathbf{u}_w)$ and $\mathrm{vec}(\mathbf{u}_w) \in \mathcal{L}(s)$. Then we proceed to bound

$$(iv)^4 = \mathrm{E}[\exp(\frac{32\lambda}{n} \frac{1-c_0}{c_0} \Big| \sup_{\boldsymbol{\theta} \in \mathcal{B}_{\mathrm{con}}(\boldsymbol{\theta}^*)} \sum_{i=1}^n \epsilon_i \langle \mathbf{Z}_i, (\mathbf{\Gamma}_2 - \mathbf{\Gamma}_1)\mathbf{f}_i \rangle \langle \mathbf{Z}_i \mathbf{f}_i^T, \mathbf{u}_j \rangle_F \Big|)]$$
$$\leq \mathrm{E}[\exp(\frac{64\lambda}{n} \frac{1-c_0}{c_0} \Big| \sup_{\boldsymbol{\theta} \in \mathcal{B}_{\mathrm{con}}(\boldsymbol{\theta}^*)} \sum_{i=1}^n \epsilon_i \langle \mathbf{Z}_i, \mathbf{\Gamma}_2 \mathbf{f}_i \rangle \langle \mathbf{Z}_i \mathbf{f}_i^T, \mathbf{u}_j \rangle_F \Big|)]^{1/2} \cdot$$
$$\mathrm{E}[\exp(\frac{64\lambda}{n} \frac{1-c_0}{c_0} \Big| \sup_{\boldsymbol{\theta} \in \mathcal{B}_{\mathrm{con}}(\boldsymbol{\theta}^*)} \sum_{i=1}^n \epsilon_i \langle \mathbf{Z}_i, \mathbf{\Gamma}_1 \mathbf{f}_i \rangle \langle \mathbf{Z}_i \mathbf{f}_i^T, \mathbf{u}_j \rangle_F \Big|)]^{1/2}$$
$$\leq \underbrace{\mathrm{E}[\exp(\frac{128\lambda}{n} \frac{1-c_0}{c_0} \Big| \sup_{\boldsymbol{\theta} \in \mathcal{B}_{\mathrm{con}}(\boldsymbol{\theta}^*)} \sum_{i=1}^n \epsilon_i \langle \mathbf{Z}_i, \mathbf{\Gamma}_2^* \mathbf{f}_i \rangle \langle \mathbf{Z}_i \mathbf{f}_i^T, \mathbf{u}_j \rangle_F \Big|)]^{1/4}}_{(iv.1)} \cdot$$
$$\underbrace{\mathrm{E}[\exp(\frac{128\lambda}{n} \frac{1-c_0}{c_0} \Big| \sup_{\boldsymbol{\theta} \in \mathcal{B}_{\mathrm{con}}(\boldsymbol{\theta}^*)} \sum_{i=1}^n \epsilon_i \langle \mathbf{Z}_i, \mathbf{u}_2 \mathbf{f}_i \rangle \langle \mathbf{Z}_i \mathbf{f}_i^T, \mathbf{u}_j \rangle_F \Big|)]^{1/4}}_{(iv.2)} \cdot$$
$$\underbrace{\mathrm{E}[\exp(\frac{128\lambda}{n} \frac{1-c_0}{c_0} \Big| \sup_{\boldsymbol{\theta} \in \mathcal{B}_{\mathrm{con}}(\boldsymbol{\theta}^*)} \sum_{i=1}^n \epsilon_i \langle \mathbf{Z}_i, \mathbf{\Gamma}_1^* \mathbf{f}_i \rangle \langle \mathbf{Z}_i \mathbf{f}_i^T, \mathbf{u}_j \rangle_F \Big|)]^{1/4}}_{(iv.3)} \cdot$$
$$\underbrace{\mathrm{E}[\exp(\frac{128\lambda}{n} \frac{1-c_0}{c_0} \Big| \sup_{\boldsymbol{\theta} \in \mathcal{B}_{\mathrm{con}}(\boldsymbol{\theta}^*)} \sum_{i=1}^n \epsilon_i \langle \mathbf{Z}_i, \mathbf{u}_1 \mathbf{f}_i \rangle \langle \mathbf{Z}_i \mathbf{f}_i^T, \mathbf{u}_j \rangle_F \Big|)]^{1/4}}_{(iv.4)} \cdot$$

For $(iv.4)$, define

$$\widetilde{W}_{\mathbf{u}} := \sup_{\text{vec}(\widetilde{\mathbf{u}}) \in \mathcal{L}(s) \cap \mathbb{S}^{pq-1}} \sum_{i=1}^{n} \epsilon_i \langle \mathbf{Z}_i, \widetilde{\mathbf{u}} \mathbf{f}_i \rangle \langle \mathbf{Z}_i \mathbf{f}_i^T, \mathbf{u}_j \rangle_F$$

$$= \sup_{\text{vec}(\widetilde{\mathbf{u}}) \in \mathcal{L}(s) \cap \mathbb{S}^{pq-1}} \langle \text{vec}(\widetilde{\mathbf{u}}), (\sum_{i=1}^{n} \epsilon_i \, \text{vec}(\mathbf{Z}_i \mathbf{f}_i^T) \, \text{vec}(\mathbf{Z}_i \mathbf{f}_i^T)^T), \text{vec}(\mathbf{u}_j) \rangle,$$

where we use

$$\langle \mathbf{Z}_i, \widetilde{\mathbf{u}} \mathbf{f}_i \rangle = \mathbf{f}_i^T \widetilde{\mathbf{u}}^T \mathbf{Z}_i = \text{tr}(\mathbf{f}_i^T \widetilde{\mathbf{u}}^T \mathbf{Z}_i) = \langle \widetilde{\mathbf{u}}, \mathbf{Z}_i \mathbf{f}_i^T \rangle_F = \langle \text{vec}(\widetilde{\mathbf{u}}), \text{vec}(\mathbf{Z}_i \mathbf{f}_i^T) \rangle.$$

Let

$$\widetilde{W}_{\widetilde{\mathbf{u}}, \mathbf{u}} = \langle \text{vec}(\widetilde{\mathbf{u}}), (\sum_{i=1}^{n} \epsilon_i \, \text{vec}(\mathbf{Z}_i \mathbf{f}_i^T) \, \text{vec}(\mathbf{Z}_i \mathbf{f}_i^T)^T), \text{vec}(\mathbf{u}_j) \rangle,$$

and $\{\widetilde{\mathbf{u}}_1, \ldots, \widetilde{\mathbf{u}}_{M_{\text{net}}}\}$ be a $1/2$-net of of $\mathcal{L}(s) \cap \mathbb{S}^{pq-1}$. For any $\mathbf{v} \in \mathcal{L}(s) \cap \mathbb{S}^{pq-1}$, let $\widetilde{\mathbf{u}}_k$ be one of the closest point in the $1/2$-cover. Then by definition,

$$\frac{\mathbf{v} - \widetilde{\mathbf{u}}_k}{\|\mathbf{v} - \widetilde{\mathbf{u}}_k\|} \in \mathcal{L}(s) \cap \mathbb{S}^{pq-1}, \quad \|\mathbf{v} - \widetilde{\mathbf{u}}_k\|_2 \le \frac{1}{2}.$$

Therefore,

$$\widetilde{W}_{\mathbf{v}, \mathbf{u}} \le \widetilde{W}_{\widetilde{\mathbf{u}}_k, \mathbf{u}} + |\widetilde{W}_{\widetilde{\mathbf{u}}_k, \mathbf{u}} - \widetilde{W}_{\mathbf{v}, \mathbf{u}}|$$

$$\le \max_{k \in [M_{\text{net}}]} \widetilde{W}_{\widetilde{\mathbf{u}}_k, \mathbf{u}} + |\langle \text{vec}(\widetilde{\mathbf{u}}_k - \mathbf{v})/\|\widetilde{\mathbf{u}}_k - \mathbf{v}\|_2, (\sum_{i=1}^{n} \epsilon_i \, \text{vec}(\mathbf{Z}_i \mathbf{f}_i^T) \, \text{vec}(\mathbf{Z}_i \mathbf{f}_i^T)^T), \text{vec}(\mathbf{u}_j) \rangle| \|\widetilde{\mathbf{u}}_k - \mathbf{v}\|_2$$

$$\le \max_{k \in [M_{\text{net}}]} \widetilde{W}_{\widetilde{\mathbf{u}}_k, \mathbf{u}} + \widetilde{W}_{\mathbf{u}} \frac{1}{2},$$

which implies $\widetilde{W}_{\mathbf{u}} \le 2 \max_{k \in [M_{\text{net}}]} \widetilde{W}_{\widetilde{\mathbf{u}}_k, \mathbf{u}}$. Then

$$\sup_{\boldsymbol{\theta} \in \mathcal{B}_{\text{con}}(\boldsymbol{\theta}^*)} \sum_{i=1}^{n} \epsilon_i \langle \mathbf{Z}_i, \mathbf{u}_1 \mathbf{f}_i \rangle \langle \mathbf{Z}_i \mathbf{f}_i^T, \mathbf{u}_j \rangle_F$$

$$= \sup_{\text{vec}(\mathbf{u}_1) \in \mathcal{L}(s)} \|\mathbf{u}_1\|_F \sum_{i=1}^{n} \epsilon_i \langle \mathbf{Z}_i, \mathbf{u}_1/\|\mathbf{u}_1\|_F \mathbf{f}_i \rangle \langle \mathbf{Z}_i \mathbf{f}_i^T, \mathbf{u}_j \rangle_F$$

$$\le C_b \Omega \cdot \widetilde{W}_{\mathbf{u}} \le 2 C_b \Omega \cdot \max_{k \in [M_{\text{net}}]} \widetilde{W}_{\widetilde{\mathbf{u}}_k, \mathbf{u}},$$

where we use $\sup_{\boldsymbol{\theta} \in \mathcal{B}_{\text{con}}(\boldsymbol{\theta}^*)} \|\mathbf{u}_1\|_F = \|\text{vec}(\boldsymbol{\Gamma}_1 - \boldsymbol{\Gamma}_1^*)\|_2 \le C_b \Omega$. Then

$$(iv.4)^4 \le \text{E}[\exp(\frac{128\lambda}{n} \frac{1-c_0}{c_0} \Big| 2 C_b \Omega \cdot \max_{k \in [M_{\text{net}}]} \widetilde{W}_{\widetilde{\mathbf{u}}_k, \mathbf{u}} \Big|)]$$

$$\le \sum_{k \in [M_{\text{net}}]} \text{E}[\exp(\frac{128\lambda}{n} \frac{1-c_0}{c_0} 2 C_b \Omega \cdot \sum_{i=1}^{n} \epsilon_i \langle \mathbf{Z}_i, \widetilde{\mathbf{u}}_k \mathbf{f}_i \rangle \langle \mathbf{Z}_i \mathbf{f}_i^T, \mathbf{u}_j \rangle_F)] +$$

$$\sum_{k \in [M_{\text{net}}]} \text{E}[\exp(\frac{128\lambda}{n} \frac{1-c_0}{c_0} 2 C_b \Omega \cdot - \sum_{i=1}^{n} \epsilon_i \langle \mathbf{Z}_i, \widetilde{\mathbf{u}}_k \mathbf{f}_i \rangle \langle \mathbf{Z}_i \mathbf{f}_i^T, \mathbf{u}_j \rangle_F)].$$

To bound $\epsilon_i \langle \mathbf{Z}_i, \widetilde{\mathbf{u}}_k \mathbf{f}_i \rangle \langle \mathbf{Z}_i \mathbf{f}_i^T, \mathbf{u}_j \rangle_F$ for fixed $\widetilde{\mathbf{u}}_k$ and $\mathbf{u}_j$, we use lemma D.2 in Wang et al. (2015) that states the product of two sub-Gaussian random variables is a sub-exponential random variable. Let $\|\cdot\|_{\psi_1}$ and $\|\cdot\|_{\psi_2}$ denote the sub-exponential and sub-Gaussian norm. Note that $\langle \mathbf{Z}_i, \widetilde{\mathbf{u}}_k \mathbf{f}_i \rangle|(W_i = w)$ and $\langle \mathbf{Z}_i \mathbf{f}_i^T, \mathbf{u}_j \rangle_F|(W_i = w)$ are normal distributions with zero mean and variance less than or equal to $\|\mathbf{f}_i\|_2^2$. For $X \sim N(0, \sigma^2)$, $\|X\|_{\psi_2} \le C_{\psi_2} \sigma$. For some $C_\psi > 0$, we have

$$\|\epsilon_i \langle \mathbf{Z}_i, \widetilde{\mathbf{u}}_k \mathbf{f}_i \rangle \langle \mathbf{Z}_i \mathbf{f}_i^T, \mathbf{u}_j \rangle_F|(W_i = w)\|_{\psi_1} = \|\langle \mathbf{Z}_i, \widetilde{\mathbf{u}}_k \mathbf{f}_i \rangle \langle \mathbf{Z}_i \mathbf{f}_i^T, \mathbf{u}_j \rangle_F|(W_i = w)\|_{\psi_1}$$

$$\le C_{\psi_1} \max\{\|\langle \mathbf{Z}_i, \widetilde{\mathbf{u}}_k \mathbf{f}_i \rangle|(W_i = w)\|_{\psi_2}^2, \|\langle \mathbf{Z}_i \mathbf{f}_i^T, \mathbf{u}_j \rangle_F|(W_i = w)\|_{\psi_2}^2\}$$

$$\le C_{\psi_1} (C_{\psi_2}^2 \max_i \|\mathbf{f}_i\|_2^2 + C_{\psi_2}'^2 \max_i \|\mathbf{f}_i\|_2^2)$$

$$\le C_\psi M_4^2.$$

Note that $\mathrm{E}[\epsilon_i\langle\mathbf{Z}_i,\widetilde{\mathbf{u}}_k\mathbf{f}_i\rangle\langle\mathbf{Z}_i\mathbf{f}_i^T,\mathbf{u}_j\rangle_F|(W_i=w)]=0$ since $\epsilon_i$ is Rademacher random variable independent of $\mathbf{Z}_i$. According to Lemma 5.15 in Vershynin (2010), we obtain

$$\mathrm{E}[\exp(\frac{128\lambda}{n}\frac{1-c_0}{c_0}2C_b\Omega\cdot\epsilon_i\langle\mathbf{Z}_i,\widetilde{\mathbf{u}}_k\mathbf{f}_i\rangle\langle\mathbf{Z}_i\mathbf{f}_i^T,\mathbf{u}_j\rangle_F)]\le\exp(C\frac{128^2\lambda^2}{n^2}\frac{(1-c_0)^2}{c_0^2}4C_b^2\Omega^2M_4^4),$$

when

$$\left|\frac{128\lambda}{n}\frac{1-c_0}{c_0}2C_b\Omega\right|\le\frac{C'}{C_\psi\max_i\|\mathbf{f}_i\|_2^2}.$$

Then

$$(iv.4)^4\le2\sum_{k\in[M_{\mathrm{net}}]}\exp(\frac{\lambda^2}{n}4C\underbrace{\frac{128^2(1-c_0)^2}{c_0^2}C_b^2\Omega^2M_4^4}_{4\widetilde{C}_{14}})$$

$$\le2\exp(\frac{\lambda^2}{n}4\widetilde{C}_{14})M_{\mathrm{net}}\le2\exp(\frac{\lambda^2}{n}4\widetilde{C}_{14}+C_{\mathrm{net}}sq^3\log(\frac{p}{sq^2})).$$

For term $(iv.3)$, using Lemma D.2 (Wang et al., 2015) and Lemma 5.15 (Vershynin, 2010) again, for some positive constant $C$, we have

$$\mathrm{E}[\exp(\frac{128\lambda}{n}\frac{1-c_0}{c_0}\cdot\epsilon_i\langle\mathbf{Z}_i,\mathbf{\Gamma}_1^*\mathbf{f}_i\rangle\langle\mathbf{Z}_i\mathbf{f}_i^T,\mathbf{u}_j\rangle_F)]\le\exp(\frac{\lambda^2}{n^2}C),$$

when

$$\left|\frac{128\lambda}{n}\frac{1-c_0}{c_0}\right|\le\frac{c}{C'(1+M_b)\max_i\|\mathbf{f}_i\|_2^2}.$$

Therefore,

$$(iv.3)^4\le2\exp(\frac{\lambda^2}{n}4\bar{C}_{14}).$$

We can bound

$$(iv.3)\cdot(iv.4)\le\sqrt{2}\exp((\widetilde{C}_{14}+\bar{C}_{14})\frac{\lambda^2}{n}+C_{\mathrm{net}}/4sq^3\frac{p}{sq^2}).$$

The analysis of $(iv.1)\cdot(iv.2)$ is similar to $(iv.3)\cdot(iv.4)$ and we have

$$(iv)\le2^{1/4}\exp(C_{14}\frac{\lambda^2}{n}+C_{\mathrm{net}}/8sq^3\frac{p}{sq^2}). \tag{39}$$

Combing (36), (37), (38), and (39), the bound for $(I)$ is

$$(I)\le[(i)\cdot(ii)\cdot(iii)\cdot(iv)]^{1/2}\le\sqrt{2}\exp(C_I\frac{\lambda^2}{n}+C_{\mathrm{net}}/8sq^3\frac{p}{sq^2}), \tag{40}$$

where $C_I=(C_{11}+C_{12}+C_{13}+C_{14})/2$. It remains to bound $(II)$. Using the same argument, for some positive constant $C_{II}$, we have

$$(II)^2=\mathrm{E}[\exp(\frac{2\lambda}{n}\Big|\sup_{\boldsymbol{\theta}\in\mathcal{B}_{\mathrm{con}}(\boldsymbol{\theta}^*)}\sum_{i=1}^n\epsilon_i\pi_1\langle\mathbf{X}_i\mathbf{f}_i^T,\mathbf{u}_j\rangle_F\Big|)]$$

$$=\mathrm{E}[\exp(\frac{4\lambda}{n}\Big|\sup_{\boldsymbol{\theta}\in\mathcal{B}_{\mathrm{con}}(\boldsymbol{\theta}^*)}\sum_{i=1}^n\epsilon_i\pi_1\langle\mathbf{Z}_i\mathbf{f}_i^T,\mathbf{u}_j\rangle_F\Big|)]^{1/2}\cdot$$

$$\mathrm{E}[\exp(\frac{4\lambda}{n}\Big|\sup_{\boldsymbol{\theta}\in\mathcal{B}_{\mathrm{con}}(\boldsymbol{\theta}^*)}\sum_{i=1}^n\epsilon_i\pi_1\langle\mathbf{\Gamma}^*\mathbf{f}_i\mathbf{f}_i^T,\mathbf{u}_j\rangle_F\Big|)]^{1/2}$$

$$\le2\exp(\frac{\lambda^2}{n}2C_{II}),$$

which implies

$$(II)\le\sqrt{2}\exp(\frac{\lambda^2}{n}C_{II}). \tag{41}$$

Combining the results (40) and (41), we obtain

$$\mathrm{E}[\exp(\lambda W_{\mathbf{u}_j}^{\mathbf{U}})] \leq (I) \cdot (II) \leq 2\exp(\frac{\lambda^2}{n}C_{\mathbf{U}} + C_{\text{net}}/8sq^3 \log(\frac{p}{sq^2})),$$

where $C_{\mathbf{U}} = C_I + C_{II}$. Thus,

$$\mathrm{E}[\exp(\lambda W^{\mathbf{U}}) \leq \mathrm{E}[\exp(2\lambda \max_{j \in [M_{\text{net}}]} W_{\mathbf{u}_j}^{\mathbf{U}})]] \leq \sum_{j \in [M_{\text{net}}]} \mathrm{E}[\exp(2\lambda W_{\mathbf{u}_j}^{\mathbf{U}})]$$

$$\leq 2M_{\text{net}} \exp(\frac{4\lambda^2}{n}C_{\mathbf{U}} + C_{\text{net}}/8sq^3 \log(\frac{p}{sq^2})) \leq 2\exp(\frac{4\lambda^2}{n}C_{\mathbf{U}} + 2C_{\text{net}}sq^3 \log(\frac{p}{sq^2}))$$

$$\leq 2\exp(\frac{4\lambda^2}{n}C_{\mathbf{U}} + 2C_{\text{net}}sq^3 \log(p)).$$

Using the Chernoff bounds, we have

$$P(\sup_{\boldsymbol{\theta} \in \mathcal{B}_{\text{con}}(\boldsymbol{\theta}^*)} \|\widehat{\mathbf{U}}_1(\boldsymbol{\theta}) - \mathbf{U}_1(\boldsymbol{\theta})\|_{F,s} > t) = P(W^{\mathbf{U}} > t)$$

$$\leq \exp(-\lambda t)\mathrm{E}[\exp(\lambda W^{\mathbf{U}})] \leq 2\exp(\frac{4\lambda^2}{n}C_{\mathbf{U}} + 2C_{\text{net}}sq^3 \log(p) - \lambda t).$$

Let $\lambda = \sqrt{nsq^3 \log(p)/C_{\mathbf{U}}}$, $t = (2C_{\text{net}} + 5)\sqrt{C_{\mathbf{U}}sq^3 \log(p)/n}$. Then

$$P(W^{\mathbf{U}} > t) \leq 2\exp(-sq^3 \log(p)) \leq 2\exp(-\log(p)) = o(1).$$

Recall that we require

$$\left|\frac{128\lambda}{n}\frac{1-c_0}{c_0}\right| = \left|128\sqrt{\frac{sq^3 \log(p)}{nC_{\mathbf{U}}}}\frac{1-c_0}{c_0}\right| \leq C'.$$

Therefore as long as $n \geq C''sq^3 \log(p)$ for a sufficiently large $C''$, we have with probability at least $1 - o(1)$,

$$\sup_{\boldsymbol{\theta} \in \mathcal{B}_{\text{con}}(\boldsymbol{\theta}^*)} \|\widehat{\mathbf{U}}_1(\boldsymbol{\theta}) - \mathbf{U}_1(\boldsymbol{\theta})\|_{F,s} \lesssim \sqrt{\frac{sq^3 \log(p)}{n}}. \tag{42}$$

Note that we can get a sharper bound when $sq^2 = o(p)$,

$$\sup_{\boldsymbol{\theta} \in \mathcal{B}_{\text{con}}(\boldsymbol{\theta}^*)} \|\widehat{\mathbf{U}}_1(\boldsymbol{\theta}) - \mathbf{U}_1(\boldsymbol{\theta})\|_{F,s} \lesssim \sqrt{\frac{sq^3 \log(\frac{p}{sq^2})}{n}}.$$

**E.3. Concentration of the weights $\pi_w$**

We proceed to bound $\sup_{\boldsymbol{\theta} \in \mathcal{B}_{\text{con}}(\boldsymbol{\theta}^*)} |\widehat{\pi}_1(\boldsymbol{\theta}) - \pi_1(\boldsymbol{\theta})|$. Recall that

$$\widehat{\pi}_1 = \frac{1}{n}\sum_{i=1}^n \gamma_{1,\boldsymbol{\theta}}(\mathbf{X}_i, Y_i), \pi_1(\boldsymbol{\theta}) = \frac{1}{n}\mathrm{E}[\gamma_{1,\boldsymbol{\theta}}(\mathbf{X}_i, Y_i)],$$

and

$$\gamma_{1,\boldsymbol{\theta}}(\mathbf{X}_i, Y_i) = \frac{\pi_1}{\pi_1 + (1-\pi_1)\exp\{\underbrace{(\mathbf{X}_i - 1/2(\boldsymbol{\Gamma}_2 + \boldsymbol{\Gamma}_1)\mathbf{f}_i)^T(\boldsymbol{\Gamma}_2 - \boldsymbol{\Gamma}_1)\mathbf{f}_i}_{C_{\boldsymbol{\theta},Y}(\mathbf{X}_i)}\}}.$$

Let $W^{\pi} = \sup_{\boldsymbol{\theta} \in \mathcal{B}_{\mathrm{con}}(\boldsymbol{\theta}^*)} |\widehat{\pi}_1(\boldsymbol{\theta}) - \pi_1(\boldsymbol{\theta})|$. We have

$$\mathrm{E}[\exp(\lambda W^{\pi})]$$

$$= \mathrm{E}[\exp(\frac{\lambda}{n} \sup_{\boldsymbol{\theta} \in \mathcal{B}_{\mathrm{con}}(\boldsymbol{\theta}^*)} \Big| \sum_{i=1}^{n} \Big\{ \frac{\pi_1}{\pi_1 + (1 - \pi_1) \exp(C_{\boldsymbol{\theta}, Y}(\mathbf{X}_i))} - \mathrm{E}[\gamma_{1, \boldsymbol{\theta}}(\mathbf{X}_i, Y_i)] \Big\} \Big|)]$$

$$\leq \underbrace{\mathrm{E}[\exp(\frac{\lambda}{n} \sup_{\boldsymbol{\theta} \in \mathcal{B}_{\mathrm{con}}(\boldsymbol{\theta}^*)} \sum_{i=1}^{n} \Big\{ \frac{\pi_1}{\pi_1 + (1 - \pi_1) \exp(C_{\boldsymbol{\theta}, Y}(\mathbf{X}_i))} - \mathrm{E}[\gamma_{1, \boldsymbol{\theta}}(\mathbf{X}_i, Y_i)] \Big\})] +}_{(I)}$$

$$\underbrace{\mathrm{E}[\exp(\frac{\lambda}{n} \sup_{\boldsymbol{\theta} \in \mathcal{B}_{\mathrm{con}}(\boldsymbol{\theta}^*)} \sum_{i=1}^{n} - \Big\{ \frac{\pi_1}{\pi_1 + (1 - \pi_1) \exp(C_{\boldsymbol{\theta}, Y}(\mathbf{X}_i))} - \mathrm{E}[\gamma_{1, \boldsymbol{\theta}}(\mathbf{X}_i, Y_i)] \Big\})]}_{(II)}.$$

Apply Lemma S.5 in Wang et al. (2024) to $(I)$,

$$(I) \leq \mathrm{E}[\exp(\frac{2\lambda}{n} \Big| \sup_{\boldsymbol{\theta} \in \mathcal{B}_{\mathrm{con}}(\boldsymbol{\theta}^*)} \sum_{i=1}^{n} \epsilon_i \frac{\pi_1}{\pi_1 + (1 - \pi_1) \exp(C_{\boldsymbol{\theta}, Y}(\mathbf{X}_i))} \Big|)]$$

$$\leq \underbrace{\mathrm{E}[\exp(\frac{4\lambda}{n} \Big| \sup_{\boldsymbol{\theta} \in \mathcal{B}_{\mathrm{con}}(\boldsymbol{\theta}^*)} \sum_{i=1}^{n} \epsilon_i \Big\{ \frac{\pi_1}{\pi_1 + (1 - \pi_1) \exp(C_{\boldsymbol{\theta}, Y}(\mathbf{X}_i))} - \pi_1 \Big\} \Big|)]^{1/2}}_{(i)} \cdot$$

$$\underbrace{\mathrm{E}[\exp(\frac{4\lambda}{n} \Big| \sup_{\boldsymbol{\theta} \in \mathcal{B}_{\mathrm{con}}(\boldsymbol{\theta}^*)} \sum_{i=1}^{n} \epsilon_i \pi_1 \Big|)]^{1/2}}_{(ii)}.$$

Note that $\psi(x) = \frac{\pi_1}{\pi_1 + (1 - \pi_1) e^x} - \pi_1$ is Lipschitz with constant $\frac{1 - w}{w} \leq \frac{1 - c_0}{c_0}$ and $\psi(0) = 0$. By Lemma C.1 (Cai et al., 2019) with $g(\cdot) = 1$, we have

$$(i)^2 \leq \mathrm{E}[\exp(\frac{8\lambda}{n} \Big| \sup_{\boldsymbol{\theta} \in \mathcal{B}_{\mathrm{con}}(\boldsymbol{\theta}^*)} \sum_{i=1}^{n} \epsilon_i (\mathbf{X}_i - 1/2(\boldsymbol{\Gamma}_2 + \boldsymbol{\Gamma}_1) \mathbf{f}_i)^T (\boldsymbol{\Gamma}_2 - \boldsymbol{\Gamma}_1) \mathbf{f}_i \Big|)]$$

$$\leq \underbrace{\mathrm{E}[\exp(\frac{16\lambda}{n} \Big| \sup_{\boldsymbol{\theta} \in \mathcal{B}_{\mathrm{con}}(\boldsymbol{\theta}^*)} \sum_{i=1}^{n} \epsilon_i \langle \mathbf{Z}_i, (\boldsymbol{\Gamma}_2 - \boldsymbol{\Gamma}_1) \mathbf{f}_i \rangle \Big|)]^{1/2}}_{(i.1)} \cdot$$

$$\underbrace{\mathrm{E}[\exp(\frac{16\lambda}{n} \Big| \sup_{\boldsymbol{\theta} \in \mathcal{B}_{\mathrm{con}}(\boldsymbol{\theta}^*)} \sum_{i=1}^{n} \epsilon_i \langle \boldsymbol{\Gamma}^* \mathbf{f}_i - \frac{1}{2} (\boldsymbol{\Gamma}_2 + \boldsymbol{\Gamma}_1) \mathbf{f}_i, (\boldsymbol{\Gamma}_2 - \boldsymbol{\Gamma}_1) \mathbf{f}_i \rangle \Big|)]^{1/2}}_{(i.2)}.$$

Terms $(i.1)$ and $(i.2)$ can be bounded similarly to (37) and (38). We have

$$(i.1) \leq \sqrt{2} \exp(2C_{11} \frac{\lambda^2}{n} + C_{\mathrm{net}}/2sq^3 \log(\frac{p}{sq^2})),$$

$$(i.2) \leq \sqrt{2} \exp(2C_{12} \frac{\lambda^2}{n}).$$

Term $(ii)$ can be bounded easily using properties of sub-Gaussian random variables,

$$(ii) \leq \sqrt{2} \exp(C_{13} \frac{\lambda^2}{n}).$$

Thus

$$(I) \leq (i) \cdot (ii) \leq 2 \exp(C_I \frac{\lambda^2}{n} + C_{\mathrm{net}}/4sq^3 \log(\frac{p}{sq^2})),$$

where $C_I = C_{11} + C_{12} + C_{13}$. Term $(II)$ can be bounded similarly. Combining the results, we have

$$\mathrm{E}[\exp(\lambda W^\pi)] \le 4 \exp(C_\pi \frac{\lambda^2}{n} + C_{\mathrm{net}} sq^3 \log(\frac{p}{sq^2})).$$

Using the same Chernoff approach as before, we have

$$P(\sup_{\boldsymbol{\theta} \in \mathcal{B}_{\mathrm{con}}(\boldsymbol{\theta}^*)} |\widehat{\pi}_1(\boldsymbol{\theta}) - \pi_1(\boldsymbol{\theta})| > t) = P(W^\pi > t) \le 4 \exp(C_\pi \frac{\lambda^2}{n} + C_{\mathrm{net}} sq^3 \log(p) - \lambda t).$$

Let $\lambda = \sqrt{nsq^3 \log(p)/C_\pi}$, $t = (C_{\mathrm{net}} + 2)\sqrt{C_\pi sq^3 \log(p)/n}$. Then

$$P(W^{\mathbf{U}} > t) \le 4 \exp(-sq^3 \log(p)) \le 4 \exp(-\log(p)) = o(1).$$

Therefore, as long as $n > C' sq^3 \log(p)$ for a sufficiently large $C'$, we have with probability at least $1 - o(1)$,

$$\sup_{\boldsymbol{\theta} \in \mathcal{B}_{\mathrm{con}}(\boldsymbol{\theta}^*)} |\widehat{\pi}_1(\boldsymbol{\theta}) - \pi_1(\boldsymbol{\theta})| \lesssim \sqrt{\frac{sq^3 \log(p)}{n}}. \tag{43}$$

Still when $sq^2 = o(p)$, we have the following sharper bound

$$\sup_{\boldsymbol{\theta} \in \mathcal{B}_{\mathrm{con}}(\boldsymbol{\theta}^*)} |\widehat{\pi}_1(\boldsymbol{\theta}) - \pi_1(\boldsymbol{\theta})| \lesssim \sqrt{\frac{sq^3 \log(\frac{p}{sq^2})}{n}}.$$

### E.4. Concentration of covariance matrices

Last, we study the concentration of $\boldsymbol{\Sigma}_w$. Recall that

$$\widehat{\boldsymbol{\Sigma}}_w(\boldsymbol{\theta}) = \frac{1}{n} \sum_{i=1}^n \gamma_{1,\boldsymbol{\theta}}(\mathbf{X}_i, Y_i) \mathbf{X}_i \mathbf{X}_i^T, \quad \boldsymbol{\Sigma}_w(\boldsymbol{\theta}) = \frac{1}{n} \sum_{i=1}^n \mathrm{E}[\gamma_{1,\boldsymbol{\theta}}(\mathbf{X}_i, Y_i) \mathbf{X}_i \mathbf{X}_i^T].$$

Directly applying Lemma C.1 (Cai et al., 2019) converts the problem into bounding the product of three sub-Gaussian random variables. While it is well known the product of two sub-Gaussian variables is sub-exponential, the product of three or more sub-Gaussian random variables is not necessarily sub-exponential. Therefore, we must use another method than the one used in the concentration of $\mathbf{U}_w$. We use tail bound for unbounded random processes given in Theorem 4 Adamczak (2008).

Let $W^{\boldsymbol{\Sigma}} = \sup_{\boldsymbol{\theta} \in \mathcal{B}_{\mathrm{con}}(\boldsymbol{\theta}^*)} \|(\widehat{\boldsymbol{\Sigma}}_1(\boldsymbol{\theta}) - \boldsymbol{\Sigma}_1(\boldsymbol{\theta}))\mathbf{B}_1^*\|_{F,s}$. By definition, we have

$$W^{\boldsymbol{\Sigma}} = \sup_{\mathrm{vec}(\mathbf{u}) \in \mathcal{L}(s) \cap \mathbb{S}^{pq-1}} \sup_{\boldsymbol{\theta} \in \mathcal{B}_{\mathrm{con}}(\boldsymbol{\theta}^*)} \langle \frac{1}{n} \sum_{i=1}^n \{\gamma_{1,\boldsymbol{\theta}}(\mathbf{X}_i, Y_i) \mathbf{X}_i \mathbf{X}_i^T - \mathrm{E}[\gamma_{1,\boldsymbol{\theta}}(\mathbf{X}_i, Y_i) \mathbf{X}_i \mathbf{X}_i^T]\} \mathbf{B}_1^*, \mathbf{u} \rangle_F.$$

Let

$$W_{\mathbf{u}}^{\boldsymbol{\Sigma}} = \sup_{\boldsymbol{\theta} \in \mathcal{B}_{\mathrm{con}}(\boldsymbol{\theta}^*)} \langle \frac{1}{n} \sum_{i=1}^n \{\gamma_{1,\boldsymbol{\theta}}(\mathbf{X}_i, Y_i) \mathbf{X}_i \mathbf{X}_i^T - \mathrm{E}[\gamma_{1,\boldsymbol{\theta}}(\mathbf{X}_i, Y_i) \mathbf{X}_i \mathbf{X}_i^T]\} \mathbf{B}_1^*, \mathbf{u} \rangle_F.$$

Then $W^{\boldsymbol{\Sigma}} = \sup_{\mathrm{vec}(\mathbf{u}) \in \mathcal{L}(s) \cap \mathbb{S}^{pq-1}} W_{\mathbf{u}}^{\boldsymbol{\Sigma}}$. Let $\mathrm{vec}(\mathbf{u}_1), \ldots, \mathrm{vec}(\mathbf{u}_{M_{\mathrm{net}}})$ denotes the $1/2$-net of $\mathcal{L}(s) \cap \mathbb{S}^{pq-1}$. For any $\mathbf{v} \in \mathcal{L}(s) \cap \mathbb{S}^{pq-1}$, let $\mathbf{u}_j$ be one of the closest point in the $1/2$-cover. Then by definition,

$$\frac{\mathbf{v} - \mathbf{u}_j}{\|\mathbf{v} - \mathbf{u}_j\|} \in \mathcal{L}(s) \cap \mathbb{S}^{pq-1}, \quad \|\mathbf{v} - \mathbf{u}_j\|_2 \le \frac{1}{2}.$$

Therefore,

$$W_{\mathbf{v}}^{\boldsymbol{\Sigma}} \le W_{\mathbf{u}_j}^{\boldsymbol{\Sigma}} + |W_{\mathbf{u}_j}^{\boldsymbol{\Sigma}} - W_{\mathbf{v}}^{\boldsymbol{\Sigma}}| \le \max_{j \in [M_{\mathrm{net}}]} W_{\mathbf{u}_j}^{\boldsymbol{\Sigma}} + \frac{1}{2} W^{\boldsymbol{\Sigma}}$$

which implies $W^{\boldsymbol{\Sigma}} \leq 2 \max_{j \in [M_{\text{net}}]} W_{\mathbf{u}_j}^{\boldsymbol{\Sigma}}$. We proceed to bound $W_{\mathbf{u}_j}^{\boldsymbol{\Sigma}}$ for fixed $\mathbf{u}_j \in \mathcal{L}(s) \cap \mathbb{S}^{pq-1}$. To use Theorem 4 (Adamczak, 2008), define

$$f(\mathbf{X}_i, Y_i) = \gamma_{1,\boldsymbol{\theta}}(\mathbf{X}_i, Y_i) \langle \mathbf{X}_i \mathbf{X}_i^T \mathbf{B}_1^*, \mathbf{u}_j \rangle_F - \mathrm{E}[\gamma_{1,\boldsymbol{\theta}}(\mathbf{X}_i, Y_i) \langle \mathbf{X}_i \mathbf{X}_i^T \mathbf{B}_1^*, \mathbf{u}_j \rangle_F].$$

We have

$$W_{\mathbf{u}_j}^{\boldsymbol{\Sigma}} \leq \frac{1}{n} \sup_{\boldsymbol{\theta} \in \mathcal{B}_{\text{con}}(\boldsymbol{\theta}^*)} |\sum_{i=1}^n f(\mathbf{X}_i, Y_i)|.$$

It is clear that for every $\boldsymbol{\theta} \in \mathcal{B}_{\text{con}}(\boldsymbol{\theta}^*)$ and every $i$, $\mathrm{E}[f(\mathbf{X}_i, Y_i)] = 0$. Next we show $\langle \mathbf{X}_i \mathbf{X}_i^T \mathbf{B}_1^*, \mathbf{u}_j \rangle_F$ is sub-exponential. Note that

$$\langle \mathbf{X}_i \mathbf{X}_i^T \mathbf{B}_1^*, \mathbf{u}_j \rangle_F = \mathrm{vec}(\mathbf{u}_j)^T \mathrm{vec}(\mathbf{X}_i \mathbf{X}_i^T \mathbf{B}_1^*) = \mathrm{vec}(\mathbf{u}_j)^T (\mathbf{I}_q \otimes \mathbf{X}_i \mathbf{X}_i^T) \mathrm{vec}(\mathbf{B}_1^*)$$

$$= \sum_{k=1}^q \mathbf{u}_{j,k}^T \mathbf{X}_i \mathbf{X}_i^T \mathbf{B}_{1,k}^* = \sum_{k=1}^q \langle \mathbf{X}_i, \mathbf{u}_{j,k} \rangle \langle \mathbf{X}_i, \mathbf{B}_{1,k}^* \rangle,$$

where $\mathbf{u}_{j,k}$ and $\mathbf{B}_{1,k}^*$ are the $k$-th column of $\mathbf{u}_j$ and $\mathbf{B}_1^*$. Clearly, $\langle \mathbf{X}_i, \mathbf{u}_{j,k} \rangle$ and $\langle \mathbf{X}_i, \mathbf{B}_{1,k}^* \rangle$ are normal. Then

$$\|\langle \mathbf{X}_i, \mathbf{u}_{j,k} \rangle\|_{\psi_2} \leq \|\langle \mathbf{Z}_i, \mathbf{u}_{j,k} \rangle\|_{\psi_2} + \|\langle \boldsymbol{\psi} \mathbf{f}_i, \mathbf{u}_{j,k} \rangle\|_{\psi_2} \leq C\|\mathbf{u}_{j,k}\|_2 + C',$$

and similarly

$$\|\langle \mathbf{X}_i, \mathbf{B}_{1,k}^* \rangle\|_{\psi_2} \leq C\|\mathbf{B}_{1,k}^*\|_2 + C'.$$

Therefore, by Lemma D.2 (Wang et al., 2015)

$$\|\langle \mathbf{X}_i \mathbf{X}_i^T \mathbf{B}_1^*, \mathbf{u}_j \rangle_F\|_{\psi_1} \leq \sum_{k=1}^q \|\langle \mathbf{X}_i, \mathbf{u}_{j,k} \rangle \langle \mathbf{X}_i, \mathbf{B}_{1,k}^* \rangle\|_{\psi_1}$$

$$\leq \sum_{k=1}^q C \cdot \max\{\|\langle \mathbf{X}_i, \mathbf{u}_{j,k} \rangle\|_{\psi_2}^2, \|\langle \mathbf{X}_i, \mathbf{B}_{1,k}^* \rangle\|_{\psi_2}^2\} \leq C' < \infty.$$

The above results hold for any $\boldsymbol{\theta} \in \mathcal{B}_{\text{con}}(\boldsymbol{\theta}^*)$. Then

$$\| \sup_{\boldsymbol{\theta} \in \mathcal{B}_{\text{con}}(\boldsymbol{\theta}^*)} |f(\mathbf{X}_i, Y_i)| \|_{\psi_1}$$

$$= \| \sup_{\boldsymbol{\theta} \in \mathcal{B}_{\text{con}}(\boldsymbol{\theta}^*)} \left| \langle \gamma_{1,\boldsymbol{\theta}}(\mathbf{X}_i, Y_i) \mathbf{X}_i \mathbf{X}_i^T \mathbf{B}_1^*, \mathbf{u}_j \rangle_F \right| + \sup_{\boldsymbol{\theta} \in \mathcal{B}_{\text{con}}(\boldsymbol{\theta}^*)} \left| \mathrm{E}[\langle \gamma_{1,\boldsymbol{\theta}}(\mathbf{X}_i, Y_i) \mathbf{X}_i \mathbf{X}_i^T, \mathbf{u}_j \rangle_F] \right| \|_{\psi_1}$$

$$\leq \| \sup_{\boldsymbol{\theta} \in \mathcal{B}_{\text{con}}(\boldsymbol{\theta}^*)} \left| \langle \mathbf{X}_i \mathbf{X}_i^T \mathbf{B}_1^*, \mathbf{u}_j \rangle_F \right| \|_{\psi_1} + \| \sup_{\boldsymbol{\theta} \in \mathcal{B}_{\text{con}}(\boldsymbol{\theta}^*)} \left| \mathrm{E}[\langle \mathbf{X}_i \mathbf{X}_i^T, \mathbf{u}_j \rangle_F] \right| \|_{\psi_1}$$

$$< \infty,$$

where we use the fact $0 < \gamma_{1,\boldsymbol{\theta}}(\mathbf{X}_i, Y_i) < 1$. We verified the two conditions for Theorem 4 (Adamczak, 2008).

Define truncated function and the remaining parts of $f(\mathbf{X}_i, Y_i)$ as

$$f_1(\mathbf{X}_i, Y_i) = f(\mathbf{X}_i, Y_i) I(\sup_{\boldsymbol{\theta} \in \mathcal{B}_{\text{con}}(\boldsymbol{\theta}^*)} |f(\mathbf{X}_i, Y_i)| \leq \rho),$$

$$f_2(\mathbf{X}_i, Y_i) = f(\mathbf{X}_i, Y_i) I(\sup_{\boldsymbol{\theta} \in \mathcal{B}_{\text{con}}(\boldsymbol{\theta}^*)} |f(\mathbf{X}_i, Y_i)| > \rho),$$

where $\rho = 8 \mathrm{E}[\max_i \sup_{\boldsymbol{\theta} \in \mathcal{B}_{\text{con}}(\boldsymbol{\theta}^*)} |f(\mathbf{X}_i, Y_i)|]$. Let $Q = \max_i |\langle \mathbf{X}_i \mathbf{X}_i^T \mathbf{B}_1^*, \mathbf{u}_j \rangle_F|$. Since $\langle \mathbf{X}_i \mathbf{X}_i^T \mathbf{B}_1^*, \mathbf{u}_j \rangle_F$ is sub-exponential, $P(|\langle \mathbf{X}_i \mathbf{X}_i^T \mathbf{B}_1^*, \mathbf{u}_j \rangle_F| > x \log n) \leq 2 \exp(-cx \log n)$. Then

$$P(Q > x \log n) \leq \sum_{i=1}^n P(|\langle \mathbf{X}_i \mathbf{X}_i^T \mathbf{B}_1^*, \mathbf{u}_j \rangle_F| > x \log n) \leq 2 \exp(-cx \log n + \log n).$$

Therefore, when $\log n > 1$ we have

$$\mathrm{E}[Q] = \int_0^\infty P(Q > t)dt = \int_0^{\frac{2\log n}{c}} \underbrace{P(Q > t)}_{\leq 1} dt + \int_{\frac{2\log n}{c}}^\infty P(Q > t)dt$$

$$\leq \frac{2\log n}{c} + \int_{\frac{2}{c}}^\infty P(Q > t\log n)d(t\log n)$$

$$\leq \frac{2\log n}{c} + 2\log n \int_{\frac{2}{c}}^\infty \exp(-(ct-1)\log n)dt$$

$$\leq \frac{2\log n}{c} + 2\log n \int_{\frac{2}{c}}^\infty \exp(-ct+1)dt$$

$$= \frac{2\log n}{c} + 2\log n \frac{\exp(-1)}{c} \leq C\log n.$$

Since the above holds for any $\boldsymbol{\theta} \in \mathcal{B}_{\mathrm{con}}(\boldsymbol{\theta}^*)$, we have $\rho \leq C\log n$.

Note that

$$\sup_{\boldsymbol{\theta} \in \mathcal{B}_{\mathrm{con}}(\boldsymbol{\theta}^*)} |\sum_{i=1}^n f(\mathbf{X}_i, Y_i)| \leq \sup_{\boldsymbol{\theta} \in \mathcal{B}_{\mathrm{con}}(\boldsymbol{\theta}^*)} |\sum_{i=1}^n f_1(\mathbf{X}_i, Y_i) - \mathrm{E}[f_1(\mathbf{X}_i, Y_i)]| +$$

$$\sup_{\boldsymbol{\theta} \in \mathcal{B}_{\mathrm{con}}(\boldsymbol{\theta}^*)} |\sum_{i=1}^n f_2(\mathbf{X}_i, Y_i) - \mathrm{E}[f_2(\mathbf{X}_i, Y_i)]|,$$

where we use the fact that $E[f_1] + E[f_2] = 0$. It follows that

$$\mathrm{E}[\sup_{\boldsymbol{\theta} \in \mathcal{B}_{\mathrm{con}}(\boldsymbol{\theta}^*)} |\sum_{i=1}^n f(\mathbf{X}_i, Y_i)|] \leq \mathrm{E}[\sup_{\boldsymbol{\theta} \in \mathcal{B}_{\mathrm{con}}(\boldsymbol{\theta}^*)} |\sum_{i=1}^n f_1(\mathbf{X}_i, Y_i) - \mathrm{E}[f_1(\mathbf{X}_i, Y_i)]|] +$$

$$2\,\mathrm{E}[\sup_{\boldsymbol{\theta} \in \mathcal{B}_{\mathrm{con}}(\boldsymbol{\theta}^*)} |\sum_{i=1}^n f_2(\mathbf{X}_i, Y_i)|].$$

By Markov inequality and definition of $f_2(\mathbf{X}_i, Y_i)$, we have

$$P(\max_{k \leq n} \sup_{\boldsymbol{\theta} \in \mathcal{B}_{\mathrm{con}}(\boldsymbol{\theta}^*)} |\sum_{i=1}^k f_2(\mathbf{X}_i, Y_i)| > 0)$$

$$\leq P(\max_i \sup_{\boldsymbol{\theta} \in \mathcal{B}_{\mathrm{con}}(\boldsymbol{\theta}^*)} |f(\mathbf{X}_i, Y_i)| > \rho) \leq \frac{\mathrm{E}[\max_i \sup_{\boldsymbol{\theta} \in \mathcal{B}_{\mathrm{con}}(\boldsymbol{\theta}^*)} |f(\mathbf{X}_i, Y_i)|]}{\rho} \leq \frac{1}{8},$$

which means

$$t_0 = \inf\{t > 0; P(\max_{k \leq n} \sup_{\boldsymbol{\theta} \in \mathcal{B}_{\mathrm{con}}(\boldsymbol{\theta}^*)} |\sum_{i=1}^k f_2(\mathbf{X}_i, Y_i)| > t) \leq \frac{1}{8}\} = 0.$$

Then, by Proposition 6.8 in Ledoux & Talagrand (1991),

$$\mathrm{E}[\max_{k \leq N} \sup_{\boldsymbol{\theta} \in \mathcal{B}_{\mathrm{con}}(\boldsymbol{\theta}^*)} |\sum_{i=1}^k f_2(\mathbf{X}_i, Y_i)|] \leq \frac{1}{8}\,\mathrm{E}[\max_i \sup_{\boldsymbol{\theta} \in \mathcal{B}_{\mathrm{con}}(\boldsymbol{\theta}^*)} |f_2(\mathbf{X}_i, Y_i)|] \leq \rho \leq C\log n.$$

Thus,

$$\mathrm{E}[\sup_{\boldsymbol{\theta} \in \mathcal{B}_{\mathrm{con}}(\boldsymbol{\theta}^*)} |\frac{1}{n}\sum_{i=1}^n f_2(\mathbf{X}_i, Y_i)|] \leq \mathrm{E}[\max_{k \leq N} \sup_{\boldsymbol{\theta} \in \mathcal{B}_{\mathrm{con}}(\boldsymbol{\theta}^*)} |\frac{1}{n}\sum_{i=1}^k f_2(\mathbf{X}_i, Y_i)|] \leq C_2\frac{\log n}{n}. \tag{44}$$

When $\sup_{\boldsymbol{\theta} \in \mathcal{B}_{\mathrm{con}}(\boldsymbol{\theta}^*)} |f(\mathbf{X}_i, Y_i)| \leq \rho$, $\langle \mathbf{X}_i \mathbf{X}_i^T \mathbf{B}_1^*, \mathbf{u}_j \rangle_F$ is bounded. We proceed to bound $f_1$, by Lemma S.5 (Wang et al., 2024) and Lemma C.1 (Cai et al., 2019),

$$
\begin{aligned}
&\mathrm{E}[\sup_{\boldsymbol{\theta} \in \mathcal{B}_{\mathrm{con}}(\boldsymbol{\theta}^*)} \Big| \frac{1}{n} \sum_{i=1}^n f_1(\mathbf{X}_i, Y_i) - \mathrm{E}[f_1(\mathbf{X}_i, Y_i)]\Big|] \\
&\leq C \underbrace{\mathrm{E}[\sup_{\boldsymbol{\theta} \in \mathcal{B}_{\mathrm{con}}(\boldsymbol{\theta}^*)} \Big| \frac{1}{n} \sum_{i=1}^n \epsilon_i \langle \mathbf{X}_i - \frac{1}{2}(\boldsymbol{\Gamma}_2 + \boldsymbol{\Gamma}_1)\mathbf{f}_i, (\boldsymbol{\Gamma}_2 - \boldsymbol{\Gamma}_1)\mathbf{f}_i \rangle \langle \mathbf{X}_i \mathbf{X}_i^T \mathbf{B}_1^*, \mathbf{u}_j \rangle_F \Big|]}_{(I)} + \\
&\leq C' \underbrace{\mathrm{E}[\sup_{\boldsymbol{\theta} \in \mathcal{B}_{\mathrm{con}}(\boldsymbol{\theta}^*)} \Big| \frac{1}{n} \sum_{i=1}^n \epsilon_i \pi_1 \langle \mathbf{X}_i \mathbf{X}_i^T \mathbf{B}_1^*, \mathbf{u}_j \rangle_F]\Big|}_{(II)},
\end{aligned}
$$

where $\langle \mathbf{X}_i \mathbf{X}_i^T \mathbf{B}_1^*, \mathbf{u}_j \rangle_F$ is bounded for all $i$. Under the condition $\langle \mathbf{X}_i \mathbf{X}_i^T \mathbf{B}_1^*, \mathbf{u}_j \rangle_F$ is bounded, $\epsilon_i \langle \mathbf{X}_i - \frac{1}{2}(\boldsymbol{\Gamma}_2 + \boldsymbol{\Gamma}_1)\mathbf{f}_i, (\boldsymbol{\Gamma}_2 - \boldsymbol{\Gamma}_1)\mathbf{f}_i \rangle \langle \mathbf{X}_i \mathbf{X}_i^T \mathbf{B}_1^*, \mathbf{u}_j \rangle_F$ is sub-exponential for any $\boldsymbol{\Gamma}_1$ and $\boldsymbol{\Gamma}_2$. Define set $\mathcal{T} = \{t : t^T = (\mathrm{vec}(\boldsymbol{\Gamma}_1)^T, \mathrm{vec}(\boldsymbol{\Gamma}_2)^T), \boldsymbol{\Gamma}_1, \boldsymbol{\Gamma}_2 \in \mathcal{B}_{\mathrm{con}}(\boldsymbol{\theta}^*)\}$. Using the argument to derive $\mathcal{L}(s)$, we have $\mathcal{T} \subset C \mathrm{conv}(\bigcup_{|J| \leq d} E_J(2pq) \cap \mathbb{S}^{2pq-1})$, where $d = C_d s q^3$. By the definition of constriction basin, we have

$$
\| \mathrm{vec}(\boldsymbol{\Gamma}_1) - \mathrm{vec}(\boldsymbol{\Gamma}_1') \|_2 \leq \| \mathrm{vec}(\boldsymbol{\Gamma}_1) - \mathrm{vec}(\boldsymbol{\Gamma}_1^*) \|_2 + \| \mathrm{vec}(\boldsymbol{\Gamma}_1') - \mathrm{vec}(\boldsymbol{\Gamma}_1^*) \|_2 \leq 2C_b\Omega.
$$

Therefore,

$$
\begin{aligned}
&\|([\mathrm{vec}(\boldsymbol{\Gamma}_1) - \mathrm{vec}(\boldsymbol{\Gamma}_1')]^T, [\mathrm{vec}(\boldsymbol{\Gamma}_2) - \mathrm{vec}(\boldsymbol{\Gamma}_2')]^T)\|_2 \\
&\leq \sqrt{\| \mathrm{vec}(\boldsymbol{\Gamma}_1) - \mathrm{vec}(\boldsymbol{\Gamma}_1') \|_2 + \| \mathrm{vec}(\boldsymbol{\Gamma}_2) - \mathrm{vec}(\boldsymbol{\Gamma}_2') \|_2} \\
&\leq \| \mathrm{vec}(\boldsymbol{\Gamma}_1) - \mathrm{vec}(\boldsymbol{\Gamma}_1') \|_2 + \| \mathrm{vec}(\boldsymbol{\Gamma}_2) - \mathrm{vec}(\boldsymbol{\Gamma}_2') \|_2 \\
&\leq 4C_b\Omega.
\end{aligned}
$$

The diameter of $\mathcal{T}$

$$
D = \mathrm{diam}(\mathcal{T}) = \sup_{s,t \in \mathcal{T}} d(s,t) \leq 4C_b\Omega,
$$

where $d$ is the $\ell_2$ distance. Note that

$$
\begin{aligned}
&\sup_{t \in \mathcal{T}} \frac{1}{n} \sum_{i=1}^n \mathrm{E}[\Big| \epsilon_i \langle \mathbf{X}_i - \frac{1}{2}(\boldsymbol{\Gamma}_2 + \boldsymbol{\Gamma}_1)\mathbf{f}_i, (\boldsymbol{\Gamma}_2 - \boldsymbol{\Gamma}_1)\mathbf{f}_i \rangle \langle \mathbf{X}_i \mathbf{X}_i^T \mathbf{B}_1^*, \mathbf{u}_j \rangle_F \Big|^q] \\
&\leq \sup_{t \in \mathcal{T}} \frac{1}{n} \sum_{i=1}^n C q^q = C q^q \leq C q! e^q := \frac{q!}{2} C' e^{q-2},
\end{aligned}
$$

where we use the bound of moments of sub-exponential random variables and Stirling's approximation. By Corollary 5.2 in Dirksen (2015),

$$
(I) \leq C_3 \left( \frac{1}{\sqrt{n}} \gamma_2(\mathcal{T}, d_2) + \frac{1}{n} \gamma_1(\mathcal{T}, d_1) \right) + C_4 \left( \frac{1}{\sqrt{n}} + \frac{1}{n} \right),
$$

where $\gamma_1(\mathcal{T}, d_1)$ and $\gamma_2(\mathcal{T}, d_2)$ are Talagrand functional (see Dirksen (2015) for details). Given $t \in \mathcal{T}$, let $K_{t_i} = \epsilon_i \langle \mathbf{X}_i - \frac{1}{2}(\boldsymbol{\Gamma}_2 + \boldsymbol{\Gamma}_1)\mathbf{f}_i, (\boldsymbol{\Gamma}_2 - \boldsymbol{\Gamma}_1)\mathbf{f}_i \rangle \langle \mathbf{X}_i \mathbf{X}_i^T \mathbf{B}_1^*, \mathbf{u}_j \rangle_F$. Dirksen (2015) define

$$
d_1(s,t) = \max_i \| K_{t_i} - K_{s_i} \|_{\psi_1}, \quad d_2(s,t) = \left( \frac{1}{n} \sum_{i=1}^n \| K_{t_i} - K_{s_i} \|_{\psi_2}^2 \right)^{1/2}.
$$

Consider the two metric spaces $(\mathcal{T}, \ell_2)$ and $(\mathcal{T}, d_1)$. We have $d_1(s,t) \leq C\rho\|s - t\|_2$. Then by Theorem 1.3.6 (Talagrand, 2005), $\gamma_1(\mathcal{T}, d_1) \leq C\rho\gamma_1(\mathcal{T}, \ell_2)$. Similar result hold for $\gamma_2(\mathcal{T}, d_2)$. Therefore,

$$
(I) \leq C_3\rho \left( \frac{1}{\sqrt{n}} \gamma_2(\mathcal{T}, \ell_2) + \frac{1}{n} \gamma_1(\mathcal{T}, \ell_2) \right) + C_4 \left( \frac{1}{\sqrt{n}} + \frac{1}{n} \right). \tag{45}
$$

By equation (4) (Dirksen, 2015),

$$\gamma_\alpha(\mathcal{T}, \ell_2) \leq C_\alpha \int_0^\infty (\log N(\mathcal{T}, \ell_2, \epsilon))^{1/\alpha} d\epsilon = C_\alpha \int_0^D (\log N(\mathcal{T}, \ell_2, \epsilon))^{1/\alpha} d\epsilon,$$

where we use $\log(1) = 0$. We know that

$$N(\mathcal{T}, \ell_2, \epsilon) \leq \left(1 + \frac{2}{\epsilon}\right)^d \binom{2pq}{d} \leq \left(1 + \frac{2}{\epsilon}\right)^d \left(\frac{2epq}{d}\right)^d.$$

Then we have

$$\gamma_1(\mathcal{T}, \ell_2) \leq C \int_0^D \log N(\mathcal{T}, \ell_2, \epsilon) d\epsilon$$
$$\leq C \int_0^D sq^3 \log\left(1 + \frac{2}{\epsilon}\right) + sq^3 \log\left(\frac{2ep}{C_d sq^2}\right) d\epsilon$$
$$\leq Csq^3 \log\left(\frac{p}{sq^2}\right) \leq Csq^3 \log p,$$

when $p > C' sq^2$ for some constant. Similarly,

$$\gamma_2(\mathcal{T}, \ell_2) \leq C \int_0^D (\log N(\mathcal{T}, \ell_2, \epsilon))^{1/2} d\epsilon$$
$$\leq C \int_0^D \sqrt{sq^3 \log\left(1 + \frac{2}{\epsilon}\right) + sq^3 \log\left(\frac{2ep}{C_d sq^2}\right)} d\epsilon$$
$$\leq C\sqrt{sq^3 \log p}.$$

Then according to (45),

$$(I) \leq C_1 \log n \left(\sqrt{\frac{sq^3 \log p}{n}} + \frac{sq^3 \log p}{n}\right) \leq C_1 \sqrt{\frac{sq^3 (\log n)^2 \log p}{n}},$$

when $n > sq^3 \log(p)$. Combining with (44), we have

$$\mathrm{E}[\sup_{\boldsymbol{\theta} \in \mathcal{B}_{\mathrm{con}}(\boldsymbol{\theta}^*)} |\frac{1}{n} \sum_{i=1}^n f(\mathbf{X}_i, Y_i)|] \leq C_1 \sqrt{\frac{sq^3 (\log n)^2 \log p}{n}} + C_2 \frac{\log n}{n} \leq C\sqrt{\frac{sq^3 (\log n)^2 \log p}{n}},$$

where we use $\log(n)/n < \log(n)/\sqrt{n} \leq \sqrt{sq^3 \log p} \log(n)/\sqrt{n}$. To use Theorem 4 (Adamczak, 2008), we need to bound

$$\sigma^2 := \sup_{\boldsymbol{\theta} \in \mathcal{B}_{\mathrm{con}}(\boldsymbol{\theta}^*)} \sum_{i=1}^n \mathrm{E}[\{\frac{1}{n} f(\mathbf{X}_i, Y_i)\}^2] = \frac{1}{n^2} \sup_{\boldsymbol{\theta} \in \mathcal{B}_{\mathrm{con}}(\boldsymbol{\theta}^*)} \sum_{i=1}^n \mathrm{E}[\{f(\mathbf{X}_i, Y_i)\}^2].$$

Note that

$$\mathrm{E}[\{f(\mathbf{X}_i, Y_i)\}^2]$$
$$= \mathrm{E}[\{\gamma_{1,\boldsymbol{\theta}}(\mathbf{X}_i, Y_i)\langle \mathbf{X}_i \mathbf{X}_i^T \mathbf{B}_1^*, \mathbf{u}_j\rangle_F - \mathrm{E}[\gamma_{1,\boldsymbol{\theta}}(\mathbf{X}_i, Y_i)\langle \mathbf{X}_i \mathbf{X}_i^T \mathbf{B}_1^*, \mathbf{u}_j\rangle_F]\}^2]$$
$$= \mathrm{E}[(\gamma_{1,\boldsymbol{\theta}}(\mathbf{X}_i, Y_i)\langle \mathbf{X}_i \mathbf{X}_i^T \mathbf{B}_1^*, \mathbf{u}_j\rangle_F)^2] - \mathrm{E}[\gamma_{1,\boldsymbol{\theta}}(\mathbf{X}_i, Y_i)\langle \mathbf{X}_i \mathbf{X}_i^T \mathbf{B}_1^*, \mathbf{u}_j\rangle_F]^2$$
$$\leq \mathrm{E}[(\langle \mathbf{X}_i \mathbf{X}_i^T \mathbf{B}_1^*, \mathbf{u}_j\rangle_F)^2] + \mathrm{E}[\langle \mathbf{X}_i \mathbf{X}_i^T \mathbf{B}_1^*, \mathbf{u}_j\rangle_F]^2 \leq C,$$

since $\langle \mathbf{X}_i \mathbf{X}_i^T \mathbf{B}_1^*, \mathbf{u}_j\rangle_F$ is sub-exponential. Then $\sigma^2 \leq C/n$. The last term to bound before applying the theorem is $\|\max_i \sup_{\boldsymbol{\theta} \in \mathcal{B}_{\mathrm{con}}(\boldsymbol{\theta}^*)} |f(\mathbf{X}_i, Y_i)|\|_{\psi_1}$. Use earlier results,

$$\|\max_i \sup_{\boldsymbol{\theta} \in \mathcal{B}_{\mathrm{con}}(\boldsymbol{\theta}^*)} |\frac{1}{n} f(\mathbf{X}_i, Y_i)|\|_{\psi_1} \leq C\frac{1}{n} \log n \|\sup_{\boldsymbol{\theta} \in \mathcal{B}_{\mathrm{con}}(\boldsymbol{\theta}^*)} |f(\mathbf{X}_i, Y_i)|\|_{\psi_1}$$
$$\leq C\frac{\log n}{n} \|\sup_{\boldsymbol{\theta} \in \mathcal{B}_{\mathrm{con}}(\boldsymbol{\theta}^*)} |\langle \mathbf{X}_i \mathbf{X}_i^T \mathbf{B}_1^*, \mathbf{u}_j\rangle_F|\|_{\psi_1} \leq \frac{C' \log n}{n}.$$

Let $W_{\mathbf{u}_j}^f = \sup_{\boldsymbol{\theta} \in \mathcal{B}_{\mathrm{con}}(\boldsymbol{\theta}^*)} |\frac{1}{n} \sum_{i=1}^{n} f(\mathbf{X}_i, Y_i)|$. Then, for $0 < \eta < 1$ and $\delta > 0$, we have the following result

$$P(W_{\mathbf{u}_j}^f \geq (1 + \eta) \mathrm{E}[W_{\mathbf{u}_j}^f] + t)$$

$$\leq \exp\left(-\frac{t^2}{2(1+\delta)\sigma^2}\right) + 3 \exp\left(-\frac{t}{C \| \max_i \sup_{\boldsymbol{\theta} \in \mathcal{B}_{\mathrm{con}}(\boldsymbol{\theta}^*)} \| f(\mathbf{X}_i, Y_i)\|_{\psi_1}}\right)$$

$$\leq \exp\left(-C_5 n t^2\right) + 3 \exp\left(-\frac{C_6 n t}{\log n}\right).$$

Let $t = C_7 \sqrt{\frac{sq^3 (\log n)^2 \log p}{n}}$, where $C_7$ is sufficiently large. Using union bound, we have

$$P(\max_{j \in M_{\mathrm{net}}} W_{\mathbf{u}_j}^f \geq (1 + \eta) \mathrm{E}[W_{\mathbf{u}_j}^f] + t)$$

$$\leq M_{\mathrm{net}} \exp\left(-C_5 n t^2\right) + 3 M_{\mathrm{net}} \exp\left(-\frac{C_6 n t}{\log n}\right)$$

$$\leq \exp\left(C_{\mathrm{net}} sq^3 \log \frac{p}{sq^2} - C_5 n t^2\right) + 3 \exp\left(C_{\mathrm{net}} sq^3 \log \frac{p}{sq^2} - \frac{C_6 n t}{\log n}\right)$$

$$\leq \exp(-\log p) + 3 \exp(-\log p) = \frac{4}{p},$$

when $n > C_8 sq^3 \log p$ for sufficiently large $C_8$. This means with probability at least $1 - o(1)$,

$$\max_{j \in M_{\mathrm{net}}} W_{\mathbf{u}_j}^f \leq (1 + \eta) \mathrm{E}[W_{\mathbf{u}_j}^f] + C_7 \sqrt{\frac{sq^3 (\log n)^2 \log p}{n}} \leq C_9 \sqrt{\frac{sq^3 (\log n)^2 \log p}{n}}.$$

Recall that $W_{\mathbf{u}_j}^{\boldsymbol{\Sigma}} \leq W_{\mathbf{u}_j}^f$. Then, with probability at least $1 - o(1)$,

$$\sup_{\boldsymbol{\theta} \in \mathcal{B}_{\mathrm{con}}(\boldsymbol{\theta}^*)} \|(\widehat{\boldsymbol{\Sigma}}_1(\boldsymbol{\theta}) - \boldsymbol{\Sigma}_1(\boldsymbol{\theta}))\mathbf{B}_1^*\|_{F,s} = W^{\boldsymbol{\Sigma}} \lesssim \sqrt{\frac{sq^3 (\log n)^2 \log p}{n}} \tag{46}$$

## F. Ancillary Lemmas

We first present some technical lemmas that are used in the proof.

**Lemma F.1.** *Let* $f_1(t) = \frac{e^t}{\{w + (1-w)e^t\}^2}$, $f_2(t) = \frac{te^t}{\{w + (1-w)e^t\}^2}$, $f_3(t) = \frac{(t^2 - b^2)e^t}{\{w + (1-w)e^t\}^2}$, *and* $f_4(t) = \frac{t^3 e^t}{\{w + (1-w)e^t\}^2}$. *Then*

$$f_1(t) \leq \frac{1}{4 \min\{w^2, (1-w)^2\}}, \quad \forall t \in \mathbb{R},$$

$$\sup_{t \geq a} f_1(t) \leq \frac{1}{\min\{w, 1-w\}^2} \exp(-a), \quad \forall a \geq 0,$$

$$|f_2(t)| \leq \frac{1}{2 \min\{w^2, (1-w)^2\}}, \quad \forall t \in \mathbb{R},$$

$$\sup_{|t| \geq a} |f_2(t)| \leq \frac{2}{\min\{w, (1-w)\}^2} \exp(-3a/4), \quad \forall a \geq 0,$$

$$|f_3(t)| \leq \frac{4 + b^2}{\min\{w^2, (1-w)^2\}}, \quad \forall t \in \mathbb{R},$$

$$\sup_{|t| \geq a} |f_3(t)| \leq \frac{4 + b^2}{\min\{w, (1-w)\}^2} \exp(-a/2), \quad \forall a \geq 0,$$

$$|f_4(t)| \leq \frac{2}{\min\{w^2, (1-w)^2\}}, \quad \forall t \in \mathbb{R},$$

$$\sup_{|t| \geq a} |f_4(t)| \leq \frac{8}{\min\{w, (1-w)\}^2} \exp(-a/4), \quad \forall a \geq 0.$$

*Proof.* We use results (C.1) - (C.6) in supplement of Cai et al. (2019) to prove. Since $f_1(t) = \frac{1}{\{we^{-t/2} + (1-w)e^{t/2}\}^2} := \widetilde{f}_1(t/2)$. By (C.1) and (C.2), $f_1(t) \leq \frac{1}{4\min\{w^2,(1-w)^2\}}$, and

$$\sup_{t \geq a} f_1(t) = \sup_{t/2 \geq a/2} \widetilde{f}_1(t/2) \leq \frac{1}{\min\{w, 1-w\}^2} \exp(2 \cdot a/2).$$

Since $f_2(t) = 2\frac{t/2}{\{we^{-t/2} + (1-w)e^{t/2}\}^2} := 2\widetilde{f}_2(t/2)$, using (C.3) and (C.4), we have

$$|f_2(t)| \leq 2|\widetilde{f}_2(t/2)| \leq 2\frac{1}{4\min\{w^2, (1-w)^2\}},$$

and

$$\sup_{t \geq a} |f_2(t)| = 2\sup_{t/2 \geq a/2} |\widetilde{f}_2(t/2)| \leq \frac{2}{\min\{w, (1-w)\}^2} \exp(-\frac{3}{2}\frac{a}{2}),$$

$$\sup_{t \leq -a} |f_2(t)| = \sup_{-t \geq a} |f_2(-t/2)| \leq \frac{2}{\min\{w, (1-w)\}^2} \exp(-\frac{3}{2}\frac{a}{2}).$$

Note that $f_3(t) = 4\frac{(t/2)^2 - (b/2)^2}{\{we^{-t/2} + (1-w)e^{t/2}\}^2} := 4\widetilde{f}_3(t/2)$. Then, by (C.5) and (C.6),

$$|f_3(t)| = 4|\widetilde{f}_3(t/2)| \leq 4\frac{1 + (b/2)^2}{\min\{w^2, (1-w)^2\}} = \frac{4 + b^2}{\min\{w^2, (1-w)^2\}},$$

and

$$\sup_{t \geq a} |f_3(t)| = 4\sup_{t/2 \geq a/2} |\widetilde{f}_3(t/2)| \leq \frac{4 + b^2}{\min\{w, (1-w)\}^2} \exp(-a/2).$$

Define $\widetilde{f}_4(t) = \frac{t^3}{\{we^{-t} + (1-w)e^t\}^2}$. Then $f_4(t) = 8\widetilde{f}_4(t/2)$. Note that

$$|\widetilde{f}_4(t)| = \frac{t^3}{\{we^{-t} + (1-w)e^t\}^2} \leq \frac{t^3}{\min\{w, (1-w)\}^2(e^{-t} + e^t)^2} \leq \frac{1}{4\min\{w, (1-w)\}^2},$$

which implies

$$|f_4(t)| \leq \frac{2}{\min\{w, (1-w)\}^2}.$$

Then note that $\frac{t^3}{\{we^{-t} + (1-w)e^t\}^2} \leq \exp(-a/2)$ when $t \geq a \geq 0$. Therefore,

$$\sup_{t \geq a} |f_4(t)| = 8\sup_{t/2 \geq a/2} |\widetilde{f}_4(t/2)| \leq \frac{8}{\min\{w, (1-w)\}^2} \exp(-a/4).$$

$\square$

**Lemma F.2.** *Let $\mathcal{L}_p(s) = \mathcal{L}(s)_{1:p}$. The restrictive eigenvalue condition*

$$\inf_{\mathbf{u} \in \mathcal{L}_p(s) \cap \mathbb{S}^{p-1}} \left| \mathbf{u}^T \frac{1}{n} \sum_{i=1}^{n} \mathbf{X}_i \mathbf{X}_i^T \mathbf{u} \right| > \tau_0$$

*holds with probability at least $1 - o(1)$ when $n > Cs \log p$ for sufficiently large positive constant $C$.*

*Proof.* Let $\mathcal{C}(s) = 2\operatorname{conv}(\bigcup_{|J| \leq d} E_J(p) \cap \mathbb{S}^{p-1})$, $d = C_d s$. According to Lemma E.3, we have $\mathcal{L}_p(s) \cap \mathbb{S}^{pq-1} \subset \mathcal{C}(s)$. We show a stronger conclusion that

$$\inf_{\mathbf{u} \in \mathcal{C}(s) \cap \mathbb{S}^{p-1}} \left| \mathbf{u}^T \frac{1}{n} \sum_{i=1}^{n} \mathbf{X}_i \mathbf{X}_i^T \mathbf{u} \right| > \tau_1.$$

Since $\mathbf{X}_i \sim \pi_1^* N(\mathbf{\Gamma}_1^* \mathbf{f}_i, \mathbf{\Delta}) + \pi_2^* N(\mathbf{\Gamma}_2^* \mathbf{f}_i, \mathbf{\Delta})$, we have

$$\mathrm{E}[\mathbf{X}_i] = \mathbf{\Gamma}^* \mathbf{f}_i, \quad \mathbf{\Gamma}^* = \pi_1^* \mathbf{\Gamma}_1^* + \pi_2 \mathbf{\Gamma}_2^*,$$
$$\mathrm{E}[\mathbf{X}_i \mathbf{X}_i^T] = \mathbf{\Delta} + \pi_1^* \mathbf{\Gamma}_1^* \mathbf{f}_i \mathbf{f}_i^T (\mathbf{\Gamma}_1^*)^T + \pi_2^* \mathbf{\Gamma}_2^* \mathbf{f}_i \mathbf{f}_i^T (\mathbf{\Gamma}_2^*)^T,$$
$$\mathbf{\Delta}_{\mathbf{X}_i} := \mathrm{cov}(\mathbf{X}_i) = \mathbf{\Delta} + \pi_1^* \mathbf{\Gamma}_1^* \mathbf{f}_i \mathbf{f}_i^T (\mathbf{\Gamma}_1^*)^T + \pi_2^* \mathbf{\Gamma}_2^* \mathbf{f}_i \mathbf{f}_i^T (\mathbf{\Gamma}_2^*)^T - \mathbf{\Gamma}^* \mathbf{f}_i \mathbf{f}_i^T (\mathbf{\Gamma}^*)^T. \tag{47}$$

For any $\mathbf{v} \in \mathcal{C}(s) \cap \mathbb{S}^{p-1}$, $\mathbf{Z}_i = \mathbf{v}^T \mathbf{X}_i$ is mixture of Gaussian and thus sub-Gaussian. Then $\mathbf{Z}_i^2 = \mathbf{v}^T \mathbf{X}_i \mathbf{X}_i^T \mathbf{v}$ is sub-exponential. Thus, $\|\mathbf{v}^T \mathbf{X}_i \mathbf{X}_i^T \mathbf{v} - \mathrm{E}[\mathbf{v}^T \mathbf{X}_i \mathbf{X}_i^T \mathbf{v}]\|_{\psi_1} \leq C_i \|\mathbf{v}^T \mathbf{X}_i \mathbf{X}_i^T \mathbf{v}\|_{\psi_1} := L_i$. Let $L = \max_i L_i$. By Bernstein's inequality (Theorem 2.8.1 in Vershynin (2018)), for $t > 0$,

$$P\Big(\Big| \sum_{i=1}^n \Big\{ \frac{1}{n} \mathbf{v}^T \mathbf{X}_i \mathbf{X}_i^T \mathbf{v} - \frac{1}{n} \mathbf{v}^T \mathrm{E}[\mathbf{X}_i \mathbf{X}_i^T] \mathbf{v} \Big\} \Big| \geq t\Big) \leq 2\exp\Big(-c \min\Big\{ \frac{t^2}{\sum_{i=1}^n \frac{1}{n^2} L_i^2}, \frac{t}{\max_i \frac{1}{n} L_i} \Big\}\Big)$$

$$\leq 2\exp\Big(-c \min\Big\{ \frac{nt^2}{L^2}, \frac{nt}{L} \Big\}\Big) \leq 2\exp(-c_1 n t^2),$$

where the last inequality holds when $t \leq L$ and $c_1 = c/L^2$. Let $\epsilon < 1$, $\mathbf{v}_1, \ldots, \mathbf{v}_{|\mathcal{J}|}$ is an $\epsilon$-net of $\mathcal{C}(s) \cap \mathbb{S}^{p-1}$. Then according to Lemma E.3, $\log |\mathcal{J}| \leq C_{\mathcal{J}} s \log p$, where $C_{\mathcal{J}}$ is a positive constant. By union bound,

$$P\Big(\Big| \sum_{i=1}^n \Big\{ \frac{1}{n} \mathbf{v}_j^T \mathbf{X}_i \mathbf{X}_i^T \mathbf{v}_j - \frac{1}{n} \mathbf{v}_j^T \mathrm{E}[\mathbf{X}_i \mathbf{X}_i^T] \mathbf{v}_j \Big\} \Big| \geq t, \exists j \in \mathcal{J}\Big) \leq 2|\mathcal{J}| \exp(-c_1 n t^2)$$

$$\leq 2\exp(C_{\mathcal{J}} s \log p - c_1 n t^2).$$

Define $\mathbf{\Psi} = 1/n \sum_{i=1}^n \mathbf{X}_i \mathbf{X}_i^T$. Then, we have for all $j \in \mathcal{J}$,

$$\mathbf{v}_j^T \mathrm{E}[\mathbf{\Psi}] \mathbf{v}_j - t \leq \|\mathbf{\Psi}^{1/2} \mathbf{v}_j\|_2^2 \leq \mathbf{v}_j^T \mathrm{E}[\mathbf{\Psi}] \mathbf{v}_j + t,$$

with probability at least $1 - 2\exp(C_{\mathcal{J}} s \log p - c_1 n t^2)$.

Assume $C_1 \leq \sigma_{\min}(\mathbf{\Delta}) \leq \sigma_{\max}(\mathbf{\Delta}) \leq C_2$. For any unit vector $\mathbf{v}$, we have

$$\mathbf{v}^T \mathbf{\Gamma}_1^* \widehat{\mathbf{\Sigma}}_f (\mathbf{\Gamma}_1^*)^T \mathbf{v} \leq \|(\mathbf{\Gamma}_1^*)^T \mathbf{v}\|_2^2 M_2 \leq \sigma_{\max}(\mathbf{\Gamma}_1^*)^2 M_2 \leq M_b^2 M_2.$$

Since $\mathrm{E}[\mathbf{\Psi}] = \mathbf{\Delta} + \pi_1^* \mathbf{\Gamma}_1^* \widehat{\mathbf{\Sigma}}_f (\mathbf{\Gamma}_1^*)^T + \pi_2^* \mathbf{\Gamma}_2^* \widehat{\mathbf{\Sigma}}_f (\mathbf{\Gamma}_2^*)^T$,

$$C_1 \leq \mathbf{v}^T \mathrm{E}[\mathbf{\Psi}] \mathbf{v} \leq C_2 + M_2 M_b^2.$$

Therefore,

$$\sqrt{C_1 - t} \leq \|\mathbf{\Psi}^{1/2} \mathbf{v}_j\|_2 \leq \sqrt{C_2 + M_2 M_b^2 + t}.$$

Then, for any $\mathbf{v} \in \mathcal{C}(s) \cap \mathbb{S}^{p-1}$, there exists a $\mathbf{v}_j$ such that

$$\|\mathbf{v} - \mathbf{v}_j\|_2 \leq \epsilon, \quad \frac{\mathbf{v} - \mathbf{v}_j}{\|\mathbf{v} - \mathbf{v}_j\|_2} \in \mathcal{C}(s) \cap \mathbb{S}^{p-1},$$

and

$$\|\mathbf{\Psi}^{1/2} \mathbf{v}_j\|_2 - \|\mathbf{\Psi}^{1/2} (\mathbf{v} - \mathbf{v}_j)\|_2 \leq \|\mathbf{\Psi}^{1/2} \mathbf{v}\|_2 \leq \|\mathbf{\Psi}^{1/2} \mathbf{v}_j\|_2 + \|\mathbf{\Psi}^{1/2} (\mathbf{v} - \mathbf{v}_j)\|_2.$$

The right-hand side is upper bounded by

$$\sqrt{C_2 + M_2 M_b^2 + t} + \epsilon \sup_{\mathbf{v} \in \mathcal{C}(s) \cap \mathbb{S}^{p-1}} \|\mathbf{\Psi}^{1/2} \mathbf{v}\|_2,$$

which implies

$$\sup_{\mathbf{v} \in \mathcal{C}(s) \cap \mathbb{S}^{p-1}} \|\mathbf{\Psi}^{1/2} \mathbf{v}\|_2 \leq \frac{\sqrt{C_2 + M_2 M_b^2 + t}}{1 - \epsilon}.$$

Meanwhile, the left-hand side is lower bounded by

$$\|\mathbf{\Psi}^{1/2}\mathbf{v}_j\|_2 - \|\mathbf{\Psi}^{1/2}(\mathbf{v} - \mathbf{v}_j)\|_2 \geq \sqrt{C_1 - t} - \epsilon \sup_{\mathbf{v}\in\mathcal{C}(s)\cap\mathbb{S}^{p-1}} \|\mathbf{\Psi}^{1/2}\mathbf{v}\|_2$$

$$\geq \sqrt{C_1 - t} - \frac{\epsilon\sqrt{C_2 + M_2 M_b^2 + t}}{1 - \epsilon},$$

which means

$$\inf_{\mathbf{v}\in\mathcal{C}(s)\cap\mathbb{S}^{p-1}} \|\mathbf{\Psi}^{1/2}\mathbf{v}\|_2 \geq \sqrt{C_1 - t} - \frac{\epsilon\sqrt{C_2 + M_2 M_b^2 + t}}{1 - \epsilon}.$$

Let $C = C_2 + M_2 M_b^2$, $t = C_1/2$ and

$$\epsilon \leq \frac{\sqrt{C_1/2} - \tau_1}{\sqrt{C + C_1/2} + \sqrt{C_1/2 - \tau_1}}.$$

When $0 < \epsilon < 1$, $0 < \tau_1 < \sqrt{C_1/2}$. We have

$$\inf_{\mathbf{v}\in\mathcal{C}(s)\cap\mathbb{S}^{p-1}} \|\mathbf{\Psi}^{1/2}\mathbf{v}\|_2 \geq \tau_1,$$

with probability at least $1 - 2\exp(C_{\mathcal{J}}s\log p - c_1 n t^2)$ for $0 < \tau_1 < \sqrt{C_1/2}$. When $n > Cs\log p$ for sufficiently large $C$, $\exp(C_{\mathcal{J}}s\log p - c_1 n t^2) < 2\exp(-\log p) = 2/p = o(1)$.

$\square$

**Lemma F.3.** *Let* $\mathbf{U} = \frac{1}{n}\sum_{i=1}^{n} \mathbf{X}_i \mathbf{f}_i^T$. *The following*

$$\|\mathbf{U}\|_{F,s} = \sup_{\substack{\mathbf{v}\in\mathbb{R}^{p\times q} \\ \mathbf{v}\in\mathcal{L}(s)\cap\mathbb{S}^{pq-1}}} \langle\mathbf{U}, \mathbf{v}\rangle_F \leq M,$$

*holds with probability at least* $1 - o(1)$ *when* $n > Csq^3 \log p$ *for a sufficiently large positive constant* $C$.

*Proof.* By definition

$$\|\mathbf{U}\|_{F,s} = \sup_{\substack{\mathbf{v}\in\mathbb{R}^{p\times q} \\ \mathrm{vec}(\mathbf{v})\in\mathcal{L}(s)\cap\mathbb{S}^{pq-1}}} \langle\frac{1}{n}\sum_{i=1}^{n}\mathbf{X}_i\mathbf{f}_i^T, \mathbf{v}\rangle_F = \sup_{\substack{\mathbf{v}\in\mathbb{R}^{p\times q} \\ \mathrm{vec}(\mathbf{v})\in\mathcal{L}(s)\cap\mathbb{S}^{pq-1}}} \frac{1}{n}\sum_{i=1}^{n}\mathrm{vec}(\mathbf{v})^T(\mathbf{f}_i\otimes\mathbf{I}_p)\mathbf{X}_i.$$

Note that, for any $\mathrm{vec}(\mathbf{v})\in\mathcal{L}(s)\cap\mathbb{S}^{pq-1}$, $\mathrm{E}[\mathrm{vec}(\mathbf{v})^T(\mathbf{f}_i\otimes\mathbf{I}_p)\mathbf{X}_i] = \mathrm{vec}(\mathbf{v})^T(\mathbf{f}_i\otimes\mathbf{I}_p)\mathbf{\Gamma}^*\mathbf{f}_i$, where $\mathbf{\Gamma}^* = \pi_1^*\mathbf{\Gamma}_1^* + \pi_2\mathbf{\Gamma}_2^*$. Since $\mathrm{vec}(\mathbf{v})^T(\mathbf{f}_i\otimes\mathbf{I}_p)\mathbf{X}_i$ is sub-Gaussian, we have $\|\mathrm{vec}(\mathbf{v})^T(\mathbf{f}_i\otimes\mathbf{I}_p)\mathbf{X}_i - \mathrm{vec}(\mathbf{v})^T(\mathbf{f}_i\otimes\mathbf{I}_p)\mathbf{\Gamma}^*\mathbf{f}_i\|_{\psi_1} \leq C_1$. By Bernstein's inequality (Theorem 2.8.1 in Vershynin (2018)), for $t > 0$,

$$P\left(\left|\sum_{i=1}^{n}\left\{\frac{1}{n}\mathrm{vec}(\mathbf{v})^T(\mathbf{f}_i\otimes\mathbf{I}_p)\mathbf{X}_i - \mathrm{vec}(\mathbf{v})^T(\mathbf{f}_i\otimes\mathbf{I}_p)\mathbf{\Gamma}^*\mathbf{f}_i\right\}\right| \geq t\right)$$

$$\leq 2\exp\left(-c\min\left\{\frac{t^2}{\sum_{i=1}^{n}\frac{1}{n^2}C_1^2}, \frac{t}{\max_i \frac{1}{n}C_1}\right\}\right) \leq 2\exp\left(-c\min\left\{\frac{nt^2}{C_1^2}, \frac{nt}{C_1}\right\}\right)$$

$$\leq 2\exp(-c_1 n t^2),$$

where the last inequality holds when $t \leq C_1$ and $c_1 = c/C_1^2$. Let $\mathrm{vec}(\mathbf{v}_1),\ldots,\mathrm{vec}(\mathbf{v}_{M_{\mathrm{net}}})$ is an $1/2$-net of $\mathcal{L}(s)\cap\mathbb{S}^{pq-1}$. Then according to Lemma E.3, $\log M_{\mathrm{net}} \leq C_{\mathrm{net}}sq^3\log p$, where $C_{\mathrm{net}}$ is a positive constant. By union bound,

$$P\left(\left|\sum_{i=1}^{n}\left\{\frac{1}{n}\mathrm{vec}(\mathbf{v})^T(\mathbf{f}_i\otimes\mathbf{I}_p)\mathbf{X}_i - \mathrm{vec}(\mathbf{v})^T(\mathbf{f}_i\otimes\mathbf{I}_p)\mathbf{\Gamma}^*\mathbf{f}_i\right\}\right| \geq t, \exists\mathbf{v}_j\in\mathcal{L}(s)\cap\mathbb{S}^{pq-1}\right)$$

$$\leq 2M_{\mathrm{net}}\exp(-c_1 n t^2) \leq 2\exp(C_{\mathrm{net}}sq^3\log p - c_1 n t^2).$$

Therefore for all $\mathbf{v}_j \in \mathcal{L}(s) \cap \mathbb{S}^{pq-1}$,

$$
\begin{aligned}
\langle \frac{1}{n}\sum_{i=1}^{n}\mathbf{X}_i\mathbf{f}_i^T, \mathbf{v}_j\rangle_F &= \sum_{i=1}^{n}\frac{1}{n}\operatorname{vec}(\mathbf{v}_j)^T(\mathbf{f}_i \otimes \mathbf{I}_p)\mathbf{X}_i \\
&\leq \sum_{i=1}^{n}\frac{1}{n}\operatorname{vec}(\mathbf{v}_j)^T(\mathbf{f}_i \otimes \mathbf{I}_p)\mathbf{\Gamma}^*\mathbf{f}_i + t \leq \sum_{i=1}^{n}\frac{1}{n}\|(\mathbf{f}_i \otimes \mathbf{I}_p)\mathbf{\Gamma}^*\|_2 M_4 + t \\
&\leq M_4^2 M_b + t.
\end{aligned}
$$

Then, for any $\operatorname{vec}(\mathbf{v}) \in \mathcal{L}(s) \cap \mathbb{S}^{pq-1}$, there exists a $\mathbf{v}_j$ such that

$$
\|\mathbf{v} - \mathbf{v}_j\|_F \leq \frac{1}{2}, \quad \frac{\operatorname{vec}(\mathbf{v} - \mathbf{v}_j)}{\|\mathbf{v} - \mathbf{v}_j\|_F} \in \mathcal{L}(s) \cap \mathbb{S}^{pq-1},
$$

and

$$
\langle \frac{1}{n}\sum_{i=1}^{n}\mathbf{X}_i\mathbf{f}_i^T, \mathbf{v}\rangle_F \leq \langle \frac{1}{n}\sum_{i=1}^{n}\mathbf{X}_i\mathbf{f}_i^T, \mathbf{v}_j\rangle_F + \langle \frac{1}{n}\sum_{i=1}^{n}\mathbf{X}_i\mathbf{f}_i^T, \mathbf{v} - \mathbf{v}_j\rangle_F.
$$

Taking supremum of both sides, we have

$$
\sup_{\mathbf{v}\in\mathcal{L}(s)\cap\mathbb{S}^{pq-1}} \langle \frac{1}{n}\sum_{i=1}^{n}\mathbf{X}_i\mathbf{f}_i^T, \mathbf{v}\rangle_F \leq 2(M_4^2 M_b + t),
$$

with probability at least $1 - 2\exp(C_{\text{net}}sq^3\log p - c_1 nt^2)$. Let $t = (\frac{2C_{\text{net}}sq^3\log p}{nc_1})^{1/2}$. When $n > Csq^3\log p$, with probability at least $1 - o(1)$,

$$
\sup_{\mathbf{v}\in\mathcal{L}(s)\cap\mathbb{S}^{pq-1}} \langle \frac{1}{n}\sum_{i=1}^{n}\mathbf{X}_i\mathbf{f}_i^T, \mathbf{v}\rangle_F \leq 2(M_4^2 M_b + \frac{2C_{\text{net}}}{c_1 C}).
$$

$\square$

**Lemma F.4.** *Suppose that $\widehat{\mathbf{B}}, \mathbf{B}^* \in \mathbb{R}^{p\times q}$ with $\operatorname{rank}(\widehat{\mathbf{B}}) = \operatorname{rank}(\mathbf{B}^*) = d$. Let $\widehat{\boldsymbol{\beta}}, \boldsymbol{\beta}$ be the top-d left singular vectors of $\widehat{\mathbf{B}}, \mathbf{B}^* \in \mathbb{R}^{p\times q}$ and $\sigma_1 \geq \cdots \geq \sigma_d$ be the singular values of $\mathbf{B}^*$. Assume $\|\mathbf{B}^* - \widehat{\mathbf{B}}\|_2 \leq C_B$. Then*

$$
\|\mathbf{P}_{\boldsymbol{\beta}} - \mathbf{P}_{\widehat{\boldsymbol{\beta}}}\|_F \leq \sqrt{2d}\frac{4\sigma_1 + 2C_B}{\sigma_d^2}\|\mathbf{B}^* - \widehat{\mathbf{B}}\|_F.
$$

*Proof.* Let $\boldsymbol{\beta}^T\boldsymbol{\beta} = \mathbf{M}\mathbf{D}\mathbf{N}^T$ denote the singular value decomposition of $\boldsymbol{\beta}^T\boldsymbol{\beta}$, where $\mathbf{M}, \mathbf{N} \in \mathbb{R}^{d\times d}$ and $\mathbf{D} = \operatorname{diag}(\omega_1, \ldots, \omega_d)$. Define the principal angles between the subspaces spanned by $\widehat{\boldsymbol{\beta}}$ and $\boldsymbol{\beta}$ as $(\phi_1, \ldots, \phi_d) = (\cos^{-1}\omega_1, \ldots, \cos^{-1}\omega_d)$, where $\omega_1 \geq \cdots \geq \omega_d$ are the singular values of $\widehat{\boldsymbol{\beta}}^T\boldsymbol{\beta}$. And define $\sin\boldsymbol{\Phi}(\widehat{\boldsymbol{\beta}}, \boldsymbol{\beta}) = \operatorname{diag}(\sin\phi_1, \ldots, \sin\phi_d)$. Then according to Theorem 3 (Yu et al., 2015),

$$
\begin{aligned}
\|\sin\boldsymbol{\Phi}(\widehat{\boldsymbol{\beta}}, \boldsymbol{\beta})\|_F &\leq \frac{2(2\sigma_1 + \|\widehat{\mathbf{B}} - \mathbf{B}^*\|_2)\min\{\sqrt{d}\|\widehat{\mathbf{B}} - \mathbf{B}^*\|_2, \|\widehat{\mathbf{B}} - \mathbf{B}^*\|_F\}}{\sigma_d^2} \\
&\leq \frac{4\sigma_1 + 2C_B}{\sigma_d^2}\sqrt{d}\|\widehat{\mathbf{B}} - \mathbf{B}^*\|_F.
\end{aligned}
$$

Then

$$
\begin{aligned}
\|\mathbf{P}_{\boldsymbol{\beta}} - \mathbf{P}_{\widehat{\boldsymbol{\beta}}}\|_F &= \|\boldsymbol{\beta}\boldsymbol{\beta}^T - \widehat{\boldsymbol{\beta}}\widehat{\boldsymbol{\beta}}^T\|_F = \sqrt{\operatorname{tr}[(\boldsymbol{\beta}\boldsymbol{\beta}^T - \widehat{\boldsymbol{\beta}}\widehat{\boldsymbol{\beta}}^T)^T(\boldsymbol{\beta}\boldsymbol{\beta}^T - \widehat{\boldsymbol{\beta}}\widehat{\boldsymbol{\beta}}^T)]} \\
&= \sqrt{\operatorname{tr}(\boldsymbol{\beta}\boldsymbol{\beta}^T + \widehat{\boldsymbol{\beta}}\widehat{\boldsymbol{\beta}}^T) - 2\operatorname{tr}(\widehat{\boldsymbol{\beta}}^T\boldsymbol{\beta}\boldsymbol{\beta}^T\widehat{\boldsymbol{\beta}})} = \sqrt{2d - 2\operatorname{tr}(\mathbf{M}\mathbf{D}^2\mathbf{M}^T)} \\
&= \sqrt{2}\left(d - \sum_{i=1}^{d}\omega_i^2\right)^{1/2} = \sqrt{2}\left(\sum_{i=1}^{d}\sin^2\phi_i\right)^{1/2} = \sqrt{2}\|\sin\boldsymbol{\Phi}(\widehat{\boldsymbol{\beta}}, \boldsymbol{\beta})\|_F \\
&\leq \sqrt{2}\frac{4\sigma_1 + 2C_B}{\sigma_d^2}\sqrt{d}\|\widehat{\mathbf{B}} - \mathbf{B}^*\|_F,
\end{aligned}
$$

where we use $\sin(\cos^{-1}\omega_i) = \sqrt{1 - \omega_i^2}$.

$\square$

**Lemma F.5.** *Let* $a^2 = \frac{2M_2^{3/2}}{\sqrt{M_1}}$, $b = 2\sqrt{M_2} + \frac{M_2 + 2\sqrt{M_2}}{\sqrt{M_1}}$. *If* $d_F(\boldsymbol{\theta}, \boldsymbol{\theta}^*) \vee \|\mathbf{B}_1 - \mathbf{B}_1^*\|_F \vee \|\mathbf{B}_2 - \mathbf{B}_2^*\|_F < r\Omega$, $\mathrm{vec}(\boldsymbol{\Gamma}_w - \boldsymbol{\Gamma}_w^*) \in \mathcal{L}(s)$, *and* $r < |c_0 - c_\pi|/\Omega \wedge C_b \wedge \frac{1}{a}(\sqrt{C_d - 1/\sqrt{M_1} + \frac{b^2}{4a^2}} - \frac{b}{2a})$, *then* $\boldsymbol{\theta} \in \mathcal{B}_{con}(\boldsymbol{\theta}^*)$.

*Proof.* Recall that

$$\mathcal{B}_{con}(\boldsymbol{\theta}^*) = \{\boldsymbol{\theta} : \pi_w \in (c_0, 1 - c_0), \|\boldsymbol{\Gamma}_w - \boldsymbol{\Gamma}_w^*\|_F \le C_b\Omega,$$
$$(1 - C_d)\Omega^2 \le |\mathrm{tr}(\boldsymbol{\delta}_w(\boldsymbol{\Gamma})\widehat{\boldsymbol{\Sigma}}_{\mathbf{f}}(\boldsymbol{\Gamma}_2 - \boldsymbol{\Gamma}_1)^T)| \le (1 + C_d)\Omega^2,$$
$$\mathrm{vec}(\boldsymbol{\Gamma}_w - \boldsymbol{\Gamma}_w^*) \in \mathcal{L}(s), w = 1, 2\},$$

where $\boldsymbol{\delta}_w(\boldsymbol{\Gamma}) = \boldsymbol{\Gamma}_w^* - (\boldsymbol{\Gamma}_2 + \boldsymbol{\Gamma}_1)/2$, and $\Omega = \sqrt{\mathrm{tr}[(\boldsymbol{\Gamma}_2^* - \boldsymbol{\Gamma}_1^*)\widehat{\boldsymbol{\Sigma}}_{\mathbf{f}}(\boldsymbol{\Gamma}_2^* - \boldsymbol{\Gamma}_1^*)^T]}$.

Since $\pi_1^* \in (c_\pi, 1 - c_\pi)$, when $|\pi_1 - \pi_1^*| < r\Omega < |c_0 - c_\pi|$, we have $c_\pi - |c_0 - c_\pi| < \pi_1 < 1 - c_\pi + |c_0 - c_\pi|$. Using $c_0 < c_\pi$, $\pi_1 \in (c_0, 1 - c_0)$. By definition of $r$, $\|\boldsymbol{\Gamma}_w - \boldsymbol{\Gamma}_w^*\|_F \le r\Omega \le C_b\Omega$.

Note that

$$|\Omega^2 - \mathrm{tr}(\boldsymbol{\delta}_w(\boldsymbol{\Gamma})\widehat{\boldsymbol{\Sigma}}_{\mathbf{f}}(\boldsymbol{\Gamma}_2 - \boldsymbol{\Gamma}_1)^T)|$$
$$= |\mathrm{tr}[(\boldsymbol{\Gamma}_2^* - \boldsymbol{\Gamma}_1^*)\widehat{\boldsymbol{\Sigma}}_{\mathbf{f}}(\boldsymbol{\Gamma}_2^* - \boldsymbol{\Gamma}_1^*)^T] - \mathrm{tr}([\boldsymbol{\Gamma}_w^* - (\boldsymbol{\Gamma}_2 + \boldsymbol{\Gamma}_1)/2]\widehat{\boldsymbol{\Sigma}}_{\mathbf{f}}(\boldsymbol{\Gamma}_2 - \boldsymbol{\Gamma}_1)^T)|$$
$$= |\mathrm{tr}[\widehat{\boldsymbol{\Sigma}}_{\mathbf{f}}\{(\boldsymbol{\Gamma}_2^* - \boldsymbol{\Gamma}_1^*)^T(\boldsymbol{\Gamma}_2^* - \boldsymbol{\Gamma}_1^*) - (\boldsymbol{\Gamma}_2 - \boldsymbol{\Gamma}_1)^T(\boldsymbol{\Gamma}_2^* - \boldsymbol{\Gamma}_1^*)\}]| +$$
$$\quad |\mathrm{tr}[\widehat{\boldsymbol{\Sigma}}_{\mathbf{f}}\{(\boldsymbol{\Gamma}_2 - \boldsymbol{\Gamma}_1)^T(\boldsymbol{\Gamma}_2^* - \boldsymbol{\Gamma}_1^*) - (\boldsymbol{\Gamma}_2 - \boldsymbol{\Gamma}_1)^T[\boldsymbol{\Gamma}_w^* - (\boldsymbol{\Gamma}_2 + \boldsymbol{\Gamma}_1)/2]\}]|$$
$$= \underbrace{|\mathrm{tr}[\widehat{\boldsymbol{\Sigma}}_{\mathbf{f}}(\boldsymbol{\Gamma}_2^* - \boldsymbol{\Gamma}_1^* - \boldsymbol{\Gamma}_2 + \boldsymbol{\Gamma}_1)^T(\boldsymbol{\Gamma}_2^* - \boldsymbol{\Gamma}_1^*)]|}_{(I)} +$$
$$\quad \underbrace{|\mathrm{tr}[\widehat{\boldsymbol{\Sigma}}_{\mathbf{f}}(\boldsymbol{\Gamma}_2 - \boldsymbol{\Gamma}_1)^T[\boldsymbol{\Gamma}_2^* - \boldsymbol{\Gamma}_1^* - \boldsymbol{\Gamma}_w^* - (\boldsymbol{\Gamma}_2 + \boldsymbol{\Gamma}_1)/2]\}]|}_{(II)}.$$

We have

$$(I) = |\mathrm{vec}(\boldsymbol{\Gamma}_2^* - \boldsymbol{\Gamma}_1^* - \boldsymbol{\Gamma}_2 + \boldsymbol{\Gamma}_1)^T \mathrm{vec}[(\boldsymbol{\Gamma}_2^* - \boldsymbol{\Gamma}_1^*)\widehat{\boldsymbol{\Sigma}}_{\mathbf{f}}]$$
$$\le \|\boldsymbol{\Gamma}_2^* - \boldsymbol{\Gamma}_1^* - \boldsymbol{\Gamma}_2 + \boldsymbol{\Gamma}_1\|_F \cdot \|(\boldsymbol{\Gamma}_2^* - \boldsymbol{\Gamma}_1^*)\widehat{\boldsymbol{\Sigma}}_{\mathbf{f}}\|_F$$
$$\le 2r\Omega \cdot \sqrt{M_2}\Omega = 2\sqrt{M_2}r\Omega^2.$$

Since

$$\|(\boldsymbol{\Gamma}_2 - \boldsymbol{\Gamma}_1)\widehat{\boldsymbol{\Sigma}}_{\mathbf{f}}^{1/2}\|_F \le \|(\boldsymbol{\Gamma}_2^* - \boldsymbol{\Gamma}_1^*)\widehat{\boldsymbol{\Sigma}}_{\mathbf{f}}^{1/2}\|_F + \|(\boldsymbol{\Gamma}_2 - \boldsymbol{\Gamma}_2^* - \boldsymbol{\Gamma}_1 + \boldsymbol{\Gamma}_1^*)\widehat{\boldsymbol{\Sigma}}_{\mathbf{f}}^{1/2}\|_F \le \Omega + 2\sqrt{M_2}r\Omega,$$

we have

$$\|\boldsymbol{\Gamma}_2 - \boldsymbol{\Gamma}_1\|_F \le \frac{1}{\sqrt{M_1}}\|(\boldsymbol{\Gamma}_2 - \boldsymbol{\Gamma}_1)\widehat{\boldsymbol{\Sigma}}_{\mathbf{f}}^{1/2}\|_F \le \frac{1 + 2\sqrt{M_2}r}{\sqrt{M_1}}\Omega.$$

And

$$\|\boldsymbol{\Gamma}_2 + \boldsymbol{\Gamma}_1\|_F = \|\boldsymbol{\Gamma}_2 - \boldsymbol{\Gamma}_2^* + \boldsymbol{\Gamma}_1 - \boldsymbol{\Gamma}_1^* + \boldsymbol{\Gamma}_2^* + \boldsymbol{\Gamma}_1^*\|_F \le 2r\Omega + 2M_b.$$

When $\Omega > 16M_2M_b$, for $w = 1$,

$$(II) = |\mathrm{vec}(\boldsymbol{\Gamma}_2 - \boldsymbol{\Gamma}_1)^T \mathrm{vec}([\boldsymbol{\Gamma}_2^* - 2\boldsymbol{\Gamma}_1^* - (\boldsymbol{\Gamma}_2 + \boldsymbol{\Gamma}_1)/2]\widehat{\boldsymbol{\Sigma}}_{\mathbf{f}})|$$
$$\le \|\boldsymbol{\Gamma}_2 - \boldsymbol{\Gamma}_1\|_F \cdot \|[\boldsymbol{\Gamma}_2^* - 2\boldsymbol{\Gamma}_1^* - (\boldsymbol{\Gamma}_2 + \boldsymbol{\Gamma}_1)/2]\widehat{\boldsymbol{\Sigma}}_{\mathbf{f}}\|_F$$
$$\le \frac{1 + 2\sqrt{M_2}r}{\sqrt{M_1}}\Omega \cdot M_2(3M_b + r\Omega + M_b)$$
$$\le \frac{1 + 2\sqrt{M_2}r}{\sqrt{M_1}}M_2r\Omega^2 + \frac{1 + 2\sqrt{M_2}r}{\sqrt{M_1}}4M_2M_b\Omega$$
$$\le \frac{1 + 2\sqrt{M_2}r}{\sqrt{M_1}}(M_2r + \frac{1}{4})\Omega^2.$$

If $w = 2$,

$$
\begin{aligned}
(II) &= |\operatorname{vec}(\boldsymbol{\Gamma}_2 - \boldsymbol{\Gamma}_1)^T \operatorname{vec}([-\boldsymbol{\Gamma}_1^* - (\boldsymbol{\Gamma}_2 + \boldsymbol{\Gamma}_1)/2]\widehat{\boldsymbol{\Sigma}}_{\mathbf{f}})| \\
&\leq \|\boldsymbol{\Gamma}_2 - \boldsymbol{\Gamma}_1\|_F \cdot \|[\boldsymbol{\Gamma}_1^* + (\boldsymbol{\Gamma}_2 + \boldsymbol{\Gamma}_1)/2]\widehat{\boldsymbol{\Sigma}}_{\mathbf{f}}\|_F \\
&\leq \frac{1 + 2\sqrt{M_2}r}{\sqrt{M_1}}\Omega \cdot M_2(M_b + r\Omega + M_b) \\
&\leq \frac{1 + 2\sqrt{M_2}r}{\sqrt{M_1}}(M_2 r + \frac{1}{4})\Omega^2.
\end{aligned}
$$

Let $a^2 = \frac{2M_2^{3/2}}{\sqrt{M_1}}$, $b = 2\sqrt{M_2} + \frac{2M_2 + \sqrt{M_2}}{2\sqrt{M_1}}$ and assume $1/(4\sqrt{M_1}) < C_d$. Then $a^2 r^2 + br < C_d - 1/(4\sqrt{M_1})$ when $r < \frac{1}{a}(\sqrt{C_d - 1/(4\sqrt{M_1}) + \frac{b^2}{4a^2}} - \frac{b}{2a})$. Therefore, we have

$$
\begin{aligned}
|\Omega^2 - \operatorname{tr}(\boldsymbol{\delta}_w(\boldsymbol{\Gamma})\widehat{\boldsymbol{\Sigma}}_{\mathbf{f}}(\boldsymbol{\Gamma}_2 - \boldsymbol{\Gamma}_1)^T)| &\leq 2\sqrt{M_2}r\Omega^2 + \frac{1 + 2\sqrt{M_2}r}{\sqrt{M_1}}(M_2 r + \frac{1}{4})\Omega^2 \\
&\leq \left[\frac{2M_2^{3/2}}{\sqrt{M_1}}r^2 + (2\sqrt{M_2} + \frac{2M_2 + \sqrt{M_2}}{2\sqrt{M_1}})r + \frac{1}{4\sqrt{M_1}}\right]\Omega^2 \leq C_d\Omega^2.
\end{aligned}
$$

$\square$

# G. Additional Discussion

The mixture PFC method is model-based and belongs to the linear heterogeneous SDR method. In contrast, nonlinear SDR aims to find a vector-valued function $\mathbf{g}$ such that $Y \perp\!\!\!\perp \mathbf{X} \mid \mathbf{g}(\mathbf{X})$. Traditional nonlinear SDR methods often combine kernel tricks and linear SDR techniques (Wu, 2008; Hsing & Ren, 2009; Li et al., 2011). However, these approaches have common computational challenges, when computing eigenvectors or inverse of $n \times n$ or $p \times p$ matrices, making them infeasible for large-scale high-dimensional data. Deep learning, with its proven success in various domains, offers promising alternatives for nonlinear SDR. The auto-encoder (Hinton & Salakhutdinov, 2006; Zong et al., 2018) is the most representative example of deep learning for unsupervised dimension reduction. Recently, several deep SDR methods have emerged, leveraging the power of deep neural networks to address the above challenges (Banijamali et al., 2018; Liang et al., 2022; Kapla et al., 2022; Huang et al., 2024; Chen et al., 2024).

We suggest two strategies to extend the linear heterogeneous SDR to nonlinear settings through deep learning. The first strategy is inspired by Kwon et al. (2024) and addresses semi-supervised scenarios with both labeled data $\{(\mathbf{X}_i, Y_i)\}_{i=1}^n$ and unlabeled data $\{(\mathbf{X}_i)\}_{i=n+1}^N$. Then model assumes the following structure:

$$
\begin{aligned}
&Y \perp\!\!\!\perp \mathbf{X} \mid (\mathbf{g}_w(\mathbf{X}), W = w), \quad \Pr(W \mid Y, \mathbf{X}) = \Pr(W \mid Y, \mathbf{g}_w(\mathbf{X})) \\
&\mathbf{X} \mid (\widetilde{W} = \widetilde{w}) \sim N(\boldsymbol{\mu}_{\widetilde{w}}, \boldsymbol{\Sigma}_{\widetilde{w}}), w, \widetilde{w} = 1, \dots, K \\
&\Pr(W = w \mid \widetilde{W} = \widetilde{w}) = \pi_{w|\widetilde{w}}.
\end{aligned}
$$

The key idea is to use the Gaussian mixture model on the unlabeled data to infer the joint distribution of $(\mathbf{X}, \widetilde{W})$ and then apply any proposed deep SDR method to learn $\mathbf{g}_w$. The procedure is as follows.

Step 1: Learn the joint distribution of $(\mathbf{X}, \widetilde{W})$ using GMM fitted to the unlabeled data.

Step 2: Assign labeled data $\{(\mathbf{X}_i, Y_i)\}_{i=1}^n$ to the $K$ clusters defined by $\widetilde{W}$ using the estimate distribution of $\widetilde{W}|\mathbf{X}$.

Step 3: Estimate the nonlinear SDR using any deep SDR method for each cluster.

Step 4: Estimate the transition matrix $\Pi_{W|\widetilde{W}} = (\pi_{w|\widetilde{w}})$.

The second strategy combines a compression network and an estimation network, similar to the deep auto-encoding Gaussian mixture model (DAGMM) (Zong et al., 2018). The compression network is a supervised auto-encoder (Le et al., 2018),

designed to perform dimension reduction while preserving the nonlinear SDR structure. The innermost layer incorporates a supervised loss to ensure the reduced representation $\mathbf{g}(\mathbf{X})$ satisfies the conditional independence condition in nonlinear SDR. Various dependence measures can be used to construct the loss function, such as distance covariance (Székely et al., 2007), martingale difference divergence (Shao & Zhang, 2014), and generalized martingale difference divergence (Li et al., 2023). Then the estimation network uses the learned low-dimensional vector $\mathbf{g}(\mathbf{X})$ and the response to predict clusters. Unlike DAGMM, this step employs a supervised clustering model rather than the Gaussian mixture model Zong et al. (2018). To evaluate the clustering quality of the estimation network, the log-likelihood of the cluster assignments can be computed. Both strategies highlight the potential of deep learning to effectively extend SDR to nonlinear, heterogeneous, and high-dimensional settings.

