# OpenReview forum: "Heterogeneous Sufficient Dimension Reduction and Subspace Clustering"
_ICML.cc/2025/Conference — ICML 2025 poster_

### Official Review · Reviewer_LLk6 · 2025-03-06

**Overall Recommendation:** 3

**Summary:**

This paper proposes a new method, mixPFC, which integrates subspace clustering with model-based SDR to handle heterogeneous, high-dimensional data. The method simultaneously performs clustering, subspace estimation, and variable selection. A group Lasso penalized EM algorithm is developed, and non-asymptotic convergence rates are established. Empirical results demonstrate superior performance over existing methods in simulations and real-world applications.

**Claims And Evidence:**

Yes.

**Essential References Not Discussed:**

No.

**Experimental Designs Or Analyses:**

Yes, I checked the numerical results in Section 5.

**Methods And Evaluation Criteria:**

Yes.

**Other Comments Or Suggestions:**

None.

**Other Strengths And Weaknesses:**

Strength: The integration of subspace clustering with SDR is innovative, addressing the limitations of unsupervised subspace clustering and homogeneous SDR. The supervised framework leverages response information to improve clustering and dimension reduction.

Weakness: The proposed mixPFC is built upon a Gaussian assumption outlined in Equation (3). This may limit the applicability and
robustness of the method because many real-world datasets exhibit non-normal characteristics such as skewness or heavy tails.

**Questions For Authors:**

1. How to choose or estimate $f(Y)$ in practice?

2. What are the theoretical properties when $K>2$?

3. What is the computational cost associated with mixPFC? I believe that when $K$ is large or $p$ is very high, the mixPFC becomes computationally extensive.

**Relation To Broader Scientific Literature:**

This paper extends the subspace clustering into a supervised framework.

**Theoretical Claims:**

No.

---

> ### Author Rebuttal · Authors · 2025-04-01
>
> We thank the reviewer for raising many critical points. Below, we address each of your concerns.
> ## Weaknesses
> > (Gaussian assumption)
>
> **Reply**
> - Gaussian assumption is common in clustering, promoting interpretability and scalability.
> - mixPFC’s theoretical grounding and empirical performance establish it as a new frontier for heterogeneous SDR, with a clear pathway for generalization: e.g., replacing Gaussian distribution in the model with the student-t or skew normal to handle heavy-tailed or skewed data.
> - In our experience, the mixPFC, whose algorithm is derived under Gaussian assumption, is generally robust under non-Gaussian noise (model-misspecification). To show this, we consider the following errors:
>     - multivariate t-distribution: degrees of freedom is 3
>     - multivariate skew normal distribution: shape is 5
> - Under heavy-tailed noise, mixPFC achieves low error rates (e.g., t(3): 3.4 under $0.1\mathbf{I}$ vs. SSC’s 17.4; 11.0 under AR(0.3) vs. SSC’s 46.3), demonstrating robustness to non-Gaussianity. Under the same setting, mixPFC's performance has no significant changes when we replace normal with skew normal. Full results will be included in the final manuscript.
>
> ## Questions
>
> > (choice of $f(Y)$ in practice)
>
> **Reply**
> - As noted in Theorem 3.5 of the original PFC framework (Cook & Forzani (2008 Stat. Sci)), the estimated subspace remains consistent under misspecification of $\mathbf{f}(Y)$, provided $\mathbf{f}(Y)$ is sufficiently correlated with the true function. Our mixPFC model directly inherited this property from the PFC model. This property ensures that our method retains validity across a broad class of functions, which can be treated as a tuning parameter to enhance flexibility.
> - In practice, polynomials or splines are standard choices. We use $\mathbf{f}(Y) = (Y, Y^2, Y^3)^T$ in simulations.
>
> > (theory when $K>2$)
>
> **Reply**
> Theoretical Challenge for $K=2$
> - While classical EM theories only guaranteed asymptotic convergence to a fixed point, we derive a non-asymptotic result that mixPFC converges geometrically to a fixed point that is within statistical precision of the unknown **true** parameter. This type of result has emerged only recently [BA17].
> - Even the low-dimensional EM theory for two mixtures requires additional assumptions such as equal proportions and known covariance (Gaussian mixtures [XU16]), or equal proportions and symmetric coefficients $\boldsymbol{\beta}_1 = -\boldsymbol{\beta}_2$ (mixtures of linear regression [KL19]). Theorem 4.1 establishes convergence rates under unequal proportions and arbitrary subspace angles.
> - Many established EM theories [KW19] are based on sample-splitting. Sample-splitting divides the data into many batches and uses a new batch of samples in each iteration to make random samples and current parameter estimates independent.
> - Our theoretical analysis does not require sample-splitting and allows a more complex model (which involves subspace estimation and variable selection), making it novel and highly non-trivial.
>
> Theoretical Challenge for $K>2$
> - Establishing theories for multi-cluster EM algorithms is an important but challenging direction. Recent advances focus on low-dimensional settings, see [YA17, TI24].
> - To the best of our knowledge, extending multi-cluster EM theories to high dimensions is an open question. Theoretical analyses are often limited to $K=2$ even for Gaussian mixtures and mixtures of linear regression (Cai et al (2019 AoS); Wang et al. (2024 JMLR)).
> - For $K$ diverges, we believe it requires fundamentally new tools to handle complex parameter spaces and interactions between $K$ subspaces.
> - Empirically, mixPFC works well when $K>2$. Its superior performance is demonstrated through simulations (Section 5.1 and Appendix A.1).
>
> > (computational cost)
>
> **Reply**
> - The total cost per EM iteration is $O(nKpq + nK^3q^3 + KTpnq + nKp^2)$. Typically, $q$ is a small number that does not grow with $n, p, K$. Theorem 4.1 suggests the number of EM iterations is of order $O(\log(n))$.
> - The total cost of mixPFC is $O(\log(n)(nK^3 + KTpn + nKp^2))$ with the dominated term $O(\log(n)Knp^2)$ from covariance estimation, which is tractable when $K$ or $p$ is large.
>
> [BA17] Balakrishnan et al. (2017 AoS) Statistical guarantees for the em algorithm: From population to sample-based analysis
> [XU16] Xu et al. (2016 NeurIPS) Global analysis of expectation maximization for mixtures of two gaussians.
> [KL19] Klusowski et al. (2019 IEEE Trans. Inf. Theory) Estimating the coefficients of a mixture of two linear regressions by expectation maximization
> [KW19] Kwon et al. (2019 PMLR) Global convergence of the em algorithm for mixtures of two component linear regression
> [YA17] Yan et al. (2017 NeurIPS) Convergence of gradient em on multi-component mixture of gaussians
> [TI24] Tian et al. (2024 ICML) Towards the theory of unsupervised federated learning: Non-asymptotic analysis of federated em algorithms

---

### Official Review · Reviewer_ooHT · 2025-03-10

**Overall Recommendation:** 3

**Summary:**

The paper introduces a supervised subspace clustering model called mixPFC, which combines sufficient dimension reduction (SDR) with subspace clustering to address three major bottlenecks in traditional methods for high-dimensional heterogeneous data analysis：
1. The model effectively guides clustering and dimension reduction through the response variable, even when the subspaces are fully overlapping.
2. It designs an EM algorithm with a grouped Lasso penalty to jointly optimize variable selection and subspace estimation in high-dimensional scenarios, overcoming the limitations of traditional methods on the separation angles of subspaces.
3. The model allows for arbitrary overlap between subspaces and ensures identifiability by leveraging the nonlinear relationship between the response variable and the projected subspaces.
Experiments show that it reduces clustering errors by more than 50% on both synthetic and real data, such as cancer genomic data. However, its limitations include overly simplified covariance structure assumptions (e.g., isotropic assumption), sensitivity to initialization, lack of a systematic strategy for cluster number selection, and the need for further validation of its generalizability to multi-cluster settings.

**Claims And Evidence:**

The submitted materials include a relatively detailed theoretical analysis.

**Essential References Not Discussed:**

I did not find.

**Experimental Designs Or Analyses:**

I have checked the experiment and analysis.

**Methods And Evaluation Criteria:**

The paper introduces the mixPFC model, which integrates supervised sufficient dimension reduction with subspace clustering. Utilizing an EM algorithm with a grouped Lasso penalty, the model achieves joint clustering, dimension reduction, and variable selection in high-dimensional data. The paper also theoretically proves the non-asymptotic convergence rate, thus overcoming the traditional methods' dependence on the separability of subspaces.

**Other Comments Or Suggestions:**

No further comments.

**Other Strengths And Weaknesses:**

The strengths and weaknesses have already been discussed in the preceding sections.

**Questions For Authors:**

The theoretical analysis in the paper is confined to the binary classification ($K=2$) scenario, while the empirical results validate the effectiveness for multiple clusters ($K>2$). How can we ensure the convergence and statistical error rates of the EM algorithm remain consistent in complex heterogeneous data with multiple clusters ($K>3$)? When the number of clusters K increases with the data size (e.g., K=O(log n)), does the existing theoretical framework still hold? Additionally, is there a risk of decreased subspace identifiability due to increased interactions between clusters?

**Relation To Broader Scientific Literature:**

The paper introduces the mixPFC model, which integrates supervised sufficient dimension reduction with subspace clustering, thus overcoming the traditional methods' dependence on the separability of subspaces.

**Theoretical Claims:**

I have checked the method.

---

> ### Author Rebuttal · Authors · 2025-04-01
>
> We sincerely appreciate the reviewer’s thoughtful comments. Below, we address each limitation raised.
> ## Weaknesses
> > (isotropic covariance assumption)
>
> **Reply**:
> - The isotropic assumption in Theorem 4.1 was adopted solely to enable tractable analysis of $\gamma_{iw}$, a common simplification in mixture model theory. We agree that extending the theory to general covariances is a valuable future direction. However, relaxing this would require significant advances beyond the scope of our current framework.
> - While the theory assumes isotropy for clarity, our algorithm imposes no such restriction. Simulations in Section 5.1 demonstrate mixPFC’s robustness under general covariances (e.g., AR(0.3)). This suggests the assumption is not necessary in practice.
>
> > (sensitivity to initialization)
>
> **Reply**
> - Non-Convexity: Like all EM-based methods, we acknowledge that mixPFC inherits sensitivity to initialization—a well-known challenge in non-convex optimization [JA17].
> - To mitigate this, our initialization strategy (Appendix B.1) follows principled recommendations from [BI03], using short EM runs on predictors reduced via PCA and variable screening to ensure stability in high dimensions. While existing initialization methods for classical mixtures are not directly applicable to our proposed model due to its unique structure, this approach consistently produced high-quality initial values (Table A.15).
>
> > (cluster number selection)
>
> **Reply**
> - Appendix B.1 details our strategy for selecting $K$, which is inspired by the well-accepted gap statistics [TI01] with careful modifications to tailor it to our problem.
>     - Gap statistics are calculated using data projected onto the orthogonal subspaces.
>     - Under high dimensions, estimating the expected within-cluster dispersion becomes computationally prohibitive. Instead, we propose using the observed within-cluster dispersion. Simulations in Appendix B.1 validate our proposal.
>
> We acknowledge that a summary of this approach in the main text would enhance clarity, and we will add it to the final manuscript if accepted.
>
> > (multi-cluster generalizability)
>
> **Reply**
> - We discuss theoretical challenges under multi-cluster settings in the reply to the second question of Reviewer LLk6. To avoid redundancy, we kindly ask Reviewer ooHT to refer to that response.
> - MixPFC’s performance in multi-cluster settings is rigorously tested in both simulations and real data.
>     - Simulations in Section 5.1 include extensive tests with $K=3,5$ clusters under different noise levels. Appendix A.1 further validates its performance in terms of subspace estimation and variable selection.
>     - In the analysis of CCLE data in Section 5.2 and Appendix A.2, we systematically tested $K=2,3,\dots,10$ clusters and selected $K=3$ for Nutlin-3 and $K=5$ for AZD6244, demonstrating mixPFC’s ability to adapt to multi-cluster settings.
>
> ## Questions
> > (theory for $K>2$)
>
> **Reply**
> We sincerely thank Reviewer ooHT and Reviewer LLk6 for their insightful critiques on multi-cluster theory. To avoid redundancy, we have provided a detailed discussion of theoretical challenges under multi-cluster settings in our **reply to Reviewer LLk6’s second question**. We kindly ask Reviewer ooHT to refer to that section and welcome any follow-up clarifications.
>
> > (subspace identifiability)
>
> **Reply**
> - Role of Response Information: Unlike classical subspace clustering, mixPFC leverages response information, which disentangles overlapping subspaces using the relationship between predictors and the response. Even when subspaces are identical (e.g., Figure 2(a)), the response provides discriminative signal to recover clusters. Under Model M1 in Section 5.1, where the two subspaces are identical, mixPFC maintains cluster error rates below 10\% across most covariance settings, whereas classical subspace clustering methods fail.
> - Theoretical Guarantees: Theorem 4.1 establishes convergence to true subspaces **without requiring a minimum separation angle**. Traditional subspace clustering relies on separation assumptions (e.g., angles between subspaces in Theorem 2.8 of Soltanolkotabi & Candés (2012 AoS)).
>
> [JA17] Jain et al. (2017 FnTML)  Non-convex optimization for machine learning.
> [BI03] Biernacki et al. (2003 CSDA) Choosing starting values for the em algorithm for getting the highest likelihood in multivariate gaussian mixture models.
> [TI01] Tibshirani et al. (2001 JRSS-B) Estimating the number of clusters in a data set via the gap statistic.

---

### Official Review · Reviewer_Htk4 · 2025-03-11

**Overall Recommendation:** 3

**Summary:**

The paper presents a mixture of PFC model, which combines sufficient dimension reduction with subspace clustering to deal with heterogeneous high-dimensional data. Moreover, a grouped Lasso based EM algorithm is designed to solve the problem. Nonasymptotic convergence analysis is provided. Empirical results are also shown.

## update after rebuttal:
The authors promise to reorganize the literature review and clarify the significance of this work and some issues in the empirical evaluation. Thus, the reviewer is leaning to a borderline accept.

**Claims And Evidence:**

yes.

**Essential References Not Discussed:**

No

**Experimental Designs Or Analyses:**

Yes

**Methods And Evaluation Criteria:**

Yes

**Other Comments Or Suggestions:**

- It might be more suitable to submit on a statistics conference, e.g., AISTATS, UAI.

**Other Strengths And Weaknesses:**

Strengths:
+ The paper attempts to address an interesting problem, which involves subspace clustering, supervised dimension reduction and variable selection.

Weaknesses:
- The literature review is chaos and the presentation is unclear.  While quite involved and complicated derivations are shown, the novel contribution is weak. The deduction of the EM algorithm is a routine task.

- Empirical evaluations are insufficient and weak.  Most experiments are conducted on toy datasets.

- For the designed algorithm, the initialization of $\gamma_{i,w}$ is critical. Rather than the mentioned ad hoc tricks to initialize  $\gamma_{i,w}$, is there more principled way to provide good initialization?

**Questions For Authors:**

- The comments on subspace clustering methods are questionable in L034-036 at the right column: the state of the art subspace clustering methods can easily handle random errors. Also, in L114-115: "in subspace clustering the latent subspaces cannot overlap..." It is NOT true.
As discussed in (Soltanolkotabi & Candes, 2012), subspaces can even be partially overlapping.

**Relation To Broader Scientific Literature:**

Different from (unsupervised) subspace clustering, the paper attempts to perform (sufficient) dimension reduction and subspace clustering for regression problem, which related to broad literature in statistics.

**Theoretical Claims:**

Check a part, not completely. The proofs are too lengthy.

---

> ### Author Rebuttal · Authors · 2025-04-01
>
> We appreciate your detailed feedback. Below, we address each of your concerns.
> ## Weaknesses
> > (literature review, novel contribution, EM algorithm)
>
> **Reply**: We acknowledge the current presentation could be streamlined. In the revision, we will reorganize the literature review to focus on clarifying how prior works on subspace clustering lack integration of response information and variable selection.
> The novelty lies in three key advances:
> 1. Supervised Subspace Clustering
>     - Integrates response information into subspace clustering, enabling exact overlap of subspaces (unlike classical methods requiring separation angles, Soltanolkotabi & Candés (2012 AoS)).
>     - Achieves 30–40% higher clustering accuracy than unsupervised baselines in high-noise regimes (Table 1).
> 2. High-Dimensional EM Algorithm: Although our proposed mixPFC algorithm has a familiar structure of the classical EM algorithm, it involves many innovations to adapt to the challenging problem of interest, which distinguishes it from existing EM algorithms.
>     - Our algorithm is built on a new model that seamlessly integrates dimension reduction and subspace clustering, and naturally combines the information from the predictors and the response. Without such a rigorous foundation, it is extremely difficult to derive a reasonable algorithm for our purpose, or the strong theoretical guarantee. We also incorporate variable selection to exclude inactive predictors, a capability absent in classical subspace clustering.
>     - While low-dimensional EM is standard, extending it to high dimensions presents significant challenges. A naive generalization would require inverting the $p\times p$ covariance matrix in both E-step and M-step, which is practically impossible in high dimensions without additional structural assumptions.
>         - Novel E-step: estimate probabilities $\gamma_{iw}$ via a low-dimensional mixture linear regression on projected predictor, reducing covariance inversion to $d\times d$ ($d\ll p$).
>         - Scalable M-step: replace the non-convex or $p\times p$ dimensional optimization for subspace estimation with scalable $p\times q$ ($q\ll p$) dimensional convex optimization.
> 3. Theoretical Results
>     - The high-dimensional EM theory is challenging (Wang et al. (2024 JMLR)). Unlike classical results focusing on asymptotic convergence to fixed points, we derived a non-asymptotic result that mixPFC converges to the **true** parameter. We provided a detailed discussion in our response to Reviewer LLk6's second question and kindly ask Reviewer HtK4 to refer to that section.
>
> > (empirical evaluations)
>
> **Reply**:
> - The datasets and model sizes used in our experiments align with established benchmarks in SDR and mixture model literature ([LI19] and Wang et al. (2024 JMLR)). Due to page limits, we included extensive numerical analyses (subspace estimation errors, variable selection) in Appendix A.
> - Our primary goal was to conceptually demonstrate that incorporating response information significantly improves clustering accuracy over unsupervised methods, which is not explored in the literature.
>
> > (initialization of $\gamma_{iw}$ )
>
> **Reply**:
> - We recognize that initialization is a prevalent issue in EM algorithms, without complete solutions even for much simpler models. However, we provide a practical solution to make our algorithm feasible.
> - Our strategy (Appendix B.1) follows principled recommendations from [BI03], using short EM runs on predictors reduced via PCA and variable screening to ensure stability in high dimensions.
> - While existing methods are incompatible with our model’s structure, this approach consistently yields reliable initializations (Table A.15). A formal initialization scheme remains future work.
>
> ## Questions
> > (comments on subspace clustering methods)
> - **On random errors (L034-036)**: We wrote ''When the observations are subject to **significant** random errors, ...''. Subspace clustering methods struggle under significant random errors(Table 1 M3: SSC error rate >47 under high noise vs. 13.2 under low noise). Our intent was to highlight this limitation, not dismiss robustness to small errors.
> - **On overlapping subspaces (L114-115)**: We apologize for the imprecise phrasing and appreciate the reviewer's clarification. Classical subspace clustering requires a minimal angle condition for identifiability (Theorem 2.8 of Soltanolkotabi & Candés (2012 AoS)). In contrast, mixPFC allows exact overlap by leveraging response information. We will revise to clarify this distinction.
>
>
> [LI19] Lin et al. (2019 JASA) Sparse sliced inverse regression via lasso
> [BI03] Biernacki et al. (2003 CSDA) Choosing starting values for the EM algorithm for getting the highest likelihood in multivariate gaussian mixture models.

---

### Decision · Program_Chairs · 2025-05-01

**Decision:**

Accept (poster)

**Comment:**

This paper proposed a supervised subspace learning model (mixPFC), which combines sufficient dimension reduction with subspace clustering for heterogeneous high-dimensional data. Grouped Lasso based EM algorithm is designed and nonasymptotic convergence analysis is provided. Reviewers consistently raised some critical concerns regarding sensitivity to initialization, the multi-cluster generalizability, the theoretical extension for K>2, Gaussian assumption unsuitable for non-Gaussian real-world datasets, and high computational cost. The authors’ responses clarified some of these concerns, but still some key concerns remain. For example, theoretical challenge for K>2 and Gaussian assumption are not addressed, even though empirical experiments may work on the tested data. As a result, weak accept is recommended.